# LipidIN: a comprehensive repository for flash platform-independent annotation and reverse lipidomics

Hao Xu [1,2,12], Tianhang Jiang [1,12], Yuxiang Lin [3,12], Lei Zhang[1,12], Huan Yang[1,4], Xiaoyun Huang[5], Ridong Mao[1], Zhu Yang [6], Changchun Zeng [7], Shuang Zhao [8], Lijun Di [9], Wenbin Zhang[10], Jun Zeng [5] ✉, Zongwei Cai [6,11] ✉ & Shu-Hai Lin [1,2] ✉

Improving annotation accuracy, coverage, speed and depth of lipid profiles remains a significant challenge in traditional lipid annotation. We introduce LipidIN, an advanced framework designed for flash platform-independent annotation. LipidIN features a 168.5-million lipid fragmentation hierarchical library that encompasses all potential chain compositions and carbon-carbon double bond locations. The expeditious querying module achieves speeds exceeding one hundred billion queries per second across all mass spectral libraries. The lipid categories intelligence model is developed using three relative retention time rules, reducing false positive annotations and predicting unannotated lipids with a 5.7% estimated false discovery rate, covering 8923 lipids cross various species. More importantly, LipidIN integrates a Wide-spectrum Modeling Yield network for regenerating lipid fragment fingerprints to further improve accuracy and coverage with a 20% estimated recall boosting. We further demonstrate the utility of LipidIN in multiple tasks for lipid annotation and biomarker discovery in clinical cohorts.

Lipid structural information encompasses the subclass (head group), fatty acyl and alkyl/alkenyl composition (chain length and degree of unsaturation), C=C double-bond location, fatty acyl positional specificity, various substitutions and modifications, and geometric configuration[1–3]. Despite multiple annotation methods in liquid chromatography-mass spectrometry (LC-MS)-based untargeted metabolomics and lipidomics have been used to improve the coverage, ~90% of the metabolic features cannot be annotated[4]. The annotation strategies primarily rely on three key aspects: (1) forward prediction that involves constructing a molecular library via standard analysis or in silico fragmentation to generate synthetic data[5,6], (2) reverse prediction that entails inferring molecular properties or structures directly from the spectra[7,8], and (3) networking prediction that clusters similar spectra to identify neighborhoods of structurally

[1]The First Affiliated Hospital of Xiamen University, State Key Laboratory of Cellular Stress Biology, School of Life Sciences, XMU-HBN Skin Biomedical Research Center, Xiamen University, Xiamen, Fujian, China. [2]School of Medicine, National Institute for Data Science in Health and Medicine, Xiamen University, Xiamen, Fujian, China. [3]Department of Breast Surgery, Fujian Medical University Union Hospital, Fuzhou, Fujian Province, China. [4]School of Pharmaceutical Sciences, Xiamen University, Xiamen, Fujian, China. [5]College of Ocean Food and Biological Engineering, Jimei University, Xiamen, China. [6]State Key Laboratory of Environmental and Biological Analysis, Department of Chemistry, Hong Kong Baptist University, Kowloon, Hong Kong, China. [7]Department of General Medicine, Shenzhen Longhua District Central Hospital, Shenzhen, China. [8]Xiamen Meliomics Co., Ltd., Xiamen, Fujian, China. [9]Department of Biological Sciences, Faculty of Health Sciences, University of Macau, Macau, China. [10]Department of Occupational and Environmental Health and the Ministry of Education Key Lab of Hazard Assessment and Control in Special Operational Environment, School of Public Health, Fourth Military Medical University, Xi'an, China. [11]Eastern Institute of Technology, Ningbo, China. [12]These authors contributed equally: Hao Xu, Tianhang Jiang, Yuxiang Lin, Lei Zhang. ✉ e-mail: junzeng@jmu.edu.cn; zwcai@hkbu.edu.hk; shuhai@xmu.edu.cn

analogous compounds[9,10]. Based on the structural similarity of molecules within each lipid subclass, theoretical predictions of MS/MS spectral patterns can be derived by interpreting the fragmentation relationships. Consequently, automated lipid annotation currently depends on MS/MS similarity calculation and decision tree annotation mass spectra to spectral libraries, such as MS-DIAL[11], LipidSearch[12], Spectral Entropy[13,14], and LipidMatch[8] (Supplementary Table 1).

However, several problems exist in current lipidomics annotation. First, there are limitations in the matching algorithms. Both classic dot-product (cosine) and spectral entropy similarity algorithms overlook actual significance of feature peaks, causing more or less feature redundancy[15]. Second, it is challenging to obtain definitive information from low-abundance signals, particularly when it is necessary to confirm the presence of characteristic fragments, such as the neutral loss of specific fragments. Third, beyond subclasses (head groups) and chain compositions, more in-depth structural information, such as double-bond locations, cannot be discerned by most current annotation tools. Fourth, considering the potential discrepancies between theoretical and actual spectra generated from different sample matrices, instruments, and analytical methods, researchers have to use personalized local databases for the effective accumulation of empirical knowledge. The above issues continue to pose a significant hindrance to improving annotation accuracy and coverage in lipidomics studies.

Retention time (RT) or retention orders (RO) in LC-MS analysis is associated with substructures within a molecule, chromatographic columns, the composition of eluents, gradient, and column temperature[16]. We hypothesized that deep learning models could extract the mapping relationship between lipid structure features (e.g., head group, chain length, and degree of unsaturation of each fatty acyl or alkyl/alkenyl composition) and RT (or RO), and that combining MS/MS- and RT-based scores could significantly improve annotation performance in LC-MS/MS-based lipidomics analyses. The advances of artificial intelligence (AI) promote metabolite annotation from complex mass spectrometric data particularly MS/MS fragmentation search and MS/MS-explainable formula candidates[17–20]. For instance, a multi-layer perceptron (MLP) is a type of artificial neural network consisting of multiple layers of neurons[21]. Ziming Liu et al. recently reported a neural network called Kolmogorov-Arnold networks (KAN) as alternatives to MLPs[22,23], inspired by Kolmogorov-Arnold representation theorem[24].

In this work, we introduce LipidIN, namely lipidomics integration, an advanced tool for rapid lipid annotation and high-accuracy reverse lipid fingerprint spectrogram regeneration. Compared to existing tools, LipidIN demonstrates superior performance in lipid coverage, including 121 subclasses, mass spectrometry library querying speed, prediction accuracy, false positive removal rate, and comprehensive annotation of mass spectrometric data, including lipid molecules and isomers with different C=C locations. Furthermore, LipidIN incorporates a Wide-spectrum Modeling Yield network (WMYn) as "reverse lipidomics", which regenerates highly accurate fingerprint spectrograms and easily transferable pre-trained models, independent of sample matrices, instruments, and analytical methods. In a word, LipidIN provides an efficient and reliable method for lipidomics research.

## Results
### Overview of LipidIN framework
LipidIN contains a five-level spectral fragmentation tree encompassing 168.5 million lipids, including both the Paternò-Büchi (P-B) reaction[25,26] and electron-activated dissociation (EAD)[27] for lipid isomers with different C=C locations (Fig. 1a, Supplementary Fig. 1 and 2). The pre-processing stage involves the use of MSconvert[28] to convert raw data to .mzML files, and the creation of mass spectrometric information lists using RaMS[29], which is a reverse lipidomics approach allowing

annotation without peak picking first. After this step, an expeditious querying (EQ) module, based on a non-informative prior greedy algorithm to search theoretical spectra at an ultra-fast speed, was used to calculate $Score_{matched}$ and $Score_{ratio}$ for decision trees (Fig. 1b, c). This enables us to match and highlight the importance of mass spectrometric features within the 168.5-million theoretical lipid library at a querying speed of over one hundred billion times per second. Utilizing $Score_{matched}$ and $Score_{ratio}$ as a priori information, we further leveraged a lipid categories intelligence (LCI) module to extrapolate relationships between carbon number, double bond equivalents (DBEs), and intraclass relative RT. After training, a multi-simulation model is used to reduce false positives and predict candidate annotations without MS/MS fragments (Figs. 1d and 1f). Building on this output, we further designed the WMYn to regenerate high-accuracy fingerprint spectra, which refer to unique ion features of lipid molecules and the common characteristic spectra shared across various mass spectrometry systems, and easy-migration pre-trained model. This network maps the mass-to-charge ratios and intensities of multiple batches into a shared latent space by integrating experimental results from a large number of batches using network layers and self-attention encoder for feature learning (Fig. 1e, g).

### Fragmentation trees for a five-level hierarchical library
First of all, we need to establish fragmentation trees for a 5-level hierarchical library, which is a mass spectrometry database categorized by each $MS^2$ characteristic peak based on lipid structures. By screening fragment ions and structural characteristics of each lipid subclass in published libraries (more references listed in Supplementary Data 1)[30], we classified the self-computed feature peaks into five levels, including (1) precursor ion and head group, (2) fatty acyl and alkyl/alkenyl chain (hereafter side chain), (3) head group neutral loss (NL), (4) side chain NL, and (5) regeneration of fragment fingerprints (Fig. 1a).

The 1st-level library is the precursor ion and the lipid head group fragment ion. Derived from the two characteristic peaks, the lipid subclasses and the lipid adduct ions, as well as the total carbon chains and total DBEs can be determined. The 2nd-level is the side chains and other peaks representing specific chain compositions. The 3rd-level of head group NL is complementary to the 1st-level. The 4th-level of side chain NL is complementary to the 2nd-level. In our analysis of C=C locations, we performed the P-B reaction using LC-MS/MS analysis[26] (Supplementary Fig. 1) and employed EAD in SCIEX ZenoTOF 7600 system for lipidomics[31,32] (Supplementary Fig. 2). The fragmentation trees of lipids generated from both methods are integrated into 1st- to 4th-level hierarchical library.

The 5th-level library represents for molecular fingerprint peaks, by summarizing the $MS^1$- and $MS^2$-features in different ionization modes and chromatographic conditions. To this end, we designed the WMYn to regenerate highly accuracy fingerprint spectra and execute cross-platform migration, resulting in enhanced subsequent annotations. We conceptualized it as "reverse lipidomics" and applied this strategy for regenerating the 5th-level library during data processing by small-sample learning, which consists of three training stages.

Overall, the generalized fragmentation rules described above have been used to construct libraries for 121 lipid subclasses, containing 168.5 million theoretical lipid fragments (Supplementary Data 1).

### Relative retention time regularities of lipid subclasses
In principle, the intraclass relative RT played an important role in lipid annotation. Thereby, three rules have been statistically analyzed and defined by using over 100 published datasets, covering various biological samples[11,33–36] (Fig. 2 and Supplementary Data 2): (1) The intrasubclass relative RT within same DBEs show a second-order polynomial trend line with carbon numbers, which was generally

recognized as the equivalent carbon number (ECN), in line with previous reports[37,38]; (2) The fitted function of intra-subclass lipids within subclasses with different DBEs indicates a parallel relationship, and the degree of unsaturation is positively correlated with the intercept of the fitted equation, defined as intra-subclass unsaturation parallelism (IUP); (3) Isomers with different separated chain compositions can be fitted by different functions, defined as equivalent separated carbon number (ESCN).

Next, we took a published dataset as an example to illustrate the above rules[34] (Fig. 2a). After polynomial fitting, FAs with different levels of DBEs show a clear trend of increasing functions (Fig. 2b). By setting appropriate confidence intervals, we observed that these intervals do not intersect, which is characteristic of ECN and IUP. For instance, PCs with four DBEs were highly consistent with ESCN (Fig. 2c). In this dataset, two DBE combinations with 0:4 and 1:3 were separately annotated and well fitted. We also examined the consistency of three

rules with the change in carbon chain and unsaturation levels. Furthermore, we analyzed the >100 published datasets and found that, most datasets highly satisfy these three rules, with a median conformity of about 90%, for both positive and negative ionization modes (Fig. 2d–f and Supplementary Fig. 3). The results show that whatever the lipid subclasses, number of side chains, carbons and DBEs exhibit high agreement with three rules. We further fitted a linear relationship between the number of annotations and the rules' compliance (Supplementary Fig. 3g), showing that only few data training of lipid annotation cannot meet these three rules. By using the absolute errors in RT by the three rules, the average absolute RT deviation rates for the ECN and ESCN rules are 1.44% and 1.28%, respectively (Supplementary Fig. 4a–c). This finding further supports the universality of these rules, although the lipid subclasses BA (bile acids) and ST (sterols) have larger deviation values due to insufficient detailed classification in the published datasets.

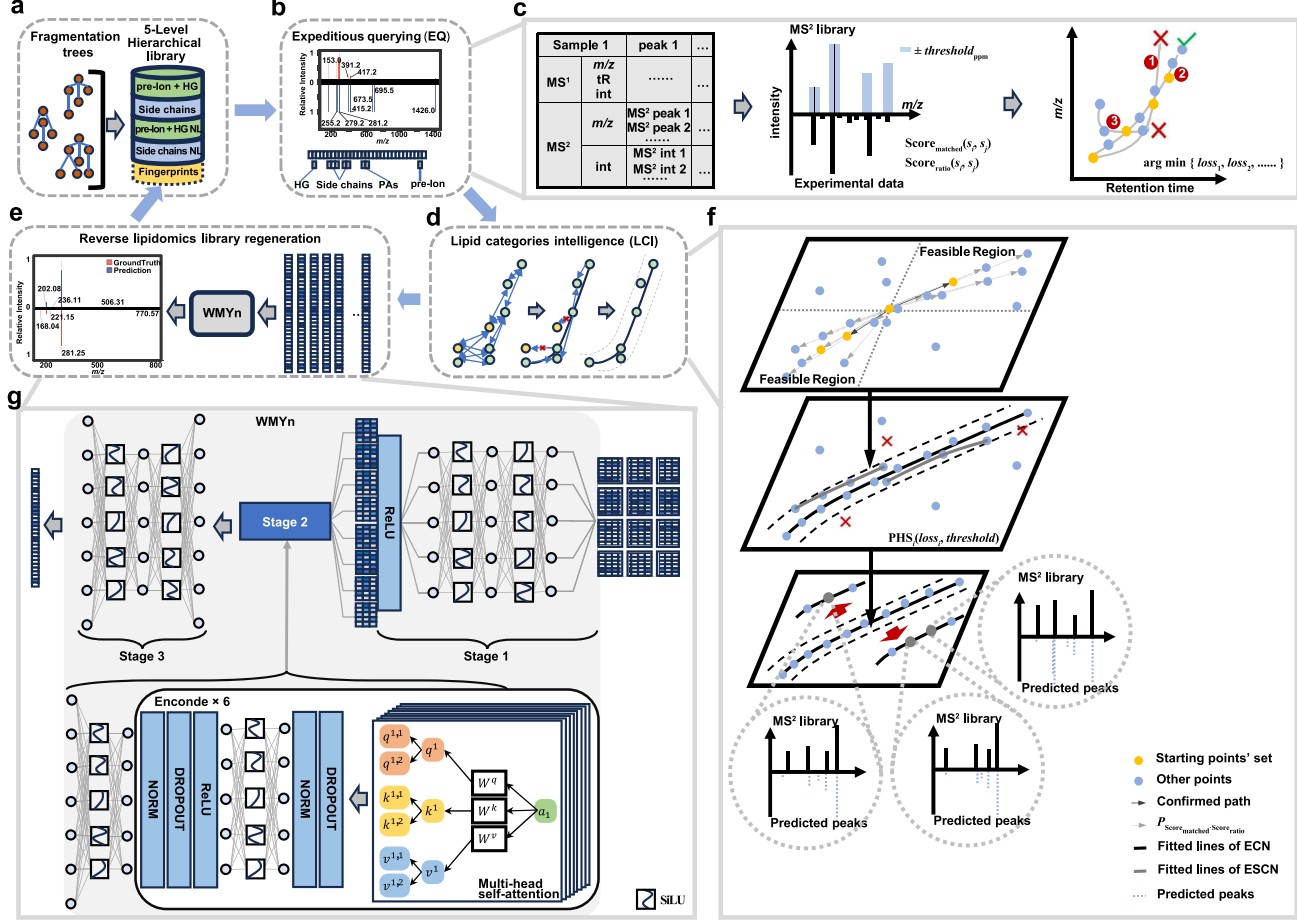

**Fig. 1 | Overview of LipidIN framework. a** Five-level spectral fragmentation tree of 168.5 million lipid fragments, including the Paternò-Büchi reaction and electron-activated dissociation for in-depth C=C positions. **b** Expeditious querying (EQ) module focuses on characteristic peak fragmentation annotations. **c** Detailed description of the inputs to the EQ module and the metrics on characteristic peak fragmentation annotations at threshold (ppm), as well as the optimal point search to the next module. **d** Lipid categories intelligence (LCI) model extrapolates relationships between carbon number and intraclass relative retention time, utilizing multi-simulations to eliminate false positives. **e** The Wide-spectrum Modeling Yield network (WMYn) consists of three stages for reverse lipidomics to regenerate a high-accuracy and easy-migration fingerprints from high-confidence annotation and execute cross-platform migration, resulting in enhanced subsequent recall.

**f** Detailed description of the LCI model. Based on the optimal point, LCI model constructs a feasible domain for performing both directions searching, and using the fitting curve confidence interval judgment for false-positive removal. In addition, spectra without MS² were predicted by taking proximity points information into consideration. **g** Detailed description of the WMYn model. The first stage maps MS² sparse matrix of mass-to-charge ratios and intensities into a same-dimensional space with 512 rows for each lipid. The second stage consists of an encoder and a separate layer, in which the encoder consists of multi-head self-attention of transformer architecture and one network layer. The third stage enables to performance downsampling and upsampling sequentially for regenerating lipid fingerprints that are the 5th-level fragmentation library.

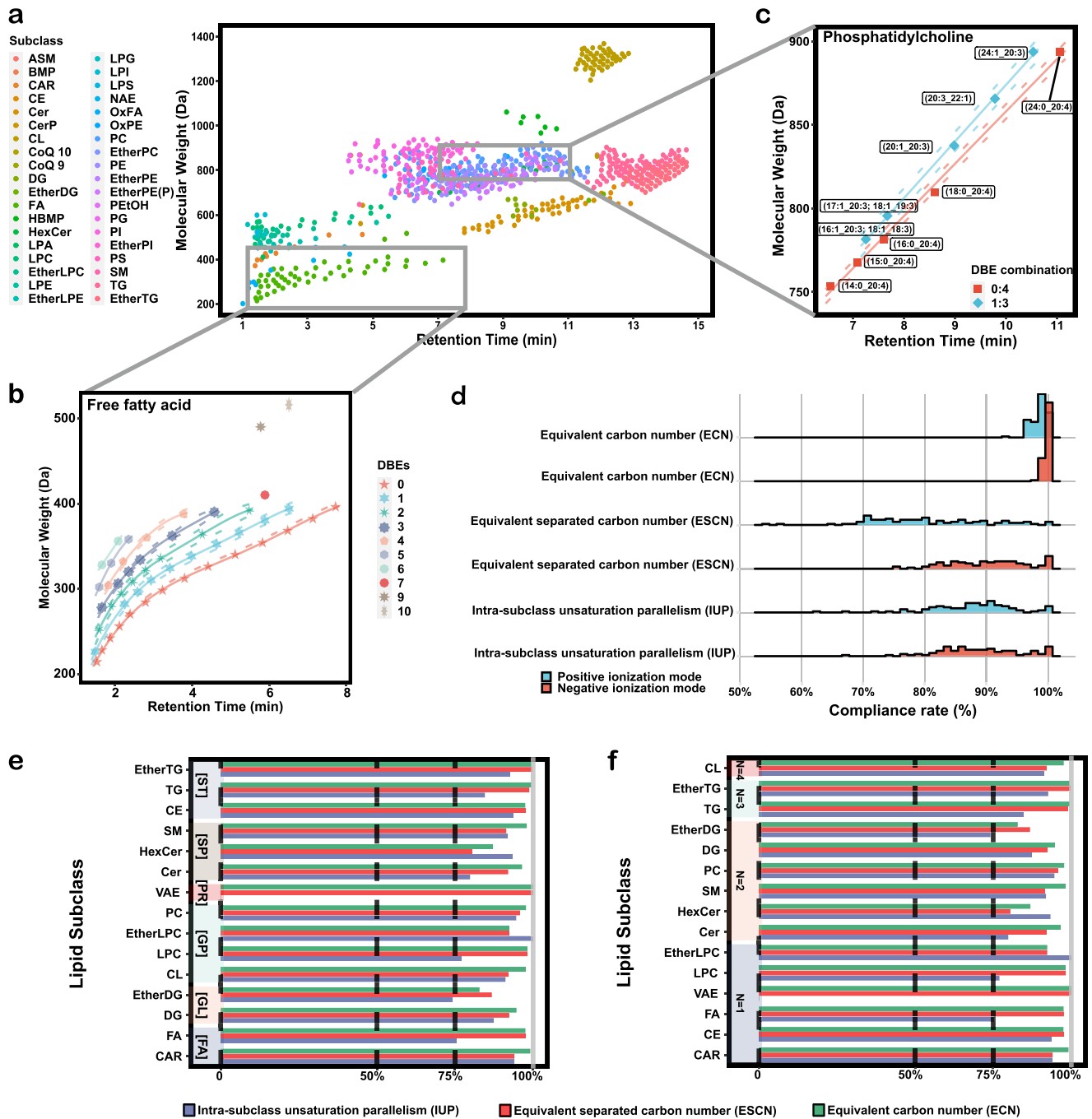

**Fig. 2 | Relative retention time regularities of lipid subclasses. a** Scatter plots of molecular weight and retention time, where lipid subclasses were distinguished by different colors. **b** Scatter plots of molecular weight and retention time for free fatty acids, poly-unsaturations were distinguished by different colors, using a polynomial fit for each poly-unsaturation and plotting the fitted curves and 95% confidence intervals, the centerline of the fitted trend is the line connecting the fitted values, some poly-unsaturations could not be fit owing to too few annotations. The red pentagram represents a double bond equivalents (DBEs) of 0, the blue hexagram represents a DBEs of 1, the green heptagram represents a DBEs of 2, the dark blue octagon represents a DBEs of 3, the flesh-colored pentagon represents a DBEs of 4, the light blue-purple hexagon represents a DBEs of 5, the light green heptagon represents a DBEs of 6, the dark red octagon represents a DBEs of 7, the dark brown octagram represents a DBEs of 9, and the light brown flat octagram represents a DBEs of 10. **c** Scatter plots of molecular weight and retention

time for phosphatidylcholines (PCs) with four DBEs, distinguish unsaturated compositions by color, and plot 95% confidence intervals using linear fitting, and the centerline of the fitted trend is the line connecting the fitted values. The red square represents a DBEs composition of 0:4, and the blue square rotated 90° represents a DBEs composition of 1:3. **d** Statistical results of equivalent carbon number (ECN), intraclass unsaturation parallelism (IUP), and equivalent separated carbon number (ESCN) for positive or negative ionization mode by using >100 published datasets. Blue and red represent positive ionization and negative ionization modes, respectively. **e, f** Statistical results of ECN, IUP, ESCN in >100 published datasets classified by different lipid category and number of separated chains. Deep purple, red, and green represent Intra-subclass Unsaturation Parallelism (IUP), Equivalent Separated Carbon Number (ESCN), and Equivalent Carbon Number (ECN), respectively.

## Performance of EQ and LCI modules

Flash entropy search seems to be the fastest tool to query all mass spectral libraries thus far, having a squared time complexity of $O(nm\Delta)$. In contrast, EQ module has linear time complexity of $O(m\Delta)$, displaying a faster querying speed. To verify the performance of spectrum size on spectral matching speed, the EQ and flash entropy were computed on a published dataset[39] (MetabolomicsWorkbench ST001794) against different sizes of theoretical library from MassBank, and all methods were tested on a same personal low-memory computer, using a single thread and CPU. It can be seen that as the number of spectral increases, the EQ matching time remains virtually unchanged even in a ten-million library querying task, suggesting that it took only about 2.3 μs to complete 10,000,000 spectral library queries. Flash Entropy also showed good results with small size libraries, but once the library size reached millions, the time consumption spikes, causing speed down to 0.14 s for each MS[2] querying 1,000,000 spectra library (Fig. 3a). Promoted by the use of hash tables and bisection methods in EQ, users can compute one billion times for MS[2]-spectral comparison in 0.23 ms (over four thousand billion queries per second), which is around 60,000-time faster than flash entropy using classic low-memory personal computers by searching against one million spectra from MassBank.

To verify the performance of the algorithm, we used published datasets[34,39] and further employed various commonly used tools for small molecular compound annotations, including MS-DIAL[11], LipidSearch[12], Spectral Entropy[13,14], and LipidMatch[8] (Fig. 3b for lipidomics and Supplementary Fig. 5a, b for metabolomics), and all parameters are shown in Supplementary Table 2. MS entropy neither identifies lipids with single characteristic peaks without daughter ions, such as FA (Supplementary Data 3), nor discriminates partial lipid isomers by the similarity scores. For example, flash entropy annotated SM d-38:2 (SM 18:2;2O/20:0) to be SM molecules with multiple fatty acyl compositions by ranking the same similarity score (Supplementary Data 4), whereas EQ can highlight the highest score for SM d-38:2 (SM 18:2;2 O/20:0) (Supplementary Data 7). In our LipidIN framework, EQ module with MS-DIAL public database (https://systemsomicslab.github.io/compms/msdial/main.html#MSP), by contrast, achieved a querying result with ~70% recall@Top-20. Of note, here recall is defined as intersection count of tool and article annotations/total article annotations. Furthermore, based on the 1st- to 4th-level hierarchical library, EQ combining with LCI modules achieved over 90% recall@Top-20, outperforming the above tools by taking advantages of three relative RT rules (ECN, IUP and ESCN) and the 1st- to 4th-level lipid fragmentation library (Fig. 3b). Similarly, we also tested LipidIN in small-molecule compound annotation on another published dataset, showing that LipidIN still achieved 88.26%, which was 1.4688-time higher than entropy search (Supplementary Fig. 5a). In addition, we statistically analyzed recall by lipid subclasses and found that LipidIN had an advantage in the annotation of cardiolipin (CL), N-acyl ethanolamine (NAE), oxidized fatty acid (OxFA), oxidized phosphatidylethanolamine (OxPE), and triacylglycerol (TG) subclasses, although LipidSearch showed superior performance in the annotation of lysophosphatidylserine (LPS) and ceramides phosphate (CerP) subclasses. Interestingly, EQ with or without LCI module using MS-DIAL published database outperformed MS-DIAL program (https://zenodo.org/records/12589462), in particular TG annotation, suggesting that EQ module encompassing MS[2] fragment annotations surpassed similarity algorithm in MS-DIAL program. Moreover, LipidIN showed the best performance by using our 168.5-million lipid fragmentation hierarchical library, including 121 lipid subclasses, indicating higher coverage in home-made hierarchical library than MS-DIAL public library (Fig. 3c).

Lipid Data Analyzer[37] (LDA; http://genome.tugraz.at/lda2), a platform-independent lipid annotation method, used a RT model to filter incorrect species identifications. To validate the effectiveness of the LCI module in removing false positives based on aforementioned relative RT rules, we compared it with RT prediction algorithm proposed by LDA (Supplementary Data 5). As a result, LCI module in LipidIN framework exhibited significantly superior performance to LDA across almost lipid subclasses in terms of accuracy (Fig. 3d).

For those lipid candidates without MS/MS fragmentation, we manually built the test set by removing the MS[2] information of 10% spectra in the annotations. By performing LCI module using 10-fold random sampling, the accuracies of spectral prediction were 75.0% and 82.6% recall@1 in positive and negative ionization modes, respectively, and nearby 85% recall@10 in both ionization modes (Supplementary Fig. 5c).

In addition to ultra-fast performance of EQ and false-positive removal capability of LCI, the combination of EQ and LCI modules can also improve the coverage of lipid annotation compared to MS-DIAL, LipidSearch, and LipidMatch methods. Our approach using EQ and LCI modules annotated 471 unique lipids from the mixture dataset compared with other tools (Supplementary Fig. 5d and Supplementary Data 6). We further validated these 471 lipids by manually checking their MS/MS fragmentation and ensuring the agreement of their relative RT with aforementioned rules (Supplementary Fig. 5e). Therefore, the combination of EQ and LCI modules can not only cover most lipids highly recognized by multiple methods, but also annotate more high-confidence lipids, which were unannotated by all other methods we tested.

Of great interest, based on the rules of relative RT (ECN, IUP and ESCN), LCI shows a higher recall after EQ module performance by using datasets from various chromatographic and mass spectrometric conditions previously reported[35,40–42] (Supplementary Fig. 5e). By taking false discovery rate (FDR) into account, EQ plus LCI achieved average FDR of 5.69% at cutoff threshold at 2.4 under strict criteria, by annotating 8923 lipids in four datasets including RBL-2H3 cells, mixture dataset, human sera, and zebrafish tissues (Fig. 3e–h and Supplementary Data 7). We set the annotations that not only complied with the ECN rule in tolerance 0.5 min but also contained all feature peaks in high intensity to be correct. In a word, LipidIN is an ultra-fast, highly accurate, and coverage platform-independent framework for lipid annotation.

## Reliability and flexibility of reverse lipidomics

To strengthen high-confidence and high-coverage annotation, we still need to regenerate fragmentation tree hierarchical library from authentic lipid spectra. Therefore, we established the WMYn, which consists of three stages for reverse lipidomics. The first stage is designed for intra-feature learning, mapping MS[2] sparse matrix of mass-to-charge ratios and intensities into a same dimensional space with 512 rows for each lipid. The second stage is designed to regenerate higher mass spectrometric resolution by incorporating an encoder with one separate layer, in which the encoder consists of multi-head self-attention and one network layer functions inter-feature learning. The third stage aims to enhance prediction of fragment ions with narrow tolerance, performing downsampling and upsampling sequentially for regenerating lipid fingerprints that are encompassed in the 5th-level hierarchical library (Fig. 1g). We also compared two activation functions in WMYn of LipidIN: ReLU and Sigmoid Linear Unit (SiLU). As a result, we found that SiLU is more flexible and has higher accuracy than ReLU activation function by small-sample learning (Supplementary Fig. 6).

Next, we tested the regenerative capacity of WMYn. By utilizing lipid reference standards (9 in positive ion mode and 6 in negative ion mode, Supplementary Data 8) continuously collected three times under the same experimental conditions, and conducted "spectral entropy similarity"[14] to evaluate prediction results from WMYn. Firstly, the fingerprints are regenerated from WMYn with 1-injection of reference standards that were regarded as GroundTruth. Secondly, we

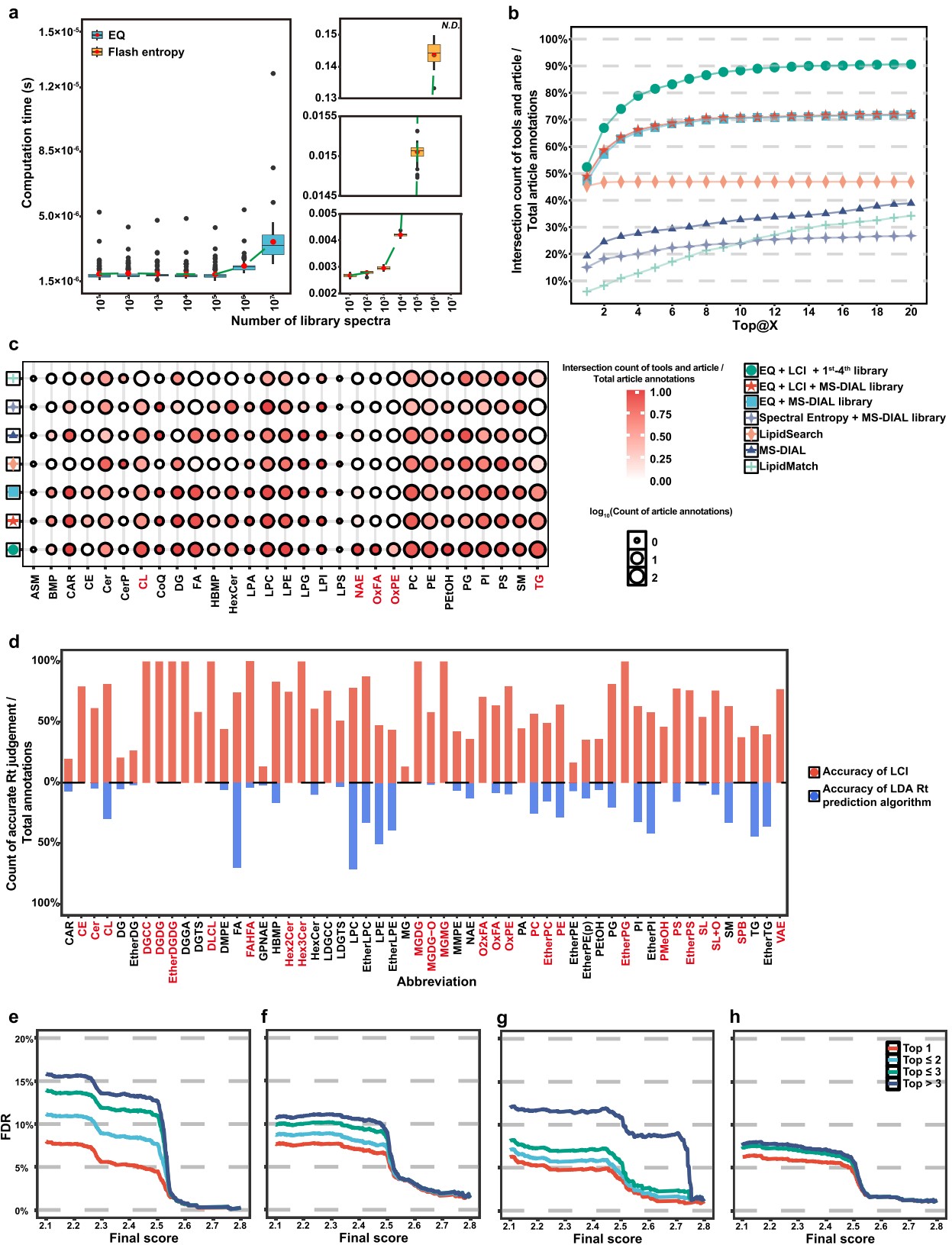

employed liquid chromatography tandem mass spectrometry (LC-MS/MS) analysis of 333 clinical sera samples from breast patients and healthy subjects that were collected on a LC-Orbitrap Exploris 240 MS system (the biomarker discovery of this cohort is discussed below), and then annotate lipids by EQ and LCI modules with 1st- to 4th-level hierarchical library. Thirdly, to regenerate 5th-level fingerprints of the

library and promote transferring of lipid annotation in different platforms, data training of 333 samples was performed in WMYn. The entropy similarity was then used to evaluate the difference between predicted fingerprints and three injections of reference standards, respectively, resulting in a similarity average of 0.9826 (Fig. 4a and Supplementary Figs. 7 and 8). To further verify the lipid annotation

**Fig. 3 | Performance of EQ and LCI modules. a** The time cost of EQ and flash entropy on a published dataset (MetabolomicsWorkbench ST001794) against different size theoretical library from MassBank, and all methods were tested on a same personal low-memory computer, using a single thread and CPU (N.D. denotes no data), all tests are repeated 100 times, in the box plot the median as a center line, with the box representing the interquartile range (IQR) between the upper and lower quartiles. Whiskers extend to 1.5 times the IQR, and points beyond this range indicate outliers. The blue boxplot on the left represents the performance of EQ, while the yellow boxplot on the right represents the performance of Flash entropy. **b** Top-20 recall rates of seven methods tested on published dataset. The dark green circle represents the performance of EQ + LCI using the 1–4 level library, the red pentagram represents the performance of EQ + LCI using the MS-DIAL library, the blue square represents the performance of EQ using the MS-DIAL library, the light blue-purple quadrilateral star represents the performance of Spectral Entropy using the MS-DIAL library, the flesh-colored rhombus represents the performance of Lipidsearch, the dark blue triangle represents the performance of MS-DIAL, and the light green cross represents the performance of LipidMatch. **c** Detial statistics of recall@Top-20 rates of seven methods classified by lipid subclass. The size of the circle reflects the number of annotations for lipid subclass, and the color shade reflects the recall within the subclass. **d** Accuracy of LCI and LDA retention time prediction algorithm for different lipid subclasses annotated in published datasets. The red bar chart in the upper part represents the accuracy of LCI, while the blue bar chart in the lower part represents the accuracy of the LDA retention time algorithm. **e**–**h** FDR of EQ + LCI, by annotating 8923 lipids in four datasets, including RBL-2H3 cells, mixture dataset, human sera, and zebrafish tissues. Herein, the annotations that not only complied with the ECN rule in tolerance 0.5 min but also contained all feature peaks in high intensity to be correct.

with reference standards, the RT deviations were calculated to be around 0.03 min, confirming the accuracy of the LipidIN framework (Fig. 4b). More importantly, to test platform transfer, we used the lipidomics data from 105 clinical sera samples in an Agilent LC-quadrupole time-of-flight (qTOF) MS system, and figured out the 15 lipid reference standards in both positive and negative ionization modes at around 0.9 similarity score, exhibiting the LipidIN's flexibility (Fig. 4c).

As mentioned above, the second stage of WMYn is designed for regenerating higher mass spectrometric resolution. Herein, we compared the predictions of WMYn with other fitting methods, including mean method, linear fitting, polynomial fitting, and exponential fitting, and used entropy similarity to measure the difference between the predicted spectra and the standardized spectra. On one hand, the low resolution of MS values with two decimals was achieved by down-sampling the corresponding high resolution data regenerated from 333 samples in Orbitrap Exploris 240 MS system. The fitting results showed that all methods had good similarity and WMYn still obtained the highest similarity scores (Fig. 4d). On the other hand, when we kept high resolution MS data with four decimals, WMYn was much better than other fitting methods (Fig. 4e). To extend reverse lipidomics applications, we utilized MS-DIAL public database with or without the 5th-level library for lipid annotation in the Entropy Search environment (https://github.com/YuanyueLi/FlashEntropySearch), and obtained higher recall at different cutoff thresholds with the 5th-level library (Fig. 4f). In particular, higher recalls for most of lipid subclasses were achieved by theoretical spectral library from MS-DIAL at a modest cutoff threshold at 0.75 similarity score with the 5th-level library (Fig. 5g). The obtained results suggest that the 5th-level library confers powerful potential of the Entropy Search in lipid annotation, although LipidIN can perform lipid annotation independently.

## Analysis of aging-associated lipidome atlas in mice and NIST SRM 1950

To verify the robustness of LipidIN, we applied the framework to a series of datasets from a recent report to map aging-associated lipidome atlas in mice[33] and NIST SRM 1950[43]. We investigated the recall of the reported 2704 lipids in the MS-DIAL public database, as well as in the 1st- to 4th-level hierarchical library and 1st- to 5th-level hierarchical library, respectively (Supplementary Fig. 9a). The results showed clearly that the hierarchical library is advantageous in lipid annotation over the MS-DIAL public database. Specifically, LipidIN achieved a 93.64% recall using the 5-level hierarchical library. Furthermore, we statistically counted the recall by lipid subclasses, highlighting a strong contribution of reverse lipidomics to the annotation of almost lipid species (Supplementary Fig. 9b). Of note, some rare lipid subclasses such as N-acyl glycine serine (NAGlySer), N-acyl ornithine (NAOrn), and N-glycolyl GM3 (NGcGM3) were also identified by LipidIN.

As a platform-independent framework, LipidIN achieved much higher accuracy in dozens of lipid subclasses than LDA by using the same datasets of aging-associated lipidome atlas in mice (Supplementary Fig. 9c), which was similar to above observation (Fig. 3d). Specifically, we showed the mass spectrometric spectra of 12 lipids identified by LipidIN but not in the article of aging-associated lipidome atlas, derived from the reported raw data for confirmation (Supplementary Figs. 10 and 11). Together, we demonstrated that LipidIN, including EQ, LCI, and reverse lipidomics modules, is also suitable for such published lipidomic datasets from the raw mass spectrometric data, exhibiting more powerful in hierarchical library and computation.

In the benchmarking experiment of NIST SRM 1950, we compared annotations using LipidIN, MS-DIAL (version 5.1), Entropy Search, and article annotations, along with the additional use of Lipid Hunter[44] and Lipid Annotator[45]. In this benchmark experiment, we focused on the Top@1 results. LipidIN demonstrated the highest total number of 434 annotations. This achievement reflects deeper identification of fatty acid (FA) compositions. However, we noted that while all other methods identified a significant number of phosphatidylcholines (PCs) and sphingomyelins (SMs), they often lacked precise FA information and were therefore excluded from this analysis. In the context of annotated intersections, LipidIN not only achieved the highest number of intersections but also annotated more lipids that were not annotated by other tools within the Top@1 category, thereby demonstrating its robust functionality. Notably, only 30 lipid molecules remained unannotated by LipidIN but were identified by at least two other tools. We further ascertained why these 30 lipids were overlooked by LipidIN. Among them, PC 18:0_22:5, displaying low intensities of FA chains, is omitted by LipidIN. The other 29 lipids were not identified by LipidIN due to large RT deviation, fail feature extraction by RaMS, lacking of critical information of characteristic ions, or totally misannotated by other software.

As aforementioned above, 15 lipid standards were used for validation of the WMYn model (Fig. 4). We further utilized the known lipids from NIST SRM 1950 as lipid reference, and were able to select 87 lipids from both positive and negative ion modes, encompassing most common lipid subclasses. These lipids were annotated in previous experiments annotated in at least two tools and achieved a top rank of 1 (Supplementary Fig. 9d–g). WMYn was trained on all samples from cohort 3 of 105 samples, with 3000 epochs set and an $MS^2$-tolerance of 0.01 Da for calculating cosine similarity. The final cosine similarity between the predicted and experimental spectra reveals a mean spectral similarity above 0.96 in both positive and negative ionization modes (Supplementary Fig. 12), demonstrating the universality of the WMYn model.

## Establishment of breast cancer lipid marker panels in clinical cohorts

Regarding the clinical applications, we further performed various liquid chromatography (LC) coupled high-resolution mass spectrometry methods for sera lipidomics from breast cancer patients and

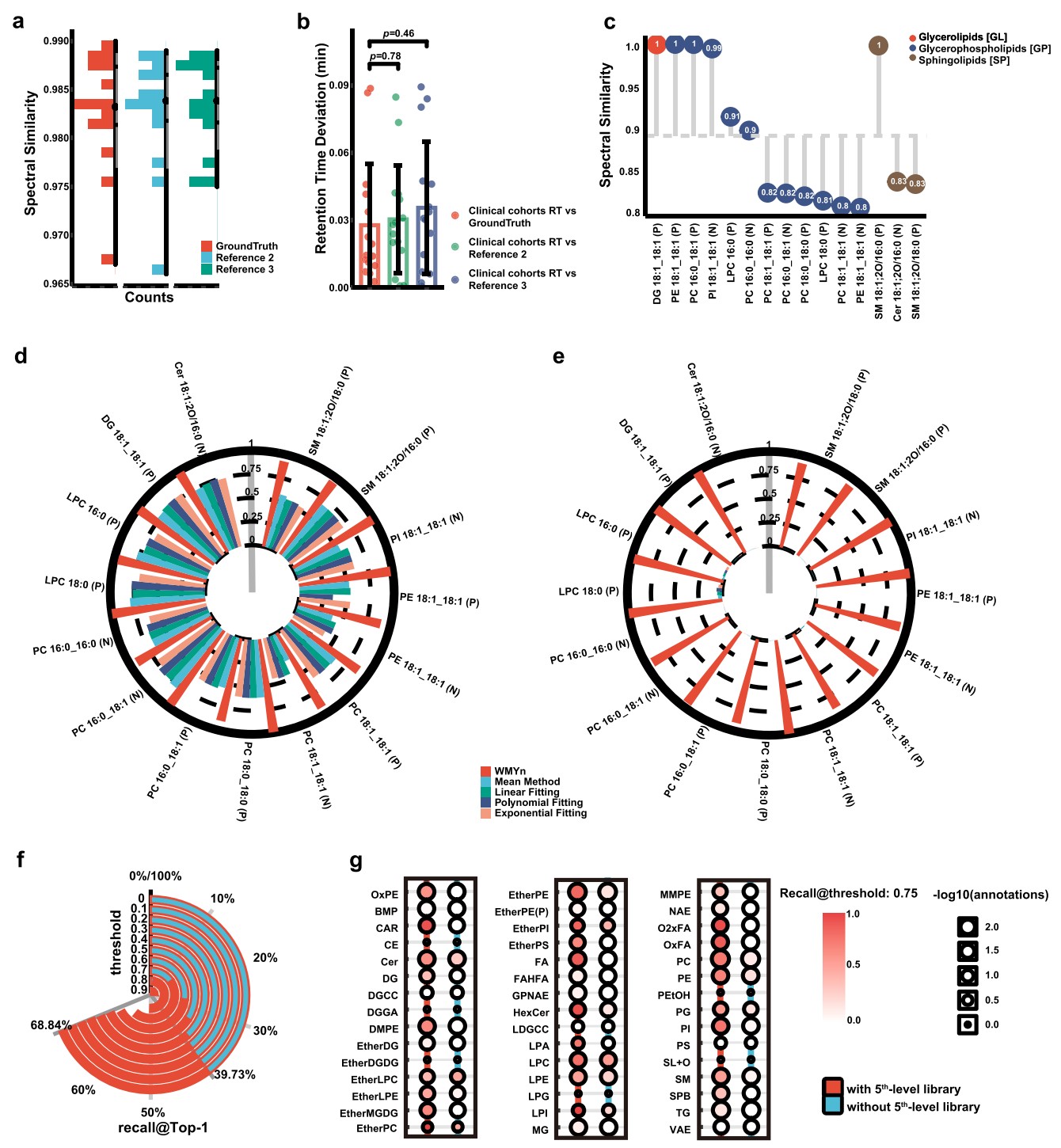

healthy subjects. We harvested the samples from two independent cohorts, including 1393 samples and 333 samples, respectively. We applied LipidIN and annotated 4747 lipids, covering 53 lipid subclasses, with average 133.47 billion times querying per second (Fig. 5a and Supplementary Data 9). Through the importance selection of random forest model (Fig. 5b), we screened 10 featured lipids and constructed Light Gradient Boosting Machine model (LightGBM)[46]. Using this lipid marker panel to distinguish breast cancer patients from healthy subjects, the model achieved an accuracy of 96.93% in the first cohort of 1393 samples. The accuracy rate of the same lipid marker

panel in 333 cases of the second cohort achieved 79.61%, indicating that the selected biomarker has a certain degree of credibility (Fig. 5c and Supplementary Fig. 13). We utilized weighted correlation network analysis (WGCNA)[47] to correlate the changes in clinical manifestations and lipid levels (Fig. 5d, e, f, and Supplementary Data 10). In the correlation analysis of the first cohort (Fig. 5d), we figured out that the levels of hexosylceramide (HexCer) and ceramide (Cer) were associated with diabetes mellitus in breast cancer patients (Fig. 5e). We found that several Cer and lysophosphatidylcholine (LPC) species are positively correlated with tumor grading and tumor size (Fig. 5f).

**Fig. 4 | Performance of WMYn. a** Sea stack plots of entropy similarity between WMYn predictions and GroundTruth (1st injection of lipid reference), 2nd injection of lipid references, and 3rd injection of lipid reference, statistically based on an interval length of 0.05, with the *x*-axis denoting the number of intervals within each interval. GroundTruth, Reference 2, and Reference 3 are lipid reference standards (9 in positive ion mode and 6 in negative ion mode) continuously collected three times on Orbitrap Exploris 240 MS system. **b** Retention time deviation of LipidIN framework annotations and GroundTruth, Reference 2, and Reference 3, confirming the accuracy of the LipidIN framework. We used 15 Lipid reference data from three injections to perform a bilateral *t*-test, and error bars are drawn using the mean and standard deviation (SD) to represent variability in the data. **c**. Platform

transfer performance of WMYn from Orbitrap Exploris 240 MS system to Agilent qTOF MS system. **d**–**e** Entropy similarity comparison of WMYn predictions with other fitting methods, including mean method, linear fitting, polynomial fitting, and exponential fitting in low-resolution MS data with two decimals and high-resolution MS data with four decimals. **f** Theoretical spectra from MS-DIAL public database and reverse lipidomics spectra were tested separately in Entropy Search environment. The recall@Top1 at different thresholds was counted separately, and colors were used to distinguish the two libraries (red: with 5th-level library, blue: without 5th-level library). **g** The recall@Top-1 of different lipid classes under two libraries was counted separately at a modest cutoff threshold at 0.75 similarity score.

Therefore, LipidIN can empower the lipidomic data associated with clinical manifestations. It should be noted that EAD in SCIEX ZenoTOF 7600 system was also applied for lipidomic analysis of 333 cases in the second cohort. Thereby, we further performed annotation of triglycerides (TGs) in the comparison of LipidIN and MS-DIAL (version 5.1). For the C:DB annotations, we identified more TG isomers with C=C locations than MS-DIAL (Supplementary Fig. 14). Interestingly, one TG molecule without C=C location information in the mass spectrum was predicted by MS-DIAL, but our LipidIN can successfully figure out this wrong annotation (Supplementary Fig. 14d).

### Identification of lipid markers associated with breast cancer lung metastasis

In third clinical cohort with 105 human sera samples, including 31 cases of breast nodules, 32 breast cancer without lung metastasis (hereafter breast cancer), 22 breast cancer lung metastases, and 20 female lung cancer, we identified a total of 4854 lipids covering 52 subclasses. Of great interest, we performed volcano plots for two-group comparisons, but could not highlight the potential biomarkers for discriminating these four groups with 1.4-fold change in the comparisons (Fig. 6a and Supplementary Fig. 15). Thereby, we further performed an in-depth double-bond positional resolution of phospholipids. By conducting Paternò-Büchi reaction in lipidomics analysis, we found that the double bond position of PC 18:1_20:1 could be effectively differentiated by fine double-bond positional resolution (Fig. 6b). Patients with breast cancer lung metastasis had higher level of side chain C18:1 (delta15) than the other three groups, with breast nodules had a higher level of side chain C20:1 (delta11) than the other three groups, with female lung cancer had a higher level of side chain C20:1 (delta14) than the other three groups. The obtained results suggest that LipidIN is suitable for phospholipid isomers with in-depth C=C locations for biomarker discovery.

## Discussion

Similarity algorithms such as MS-DIAL, LipidSearch, and MS entropy might cause feature redundancy. In contrast, EQ combined with a hierarchical library consisting of 5-level hierarchical library of MS² fragment annotations, on one hand, could avoid feature redundancy. On the other hand, data structure optimization based on linear time complexity in EQ module promoted by applying hash tables and bisection methods, outperforms squared time complexity in flash entropy search, thereby profoundly enhancing recall and spectral querying speed (Fig. 3a). Furthermore, the combination of EQ and MS-DIAL public database also showed superior performance to MS-DIAL program, suggesting that EQ module encompassing MS² fragment annotations surpassed similarity algorithm in MS-DIAL program.

By statistically analyzing over 100 published datasets, we summarized three relative RT rules: ECN, IUP, and ESCN. Based on these three rules, we developed LCI module by using heuristic search methods. Here, a heuristic search method uses heuristic information to define a route that seems more plausible than the rest, and is designed for problem solving more quickly, suggesting LCI can

profoundly decrease FDR in the highly complex datasets. Notably, heuristic search techniques can be classified into two broad categories: depth-first search (DFS) and best-first search (BFS). We utilized BFS in this work to sort the sequence of node expansions according to a heuristic function, thus exhibiting much higher accuracy than RT prediction in LDA for lipidomic data (Fig. 3d and Supplementary Fig. 9c).

We also created WMYn as a regenerative model for reverse lipidomics (Fig. 4), and demonstrated that WMYn can exhibit four advantages: (1) regeneration of lipid fingerprints as the 5th-level library for high-confidence and high-coverage annotation, (2) platform-independent lipidomic analysis for enhanced platform transferability, (3) enhancement of high resolution MS data for higher accuracy annotation, and (4) an interactive interface that empowers the broad exploration of reverse lipidomics with other spectral querying environments like entropy search. Of great importance, the WMYn is capable of learning the differences between MS platforms and effectively extracting spectral features, thereby facilitating the inter-platform migration and improving annotation accuracy and coverage. Taken together, LipidIN, including EQ, LCI, and reverse lipidomics modules, generally surpasses the performance of existing methods in mass spectrometry-based lipidomics, exhibiting three capacities of querying all mass spectral libraries in real time, improving accuracy and coverage of lipid annotation, regenerating lipid fragment fingerprints for higher accuracy and coverage, respectively.

Future research directions include pre-training on a larger and more diverse dataset to extend reverse lipidomics. Furthermore, investigating the deeper relationship between metabolites and relative RT will provide a robust foundation for metabolite identification, prediction, and causal network analysis.

## Methods
### Ethical statement

Written informed consent was obtained from each participant or from the participant's parents or legal guardians in the discovery and validation cohorts, and the study was conducted according to the Declaration of Helsinki. Ethical permission was granted by Xiamen University Ethics Committee, Fujian Province, China (Approval number: XDYX202302K08), and Fujian Medical University Union Hospital, Fuzhou, Fujian Province, China (Approval number: 2022KY111).

### Lipid extraction

The sera samples from breast cancer patients and healthy subjects were harvested and stored at −80 °C till analysis. We employed three distinct based liquid-liquid extraction methods to extract lipids from these sera samples. The lipid mixture data sets extracted from brown adipose tissue, brain, colon, heart, kidney, liver, lung, pancreas, soleus muscle, spleen, testis, and white adipose tissues of mice are based on methyl tert-butyl ether (MTBE) extraction method. The LC coupled with Orbitrap ID-X Tribrid mass spectrometric system was employed for lipidome analysis. The mobile phase A consists of acetonitrile and water (60:40, v/v) containing 10 mM NH4Ac, while mobile phase B consists of isopropyl alcohol and acetonitrile (90:10, v/v).

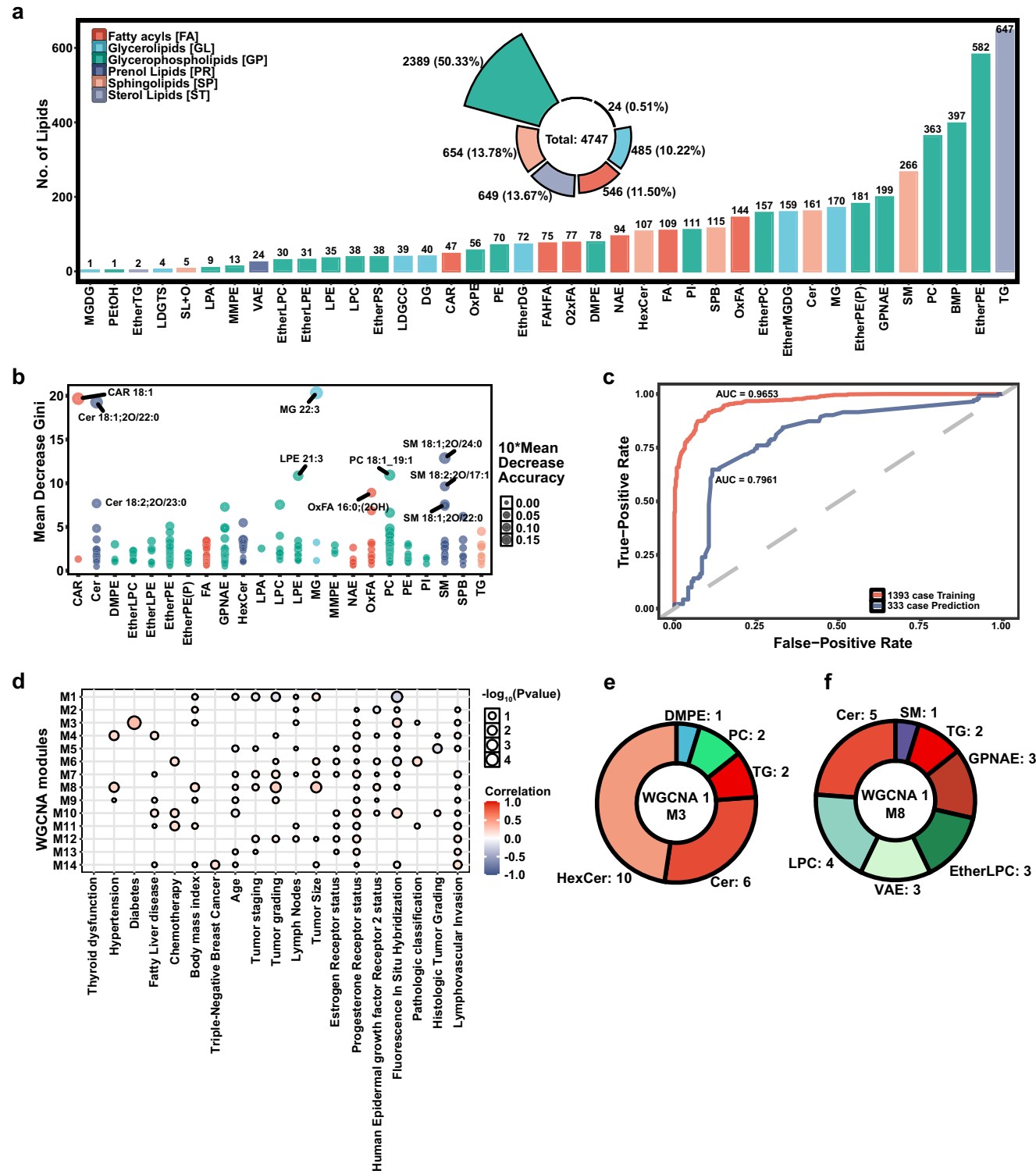

**Fig. 5 | Application of lipidIN to breast cancer clinical data. a** Statistics of all annotations. **b** Results of Randomized Forest Important Indicators by Lipid sub-class. **c** Receiver Operating Characteristic (ROC) of the first cohort as a training set and the second cohort as a test set. **d** Weighted correlation network analysis (WGCNA) of the first cohort with clinical indicators, the correlation was calculated with Pearson correlation coefficient, and cluster method was average hierarchical clustering. **e**, **f** Composition of several important modules of lipids in WGCNA of the first cohort.

## MTBE (methyl-tert-butyl ether) method

In the extraction process, the serum samples were harvested from two independent cohorts consisting of 1393 clinical samples (698 breast cancer patients and 695 healthy subjects) and 333 clinical samples (142 breast cancer patients and 191 healthy subjects), respectively. The extraction protocol started with the dispensing of 50 μL of serum into

centrifuge tubes, followed by the addition of 400 μL of pre-cooled pure methanol containing internal standards and 1 mL of MTBE to each sample. Subsequently, the samples were thoroughly vortexed (1000 rpm, 10 °C, 30 min). Phase separation was induced by adding 400 μL of Milli-Q water and centrifugation at 15,000 × g for 15 min at 10 °C. From this lipid-rich upper layer, we aspirated 300 μL of

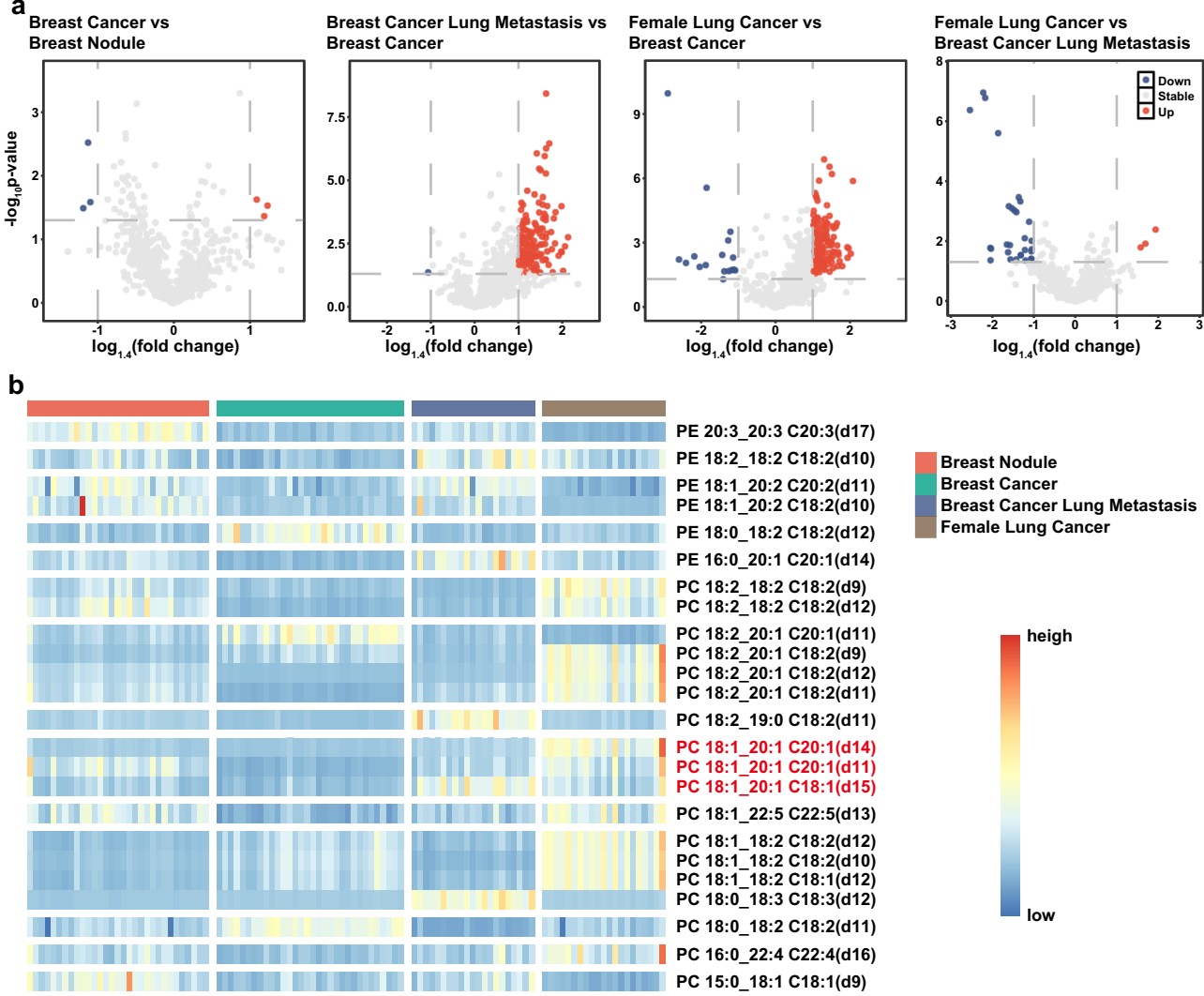

**Fig. 6 | Analysis of Breast cancer and Lung cancer Clinical data. a** Volcano plots showing selected biomarkers changes in patients with breast nodules, breast cancer, breast cancer lung metastases, and female lung cancer. The $x$ and $y$ axes show the log1.4 (fold change) and −log10 ($p$ value), respectively, and $p$ values were calculated using bilateral $t$-tests. Biomarkers differed significantly in comparisons of patients with breast cancer and breast cancer lung metastases, and in comparisons of patients with breast cancer and female lung cancer. **b** Characterization of the double bond positions of PC and PE using the Paternò-Büchi reaction. The PC 18:1_20:1 double bond position had the potential to be a biomarker for the differentiation of breast nodules, breast cancer, breast cancer lung metastases, and female lung cancer. High expression of C18:1 (delta 15) was suggestive of breast cancer lung metastasis, C20:1 (delta 11) was suggestive of breast nodules, and C20:1 (delta 14) was suggestive of lung cancer in females, whereas the absence of significant differences was suggestive of breast cancer.

supernatant from each sample. To guarantee long-term preservation, the supernatants were dried using a nitrogen blower maintained at room temperature. The dried lipids were then stored at −80 °C until further analysis was required. Prior to LC-MS analysis, the lipids were resuspended in a solvent mixture. This solvent consisted of 20 μL $CH_2Cl_2$/MeOH mixture (2:1, v/v) and 130 μL acetonitrile/isopropyl alcohol/$H_2O$ containing 5 mM ammonium acetate mixture (65:30:5, v/v/v). Each resuspended sample underwent vortexing for 30 s, followed by centrifugation at $15,000 \times g$ for 10 min at 6 °C. Additionally, a pooled aliquot of supernatant from every sample was combined as a quality control (QC) reference sample.

**Modified Folch method**

In the extraction process of 105 clinical cohort samples (31 breast nodule patients, 32 breast cancer without lung metastasis patients, 22 breast cancer lung metastasis patients, 20 female lung cancer patients), serum sample preparation was conducted rigorously according to the Standard Operating Procedure (SOP) based on the modified Folch liquid-

liquid extraction method. A lipid internal standard mixture consisting of 15 deuterated lipids and methanol was added to each sample. Following this, the samples were vortexed, and dichloromethane and water were introduced for extraction. The samples were then allowed to equilibrate at room temperature for 10 min before being centrifuged at 4 °C and $15,000 \times g$ for 10 min. After centrifugation, the organic layer was carefully transferred to a designated centrifuge tube and dried using a nitrogen blower. The residue was subsequently redissolved with mobile phase B (2-propanol/water in a ratio of 95:5), vortexed for 1 min, and diluted with Mobile Phase A (methanol/acetonitrile/water in a ratio of 50:40:10). To ensure QC, a pool of all samples was prepared. This mixture was divided into equal aliquots and stored at −80 °C. This pool sample served as the QC for that specific batch of samples, ensuring the consistency and reliability of the extraction process.

**Lipid extraction for Paternò-Büchi reaction method**

In the clinical cohort study involving 105 samples (31 breast nodule patients, 32 breast cancer without lung metastasis patients, 22 breast

cancer lung metastasis patients, 20 female lung cancer patients) for the Paternò-Büchi reaction, each 50 μL of serum sample was diluted with 1 mL of water, and subsequently, 1 mL of methanol and 2 mL of chloroform were added. The resulting mixture was vortexed vigorously for 10 min to ensure thorough mixing, followed by centrifugation at 12,000 rpm for 12 min. Following centrifugation, the lower organic layer was carefully collected, and the extraction process was repeated on the upper aqueous layer to maximize lipid recovery. The lower organic layers from both extractions were then combined, and the solvent was gently evaporated using a nitrogen blower. The residue obtained after solvent evaporation was reconstituted in 500 μL of methanol. To ensure sample purity and remove any particulate matter, the reconstituted solution was filtered through a 0.22 μm filter membrane. The resulting filtrate represented the lipid extraction original solution, ready for further analysis in the Paternò-Büchi reaction.

## Lipidomics data acquisition
Lipidomics data were obtained through the utilization of five unique mass spectrometer platforms. The details are shown as follows.

### Thermo Scientific Orbitrap Exploris 240 MS system
Lipidomics data (1393 clinical samples and 333 clinical samples) were acquired using a Vanquish Flex UPLC system coupled with Thermo Scientific Orbitrap Exploris 240 MS system equipped with a heated electrospray ionization (H-ESI) source. Samples were separated through a BEH C8 column (2.1 × 100 mm with 1.7 μm particle size, Waters, Milford, MA, USA) with column temperature maintained at 55 °C and mobile phases consisting of 2 mM ammonium formate in mobile phase A (40% water and 60% acetonitrile) and mobile phase B (90% isopropanol and 10% acetonitrile). The gradient started with 1.5 min of isocratic elution with 32% B (and 68% A). B was increased to 85% over the next 15.5 min and then from 85% B to 97% B in only 0.1 min. Maintained at 97% B for 2.4 min. Rapidly, the mobile phase composition was returned to 32% B within 0.1 min and maintained 5 min for column post-equilibration. The flow rate for mobile phases was set at 0.26 ml/min. The injection volume was 2 μL for positive ions and 5 μL for negative ions. The mass spectrometer was operated in positive or negative modes using a full scan/data-dependent secondary scan (Full-ddMS2) in the scan range $m/z$ 100–1500 Da. Capillary voltage of 3400 V for positive and 3000 V for negative. Ion Transfer Tube Temp: 320 °C, Vaporizer Temp: 350 °C, sheath gas: 40 Arb, Aux gas: 10 Arb, Sweep Gas: 10 Arb. Orbitrap resolution was set 120,000 in MS1 and 15,000 in MS2. The normalized CE type was selected.

### Agilent 6546 Q-TOF MS system
Lipidomics data from 105 clinical cohort samples (31 breast nodule patients, 32 breast cancer without lung metastasis patients, 22 breast cancer lung metastasis patients, 20 female lung cancer patients) were acquired using an Agilent 1290 LC coupled with Agilent 6546 Q-TOF Mass Spectrometer equipped with ESI source. Samples were separated through a CSH C18 column (2.1 × 100 mm with 1.7 μm particle size, Waters, Milford, MA, USA) with column temperature maintained at 40 °C and mobile phases consisting of 10 mM ammonium formate in mobile phase A (methanol/acetonitrile/water, v/v/v = 50:40:10) and mobile phase B (2-propanol/water, v/v = 95:5). The gradient started with 1.5 min of isocratic elution with 32% B. B was increased to 85% over the next 15.5 min and then from 85% B to 97% B in only 0.1 min. Maintained at 97% B for 2.4 min. Rapidly, the mobile phase composition was returned to 32% B within 0.1 min and maintained 5 min for column post-equilibration. The flow rate for mobile phases was set at 0.26 ml/min. The injection volume was 4 μL for positive ions and 12 μL for negative ions. The mass spectrometer was operated in positive or negative modes in the scan range $m/z$ 150–1500 Da. Capillary voltage of 3400 V for positive and 3000 V for negative. Ion Transfer Tube Temp: 320 °C, Vaporizer Temp: 350 °C, sheath gas: 40 Arb, Aux gas: 10 Arb, Sweep Gas: 10 Arb. The CE was 10–60 eV.

### Xevo G2-XS Q-TOF MS system
Lipidomics data from 105 clinical cohort samples (31 breast nodule patients, 32 breast cancer without lung metastasis patients, 22 breast cancer lung metastasis patients, 20 female lung cancer patients) were acquired using an ACQUITY UPLC I-Class PLUS coupled with Xevo G2-XS Q-TOF Mass Spectrometer. Samples were separated through a BEH HILIC column (2.1 × 100 mm with 1.7 μm particle size, Waters, Milford, MA, USA) with column temperature maintained at 40 °C and mobile phases consisting of mobile phase A (10 mM ammonium formate and 0.2% acetic acid in water) and mobile phase B (acetonitrile/acetone/isopropanol, v/v/v = 50:48:2). The gradient elution program was as follows: 0 to 2.4 min, 90% to 85% B; 2.4 to 3.2 min, 85% to 80% B; 3.2 to 5 min, 80% B; 5.0 to 5.1 min, 80% to 70% B; 5.1 to 6 min, 70% B; 6 to 6.1 min, 70% to 90% B; 6.1 to 10.0 min, 90% B. The flow rate for mobile phases was set at 0.35 ml/min. The mass spectrometer was operated in positive or negative modes in the scan range $m/z$ 150–1500 Da. The ESI source conditions were as follows: Source Capillary: 2.5 KV; Sampling Cone: 40 V; Source Offset: 80 V; Source Temperatures: 120 °C; Desolvation Temperatures: 500 °C; Cone Gas Flow: 50 L/h; Desolvation Gas Flow: 800 L/h; MS1 scan ranges: $m/z$ 400–1000. Raw data extraction and processing. Any raw data in whatever format is converted to mzML format using MSConvert and quantized using XCMS.

### SCIEX ZenoTOF 7600 MS system
For EAD data acquisition, an Exion LC coupled with a quadrupole time-of-flight MS system (ZenoTOF 7600, SCIEX, Framingham, MA, USA). The mobile phase A consists of methanol, acetonitrile, and water (1:1:1, v/v/v) containing 5 mM ammonium acetate. The mobile phase B consists of isopropanol containing 5 mM ammonium acetate. The flow rate is 0.3 mL/min with gradient program of 17 min. The Kinetex C18 column (2.1 × 100 mm, 2.6 μm) was used for separation. A targeted MS/MS scanning mode, referred to as "MRM HR" by SCIEX, was employed. For fragmentation conditions, collision-induced dissociation (CID) mode with collision energy (CE) set at 10, 20, and 40 volts (V) with no CE spread, and EAD mode with CE set at 10 V and electron kinetic energy (KE) at 10, 15, and 20 electron volts (eV) were conducted.

### Normalization of lipid intensities
In each tested sample group, we carefully adjusted the lipid intensity using internal standards as references. This was particularly crucial for the extensive data sets from 1393 cases and 333 cases in the first and second clinical cohorts, respectively. To manage this data efficiently, we processed the lipid testing in batches and removed batch effects with statTarget (https://stattarget.github.io/). Additionally, we applied the natural logarithm to all biomarker screening values to normalize the data distribution.

### Raw data extraction and processing
Raw data, regardless of its original format, undergoes conversion to the mzML format via MSConvert and subsequent quantization through XCMS. Annotation of lipid content was performed with LipidIN. Data from the Orbitrap Exploris 240 utilized MS1 tolerance of 10 ppm and MS2 tolerance of 20 ppm, while data from the Agilent 6546 Q-TOF was set to MS1 tolerance of 20 ppm and MS2 tolerance of 40 ppm. By taking the relatively low precision associated with the Paternò-Büchi reaction into account, data from the Xevo G2-XS Q-TOF was configured with MS1 tolerance of 1000 ppm and $MS^2$ tolerance of 40 ppm.

### Lipidomics data filtering
To ensure the accuracy and reliability of our analysis, we only included lipids in our lipidomics analysis that had a ScoreMatched of over 0.75 and a final score of 2.1 or higher in LipidIN annotations. When multiple

plausible annotations existed under the same peak, we selected the one with the highest score. Furthermore, we excluded biological samples from our analysis if they had missing data exceeding 20%. Additionally, we removed lipids with coefficient of variation (CV) higher than 30% in QC samples to maintain data consistency and reliability.

## Expeditious querying (EQ) module

The EQ module consists of two main components: (1) MS$^2$ fragment annotations, and (2) data structure optimization.

MS$^2$ fragments annotations. This algorithm emphasizes the matching degree between theoretical spectrum ($n_1$) and measured spectrum ($n_2$) and highlights importance of mass spectrometric features. Based on this concept, two indicators Score$_{matched}$ and Score$_{ratio}$ were defined respectively. The Score$_{matched}$ is defined as:

$$\text{Score}_{matched} = \frac{\sum_{i=1}^{n_1}\sum_{j=1}^{n_2} I\left(\frac{||s_i - s_j'||_1}{s_i} \leq threshold_{ppm}\right)}{n_1}, \quad (1)$$

$s_i$ and $s_j'$ denote two peaks, with a representing the $j$th peak from the measured spectrum and $i$th peak of theoretical library. $||s_i - s_j'||_1$ is the L1 norm of any two peaks I($x$) is an indicator function that takes the value of 1 when a given condition is satisfied and 0 otherwise. Herein, the indicator function determines whether the percentage difference between any two peaks exceeds $threshold_{ppm}$.

The Score$_{ratio}$ is defined as:

$$\text{Score}_{ratio} = \frac{\sum_{i=1}^{n_1}\sum_{j=1}^{n_2} I\left(\frac{||s_i - s_j'||_1}{s_i} \leq threshold_{ppm}\right) \times intensity_j}{\sum_{j=1}^{n_2} intensity_j}, \quad (2)$$

$intensity_j$ illustrates the intensity of the $j$th peak of the measured spectrum. Score$_{ratio}$ Calculates the importance of measured characteristic peaks. It is worth noting that when multiple measured spectra match a single peak in the theoretical library, we will only calculate one successful match and select the one with the highest response.

Data structure optimization. During EQ in a task with $n$ peaks to be annotated, we used a combination of bisection and hash tables to shrink time complexity $O(n)$ into $O(1)$, achieving a final time complexity of $O(m\Delta)$, where $m$ is the total number of peaks in the library and $\Delta$ is the querying tolerance.

## Lipid categories intelligence (LCI) model

Following by Score$_{matched}$ and Score$_{ratio}$ as a prior information, we further leveraged LCI model to reduce false positives and predict candidate annotations without MS/MS fragments. To delineate ECN and ESCN rules, we utilized loss functions to get starting points' sets. Modern mass spectrometers are capable of rapid switching, with the delay from MS1 scans to MS2 scans typically ranging from milliseconds to a few seconds. To streamline the extraction of primary peaks, we opted not to perform peak extraction during the lipid annotation step. Instead, we utilized the MS/MS scan times in the LCI model.

Get starting points' sets. We transferred the abstract problems of relative RT into path optimization problems. Through prior heuristic search (PHS) framework, appropriate amount of the points was randomly selected as the optimal starting point by using Score$_{matched}$ and Score$_{ratio}$. Then we used the loss to judge whether these points can construct a starting points set. The $loss_1'$ is defined as:

$$loss_1' = I\left(R_{adj}^2 \geq 0.9\right), \quad (3)$$

$R_{adj}^2$ measures the adjusted coefficient of determination of a regression model, we set Eq. (3) to restrict the adjusted coefficient of

determination to be greater than 0.9. $loss_2'$ measures the monotonicity of the curve, defined as:

$$loss_2' = I\left(\forall rt_i > rt_j, rt_i, rt_j \in [rt_1, rt_2], \frac{mz_i - mz_j}{rt_i - rt_j} \geq 0\right), \quad (4)$$

$rt_i, rt_j$ and $mz_i, mz_j$ represent the RT and $m/z$ of the $i$th spectrum and $j$th, within closed interval $[rt_1, rt_2]$. Based on ECN and ESCN, it is evident that these points exhibit a clear increasing trend. Equation (4) is defined to determine whether the selected points have a monotonically non-decreasing trend. $loss_3'$ ensures that the fitted curve is relatively smooth and continuous, preventing sudden jumps or oscillations in the function, defined as:

$$loss_3' = I(\forall rt_i, rt_j \in [rt_1, rt_2], \forall \varepsilon > 0 \ s.t. \lim_{rt_j \to rt_i} \frac{f(rt_j) - f(rt_i)}{rt_j - rt_i} < \varepsilon), \quad (5)$$

$f(rt_i)$ is the analytical expression after fitted, we require that the fitted curve must be continuous in the closed interval. Equation (5) was used to guarantee two arbitrarily close points in the closed interval, the deviation between their fitted values that is infinitely close. The interconnection of the three loss functions is established through multiplication, simultaneously satisfying ECN and ESCN rules. Ultimately, we calculated the combined loss function using the following equations:

$$loss' = \prod_{i=1}^{3} loss_i', \quad (6)$$

$$loss_i = \frac{n \times ||\widetilde{mz}_t - mz_i||_1}{\sum_{i=1}^{n} mz_i} \times I(loss' = 1), \quad (7)$$

where $\widetilde{mz}_t$ is the predicted $m/z$.

Global annotation judgment. For Global annotation judgment, we construct feasible regions to reduce computational complexity. For points within feasible regions, an exhaustive method was adopted, and Eq. (7) was used to calculate the $loss_i$. Finally, we used PHS$_i$ function to normalize the score:

$$\text{PHS}_i = \begin{cases} 1 - \frac{loss_i}{4 \times threshold}, & if\ loss' = 1, loss_i \leq threshold \\ 0.5, & if\ loss' = 0 \\ \frac{\max(loss_i) - loss_i}{4 \times (\max(loss_i) - \min(loss_i))}, & if\ loss' = 1, loss_i > threshold \end{cases}, \quad (8)$$

where 4 is a constant for data scaling, and parameter $threshold$ is used to quantify the percentage deviation between the measured $m/z$ and expected $m/z$. Piecewise function is determined by comparing whether $PHS_i$ meets the specified threshold.

Intra-subclass curve translation. To delineate IUP rule, we constructed translation equations to annotate underfitting DBEs of lipid species. Taking $n$-degree polynomial as an example, the translation equation is defined as follows:

$$\forall x \in [x_1, x_2] \ s.t. \begin{cases} |f(rt_i) - mz_i| \leq \varepsilon, \forall \varepsilon > 0 \\ g(x) = q(x)f(x) + C, \partial(q(x)) \geq 0 \end{cases}, \quad (9)$$

$f(x)$ is the fitting equation after translation, $rt_i$ and $mz_i$ are the RT and $m/z$ of points to be fitted. Furthermore, $f(x)$ is also required to satisfy the equation after substituting the points, ensuring that there is only a slight deviation of $\varepsilon$, which can be any value greater than zero. This requires $f(x)$ to be a curve passing through point $(rt_i, mz_i)$, or infinitely close to point $(rt_i, mz_i)$. Where $g(x)$ is the fitting equation of another DBEs, requiring that $f(x)$ is a component of $g(x)$ with the quotient of any constant $C$ remainder, $q(x)$ is a non-zero polynomial

divisor. $x_1$ and $x_2$ are the endpoints of a given interval. This requires that $f(x)$ has no intersection with $g(x)$ after being translated in the $y$ and $x$ direction, which is the analytical expression of IUP.

## Reverse lipidomics

To delineate WMYn termed "reverse lipidomics", we utilized feature learning, including intra- and inter-feature learning.

Intra-feature learning. The feature extractor consists of two SiLU-activated layers followed by a Rectified Linear Unit (ReLU) activation. The first layer is defined as:

$$H_1 = \alpha_1\left(\text{SiLU}(\mathbf{W}_1\mathbf{X} + \mathbf{B}_1)\right) + \beta_1 = \begin{pmatrix} h^{(1)}_{1,1} & \cdots & h^{(1)}_{1,n} \\ \vdots & \ddots & \vdots \\ h^{(1)}_{512,1} & \cdots & h^{(1)}_{512,n} \end{pmatrix}, \quad (10)$$

where $\mathbf{X}$ represents the input matrix, consisting of $\mathbf{x_1}, \cdots, \mathbf{x_n}$, where each $\mathbf{x_i}$ is a spectrum, and mass spectrometry data are discretized in the $m/z$ dimension at intervals of 0.0001 Da, based on the mass spectral resolution. Specifically, the raw continuous $m/z$ values are rounded to the nearest multiple of 0.0001 Da. $H_1$ is the latent matrix, $\mathbf{W}_1$ and $\mathbf{B}_1$ are the weight and bias matrices, $\alpha_1$ is a learnable scaling parameter, and $\beta_1$ is a learnable bias term applied to the output of the activation function.

The result in the $i$ row and $j$ column after the first layer of processing is represented as:

$$h^{(1)}_{i,j} = \alpha_1\text{SiLU}\left(\sum_{p=1}^{m}\mathbf{W}^{(1)}_{i,p}\mathbf{X}_{p,j} + \mathbf{B}^{(1)}_i\right) + \beta_1 \quad (11)$$

where $\mathbf{X}_{p,j}$ represents the $p$ row and $j$ column of $\mathbf{X}$. $\mathbf{W}^{(1)}_{i,p}$ represents the element in the $i$ row and $p$ column of the weight matrix $\mathbf{W}_1$. $\mathbf{B}^{(1)}_i$ represents the $i$ row in $\mathbf{B}_1$ bias term.

The second layer is defined as:

$$H_2 = \alpha_2\left(\text{SiLU}(\mathbf{W}_2H_1 + \mathbf{B}_2)\right) + \beta_2 = \begin{pmatrix} h^{(2)}_{1,1} & \cdots & h^{(2)}_{1,n} \\ \vdots & \ddots & \vdots \\ h^{(2)}_{512,1} & \cdots & h^{(2)}_{512,n} \end{pmatrix}, \quad (12)$$

where $H_1$ is the output matrix obtained from the first layer, $H_2$ is the latent matrix, $\mathbf{W}_2$ and $\mathbf{B}_2$ are the weight and bias matrices, respectively. $\alpha_2$ is a learnable scaling parameter, and $\beta_2$ is a learnable bias term applied to the output of the activation function.

The result in the $i$ row and $j$ column after the second layer of processing is represented as:

$$h^{(2)}_{i,j} = \alpha_2\text{SiLU}\left(\sum_{q=1}^{512}\mathbf{W}^{(2)}_{i,q}(h^{(1)}_{i,j}) + \mathbf{B}^{(2)}_i\right) + \beta_2 \quad (13)$$

where $h^{(2)}_{i,j}$ is the $i$ row and $j$ column of output matrix obtained from the second layer $\mathbf{W}^{(2)}_{i,q}$ represent the element in the $i$ row and $q$ column of the weight matrix $\mathbf{W}_2$. The bias term $\mathbf{B}^{(2)}_i$ allows the model to independently shift the activation output of the $i$ row, enhancing the network's flexibility. The output matrix from the first layer is passed through the second layer to produce the latent matrix in the feature space.

Activated by the ReLU function as follows, $H_2$ was transferred to matrix $H_3$ with 512 columns:

$$H_3 = \text{ReLU}(H_2) = \begin{pmatrix} h^{(3)}_{1,1} & \cdots & h^{(3)}_{1,n} \\ \vdots & \ddots & \vdots \\ h^{(3)}_{512,1} & \cdots & h^{(3)}_{512,n} \end{pmatrix}, \quad (14)$$

the result in the $i$ row and $j$ column of $H_3$ is represented as:

$$h^{(3)}_{i,j} = \text{ReLU}\left(h^{(2)}_{i,j}\right), \quad (15)$$

where $h^{(2)}_{i,j}$ is the output obtained from the second layer, and $H_3$ is the final feature matrix. The intra-feature learning in stage 1 of WMYn and the transformation in a network layer can finally be described as:

$$f(X_{1,j}, \ldots, X_{n,j}) = \text{ReLU}(\alpha_2\text{SiLU}\left(\sum_{q=1}^{512}\Phi_{q,i,j}\left(\sum_{p=1}^{m}\psi_{p,q}(\mathbf{X}_{p,j})\right) + b^{(2)}_i\right) + \beta_2), \quad (16)$$

$$\psi_{p,q}(X_{p,j}) = \text{I}(q \in \{1, \ldots, 512\})\mathbf{W}^{(1)}_{i,p}\mathbf{X}_{p,j}, \quad (17)$$

$$\Phi_{q,i,j}\left(\sum_{p=1}^{m}\psi_{p,q}(\mathbf{X}_{p,j})\right) = \mathbf{W}^{(2)}_{i,q}(\alpha_1\text{SiLU}\left(\sum_{p=1}^{m}\psi_{p,q}(\mathbf{X}_{p,j}) + b^{(1)}_i\right) + \beta_1) \quad (18)$$

where $I(q \in \{1, \ldots, 512\})$ is an indicator function, which equals 1 when $q \in \{1, \ldots, 512\}$, and 0 otherwise.

Inter-feature learning and resolution improvement. Our model uses 6 encoder layers along with 8-head self-attention in each encoder for data processing. Multi-head self-attention function is:

$$\text{Attention}(\mathbf{Q}, \mathbf{K}, \mathbf{V}) = \text{softmax}(\frac{\mathbf{Q}\mathbf{K}^{\text{T}}}{\sqrt{D_k}})\mathbf{V}, \quad (19)$$

where $\mathbf{Q} = H_3\mathbf{W}_Q$, $\mathbf{K} = H_3\mathbf{W}_k$ and $\mathbf{V} = H_3\mathbf{W}_V$. $\mathbf{W}_Q, \mathbf{W}_k, \mathbf{W}_V \in \mathbb{R}^{512 \times D_k}$ are projection matrices, and $D_k$ is the dimension of the subspaces for keys and queries in each attention head.

$$\text{MultiHead}(\mathbf{Q}, \mathbf{K}, \mathbf{V}) = \text{Concat}(\text{head}_1, \ldots, \text{head}_h)\mathbf{W}^O, \quad (20)$$

where $\text{head}_i = \text{Attention}(\mathbf{Q}\mathbf{W}^{\mathbf{Q}}_i, \mathbf{K}\mathbf{W}^{\mathbf{K}}_i, \mathbf{V}\mathbf{W}^{\mathbf{V}}_i)$. $\mathbf{Q}\mathbf{W}^{\mathbf{Q}}_i, \mathbf{K}\mathbf{W}^{\mathbf{K}}_i, \mathbf{V}\mathbf{W}^{\mathbf{V}}_i \in \mathbb{R}^{512 \times D_k}$, $\mathbf{W}^O \in \mathbb{R}^{HD_k \times 512}$. In this work, we employ $h = 8$ parallel attention heads.

The output is processed through a dropout, residual connection, and layer normalization:

$$H_4 = \text{LayerNorm}(H_3 + \text{Dropout}(\text{MultiHead}(\mathbf{Q}, \mathbf{K}, \mathbf{V}))), \quad (21)$$

Instead of a traditional feed-forward network. The output of the self-attention mechanism is passed through this custom layer, followed by ReLU activation, dropout, and layer normalization:

$$H_5 = \text{LayerNorm}(H_4 + \text{Dropout}(\text{ReLU}(\text{Layer}(H_4)))). \quad (22)$$

Then the $H_5$ passed through network layer to generate higher mass spectrometric resolution $H_6$.

Lipid fingerprint regenerating. The third stage integrates higher mass spectrometric features, effectively regenerating lipid fingerprints through feature integration:

$$\widehat{y} = \text{ReLU}(\text{Layer}_{\text{upsampling}}(\text{Layer}_{\text{downsampling}}(H_6))). \quad (23)$$

Fine-tuning objectives. We used the Mean Squared Error (MSE) loss function for parameter adjusting:

$$\text{MSE} = \frac{1}{n}\sum_{i=1}^{n}(y_i - \widehat{y}_i)^2 \quad (24)$$

where $n$ is the number of elements in the target vector, $\hat{y}_i$ is the predicted value, and $y_i$ is the ground truth value.

Additionally, a custom learning rate scheduler adjusts the learning rate dynamically based on the training loss to optimize the model's performance.

WMYn Model Training. The model was trained using the Adam optimizer with a learning rate of 0.01[48]. To ensure reproducibility, a random seed of 42 was set, and model parameters were initialized accordingly. The training process includes the following steps: (1) During each epoch, the model predictions were computed through a forward pass, followed by the calculation of MSE loss between the predictions and the true values. (2) Gradients were then computed via backpropagation, and the model parameters were updated using the Adam optimizer. (3) A custom learning rate scheduler was employed to adjust the learning rate dynamically based on the validation loss, with learning rates set at [0.01, 0.01, 0.001, 0.001, 0.0001] and a patience of 500 epochs. The training was conducted for a maximum of 3000 epochs. Early stopping was implemented by monitoring the loss during training and halting the process if the loss fell below a predefined threshold of 40, thus preventing overfitting and unnecessary computation. The model with the lowest validation loss after epoch 60 was selected as the optimal model, and its parameters were saved. Loss values were recorded and saved for further analysis. For each dataset, the model was trained individually. The best model for each dataset was saved and used for prediction. The predictions were then evaluated against the ground truth values to ensure the robustness and accuracy of the model. All computations were carried out using PyTorch on a high-performance computing server (https://pytorch.org/).

### Benchmark

The MS-DIAL published library was download at https://systemsomicslab.github.io/compms/msdial/main.html#MSP. Additional lipid is a hierarchical library calculated using an iterative algorithm with a total number of 168.5 million. All benchmark tests were performed on a personal computer with 13th Gen Intel® Core™ i7-13700F × 16-Core Processor, 64 GB memory, and installed with Windows11 operation system, R-4.2.3, and Python v.3.9. MS entropy and Flash entropy download from GitHub at https://github.com/YuanyueLi/SpectralEntropy, and https://github.com/YuanyueLi/FlashEntropySearch. LipidMatch was downloaded from https://github.com/GarrettLab-UF/LipidMatch. In all testing, we used MS-DIAL version v4.9.221218 and LipidSearch V4.2.

In recall testing, all methods set precursor ion matching tolerances <0.01 Da (or 5 ppm) and the MS/MS ion querying tolerance <0.025 Da (or 10 ppm). In computation time test, comparison was conducted on MS-DIAL public library, and spectra were randomly selected in mixture dataset. Flash entropy only uses the identity search module, notably. We used the RT prediction algorithm proposed by LDA to perform RT-based false positive removal, filtering false annotations using the predicted RT ± 4-fold mean RT deviation as recommended. Specifically, the "statTarget" package was used to correct for batch effects in clinical cohort of 1393 samples and another clinical cohort of 333 samples. Lipid biomarker selection was done using the "randomForest" package, setting a maximum depth of 5. Differentiation between breast cancer and healthy volunteer models using ten biomarkers was implemented using the "lightgbm" package, setting a learning rate of 0.01 and a maximum depth of 5. Clinical metrics and lipid analysis were done using the "WGCNA" package. In the analysis of expression levels at different C=C positions, the abundance ratios of the counted diagnostic ion pairs were calculated to represent the relative content ratios of the isoforms with the following equation:

$$RPA_j = PA \times \frac{IA_j + IB_j}{\sum_{i=1}^{n}(IA_i + IB_i)} \quad (25)$$

where $RPA_j$ denotes the relative peak area of the $j$th C=Cs position isomer and $PA$ denotes the peak area of all tautomers, a value generally obtained in LC-MS/MS quantitative results. $IA_j$ and $IB_j$ denote the

intensity of the paired peaks of the $j$th isomer breaks at C=C position, respectively[49,50]. Final normalization of $RPA_j$ by lipid annotations in heat maps. All the Terminology and Definitions Summary is available in Supplementary Data 11.

### Reporting summary

Further information on research design is available in the Nature Portfolio Reporting Summary linked to this article.

## Data availability

The mass spectrometry data of lipidomics were deposited to Metabolomics Workbench under accession code MTBLS10170[51] and also in National Genomics Data Center and are accessible with identifier PRJCA028507. The datasets in the three rule validations are from the following addresses, ST002384, ST001794 and ST003514 in Metabolomics Workbench [https://www.metabolomicsworkbench.org][34,39,43], DM0031 and DM0044 [https://prime.psc.riken.jp/menta.cgi/prime/drop_index][11,33], Metabolomics Workbench identifier MTBLS4684, MTBLS6965, MTBLS1369, MTBLS4654, and MTBLS6511 [https://www.ebi.ac.uk/metabolights/][35,36,40–42]. The MS-DIAL published library was download at MS-DIAL website [https://systemsomicslab.github.io/compms/msdial/main.html#MSP][11]. Additional lipid hierarchical library calculated using an iterative algorithm with a total number of 168.5 million has been uploaded in Zenodo [https://doi.org/10.5281/zenodo.14824498][52]. All data supporting the results of this study are available in the article, supplementary materials, and source data files. Source data are provided with this paper.

## Code availability

The code for LipidIN can be found at GitHub [https://github.com/LinShuhaiLAB/LipidIN], Zenodo [https://doi.org/10.5281/zenodo.14824498], and CodeOcean [https://doi.org/10.24433/CO.3229548.v3]

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

## Acknowledgements

The authors would like to thank Dr. Chenchun Zhong from SCIEX (China) Co., Ltd, Dr. Jia Li from Xiamen Meliomics Co., Ltd, and Dr. Junhan Wu from PURSPEC Technology (China) Co., Ltd for technical assistance. This work was supported by grants from the National Key Research and Development Program of China (2022YFE0205800, 2022YFA1105300), the National Natural Science Foundation of China (91957120, 21974114), Major Science and Technology Special Project of Fujian Province (2022YZ036012), the Fundamental Research Funds for the Central Universities (20720220003), Project "111" sponsored by the State Bureau of Foreign Experts and Ministry of Education of China (BP0618017) as well as grant support from Guangzhou Hybribio Medicine Technology Ltd. to S.-H.L. Natural Science Foundation of Fujian Province of China (2022J01330), Natural Science Foundation of Xiamen City of China

(3502Z20227208), and China Scholarship Council (202308350047) to J.Z.

## Author contributions

H.X. and S.H.L. conceived the project. H.X. developed and implemented E.Q. module and L.C.I. module of LipidIN framework. T.J. developed and implemented the WMYn of LipidIN framework. Y.L., L.Z., H.Y., C.Z., and S.Z. collected clinical data and obtain mass spectrometry data. H.Y., X.H., and R.M. manually checked the annotation results. H.X. built the LipidIN UI platform. H.X., T.J., L.Z., and S.H.L. wrote the manuscript. Z.Y., J.Z., Z.C., and S.H.L. reviewed the manuscript. L.D., W.Z., J.Z., Z.C., and S.H.L. supervised the project and secured funding.

## Competing interests

S.Z. is the chief technology officer of Xiamen Meliomics Co., Ltd, China. The remaining authors declare no competing interests.
