## [Transparent Peer Review file · Nature Communications]

LipidIN: a comprehensive repository for flash platform-independent annotation and reverse lipidomics

Corresponding Author: Professor Shuhai Lin

Version 0:

Reviewer comments:

Reviewer #1

(Remarks to the Author)

The authors have developed LipidIN, a software program that uses machine learning technology to improve the accuracy of lipid profile annotations. The core parts of this software are the expeditious querying (EQ) module, based on a non-informative prior greedy algorithm, and the lipid category intelligence (LCI) module. LCI includes the Wide-spectrum Modeling Yield network (WMYn), which maps the mass-to-charge ratios and intensities of multiple batches into a shared latent space by integrating experimental results from a large number of batches.

Overall, this software has been shown to be faster and more accurate than existing softwares. However, it requires revision for publication in Nature Communications due to insufficient explanation and validation of some of the details of its main architecture.

The authors claim that WMYn is based on the Kolmogorov-Arnold representation theorem, allowing it to achieve greater flexibility and accuracy with fewer parameters than MLP in "reverse lipidomics." However, the formulation given in the intra-feature learning subsection of the Method section needs to be revised to show that WMYn is based on the Kolmogorov-Arnold representation theorem. If, as written in lines 612-613, f is a SiLU and the multi-variable function $\Phi(Y)$ is defined as the sum of $f_j(WX + B)$, then this is unrelated to the Kolmogorov-Arnold representation theorem and is an artificial neural network with SiLU activation function. The authors need to fix the notation that gives rise to such misunderstandings in the Method section, clearly define each variable and function, and show a precise formulation of the WMYn architecture. In particular, they need to clearly define the variables and functions that appear in Equations (10), (11), and (12) and their relationships (e.g., the definition of Y_i and what Y_i and X correspond to in this study).

From here on, I will proceed on the assumption that the text in the intra-feature learning subsection of the Method section is incorrect and will be corrected to a formulation based on the Kolmogorov-Arnold representation theorem. As mentioned above, the authors claim that WMYn is based on the Kolmogorov-Arnold representation theorem, allowing it to achieve greater flexibility and accuracy with fewer parameters than MLP in reverse lipidomics. If the authors make such a claim, they need to show their advantages quantitatively by comparing their architecture with a multi-layer perceptron.

(Remarks on code availability)

Reviewer #2

(Remarks to the Author)

This study developed LipidIN that enables users to perform instant, accurate, and in-depth searches within a hierarchical in silico library containing over 100 million lipid structures. The outcome is interesting and would attract attention from wide ranges of communities. Traditional spectrum searching methods may overestimate similarity by double-counting sub-fragments from the same structure, even when important fragments are missing. Accurate compound identification can be challenging, particularly when the reference library lacks experimental data for the specific compound. Moreover, the scarcity of spectral libraries detailing compound structures impedes lipid annotation. The study showed potential to overcome such traditional problem by constructing a 166.3-million lipid fragmentation hierarchical library, and significantly accelerating library searches through hash table and bisection algorithms. The study also provides retention time-based false positive removal and spectral regeneration capabilities. Benchmarking case studies show convincing results of

performance superiority compared to other common approaches. The study is highly expected to enhance the annotations on unknowns that measured by LipidIN in clinical cohorts. Prior to publication, authors should address the following issues.

1. Is the data in .mzML centroided, or profile? Please indicate it explicitly.
2. Is there a difference in mean matching value of highly ranked list (e.g. top 20) between EQ + LCI + 1st-4th library, EQ + LCI + MS-DIAL published library and EQ + MS-DIAL published library? If the EQ + LCI + 1st-4th library improves the average value not only recall@1 and 20, it indicates that the hierarchical in silico library approach has an advantage in suggesting rough structure (i.e. fuzzy structure) of the query rather than traditional method.
3. In the main text, there are several references to speeds that are not consistent (Line 33: over 100 billion times' querying in 1 s, Line 194: 0.14 s for querying 1,000,000 spectra library, Line 344: average 133.47 billion times querying per seconds). These discrepancies likely arise from different testing conditions, dataset sizes, or query complexities. A detailed explanation of the methodologies used in each case would help readers understand these variations in reported performance.
4. The study states that it pertains to metabolite annotations (Line 34, Line 217); however, there are no comprehensive benchmarking comparison results available, aside from those presented in Figure S4 a. Please consider adding more detailed comparisons.
5. Why a fold change of 1.4 was chosen to identify differential metabolites in third clinical cohort?
6. The Discussion section references the Kolmogorov-Arnold theorem in the context of reverse lipidomics modules. However, the integration of this theorem into the model requires further elaboration, specifically, a clearer explanation of how the Kolmogorov-Arnold theorem is applied to the reverse lipidomics modules is needed.
7. Given the increasing research focus on oxidized lipids, it is noteworthy that Table S1 lacks comprehensive representation of these important molecules. Consider expanding the lipid categories to include oxidized variants of major phospholipids, such as: Oxidized phosphatidylcholine (OxPC), Oxidized phosphatidylethanolamine (OxPE), Oxidized phosphatidylglycerol (OxPG), Oxidized phosphatidylserine (OxPS).
8. Authors should describe how to prepare query spectrum library for users.
9. Regarding Figure 1: The icon font used in the figure is too small, which may impair readability. Please increase the font size to ensure that all text is clearly legible, especially when the figure is scaled down in print or on-screen viewing. There is an instance of obscured figure text. This obstructed text compromises the clarity of the information being conveyed. Please revise the figure layout to ensure that all text is fully visible and not overlapped by other elements.
10. Regarding Figure 3, please explain why the flash entropy appears to have no data point at the 10^7 mark. The figure shows no substantial difference between EQ + LCI + MS-DIAL published library and EQ + MS-DIAL published library presented in Figure b. Is this because the benchmarks don't use a hierarchical library? Please elaborate. In addition, the legends for Fig. 3d and Fig. S7c are inconsistent. To my knowledge, the LDA software includes several modules, such as library searching and retention time judgment. It is recommended to modify the expression to 'LDA retention time prediction algorithm' for clarity.
11. Regarding Figure 5b, I find the text of the labels very small and difficult to read. Please increase the font size of these figures.
12. Figure S8 needs to be in higher resolution.
13. In the lipid categories intelligence model, the training set is not explicitly listed.
14. The manuscript should address the scenario of multiple measured spectrogram peaks falling within the same threshold interval. Please clarify the handling method for cases where multiple measured peaks match within a single threshold interval. Explain whether Equations 1 and 2 account for this multi-peak matching situation. If they do, describe how these equations incorporate multiple peak matches. If Equations 1 and 2 do not account for multiple peak matches Consider modifying the equations to handle this scenario. Alternatively, explain why this situation is not addressed and discuss any potential implications for the analysis.
15. The relevance of Equation 10 to the subsequent mathematical development is unclear. Please providing a clear explanation of how Equation 10 relates to and informs the equations that follow, including its role in the overall mathematical framework.

(Remarks on code availability)

(Remarks to the Author)

LipidIN does show great promise with highlights specifically being: large database size, algorithm speed, accuracy, and inclusion of EAD fragmentation for double bond location (which provides important biological subtleties). The ECN models are especially beneficial for increasing coverage. The main limitation of the study is:

- 1) No ready to use interface, and the approach consists of multiple programming languages to my knowledge (Python, R, etc., making it difficult to learn). Video tutorials would be very helpful showing the workflow from start to finish. This made it difficult to validate without substantial effort to learn the workflow. This will limit the number of users.
- 2) Only covers one aspect of the workflow (annotation; this will also limit the number of users), and hence a diagram or user manual showing how to integrate all other aspects of the workflow could be helpful. How should a lipidomics user incorporate this software?
- 3) The organization of the files for the reviewer is very poor, for example in the merged document Figures don't have numbers and are not contained with their captions, making it very hard to understand what is happening. Also it would be nice to find each supplementary table with a linked index as they seemed to just be dropped into the submission.
- 4) While this manuscript ideally would be of interest to the lipidomics community, the language throughout is very dense (computer programming and model jargon heavy), without much explanation of how each aspect relates to a lipid expert terminology or workflow. This was a very hard read.
- 5) The software was validated against MS-DIAL, LipidMatch, LipidSearch, etc. but the graphs and results of this comparison are also hard to understand. Label your axis and you're your captions explain things in a way where the reader does not need to read the text and/or supplemental to understand what is being compared and what recall, "percentage", etc. mean. Walking the reader through an application such as the NIST interlab study on SRM 1950 which has a set of "known lipids", and the percent coverage of each software of these lipids, false positive rate, false negative rate, and overlap between software would be very helpful. For example see Figure 6 of the Lipid Annotator paper (<https://pmc.ncbi.nlm.nih.gov/articles/PMC7142889/>) which shows shared coverage by class and C:DB. You could also show overlap by fatty acid composition as the top ranked candidate between software. You could also spike in a much large set (100s) of endogenous lipid standards and see the false positive, negative, and coverage of each software. LipiDex is another leading software worth comparing, although we know full coverage of all software may be out of the scope of this project.
- 6) Comparison between software is very challenging. How were parameters kept the same? Were different peak picking algorithms used? (Note certain software only work with their own peak picking algorithms). Where were the fault points in each where annotations were missed? Etc.
- 7) While EAD is one of the highlights of this paper, it is only briefly mentioned and the number of new lipids with different double bond positions, etc. which can be discovered in well known datasets are not well described.
- 8) While this software has a large database, sub-class coverage is actually smaller than for example LipidSearch, MS-DIAL, and LipidMatch.

Other specific comments:

Make sure to include version numbers when doing comparisons. For example, the latest versions of MS-DIAL and LipidMatch are very unique as compared to other older versions.

Make sure to cite very clearly where all fragment and class information came from.

MS-DIAL, LipidSearch, and LipidMatch, are stated as matching experimental spectra to library spectra which is very vague, and the limitations mentioned depend on the algorithm employed. For example while MS-DIAL includes dot-product type matching, retention index, etc. LipidMatch includes a rule-based approach based on fragment occurrence (intensity independent). Therefore, software such as LipidMatch, ALEX, etc. are generally instrument/matrix independent but do rely on low abundance peaks (as mentioned as a limitation). The problem is no matter what the model (machine learning or rule based) without these low abundant peaks which are sometimes the only ones necessary for structural annotation, structure cannot be assigned correctly. Furthermore, a comparison of these software suggest false positive rates are probably <5% at least at the lipid class level, so close to this reported confidence reported here, showing that the diverse algorithms perform similarly in many cases regardless of matrix/training set. <https://pmc.ncbi.nlm.nih.gov/articles/PMC7142889/>

Line 126, please reference which community libraries were used, as I believe reference 30 does not have all libraries used.

Line 129 Define "fragment fingerprints", also afterward you start talking about each level and referring to the various levels which becomes complicated to follow. Rather, I would describe each "level" in a paragraph or few sentences, and then summarize all of them.

166.3 million theoretical lipid fragments across 76 lipid sub-classes is a lot, but the coverage is actually smaller than some of the most commonly use lipidomics software. For example, MS-DIAL has the greatest coverage with 117 lipid sub-classes, and LipidMatch has over 100 different lipid sub-classes as well.

Line 175 described "median conformity" it would be good the get some easily understood metrics such as the +/- tolerance in minutes for the 95 and 99 percentile of lipids, and whether there are error rates calculated for individual retention time values.

"approach using EQ and LCI identified more 471 unique potential lipids" incorrect English

"high-confidence potential lipids." Is strange language, can you be high confident and just a potential lipid at the same time?

The WMYn model was validated against 15 lipid standards, whereas there are 100s and 100s of lipid standards available (see Avanti, Nu-Check prep, etc.), what is the rationale for using such a limited number of standards for validation, especially as these cannot cover a great majority of the lipid classes you have included in your library?

Throughout the manuscript what does “higher recall” mean in context of your lipid annotations? Please clarify all the computer program and model specific language for the average lipidomics user. Please define these terms before use. Other terms to define for the average lipidomics user “expeditious querying”, “hierarchical library”, “fragment fingerprints”, etc.

(Remarks on code availability)

I reviewed the code briefly but did not set up an environment and test due to time constraints on my side. There was a ReadMe and the code seemed well commented.

Version 1:

Reviewer comments:

Reviewer #1

(Remarks to the Author)

The authors responded to the Reviewer's comments. However, the formulation and definition of the machine learning architecture still need to be clarified. The current manuscript is not suitable for publication in Nature Communications. Please see the supplementary material for the details of my questions and comments.

(Remarks on code availability)

Reviewer #2

(Remarks to the Author)

In the revised version, these additions have effectively addressed my previous concerns. LipidIN does show great promise and would be widely used in lipidomics community. In particular, reverse lipidomics module exhibits powerfully regenerative capacity for increasing accuracy and coverage. Therefore, I am satisfied with the revised manuscript.

(Remarks on code availability)

Reviewer #3

(Remarks to the Author)

I appreciate all the in-depth responses to my comments. I especially was impressed and appreciated the table generated containing all technical terms and the corresponding definitions.

To complete the review and accept this manuscript, I would like to be able to test the software. Currently, I have not been able to successfully run the software on my own data (which I would need for just a simple validation that the resulting annotations are realistic for a new test data), nor could I get the software to work on the test data using the exact same parameters and zip file provided.

<https://www.lifemetabolomics.cn/LipidIN>

The user interface was relatively straight forward.

I am not sure how mzML conversion parameters (e.g. compressed versus not compressed files) effect how the software runs so a tutorial for mzML conversion on the page may be helpful.

Note you mention including your file as yzip format, only zip format exists, must be a misspelling?

Note “Working” in all portions of the webpage, which are the buttons to click when running the data, should be changed to “Run” or “Process Data”

Note that the multiple options (option 0, option 1...) on the top left tabs are confusing and should have titles/better descriptions somewhere

Also note that the 2GB limit makes sense given that you are potentially running the algorithms on your own server? But this limit will significantly limit the usecase as some people will have 100 of GBs worth of files. At this point maybe LipidIN does not have to support 100 of GBs worth of files because it only needs representative MS/MS files (2 GB is still to small, at least 10 GB), but for the quant workflow you mentioned it would definitely need to have higher GB available. The quant workflow could benefit from being integrated into the WebApp...

I ran one single mzML file that was not too large, and I got the following error:
“系统接口请求超时”

“System interface request timeout”

I had it formatted correctly I believe: data.zip/data/ddMS28387_Pos.mzML and I converted with MSConvert using the following parameters:

peakPicking: vendor msLevel=1-2

threshold: absolute 1000 most-intense

I also tried a parameterless conversion to mzML and your own demo data (just downloaded, reuploaded, and got the same error)

And using the demo file that you provided and got the same error. I tested my internet connection and speed and it was fine.

12/22/2024 10:39am EST was the time ran

Note the attached supplemental file has my review with screenshots of my run with all parameters ran.

(Remarks on code availability)

I only tested the web interface.

I could not get it to work, although the workflow seemed relatively straight forward.

Version 2:

Reviewer comments:

Reviewer #1

(Remarks to the Author)

First of all, I appreciate the author's sincere responses to all the comments. However, I still have many comments on the revised manuscript. Therefore, I consider the current manuscript unsuitable for publication in Nature Communications. Please see the attached file for detailed comments.

(Remarks on code availability)

Reviewer #3

(Remarks to the Author)

I have included my review as a .docx which has some comparative analysis I did and some graphs.

I was able to now run the code in RStudio given the instructions provided (which I assume are also on the website).

Pros:

1) the software ran extremely fast!

2) The software was relatively straightforward to use.

3) The software was relatively accurate: Spot checking the MS/MS spectra and RT trends, as well as benchmarking against LipidMatch, showed that the software was quite accurate in annotations.

4) Coverage was high, for example using the same file (adipose tissue) via LipidMatch 235 unique m/zs (two decimal points) were annotated whereas in LipidIN 243 unique m/zs were annotated showing similar coverage.

Cons:

1) Results are highly sensitive to the Abundance filter, and setting the abundance filter high creates many false negatives and low creates several false positives without any numbers allowed between. Coverage was relatively low when putting the abundance filter to the recommends 0.05, for example using the same file (adipose tissue) via LipidMatch 235 unique m/zs (two decimal points) were detected whereas only 160 were detected by LipidIN in positive mode. LipidIN missed some predominant lipids including PC(16:0_18:1) (one of the major common lipids) as well as all other PCs, due to their low acyl fragments. Hence the abundance filter was set to 0 (there was only one option 0.01 or 0, could not put 0.0001 as desired... At 0 several false positives occurred including MGDGs (plant lipids) in adipose (MGDG 22:2 | MGDG 8:2_14:0).

2) Retention times reported are for MS/MS scans not for peaks, with peak RTs and abundances shown

3) Most aspects of the workflow were not covered by the implementation recommended by the author to me. Peak picking and alignment across samples, gap filling, deisotoping, blank filtering, combining negative and positive mode, are some essential steps in the lipidomics workflow not covered by LipidIN. This will limit usability.

More detailed feedback:

The GUI does not like long directories (they don't paste fully and will cause errors).

You could distribute R packages and an R version with LipidIN to reduce future dependency issues.

The following installation in the R Code:

```
install.packages(  
pkgs = '~/LipidIN-main/LipidIN GUI/LipidIN_2.0.0.1.tar.gz',  
lib = .libPaths()[length(.libPaths())],  
repos = NULL,  
dependencies = T  
)
```

Errors when you put the exact directory (as mentioned in the instructions).

I needed to go to the same documents folder it was looking for and put the tar.gz there for successful installation, so this could be modified to be able to take a full path.

The retention times reported are from the MS/MS scans (time of MS/MS scan) which is not that retention time of the peak itself. For example, for the two TGs below RTs of 14.90 and 15.39 were reported, whereas the RTs at the peak maximum were not reported (14.96 and 15.31). See Figures in attached .docx.

This may be why when I plotted RTs against m/z for the same lipid class and degree of unsaturation the trend was not perfect (albeit it's not always expected to be perfect). For reference I have included RT vs m/z trends from another software for a different dataset to show how clear they often are. Are the RTs which are used for retention time analysis the retention times reported in pos_ALL_final_output.csv, because if they are these are the incorrect RTs (the MS/MS scan RT is not the RT of the peak itself). Furthermore, if the reported RTs are not the ones used, the RTs of the peaks themselves should be reported. See Figures in attached .docx.

I did a quick comparison to LipidMatch. I was able to get 160 unique annotations base on a unique m/z in lipidIN in positive mode to two decimal points. I was able to get 235 unique m/zs annotated in LipidMatch with high confidence in positive mode to two decimal points. I spot checked a few TGs and both LipidMatch and lipidIN gave the same top annotation.

(Remarks on code availability)

See main review

Version 3:

Reviewer comments:

Reviewer #1

(Remarks to the Author)

The authors responded to all comments appropriately. I recommend this manuscript for publication in Nature Communications.

High performance is valuable in itself, and it doesn't matter whether what's behind it is new or old.

(Remarks on code availability)

Reviewer #3

(Remarks to the Author)

Thank you for your in-depth response and patience in this back and forth, I am excited to see all the improvements in the software via this discussion.

1) I appreciate the emphasis of "reverse lipidomics" and focus on annotation. Indeed, this could allow for targeted peak picking afterwards. This could be emphasized in the text in the intro/conclusion, that this is a reverse lipidomics approach allowing annotation without peak pickign first (similar to LipidSearch). Then leave the community to find a good automated targeted peak picking solution, which is highly needed. I have heard good things about EI Maven, and MZMine 2 (not 3 or 4) worked very well for targeted peak picking for non drifted data, but was very slow. I am not aware of high performing targeted peak picking workflows which are mostly automated (but this is not for you to solve! But just would be helpful in the future for users of your workflow to have a complete workflow).

2) I do not know of good automated non-targeted peak picking software, I have not used XCMS except for non-targeted peak picking where it performs relatively poorly compared to MZMine and vendor peak picking solutions in multiple in-house benchmarking studies (including involving developers). But maybe it will suffice for just finding the right RTs of the peaks, make sure to evaluate the accuracy of XCMS for finding the peak center if you haven't already.

3) Using "Apex Peak Trigger" or similar should be recommended for your work along with a dynamic exclusion where only 1

scan is acquired per peak ideally. You should add a sentence or two on this in the paper along with the discussion of the limitation of the RT used. That way, at least the DDA data will likely have correct RTs most of the time depending on the performance of the vendors Apex Peak Trigger, I know Thermo and Agilent algorithms work nicely. Indeed, the data I used to benchmark your algorithm was Apex Peak Triggered (tries to find the peak apex, which is what you want). I understand now why you used the MS/MS RT to save time and not have to incorporate a peak picking algorithm.

4) Our TGs were not isomers unlike your examples, as you can see in our previous figure the RT drift is way too large, plus we pulled up examples showing it was just about using the RT from the triggered MS/MS (hence make sure the users know even with Apex trigger there will be false positives/negatives due to the MS/MS RT not being the correct RT).

Overall, other than a few minor sentences added to the manuscript as mentioned above, I am happy publishing the manuscript.

(Remarks on code availability)

Content

Reviewer #1 (Remarks to the Author):	3
Comment 1.....	3
Comment 2.....	5
Comment 3.....	7
Reviewer #2 (Remarks to the Author):	29
Comment 1.....	29
Comment 2.....	30
Comment 3.....	31
Comment 4.....	32
Comment 5.....	33
Comment 6.....	34
Comment 7.....	36
Comment 8.....	41
Comment 9.....	41
Comment 10.....	42
Comment 11.....	42
Comment 12.....	43
Comment 13.....	44
Comment 14.....	45
Comment 15.....	45
Reviewer #3 (Remarks to the Author):	47
Comment 1.....	47
Comment 2.....	51
Comment 3.....	52
Comment 4.....	53
Comment 5.....	57
Comment 6.....	63

Comment 7.....	65
Comment 8.....	79
Comment 9.....	83
Comment 10.....	83
Comment 11.....	89
Comment 12.....	91
Comment 13.....	91
Comment 14.....	92
Comment 15.....	92
Comment 16.....	92
Comment 17.....	96
Comment 18.....	96
Comment 19.....	96
Comment 20.....	101
Reviewer #3 (Remarks on code availability):	103
Comment 21.....	103
Reference.....	104

REVIEWER COMMENTS

Reviewer #1 (Remarks to the Author):

The authors have developed lipidIN, a software program that uses machine learning technology to improve the accuracy of lipid profile annotations. The core parts of this software are the expeditious querying (E0) module, based on a non-informative prior greedy algorithm and the lipid category intelligence (LCI) module. LCI includes the Wide-spectrum Modeling Yield network (WMYn), which maps the mass-to-charge ratios and intensities of multiple batches into a shared latent space by integrating experimental results from a large number of batches.

Overall, this software has been shown to be faster and more accurate than existing software. However, it requires revision for publication in Nature Communications due to insufficient explanation and validation of some of the details of its main architecture.

Comment 1. The authors claim that WMYn is based on the Kolmogorov-Arnold representation theorem, allowing it to achieve greater flexibility and accuracy with fewer parameters than MLP in "reverse lipidomics." However, the formulation given in the intra-feature learning subsection of the Method section needs to be revised to show that WMYn is based on the Kolmogorov - Arnold representation theorem. If as written in lines 612-613, f is a SiLU and the multi-variable function $o(y)$ is defined as the sum of $f_i(WX + B)$, then this is unrelated to the Kolmogorov-Arnold representation theorem and is an artificial neural network with SiLU activation function.

Response: We sincerely appreciate your thoughtful and constructive comments! We apologize for any misunderstanding regarding the basis of WMYn as being constructed solely on Kolmogorov-Arnold Representation Theorem (KART). Indeed, our work is inspired by the function representation capabilities of KART, particularly its approach to decomposing complex functions. We aim to represent spectral data, influenced by

various factors such as mass spectrometry conditions and ion sources, as a sum of multiple observational samples.

The two-layer network architecture we employed effectively implements this decomposition. Each node (or neuron) in the network can be viewed as a simple function, and the entire network serves as a combination of these simpler functions. By adjusting the network parameters, we are able to approximate any complex function.

Additionally, we integrate activation functions such as SiLU to enable the network to capture intricate nonlinear transformation patterns within the data. This combination enhances the network's ability to model the underlying complexities of the spectral data. We will provide a more detailed explanation below in relation to the equations.

WMYn is inspired by the KART for mass spectrometry representation. We propose that the true spectrum \mathbf{S} of a lipid is influenced by various factors, including mass spectrometry conditions (Y_1), chromatography conditions (Y_2), ionization voltage (Y_3), and the nature of the compound (Y_4). Consequently, the true spectrum \mathbf{S} can be expressed as a multivariate function that accounts for these variables $\mathbf{S} = \Phi(Y_1, Y_2, \dots, Y_n)$. According to KART, any multivariate continuous function $\Phi(Y_1, \dots, Y_n)$ can be represented as a finite number of single-variable continuous functions in a two-layer nested summation:

$$f(X) = f(x_1, \dots, x_n) = \sum_{q=1}^{2n+1} \Phi_q \left(\sum_{p=1}^n \phi_{q,p}(x_p) \right),$$

thus, this function can be represented as a finite unitary function $\mathbf{S} = \sum_{j=1}^m \Phi_j$, where m is a finite constant. We utilize the spectrum from a large cohort, which has undergone certain variations is \mathbf{F} , where $\mathbf{F} = \sum_{i=1}^n f_i(WX + B)$, where X is a particular observation and the f_i function is a process for mapping X . Based on the above derivation, we construct a representation of the real spectrogram in relation to the spectrogram in the cohort:

$$\Phi(Y_1, \dots, Y_n) = \sum_{j=1}^{2n+1} \Phi_j(\sum_{i=1}^n f_i(WX + B)).$$

where X represents the input matrix, consisting of x_1, \dots, x_n , where each x is a spectrum peak. W being the weight matrix, and B the bias matrix. f_i is the Sigmoid

linear unit (SiLU) activation function to enable the network to capture intricate nonlinear transformation patterns within the data, and Φ_j is a learnable nonlinear activation. By adding learned parameters in f_i , a learnable nonlinear activation is achieved.

Inspired by the KART, $\sum_{p=1}^n \phi_{q,p}(x_p)$ represents the weighted combination of features of x_1, \dots, x_n for each spectrum. In the first stage of WMYn, the linear transformation is the method that implements this weighted sum. With the learned weights $\phi_{q,p}$, each input feature x_p participates in the weighted sum:

$$\sum_{i=1}^n f_i(WX + B)$$

Φ_q is a nonlinear transformation that processes the above weighted sum as input. In the first stage of WMYn, a learnable nonlinear activation function is implemented based on the learned parameters. For each spectrum, a different activation function is learned, and these spectra are mapped to the latent feature space through the learned activation functions, ultimately producing the final feature matrix.

It is true that the SiLU function and the multivariable function are not directly related to KART as you mentioned. However, we constructed the representation function of the spectrogram based on KART principle, enabling the two nonlinear layers in the first stage for intra-feature learning. We also revised manuscript in Line 678 and Equation 11.

Comment 2. The authors need to fix the notation that gives rise to such misunderstandings in the Method section, clearly define each variable and function, and show a precise formulation of the WMYn architecture. In particular, they need to clearly define the variables and functions that appear in Equations (10), (11), and (12) and their relationships (e.g., the definition of Y_i and what Y_i and X correspond to in this study). From here on, I will proceed on the assumption that the text in the intra-feature learning subsection of the Method section is incorrect and will be corrected to a formulation based on the Kolmogorov-Arnold representation theorem.

Response: Thank you for your constructive comments! We have revised the notation in the Methods section, providing clear definitions for each variable and function, as well as clarifying the relationships between them. Additionally, we present a more precise formulation of the WMYn architecture, as detailed below:

$$\Phi(Y_1, \dots, Y_n) = \sum_{j=1}^{2n+1} \Phi_j \left(\sum_{i=1}^n f_i(WX + B) \right)$$

where X representing the input matrix, consisting of x_1, \dots, x_n for mass spectra of real samples. W being the weight matrix, and B is the bias matrix. f is the Sigmoid linear unit (SiLU) activation function, and Φ_j is a learnable nonlinear activation. Y_1, \dots, Y_n denote the output matrix, corresponding to the theoretical spectra in the chromatographic and mass spectrometric conditions.

The details of each layer in the first stage of the WMYn architecture are as follows: The feature extractor consists of two SiLU-activated layers followed by a Rectified Linear Unit (ReLU) activation. The first layer is defined as:

$$H_1 = \alpha_1 \left((W_1 X + B_1) \sigma(W_1 X + B_1) \right) + \beta_1$$

where X represents the input matrix, consisting of x_1, \dots, x_n , where each x is a spectrum peak. H_1 is the latent matrix, W_1 and B_1 are the weight and bias matrices, α_1 is a learnable scaling parameter, and β_1 is a learnable bias term applied to the output of the activation function. σ denotes the Sigmoid function, representing the SiLU activation function. The input matrix X , composed of multiple batches of spectra, is mapped to the latent feature space through the first layer, producing the latent matrix H_1 as the output.

The second layer is defined as:

$$H_2 = \alpha_2 \left((W_2 H_1 + B_2) \sigma(W_2 H_1 + B_2) \right) + \beta_2$$

where H_1 is the output matrix obtained from the first layer, H_2 is the latent matrix, W_2 and B_2 are the weight and bias matrices, α_2 is a learnable scaling parameter, and β_2 is a learnable bias term applied to the output of the activation function. The output matrix H_1 from the first layer is passed through the second layer to produce the latent

matrix H_2 in the feature space.

Activated by the ReLU function as follows, H_2 was transferred to matrix H_3 :

$$H_3 = \text{ReLU}(H_2).$$

where H_2 is the output matrix obtained from the second layer, and H_3 is the final feature matrix. We also revised manuscript in Line 680, Equation 11 and Equation 12.

Comment 3. As mentioned above, the authors claim that WMYn is based on the Kolmogorov-Arnold representation theorem, allowing it to achieve greater flexibility and accuracy with fewer parameters than MLP in reverse lipidomics, If the authors make such a claim, they need to show their advantages quantitatively by comparing their architecture with a multi-layer perceptron.

Response: Thank you for your valuable suggestion! As mentioned above, the intra-feature learning in WMYn is inspired by KART. We revised the manuscript to emphasize small-sample learning in WMYn, rather than focusing solely on fewer parameters. Additionally, we explained the superior performance of WMYn (KART) compared to WMYn (MLP), where the layers of WMYn (KART) are replaced with multilayer perceptron (MLP) layers that do not incorporate KART.

We anticipate that WMYn will outperform the MLP in these scenarios. To evaluate this, we performed random sampling on the dataset to reduce the input size, selecting sample sizes of 10, 75, and 150. Each sampling was repeated 10 times, and we retrained both WMYn (KART) and WMYn (MLP) with the number of epochs set to 100. For each random sample, we computed the cosine similarity¹ between the predicted results and the true spectral data. The cosine similarities for each sampling were averaged, and the results are visualized in Fig. R1.a.

We utilized ridge plots to assess the cosine similarity between experimentally detected spectra and predicted spectra. The red and blue colors represent WMYn (MLP) and WMYn (KART), respectively, with mean values indicated by dotted lines. Notably, even with a small training sample size ($n=10$), the average spectral cosine similarity for the WMYn predicted spectra is approximately 0.8, indicating a high degree of accuracy.

However, when the sample size increases to 75, the predicted cosine similarity for WMYn (MLP) does not achieve the performance level observed with the smaller sample size (n=10). Even with a doubled sample size of 150 (n=150), the prediction performance of WMYn (MLP) still does not match that of WMYn (KART) with a smaller sample size of 75 (n=75).

Furthermore, as the sample size increases, the prediction accuracy of WMYn (KART) also improves. This indicates that the combination of WMYn and KART outperforms MLP in terms of prediction accuracy with smaller sample sizes. This finding aligns with practical use cases, as detecting MS2 patterns across a large number of samples can be costly in experimental settings. Additionally, the specific experimental results are presented in Table R1. We also revised manuscript in Line 89, 291, 464 and Supplementary Fig. 6 and Source Data Supplementary Supplementary Fig.6.

Fig. R1. Cosine similarity performance comparison between WMYn and WMYn(MLP). **a.** The cosine similarity between predicted spectra trained with different sample sizes and experimental detection spectra. The red bars represent WMYn (MLP), the blue bars represent WMYn (KART), and the dotted line indicates the mean cosine similarity. **b.** The cosine similarity between the predicted spectra and experimental spectra for 15 lipid reference standards under epoch = 3000, with an input sample size of 10. Error bars represent standard deviation, and a T-test was used for significance testing.

Table R1. Cosine similarity results for different models (epoch = 100).

Method: WMYn (MLP)			Method: WMYn (KANRT)		
Lipid reference standards	Used sample count	Cosine similarity	Lipid reference standards	Used sample count	Cosine similarity
Cer 18:1;2O/16:0	10	0.33	Cer 18:1;2O/16:0	10	0.19
LPC 16:0	10	0.00	LPC 16:0	10	1.00
PC 16:0_16:0	10	0.67	PC 16:0_16:0	10	0.99
PC 16:0_18:1	10	0.90	PC 16:0_18:1	10	0.06
PC 18:0_18:0	10	0.00	PC 18:0_18:0	10	0.09
PC 18:1_18:1	10	0.07	PC 18:1_18:1	10	0.99
PE 18:1_18:1	10	0.47	PE 18:1_18:1	10	1.00
PI 18:1_18:1	10	0.33	PI 18:1_18:1	10	0.99
SM 18:1;2O/16:0	10	0.62	SM 18:1;2O/16:0	10	0.07
Cer 18:1;2O/16:0	10	0.84	Cer 18:1;2O/16:0	10	1.00
LPC 16:0	10	0.65	LPC 16:0	10	0.97
PC 16:0_16:0	10	0.97	PC 16:0_16:0	10	0.04
PC 16:0_18:1	10	0.33	PC 16:0_18:1	10	1.00
PC 18:0_18:0	10	0.28	PC 18:0_18:0	10	0.99
PC 18:1_18:1	10	0.08	PC 18:1_18:1	10	0.99
PE 18:1_18:1	10	0.54	PE 18:1_18:1	10	1.00
PI 18:1_18:1	10	0.02	PI 18:1_18:1	10	1.00
SM 18:1;2O/16:0	10	0.52	SM 18:1;2O/16:0	10	0.99
Cer 18:1;2O/16:0	10	0.52	Cer 18:1;2O/16:0	10	0.99
LPC 16:0	10	0.00	LPC 16:0	10	0.08
PC 16:0_16:0	10	0.97	PC 16:0_16:0	10	0.99
PC 16:0_18:1	10	0.96	PC 16:0_18:1	10	0.05
PC 18:0_18:0	10	0.84	PC 18:0_18:0	10	0.99
PC 18:1_18:1	10	0.15	PC 18:1_18:1	10	0.99
PE 18:1_18:1	10	0.93	PE 18:1_18:1	10	1.00
PI 18:1_18:1	10	0.89	PI 18:1_18:1	10	0.13
SM 18:1;2O/16:0	10	0.93	SM 18:1;2O/16:0	10	0.07
Cer 18:1;2O/16:0	10	1.00	Cer 18:1;2O/16:0	10	1.00
LPC 16:0	10	0.00	LPC 16:0	10	0.40
PC 16:0_16:0	10	0.97	PC 16:0_16:0	10	0.02
PC 16:0_18:1	10	0.97	PC 16:0_18:1	10	0.09
PC 18:0_18:0	10	0.89	PC 18:0_18:0	10	0.48
PC 18:1_18:1	10	1.00	PC 18:1_18:1	10	0.08
PE 18:1_18:1	10	0.27	PE 18:1_18:1	10	0.01
PI 18:1_18:1	10	0.30	PI 18:1_18:1	10	0.99
SM 18:1;2O/16:0	10	0.92	SM 18:1;2O/16:0	10	1.00
Cer 18:1;2O/16:0	10	0.29	Cer 18:1;2O/16:0	10	0.18
LPC 16:0	10	0.00	LPC 16:0	10	1.00
PC 16:0_16:0	10	0.97	PC 16:0_16:0	10	0.99

PC 16:0_18:1	10	0.00	PC 16:0_18:1	10	1.00
PC 18:0_18:0	10	0.93	PC 18:0_18:0	10	0.99
PC 18:1_18:1	10	0.00	PC 18:1_18:1	10	0.79
PE 18:1_18:1	10	0.02	PE 18:1_18:1	10	0.06
PI 18:1_18:1	10	0.88	PI 18:1_18:1	10	0.99
SM 18:1;2O/16:0	10	0.91	SM 18:1;2O/16:0	10	0.00
Cer 18:1;2O/16:0	10	0.46	Cer 18:1;2O/16:0	10	0.16
LPC 16:0	10	0.65	LPC 16:0	10	0.97
PC 16:0_16:0	10	0.81	PC 16:0_16:0	10	0.99
PC 16:0_18:1	10	0.94	PC 16:0_18:1	10	1.00
PC 18:0_18:0	10	0.93	PC 18:0_18:0	10	0.99
PC 18:1_18:1	10	0.00	PC 18:1_18:1	10	0.99
PE 18:1_18:1	10	0.81	PE 18:1_18:1	10	0.98
PI 18:1_18:1	10	0.16	PI 18:1_18:1	10	0.99
SM 18:1;2O/16:0	10	0.55	SM 18:1;2O/16:0	10	0.99
Cer 18:1;2O/16:0	10	0.89	Cer 18:1;2O/16:0	10	1.00
LPC 16:0	10	0.39	LPC 16:0	10	1.00
PC 16:0_16:0	10	0.96	PC 16:0_16:0	10	1.00
PC 16:0_18:1	10	0.97	PC 16:0_18:1	10	0.91
PC 18:0_18:0	10	0.77	PC 18:0_18:0	10	0.16
PC 18:1_18:1	10	0.99	PC 18:1_18:1	10	0.99
PE 18:1_18:1	10	0.00	PE 18:1_18:1	10	1.00
PI 18:1_18:1	10	0.32	PI 18:1_18:1	10	1.00
SM 18:1;2O/16:0	10	0.92	SM 18:1;2O/16:0	10	0.99
Cer 18:1;2O/16:0	10	0.99	Cer 18:1;2O/16:0	10	1.00
LPC 16:0	10	0.00	LPC 16:0	10	0.96
PC 16:0_16:0	10	0.07	PC 16:0_16:0	10	0.85
PC 16:0_18:1	10	0.53	PC 16:0_18:1	10	0.99
PC 18:0_18:0	10	0.92	PC 18:0_18:0	10	0.99
PC 18:1_18:1	10	0.83	PC 18:1_18:1	10	0.20
PE 18:1_18:1	10	0.01	PE 18:1_18:1	10	0.06
PI 18:1_18:1	10	0.52	PI 18:1_18:1	10	0.62
SM 18:1;2O/16:0	10	0.38	SM 18:1;2O/16:0	10	0.99
Cer 18:1;2O/16:0	10	0.83	Cer 18:1;2O/16:0	10	0.99
LPC 16:0	10	0.00	LPC 16:0	10	1.00
PC 16:0_16:0	10	0.97	PC 16:0_16:0	10	0.97
PC 16:0_18:1	10	0.74	PC 16:0_18:1	10	1.00
PC 18:0_18:0	10	0.28	PC 18:0_18:0	10	0.99
PC 18:1_18:1	10	0.27	PC 18:1_18:1	10	0.99
PE 18:1_18:1	10	0.84	PE 18:1_18:1	10	1.00
PI 18:1_18:1	10	0.94	PI 18:1_18:1	10	0.99
SM 18:1;2O/16:0	10	0.91	SM 18:1;2O/16:0	10	0.09
Cer 18:1;2O/16:0	10	0.99	Cer 18:1;2O/16:0	10	1.00
LPC 16:0	10	0.93	LPC 16:0	10	0.09

PC 16:0_16:0	10	0.57	PC 16:0_16:0	10	0.14
PC 16:0_18:1	10	0.77	PC 16:0_18:1	10	1.00
PC 18:0_18:0	10	0.94	PC 18:0_18:0	10	0.99
PC 18:1_18:1	10	0.61	PC 18:1_18:1	10	0.79
PE 18:1_18:1	10	0.02	PE 18:1_18:1	10	0.14
PI 18:1_18:1	10	0.94	PI 18:1_18:1	10	0.99
SM 18:1;2O/16:0	10	0.00	SM 18:1;2O/16:0	10	0.99
DG 18 1_18 1	10	0.98	DG 18 1_18 1	10	1.00
LPC 16:0	10	0.28	LPC 16:0	10	0.09
LPC 18 0	10	0.11	LPC 18 0	10	0.86
PC 16:0_18:1	10	0.18	PC 16:0_18:1	10	1.00
PC 18:0_18:0	10	0.44	PC 18:0_18:0	10	0.21
PC 18:1_18:1	10	0.56	PC 18:1_18:1	10	0.47
PE 18:1_18:1	10	0.73	PE 18:1_18:1	10	0.99
SM 18:1;2O/16:0	10	0.40	SM 18:1;2O/16:0	10	1.00
SM 18 1;2O 18 0	10	0.84	SM 18 1;2O 18 0	10	0.96
DG 18 1_18 1	10	0.30	DG 18 1_18 1	10	1.00
LPC 16:0	10	0.00	LPC 16:0	10	1.00
LPC 18 0	10	0.64	LPC 18 0	10	0.88
PC 16:0_18:1	10	0.93	PC 16:0_18:1	10	1.00
PC 18:0_18:0	10	0.44	PC 18:0_18:0	10	0.21
PC 18:1_18:1	10	0.92	PC 18:1_18:1	10	1.00
PE 18:1_18:1	10	0.98	PE 18:1_18:1	10	1.00
SM 18:1;2O/16:0	10	0.91	SM 18:1;2O/16:0	10	1.00
SM 18 1;2O 18 0	10	0.84	SM 18 1;2O 18 0	10	0.96
DG 18 1_18 1	10	1.00	DG 18 1_18 1	10	0.99
LPC 16:0	10	0.98	LPC 16:0	10	0.99
LPC 18 0	10	0.36	LPC 18 0	10	0.02
PC 16:0_18:1	10	0.99	PC 16:0_18:1	10	1.00
PC 18:0_18:0	10	0.11	PC 18:0_18:0	10	0.81
PC 18:1_18:1	10	0.69	PC 18:1_18:1	10	1.00
PE 18:1_18:1	10	0.29	PE 18:1_18:1	10	1.00
SM 18:1;2O/16:0	10	0.03	SM 18:1;2O/16:0	10	0.03
SM 18 1;2O 18 0	10	0.94	SM 18 1;2O 18 0	10	1.00
DG 18 1_18 1	10	0.56	DG 18 1_18 1	10	1.00
LPC 16:0	10	0.93	LPC 16:0	10	1.00
LPC 18 0	10	0.68	LPC 18 0	10	0.98
PC 16:0_18:1	10	0.02	PC 16:0_18:1	10	0.96
PC 18:0_18:0	10	0.43	PC 18:0_18:0	10	0.52
PC 18:1_18:1	10	0.81	PC 18:1_18:1	10	0.37
PE 18:1_18:1	10	0.48	PE 18:1_18:1	10	1.00
SM 18:1;2O/16:0	10	0.00	SM 18:1;2O/16:0	10	0.83
SM 18 1;2O 18 0	10	1.00	SM 18 1;2O 18 0	10	1.00
DG 18 1_18 1	10	1.00	DG 18 1_18 1	10	1.00

LPC 16:0	10	0.92	LPC 16:0	10	0.00
LPC 18 0	10	0.00	LPC 18 0	10	0.96
PC 16:0_18:1	10	0.99	PC 16:0_18:1	10	1.00
PC 18:0_18:0	10	0.44	PC 18:0_18:0	10	0.81
PC 18:1_18:1	10	0.75	PC 18:1_18:1	10	0.98
PE 18:1_18:1	10	0.97	PE 18:1_18:1	10	1.00
SM 18:1;2O/16:0	10	0.90	SM 18:1;2O/16:0	10	1.00
SM 18 1;2O 18 0	10	0.89	SM 18 1;2O 18 0	10	1.00
DG 18 1_18 1	10	0.22	DG 18 1_18 1	10	1.00
LPC 16:0	10	0.20	LPC 16:0	10	1.00
LPC 18 0	10	0.78	LPC 18 0	10	0.97
PC 16:0_18:1	10	0.10	PC 16:0_18:1	10	1.00
PC 18:0_18:0	10	0.44	PC 18:0_18:0	10	0.21
PC 18:1_18:1	10	0.75	PC 18:1_18:1	10	0.98
PE 18:1_18:1	10	1.00	PE 18:1_18:1	10	0.93
SM 18:1;2O/16:0	10	0.72	SM 18:1;2O/16:0	10	1.00
SM 18 1;2O 18 0	10	0.16	SM 18 1;2O 18 0	10	0.59
DG 18 1_18 1	10	0.99	DG 18 1_18 1	10	1.00
LPC 16:0	10	0.97	LPC 16:0	10	0.99
LPC 18 0	10	0.57	LPC 18 0	10	0.98
PC 16:0_18:1	10	0.62	PC 16:0_18:1	10	0.91
PC 18:0_18:0	10	0.13	PC 18:0_18:0	10	0.82
PC 18:1_18:1	10	0.75	PC 18:1_18:1	10	0.98
PE 18:1_18:1	10	1.00	PE 18:1_18:1	10	1.00
SM 18:1;2O/16:0	10	0.00	SM 18:1;2O/16:0	10	1.00
SM 18 1;2O 18 0	10	0.00	SM 18 1;2O 18 0	10	1.00
DG 18 1_18 1	10	1.00	DG 18 1_18 1	10	0.16
LPC 16:0	10	0.44	LPC 16:0	10	0.96
LPC 18 0	10	0.13	LPC 18 0	10	0.98
PC 16:0_18:1	10	0.89	PC 16:0_18:1	10	0.98
PC 18:0_18:0	10	0.43	PC 18:0_18:0	10	0.68
PC 18:1_18:1	10	0.99	PC 18:1_18:1	10	1.00
PE 18:1_18:1	10	0.96	PE 18:1_18:1	10	0.99
SM 18:1;2O/16:0	10	0.91	SM 18:1;2O/16:0	10	1.00
SM 18 1;2O 18 0	10	0.00	SM 18 1;2O 18 0	10	1.00
DG 18 1_18 1	10	1.00	DG 18 1_18 1	10	0.16
LPC 16:0	10	0.93	LPC 16:0	10	1.00
LPC 18 0	10	0.72	LPC 18 0	10	0.98
PC 16:0_18:1	10	0.99	PC 16:0_18:1	10	1.00
PC 18:0_18:0	10	0.14	PC 18:0_18:0	10	0.83
PC 18:1_18:1	10	1.00	PC 18:1_18:1	10	0.99
PE 18:1_18:1	10	0.91	PE 18:1_18:1	10	0.53
SM 18:1;2O/16:0	10	0.35	SM 18:1;2O/16:0	10	0.06
SM 18 1;2O 18 0	10	0.84	SM 18 1;2O 18 0	10	0.96

DG 18 1_18 1	10	0.98	DG 18 1_18 1	10	1.00
LPC 16:0	10	0.51	LPC 16:0	10	1.00
LPC 18 0	10	0.20	LPC 18 0	10	0.97
PC 16:0_18:1	10	0.82	PC 16:0_18:1	10	1.00
PC 18:0_18:0	10	0.43	PC 18:0_18:0	10	0.78
PC 18:1_18:1	10	0.02	PC 18:1_18:1	10	1.00
PE 18:1_18:1	10	0.12	PE 18:1_18:1	10	0.99
SM 18:1;2O/16:0	10	0.00	SM 18:1;2O/16:0	10	0.99
SM 18 1;2O 18 0	10	0.94	SM 18 1;2O 18 0	10	1.00
Cer 18:1;2O/16:0	75	0.00	Cer 18:1;2O/16:0	75	0.99
LPC 16:0	75	1.00	LPC 16:0	75	1.00
PC 16:0_16:0	75	0.01	PC 16:0_16:0	75	0.99
PC 16:0_18:1	75	1.00	PC 16:0_18:1	75	1.00
PC 18:0_18:0	75	0.99	PC 18:0_18:0	75	1.00
PC 18:1_18:1	75	0.99	PC 18:1_18:1	75	0.08
PE 18:1_18:1	75	1.00	PE 18:1_18:1	75	1.00
PI 18:1_18:1	75	0.00	PI 18:1_18:1	75	1.00
SM 18:1;2O/16:0	75	0.02	SM 18:1;2O/16:0	75	1.00
Cer 18:1;2O/16:0	75	0.01	Cer 18:1;2O/16:0	75	1.00
LPC 16:0	75	1.00	LPC 16:0	75	1.00
PC 16:0_16:0	75	1.00	PC 16:0_16:0	75	1.00
PC 16:0_18:1	75	1.00	PC 16:0_18:1	75	0.99
PC 18:0_18:0	75	0.00	PC 18:0_18:0	75	1.00
PC 18:1_18:1	75	1.00	PC 18:1_18:1	75	0.99
PE 18:1_18:1	75	1.00	PE 18:1_18:1	75	1.00
PI 18:1_18:1	75	1.00	PI 18:1_18:1	75	1.00
SM 18:1;2O/16:0	75	1.00	SM 18:1;2O/16:0	75	1.00
Cer 18:1;2O/16:0	75	0.03	Cer 18:1;2O/16:0	75	1.00
LPC 16:0	75	1.00	LPC 16:0	75	1.00
PC 16:0_16:0	75	1.00	PC 16:0_16:0	75	0.99
PC 16:0_18:1	75	1.00	PC 16:0_18:1	75	1.00
PC 18:0_18:0	75	1.00	PC 18:0_18:0	75	1.00
PC 18:1_18:1	75	0.99	PC 18:1_18:1	75	0.99
PE 18:1_18:1	75	1.00	PE 18:1_18:1	75	1.00
PI 18:1_18:1	75	0.94	PI 18:1_18:1	75	1.00
SM 18:1;2O/16:0	75	1.00	SM 18:1;2O/16:0	75	1.00
Cer 18:1;2O/16:0	75	1.00	Cer 18:1;2O/16:0	75	1.00
LPC 16:0	75	0.99	LPC 16:0	75	1.00
PC 16:0_16:0	75	0.00	PC 16:0_16:0	75	0.99
PC 16:0_18:1	75	1.00	PC 16:0_18:1	75	1.00
PC 18:0_18:0	75	1.00	PC 18:0_18:0	75	1.00
PC 18:1_18:1	75	1.00	PC 18:1_18:1	75	0.87
PE 18:1_18:1	75	0.00	PE 18:1_18:1	75	1.00
PI 18:1_18:1	75	1.00	PI 18:1_18:1	75	1.00

SM 18:1;2O/16:0	75	1.00	SM 18:1;2O/16:0	75	1.00
Cer 18:1;2O/16:0	75	1.00	Cer 18:1;2O/16:0	75	1.00
LPC 16:0	75	1.00	LPC 16:0	75	1.00
PC 16:0_16:0	75	1.00	PC 16:0_16:0	75	0.99
PC 16:0_18:1	75	1.00	PC 16:0_18:1	75	0.87
PC 18:0_18:0	75	1.00	PC 18:0_18:0	75	1.00
PC 18:1_18:1	75	0.79	PC 18:1_18:1	75	0.87
PE 18:1_18:1	75	0.06	PE 18:1_18:1	75	1.00
PI 18:1_18:1	75	1.00	PI 18:1_18:1	75	1.00
SM 18:1;2O/16:0	75	0.00	SM 18:1;2O/16:0	75	1.00
Cer 18:1;2O/16:0	75	0.99	Cer 18:1;2O/16:0	75	1.00
LPC 16:0	75	1.00	LPC 16:0	75	1.00
PC 16:0_16:0	75	1.00	PC 16:0_16:0	75	0.97
PC 16:0_18:1	75	1.00	PC 16:0_18:1	75	1.00
PC 18:0_18:0	75	1.00	PC 18:0_18:0	75	0.99
PC 18:1_18:1	75	0.99	PC 18:1_18:1	75	0.99
PE 18:1_18:1	75	0.01	PE 18:1_18:1	75	0.99
PI 18:1_18:1	75	0.08	PI 18:1_18:1	75	1.00
SM 18:1;2O/16:0	75	0.01	SM 18:1;2O/16:0	75	1.00
Cer 18:1;2O/16:0	75	1.00	Cer 18:1;2O/16:0	75	1.00
LPC 16:0	75	0.01	LPC 16:0	75	1.00
PC 16:0_16:0	75	0.00	PC 16:0_16:0	75	0.99
PC 16:0_18:1	75	0.34	PC 16:0_18:1	75	1.00
PC 18:0_18:0	75	1.00	PC 18:0_18:0	75	1.00
PC 18:1_18:1	75	0.99	PC 18:1_18:1	75	0.87
PE 18:1_18:1	75	0.03	PE 18:1_18:1	75	1.00
PI 18:1_18:1	75	0.00	PI 18:1_18:1	75	1.00
SM 18:1;2O/16:0	75	1.00	SM 18:1;2O/16:0	75	1.00
Cer 18:1;2O/16:0	75	1.00	Cer 18:1;2O/16:0	75	1.00
LPC 16:0	75	0.99	LPC 16:0	75	0.08
PC 16:0_16:0	75	1.00	PC 16:0_16:0	75	0.99
PC 16:0_18:1	75	1.00	PC 16:0_18:1	75	0.99
PC 18:0_18:0	75	0.91	PC 18:0_18:0	75	1.00
PC 18:1_18:1	75	0.99	PC 18:1_18:1	75	0.98
PE 18:1_18:1	75	0.99	PE 18:1_18:1	75	0.06
PI 18:1_18:1	75	1.00	PI 18:1_18:1	75	1.00
SM 18:1;2O/16:0	75	0.93	SM 18:1;2O/16:0	75	1.00
Cer 18:1;2O/16:0	75	0.99	Cer 18:1;2O/16:0	75	0.91
LPC 16:0	75	0.99	LPC 16:0	75	1.00
PC 16:0_16:0	75	0.99	PC 16:0_16:0	75	0.99
PC 16:0_18:1	75	1.00	PC 16:0_18:1	75	1.00
PC 18:0_18:0	75	1.00	PC 18:0_18:0	75	0.96
PC 18:1_18:1	75	0.99	PC 18:1_18:1	75	0.99
PE 18:1_18:1	75	0.98	PE 18:1_18:1	75	1.00

PI 18:1_18:1	75	0.95	PI 18:1_18:1	75	0.15
SM 18:1;2O/16:0	75	1.00	SM 18:1;2O/16:0	75	1.00
Cer 18:1;2O/16:0	75	0.99	Cer 18:1;2O/16:0	75	1.00
LPC 16:0	75	1.00	LPC 16:0	75	0.13
PC 16:0_16:0	75	1.00	PC 16:0_16:0	75	0.99
PC 16:0_18:1	75	1.00	PC 16:0_18:1	75	0.99
PC 18:0_18:0	75	1.00	PC 18:0_18:0	75	1.00
PC 18:1_18:1	75	0.99	PC 18:1_18:1	75	0.99
PE 18:1_18:1	75	0.99	PE 18:1_18:1	75	1.00
PI 18:1_18:1	75	1.00	PI 18:1_18:1	75	1.00
SM 18:1;2O/16:0	75	1.00	SM 18:1;2O/16:0	75	1.00
DG 18 1_18 1	75	1.00	DG 18 1_18 1	75	1.00
LPC 16:0	75	0.04	LPC 16:0	75	1.00
LPC 18 0	75	0.98	LPC 18 0	75	0.10
PC 16:0_18:1	75	1.00	PC 16:0_18:1	75	1.00
PC 18:0_18:0	75	0.80	PC 18:0_18:0	75	0.98
PC 18:1_18:1	75	1.00	PC 18:1_18:1	75	0.59
PE 18:1_18:1	75	0.15	PE 18:1_18:1	75	1.00
SM 18:1;2O/16:0	75	1.00	SM 18:1;2O/16:0	75	1.00
SM 18 1;2O 18 0	75	1.00	SM 18 1;2O 18 0	75	1.00
DG 18 1_18 1	75	1.00	DG 18 1_18 1	75	1.00
LPC 16:0	75	1.00	LPC 16:0	75	1.00
LPC 18 0	75	0.01	LPC 18 0	75	1.00
PC 16:0_18:1	75	0.00	PC 16:0_18:1	75	1.00
PC 18:0_18:0	75	0.80	PC 18:0_18:0	75	0.98
PC 18:1_18:1	75	1.00	PC 18:1_18:1	75	1.00
PE 18:1_18:1	75	1.00	PE 18:1_18:1	75	0.17
SM 18:1;2O/16:0	75	1.00	SM 18:1;2O/16:0	75	0.99
SM 18 1;2O 18 0	75	0.91	SM 18 1;2O 18 0	75	1.00
DG 18 1_18 1	75	1.00	DG 18 1_18 1	75	0.59
LPC 16:0	75	0.09	LPC 16:0	75	0.97
LPC 18 0	75	0.98	LPC 18 0	75	1.00
PC 16:0_18:1	75	1.00	PC 16:0_18:1	75	1.00
PC 18:0_18:0	75	0.00	PC 18:0_18:0	75	0.97
PC 18:1_18:1	75	0.83	PC 18:1_18:1	75	0.59
PE 18:1_18:1	75	0.99	PE 18:1_18:1	75	1.00
SM 18:1;2O/16:0	75	1.00	SM 18:1;2O/16:0	75	1.00
SM 18 1;2O 18 0	75	0.06	SM 18 1;2O 18 0	75	1.00
DG 18 1_18 1	75	1.00	DG 18 1_18 1	75	1.00
LPC 16:0	75	1.00	LPC 16:0	75	1.00
LPC 18 0	75	0.98	LPC 18 0	75	0.10
PC 16:0_18:1	75	1.00	PC 16:0_18:1	75	1.00
PC 18:0_18:0	75	0.77	PC 18:0_18:0	75	0.79
PC 18:1_18:1	75	1.00	PC 18:1_18:1	75	1.00

PE 18:1_18:1	75	0.99	PE 18:1_18:1	75	0.98
SM 18:1;2O/16:0	75	1.00	SM 18:1;2O/16:0	75	1.00
SM 18 1;2O 18 0	75	0.03	SM 18 1;2O 18 0	75	1.00
DG 18 1_18 1	75	1.00	DG 18 1_18 1	75	1.00
LPC 16:0	75	0.99	LPC 16:0	75	1.00
LPC 18 0	75	0.98	LPC 18 0	75	1.00
PC 16:0_18:1	75	1.00	PC 16:0_18:1	75	1.00
PC 18:0_18:0	75	0.79	PC 18:0_18:0	75	0.97
PC 18:1_18:1	75	1.00	PC 18:1_18:1	75	0.99
PE 18:1_18:1	75	1.00	PE 18:1_18:1	75	1.00
SM 18:1;2O/16:0	75	1.00	SM 18:1;2O/16:0	75	1.00
SM 18 1;2O 18 0	75	1.00	SM 18 1;2O 18 0	75	1.00
DG 18 1_18 1	75	1.00	DG 18 1_18 1	75	1.00
LPC 16:0	75	1.00	LPC 16:0	75	0.52
LPC 18 0	75	0.98	LPC 18 0	75	1.00
PC 16:0_18:1	75	0.00	PC 16:0_18:1	75	1.00
PC 18:0_18:0	75	0.01	PC 18:0_18:0	75	0.98
PC 18:1_18:1	75	0.83	PC 18:1_18:1	75	1.00
PE 18:1_18:1	75	0.05	PE 18:1_18:1	75	1.00
SM 18:1;2O/16:0	75	1.00	SM 18:1;2O/16:0	75	0.99
SM 18 1;2O 18 0	75	1.00	SM 18 1;2O 18 0	75	1.00
DG 18 1_18 1	75	1.00	DG 18 1_18 1	75	0.99
LPC 16:0	75	0.97	LPC 16:0	75	1.00
LPC 18 0	75	0.00	LPC 18 0	75	0.99
PC 16:0_18:1	75	0.55	PC 16:0_18:1	75	0.03
PC 18:0_18:0	75	0.79	PC 18:0_18:0	75	0.98
PC 18:1_18:1	75	0.83	PC 18:1_18:1	75	1.00
PE 18:1_18:1	75	0.99	PE 18:1_18:1	75	1.00
SM 18:1;2O/16:0	75	1.00	SM 18:1;2O/16:0	75	1.00
SM 18 1;2O 18 0	75	1.00	SM 18 1;2O 18 0	75	1.00
DG 18 1_18 1	75	1.00	DG 18 1_18 1	75	1.00
LPC 16:0	75	0.00	LPC 16:0	75	0.00
LPC 18 0	75	0.98	LPC 18 0	75	0.17
PC 16:0_18:1	75	0.03	PC 16:0_18:1	75	1.00
PC 18:0_18:0	75	0.81	PC 18:0_18:0	75	0.97
PC 18:1_18:1	75	1.00	PC 18:1_18:1	75	1.00
PE 18:1_18:1	75	0.99	PE 18:1_18:1	75	1.00
SM 18:1;2O/16:0	75	0.02	SM 18:1;2O/16:0	75	1.00
SM 18 1;2O 18 0	75	0.94	SM 18 1;2O 18 0	75	1.00
DG 18 1_18 1	75	0.99	DG 18 1_18 1	75	1.00
LPC 16:0	75	1.00	LPC 16:0	75	0.10
LPC 18 0	75	0.98	LPC 18 0	75	1.00
PC 16:0_18:1	75	1.00	PC 16:0_18:1	75	0.73
PC 18:0_18:0	75	0.01	PC 18:0_18:0	75	0.55

PC 18:1_18:1	75	1.00	PC 18:1_18:1	75	0.81
PE 18:1_18:1	75	0.99	PE 18:1_18:1	75	1.00
SM 18:1;2O/16:0	75	0.92	SM 18:1;2O/16:0	75	1.00
SM 18 1;2O 18 0	75	1.00	SM 18 1;2O 18 0	75	1.00
DG 18 1_18 1	75	1.00	DG 18 1_18 1	75	0.51
LPC 16:0	75	1.00	LPC 16:0	75	1.00
LPC 18 0	75	0.34	LPC 18 0	75	0.99
PC 16:0_18:1	75	1.00	PC 16:0_18:1	75	0.19
PC 18:0_18:0	75	0.01	PC 18:0_18:0	75	0.97
PC 18:1_18:1	75	0.56	PC 18:1_18:1	75	1.00
PE 18:1_18:1	75	1.00	PE 18:1_18:1	75	1.00
SM 18:1;2O/16:0	75	1.00	SM 18:1;2O/16:0	75	0.25
SM 18 1;2O 18 0	75	1.00	SM 18 1;2O 18 0	75	1.00
Cer 18:1;2O/16:0	150	0.98	Cer 18:1;2O/16:0	150	1.00
LPC 16:0	150	0.95	LPC 16:0	150	1.00
PC 16:0_16:0	150	0.73	PC 16:0_16:0	150	0.99
PC 16:0_18:1	150	0.97	PC 16:0_18:1	150	0.99
PC 18:0_18:0	150	1.00	PC 18:0_18:0	150	1.00
PC 18:1_18:1	150	1.00	PC 18:1_18:1	150	0.99
PE 18:1_18:1	150	1.00	PE 18:1_18:1	150	1.00
PI 18:1_18:1	150	0.00	PI 18:1_18:1	150	1.00
SM 18:1;2O/16:0	150	0.99	SM 18:1;2O/16:0	150	1.00
Cer 18:1;2O/16:0	150	1.00	Cer 18:1;2O/16:0	150	0.99
LPC 16:0	150	0.00	LPC 16:0	150	1.00
PC 16:0_16:0	150	1.00	PC 16:0_16:0	150	0.99
PC 16:0_18:1	150	0.85	PC 16:0_18:1	150	1.00
PC 18:0_18:0	150	1.00	PC 18:0_18:0	150	1.00
PC 18:1_18:1	150	1.00	PC 18:1_18:1	150	0.99
PE 18:1_18:1	150	1.00	PE 18:1_18:1	150	1.00
PI 18:1_18:1	150	0.99	PI 18:1_18:1	150	1.00
SM 18:1;2O/16:0	150	1.00	SM 18:1;2O/16:0	150	1.00
Cer 18:1;2O/16:0	150	1.00	Cer 18:1;2O/16:0	150	0.99
LPC 16:0	150	1.00	LPC 16:0	150	1.00
PC 16:0_16:0	150	0.99	PC 16:0_16:0	150	0.99
PC 16:0_18:1	150	0.98	PC 16:0_18:1	150	0.99
PC 18:0_18:0	150	0.79	PC 18:0_18:0	150	1.00
PC 18:1_18:1	150	0.00	PC 18:1_18:1	150	0.99
PE 18:1_18:1	150	1.00	PE 18:1_18:1	150	1.00
PI 18:1_18:1	150	0.91	PI 18:1_18:1	150	1.00
SM 18:1;2O/16:0	150	1.00	SM 18:1;2O/16:0	150	1.00
Cer 18:1;2O/16:0	150	1.00	Cer 18:1;2O/16:0	150	1.00
LPC 16:0	150	0.99	LPC 16:0	150	0.99
PC 16:0_16:0	150	1.00	PC 16:0_16:0	150	0.99
PC 16:0_18:1	150	0.84	PC 16:0_18:1	150	0.91

PC 18:0_18:0	150	1.00	PC 18:0_18:0	150	0.96
PC 18:1_18:1	150	0.04	PC 18:1_18:1	150	0.99
PE 18:1_18:1	150	0.54	PE 18:1_18:1	150	0.00
PI 18:1_18:1	150	0.99	PI 18:1_18:1	150	1.00
SM 18:1;2O/16:0	150	0.00	SM 18:1;2O/16:0	150	0.99
Cer 18:1;2O/16:0	150	0.98	Cer 18:1;2O/16:0	150	0.99
LPC 16:0	150	0.99	LPC 16:0	150	1.00
PC 16:0_16:0	150	1.00	PC 16:0_16:0	150	0.99
PC 16:0_18:1	150	1.00	PC 16:0_18:1	150	1.00
PC 18:0_18:0	150	1.00	PC 18:0_18:0	150	1.00
PC 18:1_18:1	150	0.99	PC 18:1_18:1	150	0.99
PE 18:1_18:1	150	1.00	PE 18:1_18:1	150	1.00
PI 18:1_18:1	150	0.96	PI 18:1_18:1	150	1.00
SM 18:1;2O/16:0	150	1.00	SM 18:1;2O/16:0	150	1.00
Cer 18:1;2O/16:0	150	0.92	Cer 18:1;2O/16:0	150	1.00
LPC 16:0	150	1.00	LPC 16:0	150	1.00
PC 16:0_16:0	150	0.99	PC 16:0_16:0	150	0.99
PC 16:0_18:1	150	1.00	PC 16:0_18:1	150	0.99
PC 18:0_18:0	150	1.00	PC 18:0_18:0	150	1.00
PC 18:1_18:1	150	0.99	PC 18:1_18:1	150	0.99
PE 18:1_18:1	150	1.00	PE 18:1_18:1	150	1.00
PI 18:1_18:1	150	0.97	PI 18:1_18:1	150	1.00
SM 18:1;2O/16:0	150	0.94	SM 18:1;2O/16:0	150	1.00
Cer 18:1;2O/16:0	150	0.89	Cer 18:1;2O/16:0	150	1.00
LPC 16:0	150	0.99	LPC 16:0	150	0.94
PC 16:0_16:0	150	0.95	PC 16:0_16:0	150	0.99
PC 16:0_18:1	150	0.39	PC 16:0_18:1	150	0.99
PC 18:0_18:0	150	0.93	PC 18:0_18:0	150	1.00
PC 18:1_18:1	150	0.86	PC 18:1_18:1	150	0.99
PE 18:1_18:1	150	1.00	PE 18:1_18:1	150	1.00
PI 18:1_18:1	150	1.00	PI 18:1_18:1	150	0.99
SM 18:1;2O/16:0	150	0.99	SM 18:1;2O/16:0	150	1.00
Cer 18:1;2O/16:0	150	0.99	Cer 18:1;2O/16:0	150	1.00
LPC 16:0	150	0.86	LPC 16:0	150	1.00
PC 16:0_16:0	150	0.99	PC 16:0_16:0	150	0.99
PC 16:0_18:1	150	0.97	PC 16:0_18:1	150	0.99
PC 18:0_18:0	150	1.00	PC 18:0_18:0	150	0.99
PC 18:1_18:1	150	0.99	PC 18:1_18:1	150	0.99
PE 18:1_18:1	150	1.00	PE 18:1_18:1	150	1.00
PI 18:1_18:1	150	0.12	PI 18:1_18:1	150	1.00
SM 18:1;2O/16:0	150	1.00	SM 18:1;2O/16:0	150	0.14
Cer 18:1;2O/16:0	150	0.93	Cer 18:1;2O/16:0	150	1.00
LPC 16:0	150	0.99	LPC 16:0	150	1.00
PC 16:0_16:0	150	1.00	PC 16:0_16:0	150	0.99

PC 16:0_18:1	150	1.00	PC 16:0_18:1	150	0.99
PC 18:0_18:0	150	1.00	PC 18:0_18:0	150	1.00
PC 18:1_18:1	150	0.99	PC 18:1_18:1	150	0.99
PE 18:1_18:1	150	1.00	PE 18:1_18:1	150	1.00
PI 18:1_18:1	150	1.00	PI 18:1_18:1	150	1.00
SM 18:1;2O/16:0	150	0.14	SM 18:1;2O/16:0	150	1.00
Cer 18:1;2O/16:0	150	0.86	Cer 18:1;2O/16:0	150	0.97
LPC 16:0	150	1.00	LPC 16:0	150	0.04
PC 16:0_16:0	150	1.00	PC 16:0_16:0	150	0.99
PC 16:0_18:1	150	0.97	PC 16:0_18:1	150	0.09
PC 18:0_18:0	150	0.79	PC 18:0_18:0	150	1.00
PC 18:1_18:1	150	0.99	PC 18:1_18:1	150	0.99
PE 18:1_18:1	150	1.00	PE 18:1_18:1	150	1.00
PI 18:1_18:1	150	0.82	PI 18:1_18:1	150	1.00
SM 18:1;2O/16:0	150	1.00	SM 18:1;2O/16:0	150	1.00
DG 18 1_18 1	150	1.00	DG 18 1_18 1	150	1.00
LPC 16:0	150	1.00	LPC 16:0	150	1.00
LPC 18 0	150	0.99	LPC 18 0	150	1.00
PC 16:0_18:1	150	0.73	PC 16:0_18:1	150	1.00
PC 18:0_18:0	150	0.03	PC 18:0_18:0	150	1.00
PC 18:1_18:1	150	0.98	PC 18:1_18:1	150	1.00
PE 18:1_18:1	150	0.99	PE 18:1_18:1	150	1.00
SM 18:1;2O/16:0	150	1.00	SM 18:1;2O/16:0	150	1.00
SM 18 1;2O 18 0	150	0.03	SM 18 1;2O 18 0	150	1.00
DG 18 1_18 1	150	0.98	DG 18 1_18 1	150	0.20
LPC 16:0	150	1.00	LPC 16:0	150	1.00
LPC 18 0	150	0.99	LPC 18 0	150	1.00
PC 16:0_18:1	150	1.00	PC 16:0_18:1	150	1.00
PC 18:0_18:0	150	0.89	PC 18:0_18:0	150	0.99
PC 18:1_18:1	150	1.00	PC 18:1_18:1	150	0.99
PE 18:1_18:1	150	1.00	PE 18:1_18:1	150	1.00
SM 18:1;2O/16:0	150	1.00	SM 18:1;2O/16:0	150	1.00
SM 18 1;2O 18 0	150	0.83	SM 18 1;2O 18 0	150	1.00
DG 18 1_18 1	150	0.78	DG 18 1_18 1	150	0.99
LPC 16:0	150	1.00	LPC 16:0	150	1.00
LPC 18 0	150	0.99	LPC 18 0	150	1.00
PC 16:0_18:1	150	0.83	PC 16:0_18:1	150	1.00
PC 18:0_18:0	150	0.20	PC 18:0_18:0	150	0.99
PC 18:1_18:1	150	0.06	PC 18:1_18:1	150	1.00
PE 18:1_18:1	150	0.94	PE 18:1_18:1	150	1.00
SM 18:1;2O/16:0	150	0.83	SM 18:1;2O/16:0	150	0.59
SM 18 1;2O 18 0	150	1.00	SM 18 1;2O 18 0	150	1.00
DG 18 1_18 1	150	0.99	DG 18 1_18 1	150	0.99
LPC 16:0	150	1.00	LPC 16:0	150	1.00

LPC 18 0	150	0.97	LPC 18 0	150	1.00
PC 16:0_18:1	150	0.98	PC 16:0_18:1	150	1.00
PC 18:0_18:0	150	0.88	PC 18:0_18:0	150	0.99
PC 18:1_18:1	150	0.43	PC 18:1_18:1	150	1.00
PE 18:1_18:1	150	1.00	PE 18:1_18:1	150	1.00
SM 18:1;2O/16:0	150	0.00	SM 18:1;2O/16:0	150	1.00
SM 18 1;2O 18 0	150	1.00	SM 18 1;2O 18 0	150	1.00
DG 18 1_18 1	150	0.99	DG 18 1_18 1	150	1.00
LPC 16:0	150	0.89	LPC 16:0	150	0.04
LPC 18 0	150	0.89	LPC 18 0	150	1.00
PC 16:0_18:1	150	0.05	PC 16:0_18:1	150	1.00
PC 18:0_18:0	150	0.89	PC 18:0_18:0	150	0.99
PC 18:1_18:1	150	0.58	PC 18:1_18:1	150	1.00
PE 18:1_18:1	150	0.93	PE 18:1_18:1	150	0.99
SM 18:1;2O/16:0	150	1.00	SM 18:1;2O/16:0	150	0.36
SM 18 1;2O 18 0	150	1.00	SM 18 1;2O 18 0	150	1.00
DG 18 1_18 1	150	0.96	DG 18 1_18 1	150	1.00
LPC 16:0	150	1.00	LPC 16:0	150	1.00
LPC 18 0	150	0.01	LPC 18 0	150	0.99
PC 16:0_18:1	150	0.00	PC 16:0_18:1	150	1.00
PC 18:0_18:0	150	0.88	PC 18:0_18:0	150	1.00
PC 18:1_18:1	150	1.00	PC 18:1_18:1	150	1.00
PE 18:1_18:1	150	0.18	PE 18:1_18:1	150	0.99
SM 18:1;2O/16:0	150	0.81	SM 18:1;2O/16:0	150	1.00
SM 18 1;2O 18 0	150	1.00	SM 18 1;2O 18 0	150	1.00
DG 18 1_18 1	150	1.00	DG 18 1_18 1	150	1.00
LPC 16:0	150	0.09	LPC 16:0	150	1.00
LPC 18 0	150	0.87	LPC 18 0	150	1.00
PC 16:0_18:1	150	1.00	PC 16:0_18:1	150	1.00
PC 18:0_18:0	150	0.02	PC 18:0_18:0	150	1.00
PC 18:1_18:1	150	1.00	PC 18:1_18:1	150	1.00
PE 18:1_18:1	150	0.96	PE 18:1_18:1	150	1.00
SM 18:1;2O/16:0	150	1.00	SM 18:1;2O/16:0	150	1.00
SM 18 1;2O 18 0	150	1.00	SM 18 1;2O 18 0	150	0.99
DG 18 1_18 1	150	0.06	DG 18 1_18 1	150	1.00
LPC 16:0	150	0.96	LPC 16:0	150	0.99
LPC 18 0	150	0.96	LPC 18 0	150	0.09
PC 16:0_18:1	150	1.00	PC 16:0_18:1	150	1.00
PC 18:0_18:0	150	0.88	PC 18:0_18:0	150	1.00
PC 18:1_18:1	150	0.27	PC 18:1_18:1	150	1.00
PE 18:1_18:1	150	0.99	PE 18:1_18:1	150	1.00
SM 18:1;2O/16:0	150	1.00	SM 18:1;2O/16:0	150	1.00
SM 18 1;2O 18 0	150	1.00	SM 18 1;2O 18 0	150	1.00
DG 18 1_18 1	150	1.00	DG 18 1_18 1	150	0.99

LPC 16:0	150	1.00	LPC 16:0	150	1.00
LPC 18 0	150	0.14	LPC 18 0	150	1.00
PC 16:0_18:1	150	1.00	PC 16:0_18:1	150	1.00
PC 18:0_18:0	150	0.20	PC 18:0_18:0	150	0.99
PC 18:1_18:1	150	1.00	PC 18:1_18:1	150	1.00
PE 18:1_18:1	150	1.00	PE 18:1_18:1	150	0.18
SM 18:1;2O/16:0	150	1.00	SM 18:1;2O/16:0	150	1.00
SM 18 1;2O 18 0	150	1.00	SM 18 1;2O 18 0	150	1.00
DG 18 1_18 1	150	1.00	DG 18 1_18 1	150	1.00
LPC 16:0	150	1.00	LPC 16:0	150	1.00
LPC 18 0	150	0.99	LPC 18 0	150	1.00
PC 16:0_18:1	150	1.00	PC 16:0_18:1	150	1.00
PC 18:0_18:0	150	0.88	PC 18:0_18:0	150	0.99
PC 18:1_18:1	150	1.00	PC 18:1_18:1	150	1.00
PE 18:1_18:1	150	0.39	PE 18:1_18:1	150	1.00
SM 18:1;2O/16:0	150	1.00	SM 18:1;2O/16:0	150	1.00
SM 18 1;2O 18 0	150	1.00	SM 18 1;2O 18 0	150	1.00

To further explore the performance differences between WMYn (KART) and WMYn (MLP) in small samples, we set the number of epochs to 3000 and conducted spectral predictions on 15 lipid reference standards, using 10 samples as input each time. The specific statistical results are presented in Figure R1.b. Notably, WMYn (KART) shows a significant advantage in spectral prediction for PC 18:0_18:0, PC 18:1_18:1, SM 18:1; 2O/16:0, and SM 18:1; while the lipid reference standards DG 18:1_18:1 and PE 18:1_18:1 perform slightly worse than WMYn (MLP). This indicates that, despite the increase in the number of model iterations, WMYn (KART) still retains a superior performance over WMYn (MLP) in overall spectral prediction. The specific experimental results are shown in Table R2.

Table R2. Cosine similarity results for different models (epoch = 3000).

Method: WMYN (MLP)		Method: WMYN (KART)	
Lipid reference standards	Cosine similarity	Lipid reference standards	Cosine similarity
Cer 18:1;2O/16:0	0.89	Cer 18:1;2O/16:0	0.99
Cer 18:1;2O/16:0	0.91	Cer 18:1;2O/16:0	0.87
Cer 18:1;2O/16:0	0.36	Cer 18:1;2O/16:0	0.99
Cer 18:1;2O/16:0	0.51	Cer 18:1;2O/16:0	0.30
Cer 18:1;2O/16:0	0.70	Cer 18:1;2O/16:0	1.00
Cer 18:1;2O/16:0	0.98	Cer 18:1;2O/16:0	0.61
Cer 18:1;2O/16:0	0.99	Cer 18:1;2O/16:0	1.00
Cer 18:1;2O/16:0	0.85	Cer 18:1;2O/16:0	1.00
Cer 18:1;2O/16:0	0.94	Cer 18:1;2O/16:0	0.44
Cer 18:1;2O/16:0	1.00	Cer 18:1;2O/16:0	1.00
DG 18:1_18:1	1.00	DG 18:1_18:1	0.92
DG 18:1_18:1	0.33	DG 18:1_18:1	1.00
DG 18:1_18:1	1.00	DG 18:1_18:1	0.00
DG 18:1_18:1	0.83	DG 18:1_18:1	1.00
DG 18:1_18:1	1.00	DG 18:1_18:1	1.00
DG 18:1_18:1	0.86	DG 18:1_18:1	1.00
DG 18:1_18:1	1.00	DG 18:1_18:1	0.99
DG 18:1_18:1	1.00	DG 18:1_18:1	1.00
DG 18:1_18:1	0.99	DG 18:1_18:1	1.00
DG 18:1_18:1	0.98	DG 18:1_18:1	0.94
LPC 16:0	0.98	LPC 16:0	1.00
LPC 16:0	0.13	LPC 16:0	1.00
LPC 16:0	1.00	LPC 16:0	1.00
LPC 16:0	0.99	LPC 16:0	0.41
LPC 16:0	0.86	LPC 16:0	1.00
LPC 16:0	1.00	LPC 16:0	1.00
LPC 16:0	1.00	LPC 16:0	0.40
LPC 16:0	0.86	LPC 16:0	0.66
LPC 16:0	1.00	LPC 16:0	1.00
LPC 16:0	0.10	LPC 16:0	0.10
LPC 16:0	0.96	LPC 16:0	0.47
LPC 16:0	0.00	LPC 16:0	1.00
LPC 16:0	1.00	LPC 16:0	1.00
LPC 16:0	0.38	LPC 16:0	0.47
LPC 16:0	0.93	LPC 16:0	1.00
LPC 16:0	0.22	LPC 16:0	1.00
LPC 16:0	0.98	LPC 16:0	1.00
LPC 16:0	0.90	LPC 16:0	0.97
LPC 16:0	0.50	LPC 16:0	1.00

LPC 16:0	0.52	LPC 16:0	0.99
LPC 18:0	0.99	LPC 18:0	1.00
LPC 18:0	0.53	LPC 18:0	0.46
LPC 18:0	1.00	LPC 18:0	1.00
LPC 18:0	0.26	LPC 18:0	1.00
LPC 18:0	0.00	LPC 18:0	0.98
LPC 18:0	0.99	LPC 18:0	1.00
LPC 18:0	0.69	LPC 18:0	1.00
LPC 18:0	0.99	LPC 18:0	1.00
LPC 18:0	0.00	LPC 18:0	0.09
LPC 18:0	1.00	LPC 18:0	0.98
PC 16:0_16:0	0.90	PC 16:0_16:0	0.99
PC 16:0_18:1	0.68	PC 16:0_18:1	1.00
PC 16:0_16:0	1.00	PC 16:0_16:0	0.28
PC 16:0_18:1	0.10	PC 16:0_18:1	1.00
PC 16:0_16:0	0.84	PC 16:0_16:0	1.00
PC 16:0_18:1	0.97	PC 16:0_18:1	1.00
PC 16:0_16:0	1.00	PC 16:0_16:0	1.00
PC 16:0_18:1	0.54	PC 16:0_18:1	0.03
PC 16:0_16:0	0.89	PC 16:0_16:0	1.00
PC 16:0_18:1	0.91	PC 16:0_18:1	1.00
PC 16:0_16:0	0.79	PC 16:0_16:0	0.08
PC 16:0_18:1	1.00	PC 16:0_18:1	1.00
PC 16:0_16:0	0.54	PC 16:0_16:0	1.00
PC 16:0_18:1	1.00	PC 16:0_18:1	0.41
PC 16:0_16:0	0.00	PC 16:0_16:0	0.85
PC 16:0_18:1	0.92	PC 16:0_18:1	1.00
PC 16:0_16:0	1.00	PC 16:0_16:0	1.00
PC 16:0_18:1	1.00	PC 16:0_18:1	0.99
PC 16:0_16:0	1.00	PC 16:0_16:0	0.99
PC 16:0_18:1	1.00	PC 16:0_18:1	1.00
PC 16:0_18:1	0.19	PC 16:0_18:1	1.00
PC 16:0_18:1	1.00	PC 16:0_18:1	1.00
PC 16:0_18:1	1.00	PC 16:0_18:1	1.00
PC 16:0_18:1	1.00	PC 16:0_18:1	1.00
PC 16:0_18:1	0.55	PC 16:0_18:1	1.00
PC 16:0_18:1	0.62	PC 16:0_18:1	0.70
PC 16:0_18:1	0.90	PC 16:0_18:1	0.98
PC 16:0_18:1	1.00	PC 16:0_18:1	1.00
PC 16:0_18:1	0.83	PC 16:0_18:1	1.00
PC 18:0_18:0	0.47	PC 18:0_18:0	0.99
PC 18:0_18:0	0.01	PC 18:0_18:0	1.00
PC 18:0_18:0	0.07	PC 18:0_18:0	1.00

PC 18:0_18:0	1.00	PC 18:0_18:0	0.87
PC 18:0_18:0	1.00	PC 18:0_18:0	1.00
PC 18:0_18:0	0.00	PC 18:0_18:0	1.00
PC 18:0_18:0	1.00	PC 18:0_18:0	1.00
PC 18:0_18:0	1.00	PC 18:0_18:0	1.00
PC 18:0_18:0	0.84	PC 18:0_18:0	1.00
PC 18:0_18:0	1.00	PC 18:0_18:0	1.00
PC 18:0_18:0	0.20	PC 18:0_18:0	1.00
PC 18:0_18:0	0.20	PC 18:0_18:0	1.00
PC 18:0_18:0	0.98	PC 18:0_18:0	1.00
PC 18:0_18:0	0.98	PC 18:0_18:0	0.83
PC 18:0_18:0	1.00	PC 18:0_18:0	1.00
PC 18:0_18:0	0.20	PC 18:0_18:0	1.00
PC 18:0_18:0	0.20	PC 18:0_18:0	1.00
PC 18:0_18:0	1.00	PC 18:0_18:0	0.83
PC 18:0_18:0	1.00	PC 18:0_18:0	0.08
PC 18:0_18:0	0.16	PC 18:0_18:0	0.09
PC 18:1_18:1	1.00	PC 18:1_18:1	0.99
PC 18:1_18:1	0.75	PC 18:1_18:1	0.99
PC 18:1_18:1	1.00	PC 18:1_18:1	0.99
PC 18:1_18:1	0.87	PC 18:1_18:1	0.89
PC 18:1_18:1	1.00	PC 18:1_18:1	0.60
PC 18:1_18:1	1.00	PC 18:1_18:1	0.98
PC 18:1_18:1	1.00	PC 18:1_18:1	0.99
PC 18:1_18:1	0.75	PC 18:1_18:1	0.99
PC 18:1_18:1	0.87	PC 18:1_18:1	0.98
PC 18:1_18:1	0.68	PC 18:1_18:1	0.60
PC 18:1_18:1	0.83	PC 18:1_18:1	0.92
PC 18:1_18:1	0.42	PC 18:1_18:1	1.00
PC 18:1_18:1	0.72	PC 18:1_18:1	1.00
PC 18:1_18:1	0.85	PC 18:1_18:1	0.64
PC 18:1_18:1	0.75	PC 18:1_18:1	0.98
PC 18:1_18:1	0.75	PC 18:1_18:1	0.98
PC 18:1_18:1	0.75	PC 18:1_18:1	0.98
PC 18:1_18:1	1.00	PC 18:1_18:1	0.88
PC 18:1_18:1	0.96	PC 18:1_18:1	1.00
PC 18:1_18:1	0.06	PC 18:1_18:1	1.00
PE 18:1_18:1	0.89	PE 18:1_18:1	1.00
PE 18:1_18:1	0.62	PE 18:1_18:1	0.42
PE 18:1_18:1	0.00	PE 18:1_18:1	0.65
PE 18:1_18:1	1.00	PE 18:1_18:1	1.00
PE 18:1_18:1	1.00	PE 18:1_18:1	1.00
PE 18:1_18:1	0.28	PE 18:1_18:1	0.99
PE 18:1_18:1	0.88	PE 18:1_18:1	1.00

PE 18:1_18:1	1.00	PE 18:1_18:1	1.00
PE 18:1_18:1	1.00	PE 18:1_18:1	0.95
PE 18:1_18:1	1.00	PE 18:1_18:1	0.14
PE 18:1_18:1	0.98	PE 18:1_18:1	0.33
PE 18:1_18:1	0.98	PE 18:1_18:1	1.00
PE 18:1_18:1	1.00	PE 18:1_18:1	1.00
PE 18:1_18:1	0.50	PE 18:1_18:1	0.99
PE 18:1_18:1	1.00	PE 18:1_18:1	1.00
PE 18:1_18:1	1.00	PE 18:1_18:1	0.93
PE 18:1_18:1	1.00	PE 18:1_18:1	1.00
PE 18:1_18:1	0.98	PE 18:1_18:1	1.00
PE 18:1_18:1	0.92	PE 18:1_18:1	0.53
PE 18:1_18:1	0.99	PE 18:1_18:1	0.99
PI 18:1_18:1	0.96	PI 18:1_18:1	1.00
PI 18:1_18:1	0.43	PI 18:1_18:1	1.00
PI 18:1_18:1	0.91	PI 18:1_18:1	1.00
PI 18:1_18:1	1.00	PI 18:1_18:1	0.79
PI 18:1_18:1	1.00	PI 18:1_18:1	1.00
PI 18:1_18:1	0.54	PI 18:1_18:1	1.00
PI 18:1_18:1	0.99	PI 18:1_18:1	1.00
PI 18:1_18:1	0.37	PI 18:1_18:1	0.63
PI 18:1_18:1	0.93	PI 18:1_18:1	1.00
PI 18:1_18:1	0.98	PI 18:1_18:1	1.00
SM 18:1;2O/16:0	0.00	SM 18:1;2O/16:0	0.98
SM 18:1;2O/16:0	0.00	SM 18:1;2O/16:0	0.08
SM 18:1;2O/16:0	0.08	SM 18:1;2O/16:0	1.00
SM 18:1;2O/16:0	1.00	SM 18:1;2O/16:0	1.00
SM 18:1;2O/16:0	0.00	SM 18:1;2O/16:0	1.00
SM 18:1;2O/16:0	0.68	SM 18:1;2O/16:0	1.00
SM 18:1;2O/16:0	1.00	SM 18:1;2O/16:0	1.00
SM 18:1;2O/16:0	0.21	SM 18:1;2O/16:0	1.00
SM 18:1;2O/16:0	0.93	SM 18:1;2O/16:0	0.98
SM 18:1;2O/16:0	0.82	SM 18:1;2O/16:0	1.00
SM 18:1;2O/16:0	0.92	SM 18:1;2O/16:0	1.00
SM 18:1;2O/16:0	0.92	SM 18:1;2O/16:0	1.00
SM 18:1;2O/16:0	1.00	SM 18:1;2O/16:0	0.91
SM 18:1;2O/16:0	0.00	SM 18:1;2O/16:0	0.83
SM 18:1;2O/16:0	0.92	SM 18:1;2O/16:0	1.00
SM 18:1;2O/16:0	0.68	SM 18:1;2O/16:0	1.00
SM 18:1;2O/16:0	0.00	SM 18:1;2O/16:0	1.00
SM 18:1;2O/16:0	1.00	SM 18:1;2O/16:0	1.00
SM 18:1;2O/16:0	0.94	SM 18:1;2O/16:0	0.09
SM 18:1;2O/16:0	0.00	SM 18:1;2O/16:0	0.94
SM 18:1;2O/18:0	0.55	SM 18:1;2O/18:0	0.96

SM 18:1;2O/18:0	0.55	SM 18:1;2O/18:0	0.96
SM 18:1;2O/18:0	0.91	SM 18:1;2O/18:0	1.00
SM 18:1;2O/18:0	0.92	SM 18:1;2O/18:0	1.00
SM 18:1;2O/18:0	0.89	SM 18:1;2O/18:0	1.00
SM 18:1;2O/18:0	0.17	SM 18:1;2O/18:0	0.59
SM 18:1;2O/18:0	0.00	SM 18:1;2O/18:0	1.00
SM 18:1;2O/18:0	0.00	SM 18:1;2O/18:0	1.00
SM 18:1;2O/18:0	0.55	SM 18:1;2O/18:0	0.96
SM 18:1;2O/18:0	0.65	SM 18:1;2O/18:0	1.00

Reviewer #2 (Remarks to the Author):

This study developed LipidIN that enables users to perform instant, accurate, and in-depth searches within a hierarchical in silico library containing over 100 million lipid structures. The outcome is interesting and would attract attention from wide ranges of communities. Traditional spectrum searching methods may overestimate similarity by double-counting sub-fragments from the same structure, even when important fragments are missing. Accurate compound identification can be challenging, particularly when the reference library lacks experimental data for the specific compound. Moreover, the scarcity of spectral libraries detailing compound structures impedes lipid annotation. The study showed potential to overcome such traditional problem by constructing a 166.3-million lipid fragmentation hierarchical library, and significantly accelerating library searches through hash table and bisection algorithms. The study also provides retention time-based false positive removal and spectral regeneration capabilities. Benchmarking case studies show convincing results of performance superiority compared to other common approaches. The study is highly expected to enhance the annotations on unknowns that measured by LipidIN in clinical cohorts. Prior to publication, authors should address the following issues.

Comment 1. Is the data in .mzML centroided, or profile? Please indicate it explicitly.

Response: Thank you for your interest in this work! Typically, centroided data allows for faster analysis due to its simplified representation, while profile data retains more detailed information about the mass spectra. In all experiments and benchmark comparisons presented in this article, we utilized raw profile data to ensure a comprehensive analysis of the lipid profiles.

It's important to highlight that LipidIN supports both centroided and profile mzML formats, providing flexibility for users depending on their analytical needs. This capability allows researchers to choose the format that best suits their workflow, whether they prioritize speed or detailed spectral information. If you have any further questions or need additional clarification, please feel free to ask!

Comment 2. Is there a difference in mean matching value of highly ranked list (e.g. top 20) between EQ + LCI + 1st-4th library, EQ + LCI + MS-DIAL published library and EQ + MS-DIAL published library? If the EQ + LCI + 1st-4th library improves the average value not only recall@1 and 20, it indicates that the hierarchical in silico library approach has an advantage in suggesting rough structure (i.e. fuzzy structure) of the query rather than traditional method.

Response: Thank you for your valuable suggestion. We agree that calculating the average annotation scores can greatly enhance our evaluation of the differences among the various methods. In our revised analysis, we specifically assessed the advantages and disadvantages of the three methods using the dataset referenced in Fig. 3b.

To strengthen our evaluation, we computed the average scores for both the Top 10 and Top 20 annotations, using the publicly available appraisal results for this dataset as a benchmark, which we present in Fig. R2. Our findings indicate that the EQ + LCI + 1st-4th library method achieves the highest average scores in both the Top 10 and Top 20 categories, demonstrating a significant improvement compared to the other methods evaluated.

Furthermore, while the EQ + LCI + MS-DIAL published library method shows an improvement over the EQ + MS-DIAL published library method, the difference is not statistically significant. This suggests that, although there are some advantages to the EQ + LCI + MS-DIAL approach, the improvements may not be substantial enough to warrant a change in methodology without further investigation.

We appreciate your feedback, which has helped us clarify these important distinctions in our analysis.

Fig. R2. Average scores for both Top10 and Top20 of three methods. a. the mean score of EQ + LCI + 1st-4th library, EQ + LCI + MS-DIAL published library and EQ + MS-DIAL published library Top@10, this methods were tested two-by-two using the t-test. **b.** the mean score of EQ + LCI + 1st-4th library, EQ + LCI + MS-DIAL published library and EQ + MS-DIAL published library Top@20, this methods were tested two-by-two using the t-test.

Comment 3. In the main text, there are several references to speeds that are not consistent (Line 33: over 100 billion times' querying in 1 s, Line 194: 0.14 s for querying 1,000,000 spectra library, Line 344: average 133.47 billion times querying per seconds). These discrepancies likely arise from different testing conditions, dataset sizes, or query complexities. A detailed explanation of the methodologies used in each case would help readers understand these variations in reported performance.

Response: Thank you for your insightful comments. I would like to clarify that the values mentioned are not contradictory. The statement in the abstract (Line 33)

regarding over 100 billion queries per second is a conservative estimate, reflecting what most users can achieve and even exceed through replication.

In contrast, the result described in Line 194 pertains to a benchmark experiment focused on metabolite annotation. In this experiment, we set the precursor parent ion tolerance to 5 ppm and the allowable error for the charge-to-mass ratio of secondary fragments to 10 ppm. We utilized a sample file to search a spectral library containing 1 million secondary messages, and we calculated the average elapsed time for each peak in the sample file.

Furthermore, the results presented in Line 344 were obtained using an Agilent 6546 Q-TOF instrument for lipid identification, with tolerances set at 20 ppm for MS1 and 40 ppm for MS2. It is important to note that this larger tolerance does result in a decrease in the speed of library searching.

I hope this clarifies the distinctions between the different results and their respective methodologies. Thank you once again for your valuable feedback.

Comment 4. The study states that it pertains to metabolite annotations (Line 34, Line 217); however, there are no comprehensive benchmarking comparison results available, aside from those presented in Figure S4 a. Please consider adding more detailed comparisons.

Response: We appreciate your interest in our work on metabolite annotation, which builds upon our EQ module. In this revision, we have incorporated relevant benchmarking data to enhance our analysis. Specifically, we integrated data from MassBank² and the MassBank of North America (MoNA, <https://mona.fiehnlab.ucdavis.edu>) to create a comprehensive library consisting of 320,531 metabolic spectra. From MoNA, we sourced 65,672 metabolic spectra obtained using the Orbitrap instrument. To ensure data quality, we applied a filter to remove peaks that were 5% or 10% below the highest peak.

This integrated library was then utilized for annotation across various m/z tolerances, effectively demonstrating the capabilities of the EQ module in metabolic

characterization. Additionally, we have updated Supplementary Fig.4f and revised the corresponding sections in this response for clarity. We also revised manuscript in Line 216, Supplementary Fig.5b. We hope these enhancements provide a clearer understanding of our methodology and findings.

Fig. R3. EQ module uses an integrated spectral library to characterize MoNA's Orbitrap instrument spectral. a. The library querying was performed using the EQ module with the MS1 tolerance set at 5 and the MS2 tolerance set at 5,10 or 20 when the Orbitrap instrument spectral was filtered for the highest 10% or 5% of peaks, respectively.

Comment 5. Why a fold change of 1.4 was chosen to identify differential metabolites in third clinical cohort?

Response: Thank you for your meticulous attention to detail. Typically, a fold change threshold of 1.5 to 2 is commonly employed, with an absolute fold change greater than 2 considered significant. In instances involving smaller sample sizes or higher data variability, a threshold of 1.5 may be utilized to capture additional biological variation.

In our study, we set the threshold at 1.4; however, we still did not identify any common differential lipids among the disease groups. This finding suggests that

differentiating between the four diseases using specific lipid profiles is quite challenging, even with more lenient criteria. As a result, we directed our focus toward the fine structure of double-bonding positions, as this approach may provide more effective differentiation among the diseases.

We appreciate your insightful feedback, which has helped us clarify our analytical approach.

Comment 6. The Discussion section references the Kolmogorov-Arnold theorem in the context of reverse lipidomics modules. However, the integration of this theorem into the model requires further elaboration, specifically, a clearer explanation of how the Kolmogorov-Arnold theorem is applied to the reverse lipidomics modules is needed.

Response: Thank you! Our work is inspired by the function representation capabilities of KART, particularly its approach to decomposing complex functions. We aim to represent spectral data, influenced by various factors such as mass spectrometry conditions and ion sources, as a sum of multiple observational samples.

The two-layer network architecture we employed effectively implements this decomposition. Each node (or neuron) in the network can be viewed as a simple function, and the entire network serves as a combination of these simpler functions. By adjusting the network parameters, we are able to approximate any complex function.

Additionally, we integrate activation functions such as SiLU to enable the network to capture intricate nonlinear transformation patterns within the data. This combination enhances the network's ability to model the underlying complexities of the spectral data. We will provide a more detailed explanation below in relation to the equations.

WMy_n is inspired by the KART for mass spectrometry representation. We propose that the true spectrum \mathbf{S} of a lipid is influenced by various factors, including mass spectrometry conditions (Y_1), chromatography conditions (Y_2), ionization voltage (Y_3), and the nature of the compound (Y_4). Consequently, the true spectrum \mathbf{S} can be expressed as a multivariate function that accounts for these variables $\mathbf{S} = \Phi(Y_1, Y_2, \dots, Y_n)$. According to KART, any multivariate continuous function

$\Phi(Y_1, \dots, Y_n)$ can be represented as a finite number of single-variable continuous functions in a two-layer nested summation:

$$f(X) = f(x_1, \dots, x_n) = \sum_{q=1}^{2n+1} \Phi_q \left(\sum_{p=1}^n \phi_{q,p}(x_p) \right),$$

thus, this function can be represented as a finite unitary function $\mathbf{S} = \sum_{j=1}^m \Phi_j$, where m is a finite constant. We utilize the spectrum from a large cohort, which has undergone certain variations is \mathbf{F} , where $\mathbf{F} = \sum_{i=1}^n f_i(WX + B)$, where X is a particular observation and the f_i function is a process for mapping X . Based on the above derivation, we construct a representation of the real spectrogram in relation to the spectrogram in the cohort:

$$\Phi(Y_1, \dots, Y_n) = \sum_{j=1}^{2n+1} \Phi_j(\sum_{i=1}^n f_i(WX + B)).$$

where X represents the input matrix, consisting of x_1, \dots, x_n , where each x is a spectrum peak. W being the weight matrix, and B the bias matrix. f_i is the Sigmoid linear unit (SiLU) activation function to enable the network to capture intricate nonlinear transformation patterns within the data, and Φ_j is a learnable nonlinear activation. By adding learned parameters in f_i , a learnable nonlinear activation is achieved.

Inspired by the KART, $\sum_{p=1}^n \phi_{q,p}(x_p)$ represents the weighted combination of features of x_1, \dots, x_n for each spectrum. In the first stage of WMYn, the linear transformation is the method that implements this weighted sum. With the learned weights $\phi_{q,p}$, each input feature x_p participates in the weighted sum:

$$\sum_{i=1}^n f_i(WX + B)$$

Φ_q is a nonlinear transformation that processes the above weighted sum as input. In the first stage of WMYn, a learnable nonlinear activation function is implemented based on the learned parameters. For each spectrum, a different activation function is learned, and these spectra are mapped to the latent feature space through the learned activation functions, ultimately producing the final feature matrix.

Thus, KART provides a theoretical foundation for inferring the relationship between the actual spectrogram and the spectrogram obtained from the cohort experiment. In contrast, WMYn serves as the transformation used to derive the series of positions within the expression. We also revised manuscript in Line 675.

Comment 7. Given the increasing research focus on oxidized lipids, it is noteworthy that Table S1 lacks comprehensive representation of these important molecules. Consider expanding the lipid categories to include oxidized variants of major phospholipids, such as: Oxidized phosphatidylcholine (OxPC), Oxidized phosphatidylethanolamine (OxPE), Oxidized phosphatidylglycerol (OxPG), Oxidized phosphatidylserine (OxPS).

Response: Thanks to your attention to detail, we have updated the lipids you mentioned above, in addition to more than 30 other lipids²⁻³⁶, as reflected in the attached Table R3. We also revised manuscript in Line 32, 93, 104, 113, 156, 242, 243, 759 and Table. S2.

Table R3. All 121 lipids brief information.

Categories	Lipid subclass	Abbreviation
Prenol Lipids [PR]	Vitamin A fatty acid ester	VAE
Sphingolipids [SP]	Trihexosylceramide	Hex3Cer
Sterol Lipids [ST]	Triacylglycerol	TG
Sphingolipids [SP]	Sulfonolipid	SL
Sphingolipids [SP]	Sphingomyelin	SM
Sphingolipids [SP]	Sphinganine	SPB
Sphingolipids [SP]	Sphingosine	SPB
Sphingolipids [SP]	Phytosphingosine	SPB
Glycerophospholipids [GP]	Phosphatidylserine	PS
Glycerophospholipids [GP]	Phosphatidylmethanol	PMeOH
Glycerophospholipids [GP]	Phosphatidylinositol	PI
Glycerophospholipids [GP]	Phosphatidylglycerol	PG
Glycerophospholipids [GP]	Phosphatidylethanolamine	PE
Glycerophospholipids [GP]	Phosphatidylethanol	PEtOH
Glycerophospholipids [GP]	Phosphatidylcholine	PC
Glycerophospholipids [GP]	Phosphatidic acid	PA
Sterol Lipids [ST]	Ether-linked triacylglycerol	TG-O
Sterol Lipids [ST]	Oxidized triglyceride	OxTG
Sphingolipids [SP]	Oxidized sulfonolipid	SL+O
Glycerophospholipids [GP]	Oxidized phosphatidylethanolamine	OxPE
Fatty acyls [FA]	Oxidized fatty acid	OxFA
Glycerophospholipids [GP]	N-Monomethylphosphatidylethanolamine	MMPE
Glycerophospholipids [GP]	N-Dimethylphosphatidylethanolamine	DMPE
Fatty acyls [FA]	N-acyl ethanolamines	NAE
Glycerolipids [GL]	Monogalactosylmonoacylglycerol	MGMG
Glycerolipids [GL]	Monogalactosyldiacylglycerol	MGDG
Glycerolipids [GL]	Monoacylglycerol	MG
Glycerophospholipids [GP]	Lysophosphatidylserine	LPS
Glycerophospholipids	Lysophosphatidylinositol	LPI

[GP]		
Glycerophospholipids	Lysophosphatidylglycerol	LPG
[GP]		
Glycerophospholipids	Lysophosphatidylethanolamine	LPE
[GP]		
Glycerophospholipids	Lysophosphatidic acid	LPA
[GP]		
Glycerophospholipids	Lysophosphatidylcholine	LPC
[GP]		
Glycerolipids [GL]	Lysodiacylglyceryl-3-O-carboxyhydroxymethylcholine	LDGCC
Glycerolipids [GL]	Lysodiacylglyceryl trimethylhomoserine	LDGTS
Glycerolipids [GL]	Lysodiacylglyceryl hydroxymethyl-N,N,N-trimethyl-β-alanine	LDGTS
Glycerophospholipids	Lysocardiolipin	MLCL
[GP]		
Sphingolipids [SP]	Hexosylceramide non-hydroxyfatty acid-sphingosine	HexCer
Sphingolipids [SP]	Hexosylceramide non-hydroxyfatty acid-dihydrosphingosine	HexCer
Sphingolipids [SP]	Hexosylceramide hydroxyfatty acid-dihydrosphingosine	HexCer
Sphingolipids [SP]	Hexosylceramide hydroxyfatty acid-sphingosine	HexCer
Glycerophospholipids	Hemibismonoacylglycerophosphate	HBMP
[GP]		
Glycerophospholipids	Glycerophospho N-acyl ethanolamine	GPNAE
[GP]		
Fatty acyls [FA]	Free fatty acid	FA
Fatty acyls [FA]	Fatty acid ester of hydroxyl fatty acid	FAHFA
Glycerophospholipids	Ether-linked phosphatidylserine	PS-O
[GP]		
Glycerophospholipids	Ether-linked phosphatidylinositol	PI-O
[GP]		
Glycerophospholipids	Ether-linked phosphatidylglycerol	PG-O
[GP]		
Glycerophospholipids	Ether-linked phosphatidylethanolamine	PE-O
[GP]		
Glycerophospholipids	Ether-linked phosphatidylethanolamine (P)	PE-P
[GP]		
Glycerophospholipids	Ether-linked phosphatidylcholine	PC-O
[GP]		
Glycerolipids [GL]	Ether-linked monogalactosyldiacylglycerol	MGDG-O
Glycerophospholipids	Ether-linked lysophosphatidylglycerol	LPG-O

[GP]		
Glycerophospholipids	Ether-linked lysophosphatidylethanolamine	LPE-O
[GP]		
Glycerophospholipids	Ether-linked lysophosphatidylcholine	LPC-O
[GP]		
Glycerolipids [GL]	Ether-linked digalactosyldiacylglycerol	DGDG-O
Glycerolipids [GL]	Ether-linked diacylglycerol	DG-O
Glycerophospholipids		
[GP]	Dilysocardiolipin	DLCL
Sphingolipids [SP]	Dihexosylceramide	Hex2Cer
Glycerolipids [GL]	Digalactosyldiacylglycerol	DGDG
Glycerolipids [GL]	Diacylglyceryl-3-O-carboxyhydroxymethylcholine	DGCC
Glycerolipids [GL]	Diacylglyceryl trimethylhomoserine/diacylglyceryl hydroxymethyl-N,N,N-trimethyl-β-alanine	DGTS
Glycerolipids [GL]	Diacylglyceryl glucuronide	DGGA
Glycerolipids [GL]	Diacylglycerol	DG
Prenol Lipids [PR]	Coenzyme Q	CoQ
Sterol Lipids [ST]	Cholesteryl ester	CE
Sphingolipids [SP]	Ceramide non-hydroxyfatty acid-phytospingosine	Cer
Sphingolipids [SP]	Ceramide hydroxy fatty acid-sphingosine	Cer
Sphingolipids [SP]	Ceramide hydroxy fatty acid-dihydrosphingosine	Cer
Sphingolipids [SP]	Ceramide non-hydroxyfatty acid-dihydrosphingosine	Cer
Sphingolipids [SP]	Ceramide non-hydroxyfatty acid-sphingosine	Cer
Sphingolipids [SP]	Ceramide 1-phosphates	CerP
Glycerophospholipids		
[GP]	Cardiolipin	CL
Glycerophospholipids		
[GP]	Bismonoacylglycerophosphate	BMP
Glycerophospholipids		
[GP]	Oxidized phosphatidylcholine	OxPC
Glycerophospholipids		
[GP]	Oxidized phosphatidylserine	OxPS
Glycerophospholipids		
[GP]	Oxidized phosphatidylinositol	OxPI
Glycerophospholipids		
[GP]	Oxidized phosphatidylglycerol	OxPG
Glycerolipids [GL]	Sulfoquinovosyl diacylglycerol	SQDG
Glycerolipids [GL]	Acyl diacylglyceryl glucuronide	ADGGA
Glycerolipids [GL]	Ether-linked Acyl diacylglyceryl glucuronide	ADGGA-O

Sterol Lipids [ST]	Acylhexosyl brassicasterol	AHexBRS-ST
Sterol Lipids [ST]	Acylhexosyl campesterol	AHexCAS-ST
Sphingolipids [SP]	Acylhexosylceramide	AHexCer-O
Sterol Lipids [ST]	Acylhexosyl cholesterol	AHexCS-ST
Sterol Lipids [ST]	Acylhexosyl sitosterol	AHexSIS-ST
Sterol Lipids [ST]	Acylhexosyl stigmasterol	AHexSTS-ST
Sphingolipids [SP]	Acylsphingomyelin	ASM
Sterol Lipids [ST]	Cholic acid sulfate	BASulfate-ST
Sterol Lipids [ST]	Brassicasterol ester	BRSE
Fatty acyls [FA]	Acylcarnitine	CAR
Sterol Lipids [ST]	Campesterol ester	CASE
Sterol Lipids [ST]	Cholesterol	Cholesterol-ST
Sterol Lipids [ST]	Esterified deoxycholic acid	DCAE-SE
Sterol Lipids [ST]	Dehydroergosterol ester	DEGSE
Sphingolipids [SP]	Ganglioside GD1a	GD1a
Sphingolipids [SP]	Ganglioside GD1b	GD1b
Sphingolipids [SP]	Ganglioside GM1	GM1
Sphingolipids [SP]	Ganglioside GM3	GM3
Sphingolipids [SP]	Ganglioside GQ1b	GQ1b
Sphingolipids [SP]	Ganglioside GT1b	GT1b
Glycerophospholipids [GP]	N-acyl-lysophosphatidylethanolamine	LNAPE
Glycerophospholipids [GP]	N-acyl-lysophosphatidylserine	LNAPS
Fatty acyls [FA]	N-acyl glycine	NAGly
Fatty acyls [FA]	N-acyl glyceryl serine	NAGlySer
Fatty acyls [FA]	N-acyl ornithine	NAOrn
Sphingolipids [SP]	Ceramide phosphoethanolamine	PE-Cer
Sphingolipids [SP]	Oxidized ceramide phosphoethanolamine	PE-Cer+O
Sphingolipids [SP]	Ceramide phosphoinositol	PI-Cer
Sphingolipids [SP]	Oxidized ceramide phosphoinositol	PI-Cer+O
Sphingolipids [SP]	Sulfatide	SHexCer
Sphingolipids [SP]	Oxidized sulfatide	SHexCer+O
Sterol Lipids [ST]	Stigmasterol hexoside	SHex-ST
Glycerolipids [GL]	Ether-linked Semino lipid	SMGDG-O
Sphingolipids [SP]	Solated cysteinolide A	Solated_cysteinolide-A
Sphingolipids [SP]	Solated cysteinolide B	Solated_cysteinolide-B
Sterol Lipids [ST]	Sterol sulfate	SSulfate-ST
Sphingolipids [SP]	Synthetic sulfur-containing amino lipids	Synthetic-SAL
Glycerolipids [GL]	Triacylglycerol estolides	TG-EST
Sterol Lipids [ST]	Vitamin D	Vitamin-D
Prenol Lipids [PR]	Vitamin E	Vitamin-E

Comment 8. Authors should describe how to prepare query spectrum library for users.

Response: Thank you for your valuable suggestion; it would indeed enhance the usability of LipidIN. We have uploaded the code on GitHub (<https://github.com/LinShuhaiLAB/LipidIN/tree/main/How%20to%20Convert%20Your%20MSP%20Format%20Spectral%20Library%20to%20RDA%20Format>) for converting spectrogram files from MSP format to RDA format, which is compatible with LipidIN. Additionally, a README file is included to provide guidance on using the code.

Fig. R4. Showing the updated MSP format to RDA files module on Github.

Comment 9. Regarding Figure 1: The icon font used in the figure is too small, which may impair readability. Please increase the font size to ensure that all text is clearly legible, especially when the figure is scaled down in print or on-screen viewing. There is an instance of obscured figure text. This obstructed text compromises the clarity of the information being conveyed. Please revise the figure layout to ensure that all text is fully visible and not overlapped by other elements.

Response: Thank you for your insightful comments regarding Figure 1. We appreciate your attention to detail. In response, we have increased the font size to enhance

readability and ensure that all text is clearly legible, even when the figure is scaled down. We also carefully revised the figure layout to eliminate any instances of obscured text and overlapping elements. These updates aim to improve the clarity and effectiveness of the information conveyed in the figure. We believe these changes will significantly enhance the viewer's understanding and overall presentation of the data.

Comment 10. Regarding Figure 3, please explain why the flash entropy appears to have no data point at the 10^7 mark. The figure shows no substantial difference between EQ + LCI + MS-DIAL published library and EQ + MS-DIAL published library presented in Figure b. Is this because the benchmarks don't use a hierarchical library? Please elaborate. In addition, the legends for Fig. 3d and Fig. S7c are inconsistent. To my knowledge, the LDA software includes several modules, such as library searching and retention time judgment. It is recommended to modify the expression to 'LDA retention time prediction algorithm' for clarity.

Response: Thank you for your question. First, during the construction of the spectral library recognizable by flash entropy, we encountered memory limitations that prevented us from testing it with 10^7 spectral libraries; thus, we labeled the data as unavailable. Second, while it is accurate that there is no significant difference between the two methods as illustrated in Fig 3, we do observe a difference in mean scores between these methods, as shown in Fig. 3b. However, this difference is not statistically significant. Finally, we acknowledge that our original legend presentation was not sufficiently rigorous, and we have revised it in accordance with your suggestions.

Comment 11. Regarding Figure 5b, I find the text of the labels very small and difficult to read. Please increase the font size of these figures.

Response: Thank you for your valuable suggestion. We acknowledge that the font size for the lipid names in Figure 5b was indeed too small. We have enlarged the font to enhance readability.

Comment 12. Figure S8 needs to be in higher resolution.

Response: Thank you for your suggestion. Supplementary Fig.8 contains a substantial amount of information, which contributed to its large file size and potential distortion.

To enhance clarity, we have split it into two separate images.

g

**PIO-13:0_20:4 [M-H]-
MS¹m/z deviation 0.0128 Da**

Page 1 of 1

h

**TG O-17:0_17:1_17:1 [M+NH4]⁺
MS¹m/z deviation 0.0025 Da**

Page 1 of 1

i

**FA 14:3 [M-H]-
MS¹m/z deviation 0.0012 Da**

Page 1 of 1

j

**PG O-15:1_16:0 [M-H]-
MS¹m/z deviation 0.0010 Da**

Page 1 of 1

k

**OxPE 16:0_20:3;O [M-H]-
MS¹m/z deviation 0.0073 Da**

Page 1 of 1

l

**TG 12:0_14:0_18:2 [M+NH4]⁺
MS¹m/z deviation 0.0012 Da**

Page 1 of 1

Fig. R5. Modified Figure 12 lipids mass spectrometric spectra identified by LipidIN but not showed in the article (Tsubawa H, et al. A lipidome landscape of aging in mice. Nat Aging. 2024).

Comment 13. In the lipid categories intelligence model, the training set is not explicitly listed.

Response: Thank you for your thoughtful question. I would like to clarify that the LCI model I proposed does not require a traditional training set. We first validate the correctness and universality of the three rules (ECN, ESCN, and IUP) using data from over 100 public datasets. Following this validation, we construct a path search model based on these rules. This model operates without relying on a conventional training set; instead, it utilizes heuristic functions and search strategies to guide the search process. The primary objective of the model is to address problems by making decisions based on the three rules rather than learning from data. I appreciate your understanding and consideration of this approach.

Comment 14. The manuscript should address the scenario of multiple measured spectrogram peaks falling within the same threshold interval. Please clarify the handling method for cases where multiple measured peaks match within a single threshold interval. Explain whether Equations 1 and 2 account for this multi-peak matching situation. If they do, describe how these equations incorporate multiple peak matches. If Equations 1 and 2 do not account for multiple peak matches Consider modifying the equations to handle this scenario. Alternatively, explain why this situation is not addressed and discuss any potential implications for the analysis.

Response: "Thank you for your question. We previously had a thorough discussion regarding the issue of multiple peak matching. In practice, when identifying multiple peaks that correspond to a library entry, we retain the peak with the highest response. However, this detail was not clearly articulated in Equations 1 and 2. To address this oversight, we have revised the expression to state: 'It is worth noting that when multiple measured spectra match a single peak in the theoretical library, we will only calculate one successful match and select the peak with the highest response.' (Lines 539-541).

Comment 15. The relevance of Equation 10 to the subsequent mathematical development is unclear. Please providing a clear explanation of how Equation 10 relates to and informs the equations that follow, including its role in the overall mathematical

framework.

Response: We appreciate your insights. As outlined in Response 6, KART serves as the theoretical foundation, while WMYn is utilized to determine the unknown mapping. Consequently, Equation 10 formally expresses the idea that the true spectral map can be discretized. The unknown transformation function f_i must be derived or approximated through the subsequent series of equations.

Reviewer #3 (Remarks to the Author):

LipidIN does show great promise with highlights specifically being: large database size, algorithm speed, accuracy, and inclusion of EAD fragmentation for double bond location (which provides important biological subtleties). The ECN models are especially beneficial for increasing coverage. The main limitation of the study is:

Comment 1. No ready to use interface, and the approach consists of multiple programming languages to my knowledge (Python, R, etc., making it difficult to learn). Video tutorials would be very helpful showing the workflow from start to finish. This made it difficult to validate without substantial effort to learn the workflow. This will limit the number of users.

Response: We sincerely appreciate your insightful comments and professional advice! You're absolutely right that a user-friendly interface would significantly enhance LipidIN's accessibility and broaden its appeal. To address this, we have now deployed LipidIN on our team's webpage (<https://www.lifemetabolomics.cn/software>), and Fig. R6 illustrates the web interface. This deployment provides users with a streamlined interface for uploading files and performing lipid annotation with simplified steps. We believe this significantly improves the user experience. We welcome any further feedback you may have on this new interface.

Regarding your question about LipidIN's use of multiple programming languages, let me provide a more detailed explanation. Our choice reflects a strategic optimization for performance and functionality.

It's well-established that the R language, while excellent for statistical analysis, can be computationally intensive when processing large datasets and complex iterative calculations, for example, LipidMatch, which is also an R-based tool, faces the same

problem³⁷. To mitigate this limitation, we've leveraged Rcpp that seamless integration of R and C++, to optimize LipidIN's data structures, resulting in reduced memory usage and improved processing speed.

Similarly, while Python offers advantages in implementing complex neural network models, R's strengths in statistical modeling and data manipulation were crucial for other aspects of LipidIN. Therefore, we strategically employed Python for specific neural network components, integrating seamlessly with the R-based core.

Any perceived complexities arising from the use of multiple languages are significantly mitigated by the user-friendly interface provided on our website. This interface abstracts away the underlying technical details, allowing users to interact with LipidIN easily and efficiently.

Additionally, we have created example videos for users who wish to utilize the code upon request. These videos are available in the introductory video folder and the README file on GitHub, as illustrated in Fig. R7. We believe these resources will enhance the user experience and facilitate easier implementation. (<https://github.com/LinShuhaiLAB/LipidIN>).

a**b**
Fig. R6. Team Web Interface. **a.** The team’s website interface features four modules: LipidIN, LipidIN (WMYn), QuanFormer, and Common Omics Analysis. These modules encompass mass spectrometry quantification, lipid identification, and histology analysis, addressing the majority of usage needs in biomedical and related fields. **b.** The LipidIN interface features a layout designed for clarity and usability. The left side presents a brief overview of the model structure and processes, accompanied by explanatory notes. On the right, users will find the operational interface, where interactions and commands take place.

a

LipidIN Public

main 1 Branch 0 Tags

Go to file Add file Code

HaroldHaoXu Update README.md 9d3e395 · 27 minutes ago 211 Commits

How to Convert Your MSP Format Spectral Libr...	Add files via upload	2 weeks ago
Intoduction Videos	Add files via upload	3 hours ago
LipidIN 4-level hierarchical library	Add files via upload	5 hours ago
LipidIN MS-DIAL published library	Add files via upload	3 months ago
Manuscript	Add files via upload	3 months ago
WMYn	Add files via upload	last month
LICENSE	Create LICENSE	4 months ago
README.md	Update README.md	27 minutes ago
modelcard.md	Create modelcard.md	3 months ago

b

README Apache-2.0 license

Demo Watch

LipidIN demo watch

LipidIN.demo.watch.mp4

```
1 FN <- 'D:/bio_inf/LipidIN-main/LipidIN 4-level hierarchical library/demo pos'
2 pt <- 'D:/bio_inf/LipidIN-main/LipidIN 4-level hierarchical library'
3 MS2_filter <- 0.05
4 ppm1 <- 5
5 ppm2 <- 10
6 ESI <- 'p'
7 # FN: Address of the *.mzML file to be tested.
8 # pt: Support code (EQ.cpp, LCI.R, etc.) address.
9 # filename: Location of .mzML file, for example './demo pos/QC_POS1.mzML'.
10 # ESI: 'p' for positive ionization mode.
11 # 'n1' for negative ionization mode [M+COOH]-.
12 # 'n2' for negative ionization mode [M+CH3COO]-.
13 # MS2_filter: a value of 0-1, MS2 fragments with intensity lower than the MS2_filter*max intensity will be de
14
15- # Preparation and installation packages -----
16 packages <- c('this.path', 'parallel', 'doParallel', 'Rams', 'Rcpp', 'tidyverse', 'dplyr')
17 installed_packages <- packages %in% rownames(installed.packages())
18- if(all(installed_packages)){
19-   print("All required packages are installed.")
20- }
```

Environment History C

Environment is empty

Console Terminal Source Cpp Background Jobs

R 4.3.2 ~ /

> |

ESI 参数指示文件是在正电离模式还是负电离模式下获得的
the ESI parameter indicates whether the file was
obtained in positive or negative ionization mode

Fig. R7. Example videos uploaded to GitHub. a. The video is available in the

Introductory videos folder and can be downloaded by users for offline learning. b. additionally, a preview video is embedded in the README file, allowing users to quickly see how to utilize it.

Comment 2. Only covers one aspect of the workflow (annotation; this will also limit the number of users), and hence a diagram or user manual showing how to integrate all other aspects of the workflow could be helpful. How should a lipidomics user incorporate this software?

Response: Thank you for your valuable advice. As you mentioned, a comprehensive note on integrating LipidIN with other tools is indeed essential. Our team has developed several complementary tools:

1. LipidIN: Designed for high-confidence, high-coverage, and rapid lipid identification for easy-used.

2. QuanFormer: Focused on accurate quantification of mass spectrometry data, which can be integrated with LipidIN for non-expert readers. And QuanFormer has been submitted to another journal.

3. Common omics analysis Module: Aids in analyzing both qualitative and quantitative results, helping users identify discrepant metabolites through a user-friendly interface. If readers have multi-omics data including lipidomics, they can perform Common omics analysis Module in conjunction with LipidIN for their data.

I would like to take a moment to introduce the collaborative use of the three modules. This process has been made straightforward, as we have provided example files for each module and have ensured their compatibility from the outset. Specifically, both LipidIN (<https://github.com/LinShuhaiLAB/LipidIN>) and QuanFormer (<https://github.com/LinShuhaiLAB/QuanFormer>) include demo videos to facilitate understanding.

In Fig. R8, we outline the workflow. After obtaining results from LipidIN, users can easily input the file containing the m/z and retention time into QuanFormer for rapid and accurate quantitative analysis. Following the acquisition of both quantitative and

qualitative results, users can organize the data format in accordance with the Common omics analysis example file. This data can then be analyzed through the visual interface. Currently, the Common omics analysis supports various analyses, including simple statistical assessments, volcano plots, PCA analysis, heatmaps, and correlation matrix plots.

Fig. R8 Workflow for integrating LipidIN, QuanFormer, and Common omics analysis. a. A concise description of the essential information required for each module.

Comment 3. The organization of the files for the reviewer is very poor, for example in the merged document Figures don't have numbers and are not contained with their captions, making it very hard to understand what is happening. Also it would be nice to find each supplementary table with a linked index as they seemed to just be dropped into the submission.

Response: We sincerely apologize for any difficulties you encountered during the review process. We want to clarify that our primary concern was the clarity of the images when embedded in the Word document, which led us to separate the images from their numbering and legends. We regret any confusion this may have caused.

In response to your valuable feedback, we have now added serial numbers and figure titles for each image in the revised version. Additionally, we will incorporate links to the supporting tables in the relevant sections to facilitate easier navigation. Thank you for your understanding and support.

Comment 4. While this manuscript ideally would be of interest to the lipidomics community, the language throughout is very dense (computer programming and model jargon heavy), without much explanation of how each aspect relates to a lipid expert terminology or workflow. This was a very hard read.

Response: Thank you for your professional advice. We have addressed the terms in the article that may be challenging for readers to understand by incorporating them into a terminology explanation table, as shown in Table R4. We hope this will help readers gain a more comprehensive understanding of the concepts and algorithms presented in the article. We also revised manuscript in Line 129, 225, 267 and Table. S11

Table R4. Terminology and Definitions Summary

Terminology and Definitions	Details
Bias Matrix	Similar to the weight matrix, the bias matrix is a set of parameters added to the weighted sum of inputs before passing the result through an activation function. Biases allow the network to shift the activation function and provide more flexibility, enabling the model to learn patterns that aren't strictly tied to zero-centered data. In most cases, each neuron has its own bias.
Confidence interval	Confidence Interval (CI) is a statistical concept that provides a range of values within which we expect a population parameter (such as a mean or proportion) to lie, based on a sample of data. It is a tool used to express the uncertainty or variability associated with a sample estimate.
Dichotomy / binary search	Dichotomy, or binary search, is an efficient algorithm for finding a target value within a sorted array. It works by repeatedly dividing the search interval in half. If the target value is less than the middle element, the search continues in the lower half; if it's greater, the search continues in the upper half. This process continues until the target value is found or the interval is empty. The time complexity of dichotomy is $O(\log n)$, making it much faster than a linear search, especially for large datasets.
Downsampling and Upsampling (U-Net Concept)	These processes modify the resolution of the data to improve model predictions. Downsampling reduces the resolution to simplify data and reduce noise, while upsampling increases the resolution to better capture fine details. U-Net is a deep learning model commonly used for these tasks, especially in image segmentation, and is adapted here for lipid fingerprint regeneration.
Encoder with multi-head self-attention	An encoder is a type of neural network layer used to process input data, especially in transformer models. Multi-head self-attention allows the model to focus on different parts of the input data simultaneously, which helps in learning complex relationships in lipid spectra for better prediction and feature extraction.
Expeditious querying	The EQ module is a fast theoretical spectra search tool capable of performing millions of searches per second. It optimizes data structures using methods such as dichotomy and hash tables. By employing these methods, the EQ module enhances its search efficiency, enabling rapid and accurate spectral matching.
Feature Learning	Feature learning refers to the process where a model automatically discovers the representations (features) that are most useful for a given task, usually through unsupervised or semi-supervised learning techniques. In deep learning, this process often happens in layers, with each layer learning increasingly abstract features from the raw input data. The goal is to enable the model to extract meaningful patterns or structures without the need for manual feature engineering.
Feed-forward Network	A feed-forward network is a type of artificial neural network where the connections between the nodes (neurons) do not form cycles. Information flows in one direction from input to output through intermediate hidden layers, without any feedback loops. This is the most basic type of neural network, used for tasks

	like classification and regression.
Fitting Parameters	Fitting parameters are the values or coefficients that are learned by a model during the training process. These parameters define the relationship between input features and output predictions. In machine learning, fitting parameters are optimized to minimize the error between the predicted outputs and the true labels in the training data. Examples of fitting parameters include weights and biases in neural networks.
Fragment fingerprints / fingerprint spectra	Fingerprint spectra refer to unique mass spectrometry profiles that capture specific features of lipid substances, in addition to the common characteristic spectra shared across various mass spectrometry systems. Unlike the widely recognized peaks that represent general lipid characteristics, fingerprint spectra include distinctive peaks that may be present in individual mass spectrometry systems. These unique peaks often exhibit lower abundance compared to the more dominant features but carry significant meaning. They can provide crucial information about specific lipid species, their structural variations, or their interactions within biological systems. Therefore, analyzing these fingerprint spectra allows researchers to gain deeper insights into the complexity of lipid compositions and their functional roles.
Hash Table	A hash table is a data structure that implements an associative array, allowing for fast data retrieval. It uses a hash function to compute an index (or hash code) into an array of buckets or slots, from which the desired value can be found. This allows for average-case time complexity of $O(1)$ for insertions, deletions, and lookups, making hash tables very efficient for scenarios where quick data access is essential. However, performance can degrade if many elements hash to the same index (collisions), which can be managed through techniques like chaining or open addressing.
Hierarchical library	The 5-level hierarchical library is a mass spectrometry database categorized by the researcher based on the analytical significance of each MS2 characteristic peak. The specific meaning of each level is detailed in the article.
Intraclass relative retention time	Refers to the variation in retention time within a group after the annotation results have been categorized in two ways: by the same subcategory with identical total unsaturation or by the same subcategory with the same combination of sub-unsaturation. Once categorized, the relative change in retention time within each category will be modeled.
Intra-feature learning	A learning process where a model focuses on understanding the relationships within individual features (e.g., mass-to-charge ratios and intensities of lipids in mass spectrometry) before considering how they relate to each other across different features. It helps in extracting useful information from each feature independently.
Latent Matrix	A latent matrix is a matrix that represents hidden or unobserved features in a model, often learned during the training process. In the context of dimensionality reduction or matrix factorization, the latent matrix captures the underlying factors or structures in the data that explain its variation. In some contexts, like collaborative filtering or autoencoders, the latent matrix can represent

	compressed features of the input data.
Linear time complexity	Linear time complexity refers to an algorithm whose runtime grows proportionally to the size of the input data. It is commonly denoted as $O(n)$, where n represents the size of the input.
Lipid categories intelligence	In this paper, we present a search algorithm designed to ensure that the retention times of annotated results adhere to the three rules: ECN, ESCN, and IUP.
Multi-simulation model	A multi-simulation model refers to a simulation framework that operates under two specific scenarios: one based on total unsaturation and the other on sub-unsaturation composition. The primary goal of this model is to identify combinations of annotation results that best satisfy three specified rules.
Non-informative prior greedy algorithm	The Non-informative Prior Greedy Algorithm is a concept that combines elements from Bayesian statistics and optimization algorithms, particularly in the context of parameter estimation or model selection. Here's a breakdown of the key components. Here, we refer to the method of matching all theoretical spectra using MS2 and precursor m/z values, without considering retention time as prior information.
Platform Transfer	Platform transfer refers to the ability of the WMYn method to perform consistently across different experimental setups or mass spectrometry systems, ensuring that lipid annotation can be reliably transferred between different instruments (e.g., Orbitrap vs. qTOF MS systems).
Querying	Querying between spectra refers to the process of searching for and retrieving specific information from spectral data, which is often used in fields like mass spectrometry, analytical chemistry, and metabolomics.
Recall (Intersection Count of Tool and Article Annotations / Total Article Annotations)	The number of elements annotated both in the tool and published result, divided by the total number of compositions shown in the article. It is important to note that this requires the annotated results to have same composition annotations, which is a more in-depth annotation than C:DB, with a retention time error of less than 0.15 minutes.
Self-attention	A mechanism within machine learning models (especially transformers) where each part of the input data (e.g., a spectrum of lipid mass) attends to every other part of the data to capture dependencies and relationships. It's essential in processing sequences or data with complex internal structures.
Spectral Entropy Similarity	This is a method used to evaluate how similar two mass spectrometry spectra are by comparing their entropy. Entropy measures the amount of uncertainty or disorder in the data. In this case, spectral entropy similarity quantifies how closely a predicted spectrum matches the reference or experimentally obtained spectrum.
Squared time complexity	Squared time complexity refers to an algorithm whose runtime grows proportionally to the square of the size of the input data. It is commonly denoted as $O(n^2)$, where n^2 represents the size of the input.
Time	Time complexity is a computational concept that describes the amount of time

complexity	an algorithm takes to complete as a function of the length of the input. It provides a way to evaluate the efficiency of an algorithm in terms of its performance and scalability.
U-Net	A deep learning architecture commonly used for image segmentation tasks but adapted here for downsampling and upsampling tasks in lipidomics. The architecture features symmetric encoder-decoder structures, making it effective for regenerating or enhancing features like lipid fingerprints.
Weight Matrix	In neural networks, the weight matrix is a collection of parameters that define the strength of connections between neurons in adjacent layers. Each element in the weight matrix corresponds to the weight applied to the connection between two neurons. During training, these weights are updated using optimization algorithms (like gradient descent) to minimize the loss function and improve the network's performance.

Comment 5. The software was validated against MS-DIAL, LipidMatch, LipidSearch, etc. but the graphs and results of this comparison are also hard to understand. Label your axis and you're your captions explain things in a way where the reader does not need to read the text and/or supplemental to understand what is being compared and what recall, "percentage", etc. mean. Walking the reader through an application such as the NIST interlab study on SRM 1950 which has a set of "known lipids", and the percent coverage of each software of these lipids, false positive rate, false negative rate, and overlap between software would be very helpful. For example see Figure 6 of the Lipid Annotator paper (<https://pmc.ncbi.nlm.nih.gov/articles/PMC7142889/>) which shows shared coverage by class and C:DB. You could also show overlap by fatty acid composition as the top ranked candidate between software. You could also spike in a much large set (100s) of endogenous lipid standards and see the false positive, negative, and coverage of each software. LipiDex is another leading software worth comparing, although we know full coverage of all software may be out of the scope of this project.

Response: Thank you very much for your valuable advice; it has been incredibly helpful. To enhance clarity, we have provided more detailed labeling of the axes in the relevant sections of the article. This should enable readers to grasp the information more easily without the need for extensive reading. We have revised the "percentage" to "Intersection Count of Tool and Article Annotations / Total Article Annotations". While

this wording is more complex, it offers greater specificity. Furthermore, we have changed “Recall@X” to “Top@X” to provide a clearer representation of the annotation results reported by individual software and articles, reflecting the consistency of annotation outcomes as the ranking increases. Additionally, we have included detailed explanations for any ambiguous terms, such as “annotation”. The updated figures illustrating these modifications can be found following. We also revised manuscript in Line225, Fig.3, Supplementary Fig.5, Supplementary Fig.9 and Source Data Supplementary Fig.9.

Thank you for your valuable suggestion regarding the use of a more accurate lipid reference. After considering the 100 lipid standards in relation to the NIST SRM 19500, we believe that employing the NIST SRM 19500 provides a more efficient, comprehensive option with broader coverage for comparison. We located the original mass spectrometry file for the NIST SRM 19500 in the Metabolomics Workbench (ST003514)³⁸, which was generated using an Agilent 1290 Infinity II. Although this instrument differs from those used in our benchmark experiments, it serves as a valuable complement that effectively illustrates the platform-independent nature of LipidIN.

In our benchmarking experiments, we compared identifications using LipidIN, MS-DIAL (version 5.1), Entropy Search, and article annotations, along with the additional use of Lipid Hunter³⁹ and Lipid Annotator⁴⁰. We sincerely apologize for the omission of LipiDex2 from our comparison; unfortunately, its file format was not recognized, and despite our attempts to contact their team both at email and github, we did not receive a response. Similarly, LipidMatch was excluded due to issues with multiple files not being recognized. In our benchmark, MS-DIAL and Entropy Search employed self-contained peak extraction methods, while LipidIN utilized the RaMS package for peak extraction. MS-DIAL relied on self-contained spectral libraries, LipidIN used libraries 1-4, and Entropy Search leveraged the MS-DIAL release libraries. All MS1 tolerances were set to 0.01 Da (5 ppm), and MS2 tolerances were set to 0.025 Da (10 ppm). The results of the comparison are displayed in Fig. R9 d-g.

In this benchmark experiment, we focused on the Top@1 results. LipidIN demonstrated the highest total number of annotations, totaling approximately 450. This

achievement reflects deeper identification of fatty acid (FA) compositions. However, we noted that while all methods identified a significant number of phosphatidylcholines (PCs) and sphingomyelins (SMs), they often lacked precise FA information and were therefore excluded from this analysis.

In the context of annotated intersections, LipidIN not only achieved the highest number of intersections but also identified lipids that were not annotated by other tools within the Top@1 category, thereby demonstrating its robust functionality. Notably, only a small subset of lipids (30) remained unannotated by LipidIN but were identified by at least two other tools. We undertook a comprehensive analysis of these lipids to ascertain the reasons for their omission by LipidIN.

Among these, three lipids were correctly annotated. One in particular, PC 18:0_22:5, exhibited a notably low secondary fragmentation response concerning fatty acid chains (see Fig. R10.b). The other two, LPC 16:0 and CAR 11:0, were identified by LipidIN; however, their retention times deviated from the results obtained from MS-DIAL and relevant literature by 0.44 min and 0.6 min, respectively.

Additionally, five lipid peaks were not extracted in RaMS, which contributed to the lack of annotation by LipidIN. We also identified 19 lipids that, despite being reported by other software, were found upon examination to be lacking critical information regarding their headgroups, fatty acid chains, or precursor ions (refer to Fig. R10.a and Fig. R10.c). This deficiency raises concerns regarding the reliability of their identification. Finally, three of these lipids were incorrectly annotated, with EtherPE(P) mistakenly labeled as EtherPE(O). We have compiled more specific information in Table R3, which we hope will provide further clarity on this matter. We also revised manuscript in Line 359, Fig.3, Supplementary Fig.5, Supplementary Fig.9 and Source Data Supplementary Fig.9.

Fig. R9. Modified Figure Establishment of aging-associated lipidome atlas in mice and NIST SRM 1950. a. recall of the reported 2704 lipids in the MS-DIAL public database, 1- to 4- level hierarchical library, and 1- to 5- level library, respectively. **b.** Detail statistics of recall@cuttoff@threshold 1.8. The size of the circle reflects the

number of annotations in lipid subclass, and the color shade reflects the recall. Hierarchical library and reverse lipidomics demonstrated advantages in the annotation of various lipid subclasses. **c.** Comparison of Lipid Categories Intelligence Modeling (LCI) and Lipid Data Analyzer (LDA) for removal of false positive annotations in aging-associated lipidome atlas in mice. The SRM 1950 was analyzed using LipidIN, MS-DIAL, Entropy Search, and article annotations (Lipid Hunter + LipidAnnotator + manual checks). **d.** The number of Top@1 annotations for different fatty acid (FA) compositions by each method in positive ionization mode. **e.** The number of intersections of FA compositions annotated by different methods in positive ionization mode. **f.** The number of Top@1 annotations for different FA compositions by each method in negative ionization mode. **g.** The number of intersections of FA compositions annotated by different methods in negative ionization mode.

Fig. R10. Annotated Performance in NIST SRM 1950. **a.** The annotated results show the measurement spectrum (blue) and reference spectrum (red) of PE P-18:0_22:6, highlighting the absence of important head group characteristic fragments (m/z : 294) in the measurement spectrum. **b.** The annotation results depict the measurement spectrum (blue) and reference spectrum (red) of PC 18:0_22:5, revealing a low response intensity of the characteristic fragments in the measured spectrum relative to the FA composition. **c.** The results illustrate the measurement spectra (blue) and reference spectra (red) of CE 22:6, emphasizing the absence of precursor ion peaks and characteristic neutral loss fragments.

Table R5. Reasons for the Annotation Failures of LipidIN

Rt	m/z	Other tools annotations	Reasons for the Annotation Failures of LipidIN
11.15	776.562	PE(P-18:0/22:6)	Questionable annotations due to missing head group (m/z: 294.316)
9.95	836.615	PC(18:0/22:5)	Correctly annotated, but FA composition-related fragmentation response is particularly low (<1%), which filtered by LipidIN
2.89	480.308	LPE(0:0/18:1)	RaMS fail to obtain this peak
2.72	454.294	LPE(0:0/16:0)	Correctly annotated out under another peak, the absolute deviation in retention time was 0.44 min
14.23	714.617	CE(22:6)	Questionable annotations due to missing precursor ion peak fragments and some neutral loss fragments
14.57	690.619	CE(20:4)	Questionable annotations due to missing precursor ion peak fragments and some neutral loss fragments
15.04	692.632	CE(20:3)	Questionable annotations due to missing precursor ion peak fragments and some neutral loss fragments
14.41	664.603	CE(18:3)	Questionable annotations due to missing precursor ion peak fragments and some neutral loss fragments
14.98	666.620	CE(18:2)	Questionable annotations due to missing precursor ion peak fragments and some neutral loss fragments
0.87	302.233	CAR(9:0)	RaMS package fail to obtain this peak
2.59	426.361	CAR(18:1)	RaMS package fail to obtain this peak
0.99	356.279	CAR(13:1)	RaMS package fail to obtain this peak
0.99	330.264	CAR(11:0)	Correctly annotated out under another peak, the absolute deviation in retention time was 0.6 min
7.80	832.584	PC(20:3/20:4)	Questionable annotation, without highly responsive FA compositional information in the MS2 spectra
12.25	812.698	HexCer(d18:1/24:0)	Questionable annotation, without highly responsive FA compositional information in the MS2 spectra
10.88	812.616	PC(18:0_20:3)	Questionable annotation, without highly responsive FA compositional information in the MS3 spectra
9.72	810.601	PC(18:0/20:4)	Questionable annotation, without highly

			responsive FA compositional information in the MS4 spectra
10.23	786.602	PC(18:1/18:1)	Questionable annotation, without highly responsive FA compositional information in the MS5 spectra
10.23	786.602	PC(18:0/18:2)	Questionable annotation, without highly responsive FA compositional information in the MS6 spectra
8.47	784.586	PC(18:1/18:2)	Questionable annotation, without highly responsive FA compositional information in the MS7 spectra
9.14	784.586	PC(16:0_20:3)b	Questionable annotation, without highly responsive FA compositional information in the MS8 spectra
7.08	782.570	PC(18:2/18:2)	Questionable annotation, without highly responsive FA compositional information in the MS9 spectra
6.70	780.553	PC(16:1/20:4)	Questionable annotation, without highly responsive FA compositional information in the MS10 spectra
9.63	760.585	PC(16:0/18:1)	Questionable annotation, without highly responsive FA compositional information in the MS11 spectra
11.50	752.559	PE(O-18:1/20:4)	Wrong annotation, should be EtherPE(P) category
9.17	734.570	PC(16:0/16:0)	Questionable annotation, without highly responsive FA compositional information in the MS11 spectra
7.54	732.548	PC(16:0_16:1)	Questionable annotation, without highly responsive FA compositional information in the MS11 spectra
11.56	728.559	PE(O-18:1/18:2)	Wrong annotation, should be EtherPE(P) category
9.06	724.526	PE(O-16:1/20:4)	Wrong annotation, should be EtherPE(P) category
5.31	337.313	FA(22:1)	RaMS fail to obtain this peak

Comment 6. Comparison between software is very challenging. How were parameters kept the same? Were different peak picking algorithms used? (Note certain software only work with their own peak picking algorithms). Where were the fault points in each where annotations were missed? Etc.

Response: Thank you for your suggestion. As you noted, comparing software can be challenging, and we strive to keep parameters as consistent as possible. In the benchmark section of the article (Line 674), we provide detailed information about the computer configuration used, including the number of cores, RAM, operating system, and more. We also indicate the version and download links for all software utilized.

Additionally, we have specified the tolerances for MS1 and MS2. However, due to the variability among the tools (e.g., some accept Da, while others accept ppm), we have done our best to ensure consistency. Unfortunately, since most software relies on their own peak lifting algorithms, we cannot guarantee that the peak lifting results from MS-DIAL, LipidSearch, and LipidMatch will match those of other tools.

Given that the benchmark experiments were conducted using a Thermo instrument, we believe that LipidSearch demonstrates superior peak lifting performance, even though it may not achieve the highest recall (recall is defined as the number of elements annotated both in the tool and published result, divided by the total number of compositions shown in the article. It is important to note that this requires the annotated results to have same composition annotations, which is a more in-depth annotation than C:DB, with a retention time error of less than 0.15 minutes). We also revised manuscript in Line 217 and Table. S1.

Table. R6. Parameters and peak picking algorithms utilized by various Tools.

Tool Names	MS1 tolerance	MS2 tolerance	peak picking algorithms / packages	Version / Download data	other parameters
LipidIN	5 ppm / 10 ppm	10 ppm /20 ppm	RaMS	-	MS2_fliter: 0.05 Other Parameters Set to Default
MS-DIAL	0.01 Da	0.025 Da	Model-based peak detection	V 5.1.23091 2	Other Parameters Set to Default
LipidMath	0.01 Da	0.025 Da	No peak detection	2.0.2	Other Parameters Set to Default
LipidSearch	5 ppm	10 ppm	Self-peak detection	V 4.2	Other Parameters Set to Default
Flash entropy	0.01 Da	0.025 Da	No peak detection	2023-9-23	Other Parameters Set to Default
Entropy Search	0.01 Da	0.025 Da	No peak detection	2024-5-3	Other Parameters Set to Default

Comment 7. While EAD is one of the highlights of this paper, it is only briefly mentioned and the number of new lipids with different double bond positions, etc. which can be discovered in well known datasets are not well described.

Response: Thank you for your interest in the LipidIN identification of EAD spectra. As you noted, EAD is indeed a highlight of this paper. We have supplemented Fig. R11 to illustrate the effect of LipidIN annotation alongside MS-DIAL (version 5.1) annotation of triglycerides (TG). For the C:DB annotations, we have included results for all TGs identified by MS-DIAL, effectively doubling the number of annotations. Regarding the deeper compositional annotations at Top@1, there was only one lipid identified by MS-DIAL that we did not identify. Upon examining the spectrogram in Fig. R12, we believe this to be a misidentification, as many peaks containing critical information about the position of the double bond are missing. Furthermore, when comparing the double key position annotation results from the two tools, LipidIN includes all correctly annotated entries from MS-DIAL at Top@3. Lastly, we would like to clarify why we did not present the EAD results in the article. Our analysis of the

333 clinical cohort data did not reveal any significant differential lipids, as shown in Fig. R11.d. We also revised manuscript in Line 409, Supplementary Fig.14 and Source Data Supplementary Fig.14.

Fig. R.11. Annotated Performance and Clinical Case Study of LipidIN on the Zeno TOF 7600 System. a. Annotated intersections between LipidIN and MS-DIAL at the C:DB level. **b.** Annotated intersections between LipidIN and MS-DIAL at the fatty acid composition level. **c.** Annotated intersections between LipidIN and MS-DIAL at the C=C position level. **d.** Analysis of differences in clinical data across 333 cases at specific double-bonded locations; BC indicates breast cancer, while HD indicates healthy donor.

Fig. R.12. Reference (blue) and measured (red) spectra of TG 16:0_16:1(9)_18:2(9,12). We can observe that the fragmentation response in the spectrogram with respect to the information about the double bond position is so low as to be almost invisible, so the annotation of this double bond position is doubtful.

Table R7. EAD annotations.

m/z	Rt (min)	Score	Top rank	sn 2	DBs position	Adduction
911.802	10.57	3.00	Top1	18:1	TG 18:0_18:0_18:1	[M+Na] ⁺
935.805	10.30	4.79	Top3	18:1(12)	TG 18:1(12)_18:1(12)_20:1(9)	[M+Na] ⁺
935.805	10.30	4.79	Top3	18:1(6)	TG 18:1(12)_18:1(6)_20:1(9)	[M+Na] ⁺
935.805	10.30	4.79	Top3	18:1(6)	TG 18:1(6)_18:1(6)_20:1(9)	[M+Na] ⁺
935.805	10.30	4.93	Top2	18:1(9)	TG 18:1(12)_18:1(9)_20:1(9)	[M+Na] ⁺
935.805	10.30	4.93	Top2	18:1(6)	TG 18:1(6)_18:1(9)_20:1(9)	[M+Na] ⁺
935.805	10.30	5.07	Top1	18:1(9)	TG 18:1(9)_18:1(9)_20:1(9)	[M+Na] ⁺
883.772	10.23	3.01	Top2	18:1	TG 16:0_18:0_18:1	[M+Na] ⁺
883.772	10.23	3.08	Top1	18:1(12)	TG 16:0_18:0_18:1(12)	[M+Na] ⁺
883.772	10.23	3.08	Top1	18:1(9)	TG 16:0_18:0_18:1(9)	[M+Na] ⁺
883.772	10.23	3.08	Top1	18:1(6)	TG 16:0_18:0_18:1(6)	[M+Na] ⁺
878.818	10.23	2.13	Top3	20:1(9)	TG 16:0_16:0_20:1(9)	[M+NH ₄] ⁺
857.755	10.20	3.00	Top1	16:0	TG 16:0_16:0_18:0	[M+Na] ⁺
907.771	9.99	3.01	Top1	18:1	TG 18:1_18:1_18:1	[M+Na] ⁺
855.741	9.92	3.23	Top3	16:0	TG 16:0_16:0_18:1(9)	[M+Na] ⁺
855.741	9.92	3.23	Top3	16:0	TG 16:0_16:0_18:1(12)	[M+Na] ⁺
855.741	9.92	3.26	Top2	16:0	TG 16:0_16:1(9)_18:0	[M+Na] ⁺
855.741	9.92	3.30	Top1	16:0	TG 16:0_16:0_18:1(6)	[M+Na] ⁺
881.756	9.89	3.00	Top3	18:1	TG 16:0_18:1_18:1	[M+Na] ⁺
881.756	9.89	3.08	Top2	18:1(6)	TG 16:0_18:1(6)_18:1	[M+Na] ⁺
881.756	9.89	3.08	Top2	18:1(9)	TG 16:0_18:1(9)_18:1	[M+Na] ⁺
881.756	9.89	3.08	Top2	18:1(12)	TG 16:0_18:1(12)_18:1	[M+Na] ⁺
881.756	9.89	3.15	Top1	18:1(6)	TG 16:0_18:1(6)_18:1(6)	[M+Na] ⁺
881.756	9.89	3.15	Top1	18:1(9)	TG 16:0_18:1(6)_18:1(9)	[M+Na] ⁺
881.756	9.89	3.15	Top1	18:1(12)	TG 16:0_18:1(12)_18:1(6)	[M+Na] ⁺
881.756	9.89	3.15	Top1	18:1(9)	TG 16:0_18:1(9)_18:1(9)	[M+Na] ⁺
881.756	9.89	3.15	Top1	18:1(12)	TG 16:0_18:1(12)_18:1(9)	[M+Na] ⁺

881.756	9.89	3.15	Top1	18:1(12)	TG 16:0_18:1(12)_18:1(12)	[M+Na] ⁺
900.802	9.64	2.22	Top3	18:2(9,12)	TG 18:1(9)_18:1(9)_18:2(9,12)	[M+NH4] ⁺
900.802	9.64	2.22	Top3	18:2(9,12)	TG 18:1(6)_18:1(9)_18:2(9,12)	[M+NH4] ⁺
900.802	9.64	2.22	Top3	18:2(9,12)	TG 18:1(6)_18:1(6)_18:2(9,12)	[M+NH4] ⁺
900.802	9.64	2.22	Top2	18:2(9,12)	TG 18:1(12)_18:1(9)_18:2(9,12)	[M+NH4] ⁺
900.802	9.64	2.22	Top2	18:2(9,12)	TG 18:1(12)_18:1(6)_18:2(9,12)	[M+NH4] ⁺
900.802	9.64	2.22	Top1	18:2(9,12)	TG 18:1(12)_18:1(12)_18:2(9,12)	[M+NH4] ⁺
879.741	9.59	3.01	Top1	18:1	TG 16:0_18:1_18:2	[M+Na] ⁺
853.724	9.47	3.08	Top3	16:1	TG 16:0_16:1_18:1(12)	[M+Na] ⁺
853.724	9.47	3.08	Top3	16:1	TG 16:0_16:1_18:1(9)	[M+Na] ⁺
853.724	9.47	3.08	Top3	16:1	TG 16:0_16:1_18:1(6)	[M+Na] ⁺
853.724	9.47	3.09	Top2	16:1(9)	TG 16:0_16:1(9)_18:1	[M+Na] ⁺
853.724	9.47	3.16	Top1	16:1(9)	TG 16:0_16:1(9)_18:1(12)	[M+Na] ⁺
853.724	9.47	3.16	Top1	16:1(9)	TG 16:0_16:1(9)_18:1(9)	[M+Na] ⁺
853.724	9.47	3.16	Top1	16:1(9)	TG 16:0_16:1(9)_18:1(6)	[M+Na] ⁺
827.708	9.47	2.26	Top3	16:1(9)	TG 16:0_16:0_16:1(9)	[M+Na] ⁺
827.708	9.47	2.27	Top2	20:1(9)	TG 14:0_14:0_20:1(9)	[M+Na] ⁺
827.708	9.47	2.30	Top1	18:1(9)	TG 14:0_16:0_18:1(9)	[M+Na] ⁺
898.786	9.33	2.47	Top3	18:2(9,12)	TG 18:1_18:2(9,12)_18:2(9,12)	[M+NH4] ⁺
898.786	9.33	2.54	Top2	18:2(9,12)	TG 18:1(9)_18:2(9,12)_18:2(9,12)	[M+NH4] ⁺
898.786	9.33	2.54	Top2	18:2(9,12)	TG 18:1(6)_18:2(9,12)_18:2(9,12)	[M+NH4] ⁺
898.786	9.33	2.61	Top1	18:2(9,12)	TG 18:1(12)_18:2(9,12)_18:2(9,12)	[M+NH4] ⁺
877.724	9.26	3.00	Top3	18:2	TG 16:0_18:2(6,9)_18:2(6,9)	[M+Na] ⁺
877.724	9.26	3.24	Top2	18:2(9,12)	TG 16:0_18:2(9,12)_18:2(6,9)	[M+Na] ⁺
877.724	9.26	3.47	Top1	18:2(9,12)	TG 16:0_18:2(9,12)_18:2(9,12)	[M+Na] ⁺

851.708	9.16	3.25	Top3	16:1(9)	TG 16:0_16:1(9)_18:2(9,12)	[M+Na] ⁺
851.708	9.16	3.25	Top2	16:1(9)	TG 16:1(9)_16:1(9)_18:1(9)	[M+Na] ⁺
851.708	9.16	3.25	Top2	16:1(9)	TG 16:1(9)_16:1(9)_18:1(6)	[M+Na] ⁺
851.708	9.16	3.32	Top1	16:1(9)	TG 16:1(9)_16:1(9)_18:1(12)	[M+Na] ⁺
904.831	9.10	2.16	Top1	18:2(9,12)	TG 18:0_18:0_18:2(9,12)	[M+NH ₄] ⁺
799.677	9.02	3.00	Top2	14:0	TG 14:0_14:0_18:1	[M+Na] ⁺
799.677	9.02	3.08	Top1	14:0	TG 14:0_14:0_18:1(6)	[M+Na] ⁺
896.770	8.99	2.23	Top3	18:1(12)	TG 18:1(12)_18:2(9,12)_18:3	[M+NH ₄] ⁺
896.770	8.99	2.31	Top2	18:2(9,12)	TG 18:2(9,12)_18:2(9,12)_18:2	[M+NH ₄] ⁺
896.770	8.99	2.47	Top1	18:2(9,12)	TG 18:2(9,12)_18:2(9,12)_18:2(9,12)	[M+NH ₄] ⁺
849.693	8.82	3.01	Top3	18:2	TG 14:0_18:2_18:2	[M+Na] ⁺
849.693	8.82	3.08	Top2	18:2(9,12)	TG 14:0_18:2(9,12)_18:2	[M+Na] ⁺
849.693	8.82	3.16	Top1	18:2(9,12)	TG 14:0_18:2(9,12)_18:2(9,12)	[M+Na] ⁺
847.677	8.49	2.16	Top2	18:2(9,12)	TG 14:0_18:2(9,12)_18:3	[M+Na] ⁺
847.677	8.49	2.41	Top1	18:2(9,12)	TG 14:0_18:2(9,12)_18:3(9,12,15)	[M+Na] ⁺

Table R8. Quantitative results of EAD.

Sample ID	TG 52:2 18:1(9)	TG 52:2 18:1(6)	TG 52:4 18:1(12)	TG 52:4 18:1(9)	TG 52:4 18:2(6,9)	TG 52:4 18:2(9,12)
BC-001	8471.57	6280.84	24278.50	4181.86	4493.99	4734.24
BC-002	10242.04	5784.07	23588.28	3226.58	3346.08	4439.88
BC-003	8892.19	7245.32	18040.02	3263.17	2793.22	4719.97
BC-004	10235.09	7489.59	22162.50	4424.42	4441.66	5372.17
BC-005	10429.89	7329.11	21085.27	3690.57	3876.38	4396.91
BC-006	7847.92	6989.12	20836.63	3450.52	3128.13	4223.59
BC-007	9031.10	6707.54	21205.90	3633.38	3986.91	6270.27
BC-008	8167.64	7346.04	21999.42	3809.50	2763.80	4110.63
BC-009	8110.13	6362.16	18152.34	3958.68	3193.64	4799.96
BC-010	9875.96	8576.05	23029.44	6562.27	3953.75	6478.96
BC-011	9103.34	7145.87	18101.72	1944.61	3187.79	4449.03
BC-012	9624.47	6483.62	24768.54	3624.03	3922.13	6998.27
BC-013	8912.43	5929.05	21390.01	3095.13	3544.03	6098.60
BC-014	11088.59	10393.24	29103.11	6686.22	4235.70	6814.49
BC-015	8723.78	6456.86	24788.18	4754.76	3323.97	4795.96
BC-016	8521.38	7178.48	26162.93	5062.24	4140.72	6073.50
BC-017	8797.83	6342.62	22917.28	3129.56	3925.90	3539.84
BC-018	9951.71	7127.37	15410.49	5032.73	1226.55	3544.14
BC-019	9582.76	6959.09	16466.23	5812.09	3256.83	4118.29
BC-020	9761.81	7796.49	24182.38	3734.54	4504.91	5720.10
BC-021	10987.80	9302.35	23978.89	2455.78	4557.34	4765.35
BC-022	8216.14	7020.67	19972.73	3024.33	3831.42	4104.97
BC-023	7449.29	5338.08	14341.70	2282.55	2837.50	3258.74
BC-024	10198.87	8313.62	24261.24	5264.96	3618.79	5051.62
BC-025	8161.15	6316.75	19149.13	3528.78	3419.06	3657.30
BC-026	8785.78	7385.08	19477.00	5636.05	2701.94	3911.54
BC-027	7947.16	5554.21	20865.82	3383.86	2923.42	4270.43
BC-028	6377.45	5807.75	23303.17	2176.56	3806.56	4302.59
BC-029	9176.11	6130.80	18324.92	2372.43	3027.63	3809.78
BC-030	8745.07	5838.73	19540.04	4000.35	3290.41	4549.12
BC-031	8274.60	8174.38	18108.29	4102.53	3820.38	4474.66
BC-032	7196.89	6325.84	21844.51	2629.84	4167.36	3602.67
BC-033	7166.59	6173.25	19849.10	1649.05	3323.95	3777.66
BC-034	7664.36	5901.01	17137.67	2558.12	3248.42	3336.38
BC-035	8360.87	6910.15	16826.61	4196.85	3401.98	3910.36
BC-036	7657.35	6689.88	20385.97	3562.89	3898.44	4164.69
BC-037	5268.37	5252.36	15425.02	3980.77	2564.41	3781.75
BC-038	7461.89	5406.08	21155.73	2760.33	3797.88	5730.41
BC-039	7709.30	5303.28	16529.90	3958.16	3490.28	4379.48
BC-040	9233.01	6519.80	25076.76	3755.46	3792.77	5138.38
BC-041	6106.23	4427.97	18820.59	2566.51	3538.15	2702.97

BC-042	8401.97	7640.27	22237.32	5006.91	3911.31	3745.79
BC-043	6704.89	6704.89	20490.74	3425.06	3140.70	5456.29
BC-044	7581.53	6505.74	19480.45	2939.90	3192.75	3519.37
BC-045	5842.58	7228.27	16997.26	3575.47	3282.02	4509.73
BC-046	7401.83	6693.54	18425.31	3587.94	2246.54	3125.84
BC-047	8305.15	6325.98	18461.39	3870.50	3876.55	4509.49
BC-048	6233.67	5923.04	19351.51	3887.31	3311.05	3808.19
BC-049	7229.11	4745.33	19480.13	1857.76	2969.29	3518.56
BC-050	8941.95	5096.93	22838.10	2056.13	3149.64	3873.71
HD-001	11314.49	6335.96	16855.28	3673.50	3499.87	4296.10
HD-002	7680.17	7160.86	25759.68	3874.01	4580.97	5030.43
HD-003	7002.38	5542.40	17067.43	2923.72	4064.85	3560.78
HD-004	8930.37	6855.45	20682.16	4294.81	2571.59	5672.74
HD-005	9701.00	6765.19	22166.84	2779.70	4162.83	3512.20
HD-006	8873.38	6757.27	26697.79	3997.18	4269.63	4597.34
HD-007	9277.15	8551.17	21283.66	3620.73	3041.77	3232.49
HD-008	8851.76	6582.20	20707.45	4967.86	3134.15	4745.78
HD-009	9582.06	6235.76	21733.24	3949.04	3886.21	5501.68
HD-010	8353.92	6315.10	18086.51	2819.71	3040.04	3408.72
HD-011	10688.10	7878.50	19138.07	3808.35	3482.45	4344.33
HD-012	9563.96	7716.23	20716.10	5971.67	3583.70	4939.05
HD-013	9600.93	5945.07	21513.16	2682.62	4753.49	5532.59
HD-014	6194.88	6395.42	16950.88	3288.94	2978.24	4147.63
HD-015	8800.63	7808.84	23270.62	4721.92	3991.00	5399.66
HD-016	10471.28	9361.23	23221.78	3836.75	3481.50	4766.30
HD-017	9987.92	7281.55	21236.39	5459.15	2665.56	6046.41
HD-018	8537.92	6328.48	17039.14	3281.52	2857.47	3615.95
HD-019	8671.63	6381.01	25583.78	2404.44	4667.08	4726.75
HD-020	7062.14	5378.64	25283.57	2200.89	4970.01	4244.66
HD-021	9859.56	7521.71	18367.67	4098.14	2649.44	4215.17
HD-022	8538.78	7555.31	21420.88	3381.73	3724.22	4714.93
HD-023	9954.66	5784.32	19803.40	3411.94	3578.46	5350.47
HD-024	7198.31	5746.32	23508.37	2238.97	3888.48	4796.10
HD-025	7097.06	4775.86	19597.20	3693.13	4004.55	3052.21
HD-026	8649.93	6275.21	16867.00	4066.88	3833.60	3160.77
HD-027	4978.26	3186.13	15009.12	2242.28	1543.52	3267.82
HD-028	11217.73	7920.47	21028.18	5490.24	3595.74	4976.72
HD-029	8982.27	6342.00	21245.42	3314.08	2422.77	4073.16
HD-030	2752.30	1633.52	15380.13	1496.11	2035.71	3448.15
HD-031	7500.72	4756.39	14249.31	2209.81	2910.61	3468.54
HD-032	7565.05	6535.63	18420.90	3260.17	3282.54	3722.84
HD-033	7112.49	6309.92	19576.10	3180.83	3979.55	3880.68
HD-034	9708.12	7094.71	24708.15	4104.06	3411.46	6345.74
HD-035	6676.83	6574.12	20498.93	3625.45	4259.14	3869.91

HD-036	6895.73	6843.95	22336.03	3007.31	3140.82	4481.78
HD-037	10844.10	7841.83	20376.51	6017.71	3675.42	6250.10
HD-038	7814.06	5646.82	21102.11	2407.16	3628.56	5368.27
HD-039	9454.44	7909.26	19646.75	4349.78	3235.81	3675.48
HD-040	6860.37	7690.91	17413.67	4204.40	3888.42	4988.82
HD-041	8266.96	6539.74	19940.82	2638.99	3034.96	2717.55
HD-042	8879.64	7731.56	22386.61	4051.73	3281.84	4086.56
HD-043	6000.64	4977.03	16502.84	3626.67	2883.63	3074.64
HD-044	9359.87	8453.00	20438.98	3708.35	4201.82	5616.00
HD-045	8582.50	6265.01	21132.60	3804.76	3249.29	3701.49
HD-046	4374.92	5010.60	17440.11	1423.91	2210.55	2264.27
HD-047	9369.54	7470.03	22338.10	5900.32	2993.34	6198.23
HD-048	7651.54	5516.85	20478.75	2926.42	3120.24	2881.41
HD-049	6419.03	5237.98	17141.81	2091.41	3341.82	3609.10
HD-050	8812.77	7802.60	18526.90	4120.19	3054.52	3898.01
QC-01	4995.87	3690.30	14486.55	1433.95	1720.17	1894.04
QC-02	4565.75	3028.53	12184.05	1574.18	1832.89	1879.62
QC-03	6306.15	4211.46	13853.73	1602.29	1432.89	2206.52
QC-04	4216.15	3337.91	11673.49	1793.18	1498.09	1581.86
QC-05	5683.80	3863.10	11981.58	1610.65	1710.17	1585.33
QC-06	4612.30	3372.22	12036.62	1474.36	1507.04	1985.43
QC-07	4574.54	3345.20	12126.06	1427.63	1672.81	2028.59
QC-08	4441.27	3738.50	11012.57	1747.53	1673.80	1757.70
QC-09	5319.55	3860.44	10367.01	1277.10	1787.07	2156.15
QC-10	4184.98	3215.34	10030.77	1580.18	1455.28	1440.19
BC-051	10059.36	7346.55	16696.90	3644.20	2391.36	2596.42
BC-052	8615.83	6218.12	23211.07	3590.63	3532.98	5860.13
BC-053	6997.65	5755.58	20695.23	3377.11	3307.58	4061.85
BC-054	8201.36	7293.76	16792.40	4019.56	2701.46	2972.25
BC-055	5356.38	4883.75	15975.34	1946.60	2746.47	3575.35
BC-056	8740.24	6412.44	17388.89	3283.86	3146.86	3438.77
BC-057	4495.84	4208.22	13714.21	2127.65	1532.72	2139.14
BC-058	7867.79	4863.93	16599.21	3681.53	3144.52	3196.22
BC-059	6671.38	5398.27	14592.15	2630.78	2727.72	2535.30
BC-060	8088.10	4804.83	19178.83	3929.59	3506.74	3438.14
BC-061	5922.30	4451.46	17427.62	1459.96	2580.26	3047.45
BC-062	8231.92	5715.46	17468.54	2281.15	2912.11	3138.81
BC-063	7678.00	6029.11	20496.77	3698.77	3915.44	3479.33
BC-064	7071.53	4839.08	14458.29	1838.48	2260.59	2917.82
BC-065	6880.69	5075.86	17980.47	3621.51	3044.65	4021.34
BC-066	9681.55	6865.67	21121.01	3182.14	2897.09	4752.09
BC-067	5789.24	5502.49	18592.52	1777.21	3276.06	4283.22
BC-068	6961.35	6829.77	19332.07	3045.75	3463.59	3796.56
BC-069	7522.62	4806.04	15318.47	3090.78	3224.62	3345.24

BC-070	8152.98	7202.49	24278.51	3360.88	3928.28	5978.21
BC-071	7032.15	5692.87	18643.44	3017.25	3392.07	3949.98
BC-072	8113.44	7035.61	19739.27	4477.97	3398.14	4348.24
BC-073	7653.77	6167.42	16176.36	2441.99	2906.77	4077.60
BC-074	7588.94	5686.73	21701.04	5175.99	3559.72	4201.81
BC-075	7601.86	6691.43	18971.21	2886.18	3096.11	3853.83
BC-076	7602.65	6211.33	19263.73	4113.26	2778.54	5195.14
BC-077	9606.15	6635.24	18467.21	3691.79	2742.51	4044.15
BC-078	8732.60	6250.94	23275.26	2047.35	4300.13	4622.20
BC-079	9644.43	6968.60	21993.40	3842.20	3727.84	3548.83
BC-080	9646.34	5760.91	19251.75	3004.16	3409.63	4164.32
BC-081	7067.48	6313.26	19272.57	2626.83	3259.87	3381.22
BC-082	8120.18	5533.47	19658.61	3164.66	2419.92	5235.25
BC-083	7923.89	5375.95	17248.39	3230.16	2349.93	2908.10
BC-084	6577.45	4860.55	14842.63	2714.72	2342.64	2222.52
BC-085	6819.96	4816.39	16095.70	3650.08	1755.34	3775.44
BC-086	5579.19	4966.17	14718.55	2612.42	2600.01	1798.37
BC-087	6325.39	5387.98	18386.04	2518.59	3951.85	3485.14
BC-088	5824.48	6302.28	16913.35	2584.70	3007.50	3301.22
BC-089	6851.27	5693.85	21634.72	4847.78	3231.20	5809.86
BC-090	6768.28	6455.55	17729.87	3261.70	3536.60	3546.14
BC-091	5913.09	5500.30	16087.41	2430.82	3540.85	3077.22
BC-092	7479.73	6090.12	20075.03	3669.95	3705.62	3847.43
BC-093	4780.87	5244.62	12212.00	1905.97	2078.99	2388.16
BC-094	5324.98	5529.97	15818.41	3628.72	2343.74	2824.71
BC-095	9375.68	7173.26	17593.78	3644.37	3297.01	5339.73
BC-096	6342.48	5325.69	16859.99	3094.74	2889.74	2803.08
BC-097	7021.72	6324.22	17675.82	2864.70	3021.52	4190.38
BC-098	5423.16	4306.58	15786.88	2343.83	1911.96	2162.07
BC-099	6574.02	4978.53	16061.62	5028.12	2748.06	2884.38
BC-100	5589.15	5943.53	20024.38	2810.12	2289.89	5073.03
HD-051	6215.51	5764.74	13044.16	3249.14	3730.37	2969.43
HD-052	6801.09	7254.03	17373.68	2356.74	2174.82	2487.81
HD-053	6000.39	6491.88	18037.80	3643.09	2281.17	3236.26
HD-054	8703.39	6382.71	20803.04	3492.65	3556.62	3283.54
HD-055	7936.35	5404.02	17679.28	3141.63	2267.97	3219.18
HD-056	8584.21	6386.61	17189.56	3337.87	2635.36	3891.31
HD-057	8850.52	6758.82	18119.28	4184.65	3261.93	5644.04
HD-058	5854.94	5781.38	15320.01	3103.21	2672.93	3880.62
HD-059	7132.03	5167.08	18270.99	2383.52	2475.84	1845.80
HD-060	5023.51	4253.29	13996.01	1443.41	1832.35	3101.37
HD-061	5755.75	4233.57	15796.89	1958.73	1987.46	2159.52
HD-062	7853.78	5660.44	17557.71	2621.35	2363.86	3031.16
HD-063	8432.51	8159.92	16856.61	4774.61	2722.94	4186.80

HD-064	6801.18	5553.48	18205.75	2734.10	1950.57	2950.19
HD-065	6477.49	5361.83	17871.41	3571.39	2459.25	2929.34
HD-066	7614.66	6696.99	18113.34	3962.17	2717.96	3911.91
HD-067	4754.52	3289.44	10448.62	1378.54	2138.56	1454.02
HD-068	10523.09	8104.02	20780.07	1760.30	3803.73	3940.05
HD-069	7371.48	6342.92	16258.65	2049.36	2824.32	3259.21
HD-070	5524.36	4038.74	18299.58	2112.52	3860.59	2424.12
HD-071	9525.92	7884.94	20258.81	4553.13	3906.20	5593.03
HD-072	9946.48	7245.70	21393.00	5213.06	3795.28	4525.90
HD-073	5850.98	4492.39	19048.29	1735.60	3433.97	2248.97
HD-074	7248.65	4787.66	15434.77	3146.10	2272.25	2618.09
HD-075	8234.12	6679.44	19293.33	3089.68	3186.77	3544.94
HD-076	6334.49	7322.51	19724.84	3722.38	3682.94	4026.72
HD-077	8729.76	7027.55	15719.52	4489.76	2957.99	3316.45
HD-078	6696.56	5525.04	15898.09	3732.77	3011.67	3081.31
HD-079	11131.39	8565.04	22284.61	4271.32	2719.51	6225.89
HD-080	5428.94	5489.23	17658.25	1912.50	3364.08	4321.06
HD-081	8969.38	7900.33	19095.75	4627.72	3472.16	5051.45
HD-082	5983.80	4580.61	14237.62	1886.70	2602.57	3032.82
HD-083	6138.50	5256.28	16356.77	1893.52	2275.04	2705.44
HD-084	6571.33	5673.94	16856.39	3492.54	2693.47	4578.16
HD-085	6840.86	4511.50	12107.09	2355.67	1963.09	2041.70
HD-086	6684.37	4704.17	13317.01	2680.63	2472.45	1931.07
HD-087	8219.45	7002.40	21776.13	3795.66	3245.72	3649.74
HD-088	7863.23	6251.94	16253.16	3298.89	2461.28	4767.04
HD-089	6916.51	5712.58	18258.22	3352.68	2719.05	2812.76
HD-090	8118.78	5604.87	19036.67	5521.49	2451.47	4010.86
HD-091	8321.52	7071.73	18211.32	3713.78	3382.53	3100.15
HD-092	5735.96	3803.69	12404.21	1318.67	1912.17	1854.39
HD-093	7643.19	5418.70	14807.22	2625.03	3879.31	2379.24
HD-094	4519.64	5459.36	14266.87	2830.07	2489.65	2156.98
HD-095	5505.26	5337.76	14843.68	3133.70	3391.37	3344.22
HD-096	5381.89	5968.82	16428.91	4828.00	3017.47	4268.59
HD-097	8871.32	6588.47	20699.50	4087.02	2219.88	3760.51
HD-098	7796.48	5681.60	13514.77	2555.86	2451.38	3394.43
HD-099	5275.50	4470.58	14556.20	2698.92	2054.19	2472.57
HD-100	7389.03	5585.69	15230.38	2794.77	2199.83	3594.90
QC-11	3873.97	3223.73	10895.14	1399.43	1588.50	1573.16
QC-12	4075.56	3184.41	10663.04	1176.70	1598.03	1963.72
QC-13	3581.07	3608.85	9898.93	1629.64	1112.24	1463.34
QC-14	5542.39	3333.67	11610.98	1263.42	1432.74	1784.28
QC-15	4597.32	3664.89	12703.73	1520.84	1648.44	2079.98
QC-16	5123.23	3516.95	10768.42	1417.03	1555.13	2291.34
QC-17	4354.56	3148.78	10099.91	1371.00	1659.76	1907.40

QC-18	4369.01	2926.08	9399.55	1240.15	1404.30	1475.25
QC-19	4172.98	3190.70	10580.75	1132.33	1550.90	1849.63
QC-20	3610.13	3020.15	8627.40	1284.93	1256.56	1726.32
BC-101	6802.81	6666.74	22152.33	3863.25	3158.38	3798.31
BC-102	9561.62	6001.61	21899.76	1762.41	3267.75	4599.80
BC-103	7925.80	6181.28	15496.93	3277.59	2609.81	3291.53
BC-104	7836.60	5916.00	17933.01	3661.59	2660.07	2780.27
BC-105	6327.49	5693.23	21997.90	1569.76	3523.72	4032.19
BC-106	5962.55	4560.58	15709.46	2162.53	1860.56	3144.78
BC-107	6182.36	6081.48	18413.53	1308.75	3220.65	3672.40
BC-108	7820.41	7269.25	18260.43	3362.97	3254.93	4248.27
BC-109	6614.81	5556.39	15746.32	2650.25	2874.07	3915.13
BC-110	8148.60	6086.10	17406.26	1542.06	2165.29	4575.88
BC-111	7706.03	5085.31	17493.22	3657.01	3705.62	2818.80
BC-112	8171.95	5914.34	18903.27	1364.89	3728.11	3143.29
BC-113	6612.36	5656.06	16387.60	2794.50	2785.90	4475.03
BC-114	5672.26	4239.44	18968.97	2638.64	2924.57	5336.05
BC-115	6432.70	5668.71	19997.52	3280.56	3537.55	4575.51
BC-116	5638.80	5490.28	17296.66	1731.10	2216.39	3432.17
BC-117	7692.28	5387.39	21218.37	1397.23	3772.30	3883.80
BC-118	6454.50	4905.16	16395.53	2032.98	3248.28	2889.99
BC-119	7214.27	5560.96	19116.76	3758.26	2688.75	3676.37
BC-120	2228.65	1351.38	3762.01	472.98		
BC-121	7361.76	5862.93	15932.42	3825.17	3015.08	3597.77
BC-122	6690.23	4305.26	16689.73	1545.47	2567.77	2712.38
BC-123	7004.78	5756.46	17372.32	3143.31	3000.75	3574.80
BC-124	7430.58	5102.48	17627.88	3940.25	1676.76	3133.51
BC-125	9485.35	6308.78	14763.93	5501.26	2182.31	3617.77
BC-126	7693.59	6451.64	21592.24	4576.57	3248.59	4255.17
BC-127	5117.49	4335.49	14009.40	3302.16	2515.62	3603.72
BC-128	6670.20	4689.17	17122.20	2670.31	3271.59	3940.10
BC-129	6803.08	5366.88	22046.18	3681.35	3158.84	4338.02
BC-130	5682.70	4347.15	17804.49	3099.35	2713.33	2550.22
BC-131	5819.19	5062.93	18229.11	4081.51	3012.02	4696.27
BC-132	5242.49	4250.37	9825.63	2344.24	1546.17	3003.14
BC-133	5216.02	3675.00	15210.20	1988.14	3381.39	3444.89
BC-134	5421.77	5651.77	14570.20	3895.38	1348.36	2533.32
BC-135	6286.14	5972.97	16965.72	3835.37	2767.13	3124.22
BC-136	5642.18	4560.03	13114.21	1513.34	2347.90	2849.57
BC-137	5266.88	5338.16	19145.40	4534.98	2572.23	3357.67
BC-138	5389.39	4062.84	14291.08	2422.12	1896.99	2427.37
BC-139	1557.64	751.48	8369.02	763.83	1237.85	1588.64
BC-140	5866.66	4211.36	15890.18	3326.86	2504.30	4803.23
BC-141	7718.20	7258.53	13232.67	3496.39	2916.22	4107.51

BC-142	6534.80	4257.65	14707.16	2959.69	2294.37	2289.10
BC-143	3938.37	3616.34	13812.37	2420.91	1571.92	2314.89
BC-144	7352.44	5286.28	15628.73	2895.07	2436.21	3365.25
HD-101	8191.79	6686.33	17582.39	2591.09	3034.74	2695.52
HD-102	9810.39	5833.05	15903.62	3180.67	1931.55	4268.85
HD-103	10338.97	6502.81	17555.33	4191.30	2972.21	3743.36
HD-104	8363.28	6442.22	18773.67	3510.86	4115.94	5130.50
HD-105	9029.61	4913.37	18429.20	2524.94	3041.93	3720.03
HD-106	7015.63	4897.43	13589.97	3694.10	2912.49	2718.41
HD-107	6831.50	5028.09	16352.70	2046.02	2942.44	3377.56
HD-108	7730.49	6099.59	21921.73	4152.81	3499.60	5418.05
HD-109	11597.27	7549.06	21378.98	3581.82	4182.79	4857.07
HD-110	8248.60	6525.42	17252.83	3939.21	2938.82	3849.53
HD-111	8605.58	7200.05	18126.20	3588.14	3280.19	4224.79
HD-112	7656.26	5826.52	18249.89	2951.60	3125.85	4699.50
HD-113	6045.07	5679.98	16828.70	3555.36	3492.12	4078.39
HD-114	9810.47	7325.87	15603.06	3131.75	2394.38	3714.91
HD-115	7956.44	5590.20	17657.76	1910.96	2488.48	3339.95
HD-116	6925.33	5494.28	18514.69	3026.36	2728.32	3220.01
HD-117	9887.89	5781.39	15731.93	3263.98	3484.10	2631.37
HD-118	5616.44	4774.61	16370.72	2250.34	2505.91	2286.08
HD-119	8290.15	6320.14	20455.28	2688.78	3703.57	4278.17
HD-120	5014.90	4422.44	14404.93	1909.05	3048.58	3222.83
HD-121	6523.01	5612.92	16223.17	2571.95	2820.77	3799.43
HD-122	5339.02	3388.00	13152.44	1172.18	3193.21	2855.31
HD-123	7263.98	5208.38	16799.95	3203.54	2675.91	3720.60
HD-124	7793.23	5197.57	12952.60	1890.93	1799.76	2879.35
HD-125	5190.62	3991.04	12864.36	2542.25	2252.60	2677.29
HD-126	6009.57	4990.18	19206.34	2480.83	2534.64	3360.19
HD-127	7628.99	6796.93	18318.68	4809.04	3220.93	6431.73
HD-128	5995.40	3135.92	14662.54	2452.45	2574.42	2959.61
HD-129	6521.09	5078.87	17265.77	1939.38	2740.87	3051.34
HD-130	5948.16	4679.08	16417.26	2558.55	3322.29	2892.51
HD-131	6624.35	4994.61	13783.05	3578.94	1782.53	2244.28
HD-132	7805.89	4749.80	14387.65	2599.78	2050.57	2816.05
HD-133	8124.28	5782.48	15409.69	3329.75	2155.99	4275.12
HD-134	5478.34	4098.33	14726.93	1479.32	2633.88	2362.62
HD-135	6366.35	5422.71	15267.97	3176.73	2851.82	3506.03
HD-136	8287.49	5112.24	17711.83	2655.63	3043.97	4160.37
HD-137	5624.20	5408.83	14146.00	1940.97	2694.96	3106.71
HD-138	5704.44	4066.82	9864.39	1453.12	2121.33	2749.18
HD-139	8848.62	4872.96	15178.51	2428.41	3219.56	2869.34
HD-140	7143.44	4798.28	19194.28	2813.99	3268.87	2521.00
HD-141	7856.76	5164.83	23608.15	1821.51	3363.50	5472.10

HD-142	8105.15	6596.00	13184.22	2962.91	2423.06	3072.57
HD-143	6848.24	5182.06	16127.27	3181.12	2709.23	3307.71
HD-144	6965.13	6831.36	18003.35	3424.51	3369.50	4566.02
HD-145	6316.37	5262.29	20962.12	1401.42	3258.84	3846.11
HD-146	6392.35	5142.27	17325.49	3656.53	3368.69	4523.64
HD-147	9633.11	6761.50	19176.20	3771.46	3655.41	5765.55
HD-148	7981.34	6469.98	14491.62	5110.88	3137.86	3308.66
HD-149	6756.42	5916.95	19812.39	2917.48	2960.38	4112.39
HD-150	7102.04	4983.02	14095.11	1982.99	2882.27	2041.78
HD-151	7027.42	6320.04	18362.68	3124.81	3522.09	4133.29
HD-152	8081.84	5016.43	19614.96	3406.01	3671.23	4276.26
HD-153	6547.62	4968.05	16135.72	2942.08	4838.19	5117.49
HD-154	5647.30	4884.65	16199.33	3452.77	2516.24	2927.63
HD-155	7277.02	6006.41	19409.17	2700.53	3188.39	4241.74
HD-156	6704.49	5578.41	17488.03	4079.24	1915.07	4548.97
HD-157	6347.13	4526.72	17756.59	2776.10	2443.02	3320.24
HD-158	5285.95	4765.66	16178.09	2075.61	2391.06	3464.64
HD-159	8225.85	5537.85	16973.24	4319.97	2521.05	3728.12
HD-160	7257.89	5909.83	19267.00	3455.72	2747.24	4563.32
HD-161	7200.53	6359.97	14974.56	3976.39	2835.12	2666.29
HD-162	5604.89	4102.90	13045.45	2323.07	2667.93	2761.80
HD-163	4264.23	3639.15	11187.37	1656.47	2500.76	2678.42
HD-164	5025.52	3759.10	14616.09	2686.41	2654.07	2366.07
HD-165	4927.84	4344.12	11176.61	1839.19	1569.30	2918.67
HD-166	6733.55	5243.91	12734.18	3445.68	2422.15	4175.99
HD-167	6047.45	5379.46	16405.25	3017.24	2226.51	3635.34
HD-168	4970.03	3931.84	12431.02	2466.54	2078.55	1879.19
HD-169	5390.03	4155.52	12282.05	2283.48	1349.17	2294.75
HD-170	7316.67	5710.65	13652.50	2299.45	2507.67	3346.55
HD-171	7048.85	6211.23	14821.84	4209.37	2281.51	4062.68
HD-172	5593.80	4514.82	15070.46	2666.16	1695.75	2555.51
HD-173	5354.19	4440.38	10147.73	3478.49	1858.46	1730.59
HD-174	7224.73	6000.14	12469.66	4764.04	1790.64	2233.41
HD-175	8609.16	7536.11	12257.75	4705.82	2201.91	3128.42
HD-176	5886.98	4308.34	13767.64	4848.70	1807.45	3400.75
HD-177	6807.29	5214.65	15459.69	3042.28	3849.77	2148.94
HD-178	8325.84	6786.59	14098.31	6364.78	2721.59	3315.98
HD-179	8094.15	7340.95	14282.28	5924.14	2493.69	3133.90
HD-180	7994.55	6833.18	13591.69	5785.52	3412.00	3244.79
HD-181	7260.95	4456.81	16503.62	2384.52	2416.34	3813.59
HD-182	9122.63	5989.60	22994.93	3310.35	3867.79	4822.53
HD-183	9872.85	7184.90	17737.36	4708.56	4033.06	4577.73
HD-184	5459.55	4348.36	10263.45	5100.28	2122.28	1649.62
HD-185	7863.33	5586.45	17957.82	2730.60	1986.39	3531.81

HD-186	4696.59	4754.73	15798.51	3004.01	2853.69	2683.33
HD-187	8788.72	6235.40	20622.41	3196.28	3130.21	4073.22
HD-188	5590.21	4640.61	18137.19	3144.09	2682.45	2205.85
HD-189	4540.89	2879.36	12456.15	3451.26	2416.30	2609.17
HD-190	6104.67	4838.88	16506.56	1536.86	2371.04	2346.08
HD-191	5995.42	5376.54	18385.70	2554.46	3142.37	3413.60
QC-21	4212.97	3461.27	10804.40	1452.27	1547.94	1911.75
QC-22	4652.96	2980.05	11081.85	1275.26	1668.02	1537.92
QC-23	3731.01	2570.37	10293.59	1322.04	1625.01	1740.15
QC-24	4333.01	3044.04	8614.36	1249.83	1364.21	1940.39
QC-25	4935.69	3111.02	10865.70	1588.42	1645.47	1929.49
QC-26	4342.40	3030.06	9073.40	1263.70	1265.57	1948.82
QC-27	4493.22	3105.97	10834.30	1430.20	1106.72	1778.72
QC-28	3618.18	3193.76	12749.28	1317.81	1552.16	1755.40
QC-29	3541.81	2916.11	9106.58	994.63	1693.57	1946.31
QC-30	4021.96	3115.13	10525.95	1335.30	1532.77	1776.94
QC-31	4598.89	2713.29	11090.56	1362.96	1644.42	2056.60
QC-32	4086.42	2907.41	10264.13	1381.86	1300.14	1847.23
QC-33	4038.63	2552.33	9791.11	1021.22	1587.15	1756.04

Comment 8. While this software has a large database, sub-class coverage is actually smaller than for example LipidSearch, MS-DIAL, and LipidMatch.

Response: Thank you for your valuable suggestion. We have expanded our database to 121 lipid subclasses, which is more 4 subclasses than MS-DIAL. These 4 subclasses are the Sulfonolipid species (as noted in the article titled Structure Revision of a Widespread Marine Sulfonolipid Class Based on Isolation and Total Synthesis). These four are synthetic, and we are committed to continuing our focus on novel lipids as we grow our database further. All 121 lipid subclasses have been updated in Table R9 of the Supporting Materials, with abbreviated information provided in the table below. We also revised manuscript in Line 32, 93, 104, 113, 156, 242, 243, 759 and Table. S2.

Table R9. All 121 lipids brief information.

Categories	Lipid subclass	Abbreviation
Prenol Lipids [PR]	Vitamin A fatty acid ester	VAE
Sphingolipids [SP]	Trihexosylceramide	Hex3Cer
Sterol Lipids [ST]	Triacylglycerol	TG
Sphingolipids [SP]	Sulfonolipid	SL
Sphingolipids [SP]	Sphingomyelin	SM
Sphingolipids [SP]	Sphinganine	SPB
Sphingolipids [SP]	Sphingosine	SPB
Sphingolipids [SP]	Phytosphingosine	SPB
Glycerophospholipids [GP]	Phosphatidylserine	PS
Glycerophospholipids [GP]	Phosphatidylmethanol	PMeOH
Glycerophospholipids [GP]	Phosphatidylinositol	PI
Glycerophospholipids [GP]	Phosphatidylglycerol	PG
Glycerophospholipids [GP]	Phosphatidylethanolamine	PE
Glycerophospholipids [GP]	Phosphatidylethanol	PEtOH
Glycerophospholipids [GP]	Phosphatidylcholine	PC
Glycerophospholipids [GP]	Phosphatidic acid	PA
Sterol Lipids [ST]	Ether-linked triacylglycerol	TG-O
Sterol Lipids [ST]	Oxidized triglyceride	OxTG
Sphingolipids [SP]	Oxidized sulfonolipid	SL+O
Glycerophospholipids [GP]	Oxidized phosphatidylethanolamine	OxPE
Fatty acyls [FA]	Oxidized fatty acid	OxFA
Glycerophospholipids [GP]	N-Monomethylphosphatidylethanolamine	MMPE
Glycerophospholipids [GP]	N-Dimethylphosphatidylethanolamine	DMPE
Fatty acyls [FA]	N-acyl ethanolamines	NAE
Glycerolipids [GL]	Monogalactosylmonoacylglycerol	MGMG
Glycerolipids [GL]	Monogalactosyldiacylglycerol	MGDG
Glycerolipids [GL]	Monoacylglycerol	MG
Glycerophospholipids [GP]	Lysophosphatidylserine	LPS
Glycerophospholipids [GP]	Lysophosphatidylinositol	LPI
Glycerophospholipids [GP]	Lysophosphatidylglycerol	LPG
Glycerophospholipids [GP]	Lysophosphatidylethanolamine	LPE
Glycerophospholipids [GP]	Lysophosphatidic acid	LPA
Glycerophospholipids [GP]	Lysophosphatidylcholine	LPC
Glycerolipids [GL]	Lysodiacylglyceryl-3-O-carboxyhydroxymethylcholine	LDGCC
Glycerolipids [GL]	Lysodiacylglyceryl trimethylhomoserine	LDGTS
Glycerolipids [GL]	Lysodiacylglyceryl hydroxymethyl-N,N,N-trimethyl- β -alanine	LDGTS
Glycerophospholipids [GP]	Lysocardiolipin	MLCL

Sphingolipids [SP]	Hexosylceramide non-hydroxyfatty acid-sphingosine	HexCer
Sphingolipids [SP]	Hexosylceramide non-hydroxyfatty acid-dihydrosphingosine	HexCer
Sphingolipids [SP]	Hexosylceramide hydroxyfatty acid-dihydrosphingosine	HexCer
Sphingolipids [SP]	Hexosylceramide hydroxyfatty acid-sphingosine	HexCer
Glycerophospholipids [GP]	Hemibismonoacylglycerophosphate	HBMP
Glycerophospholipids [GP]	Glycerophospho N-acyl ethanolamine	GPNAE
Fatty acyls [FA]	Free fatty acid	FA
Fatty acyls [FA]	Fatty acid ester of hydroxyl fatty acid	FAHFA
Glycerophospholipids [GP]	Ether-linked phosphatidylserine	PS-O
Glycerophospholipids [GP]	Ether-linked phosphatidylinositol	PI-O
Glycerophospholipids [GP]	Ether-linked phosphatidylglycerol	PG-O
Glycerophospholipids [GP]	Ether-linked phosphatidylethanolamine	PE-O
Glycerophospholipids [GP]	Ether-linked phosphatidylethanolamine (P)	PE-P
Glycerophospholipids [GP]	Ether-linked phosphatidylcholine	PC-O
Glycerolipids [GL]	Ether-linked monogalactosyldiacylglycerol	MGDG-O
Glycerophospholipids [GP]	Ether-linked lysophosphatidylglycerol	LPG-O
Glycerophospholipids [GP]	Ether-linked lysophosphatidylethanolamine	LPE-O
Glycerophospholipids [GP]	Ether-linked lysophosphatidylcholine	LPC-O
Glycerolipids [GL]	Ether-linked digalactosyldiacylglycerol	DGDG-O
Glycerolipids [GL]	Ether-linked diacylglycerol	DG-O
Glycerophospholipids [GP]	Dilysocardiolipin	DLCL
Sphingolipids [SP]	Dihexosylceramide	Hex2Cer
Glycerolipids [GL]	Digalactosyldiacylglycerol	DGDG
Glycerolipids [GL]	Diacylglyceryl-3-O-carboxyhydroxymethylcholine	DGCC
Glycerolipids [GL]	Diacylglyceryl trimethylhomoserine/diacylglyceryl hydroxymethyl-N,N,N-trimethyl-β-alanine	DGTS
Glycerolipids [GL]	Diacylglyceryl glucuronide	DGGA
Glycerolipids [GL]	Diacylglycerol	DG
Prenol Lipids [PR]	Coenzyme Q	CoQ
Sterol Lipids [ST]	Cholesteryl ester	CE
Sphingolipids [SP]	Ceramide non-hydroxyfatty acid-phytospingosine	Cer
Sphingolipids [SP]	Ceramide hydroxy fatty acid-	Cer

	sphingosine	
Sphingolipids [SP]	Ceramide hydroxy fatty acid-dihydrosphingosine	Cer
Sphingolipids [SP]	Ceramide non-hydroxyfatty acid-dihydrosphingosine	Cer
Sphingolipids [SP]	Ceramide non-hydroxyfatty acid-sphingosine	Cer
Sphingolipids [SP]	Ceramide 1-phosphates	CerP
Glycerophospholipids [GP]	Cardiolipin	CL
Glycerophospholipids [GP]	Bismonoacylglycerophosphate	BMP
Glycerophospholipids [GP]	Oxidized phosphatidylcholine	OxPC
Glycerophospholipids [GP]	Oxidized phosphatidylserine	OxPS
Glycerophospholipids [GP]	Oxidized phosphatidylinositol	OxPI
Glycerophospholipids [GP]	Oxidized phosphatidylglycerol	OxPG
Glycerolipids [GL]	Sulfoquinovosyl diacylglycerol	SQDG
Glycerolipids [GL]	Acyl diacylglyceryl glucuronide	ADGGA
Glycerolipids [GL]	Ether-linked Acyl diacylglyceryl glucuronide	ADGGA-O
Sterol Lipids [ST]	Acylhexosyl brassicasterol	AHexBRS-ST
Sterol Lipids [ST]	Acylhexosyl campesterol	AHexCAS-ST
Sphingolipids [SP]	Acylhexosylceramide	AHexCer-O
Sterol Lipids [ST]	Acylhexosyl cholesterol	AHexCS-ST
Sterol Lipids [ST]	Acylhexosyl sitosterol	AHexSIS-ST
Sterol Lipids [ST]	Acylhexosyl stigmaterol	AHexSTS-ST
Sphingolipids [SP]	Acylsphingomyelin	ASM
Sterol Lipids [ST]	Cholic acid sulfate	BASulfate-ST
Sterol Lipids [ST]	Brassicasterol ester	BRSE
Fatty acyls [FA]	Acylcarnitine	CAR
Sterol Lipids [ST]	Campesterol ester	CASE
Sterol Lipids [ST]	Cholesterol	Cholesterol-ST
Sterol Lipids [ST]	Esterified deoxycholic acid	DCAE-SE
Sterol Lipids [ST]	Dehydroergosterol ester	DEGSE
Sphingolipids [SP]	Ganglioside GD1a	GD1a
Sphingolipids [SP]	Ganglioside GD1b	GD1b
Sphingolipids [SP]	Ganglioside GM1	GM1
Sphingolipids [SP]	Ganglioside GM3	GM3
Sphingolipids [SP]	Ganglioside GQ1b	GQ1b
Sphingolipids [SP]	Ganglioside GT1b	GT1b
Glycerophospholipids [GP]	N-acyl-lysophosphatidylethanolamine	LNAPE
Glycerophospholipids [GP]	N-acyl-lysophosphatidylserine	LNAPS
Fatty acyls [FA]	N-acyl glycine	NAGly
Fatty acyls [FA]	N-acyl glyceryl serine	NAGlySer
Fatty acyls [FA]	N-acyl ornithine	NAOrn
Sphingolipids [SP]	Ceramide phosphoethanolamine	PE-Cer

Sphingolipids [SP]	Oxidized ceramide phosphoethanolamine	PE-Cer+O
Sphingolipids [SP]	Ceramide phosphoinositol	PI-Cer
Sphingolipids [SP]	Oxidized ceramide phosphoinositol	PI-Cer+O
Sphingolipids [SP]	Sulfatide	SHexCer
Sphingolipids [SP]	Oxidized sulfatide	SHexCer+O
Sterol Lipids [ST]	Stigmasterol hexoside	SHex-ST
Glycerolipids [GL]	Ether-linked Semino lipid	SMGDG-O
Sphingolipids [SP]	Solated cysteinolide A	Solated_cysteinolide-A
Sphingolipids [SP]	Solated cysteinolide B	Solated_cysteinolide-B
Sterol Lipids [ST]	Sterol sulfate	SSulfate-ST
Sphingolipids [SP]	Synthetic sulfur-containing amino lipids	Synthetic-SAL
Glycerolipids [GL]	Triacylglycerol estolides	TG-EST
Sterol Lipids [ST]	Vitamin D	Vitamin-D
Prenol Lipids [PR]	Vitamin E	Vitamin-E

Other specific comments:

Comment 9. Make sure to include version numbers when doing comparisons. For example, the latest versions of MS-DIAL and LipidMatch are very unique as compared to other older versions.

Response: Thanks to your suggestion, we have labeled the Benchmark Experiments section with the version numbers of all software used as well as the download addresses, and a more visual representation of the parameters is provided in Table R6.

Comment 10. Make sure to cite very clearly where all fragment and class information came from.

Response: Thank you for your suggestion, we have removed many reference articles on lipid structure during the previous submission process considering the article requirements, which have now been replenished. We present the other references in Table R10. We also revised manuscript in Line 132 and Table. S2 and cite all references in Supporting Information.

Table R10. Lipid Subclass and References.

Abbreviation	Reference
VAE	Analysis of Nonvolatile Lipids by Mass Spectrometry
Hex3Cer	Photoinduced Online Enrichment–Deglycosylation of Glycolipids for Enhancing Lipid Coverage and Identification in Single-Cell Mass Spectrometry
TG	Advances in Liquid Chromatography Mass Spectrometry-Based Lipidomics: A Look Ahead
SL	A lipidome atlas in MS-DIAL 4
SM	Advances in Liquid Chromatography Mass Spectrometry-Based Lipidomics: A Look Ahead
SPB	Sphingolipidomics: High-throughput, structure-specific, and quantitative analysis of sphingolipids by liquid chromatography tandem mass spectrometry
SPB	Sphingolipidomics: High-throughput, structure-specific, and quantitative analysis of sphingolipids by liquid chromatography tandem mass spectrometry
SPB	Sphingolipidomics: High-throughput, structure-specific, and quantitative analysis of sphingolipids by liquid chromatography tandem mass spectrometry
PS	Multi-dimensional mass spectrometry-based shotgun lipidomics and novel strategies for lipidomic analyses
PMeOH	A lipidome atlas in MS-DIAL 4
PI	Multi-dimensional mass spectrometry-based shotgun lipidomics and novel strategies for lipidomic analyses
PG	Multi-dimensional mass spectrometry-based shotgun lipidomics and novel strategies for lipidomic analyses
PE	Multi-dimensional mass spectrometry-based shotgun lipidomics and novel strategies for lipidomic analyses
PEtOH	A lipidome atlas in MS-DIAL 4
PC	Multi-dimensional mass spectrometry-based shotgun lipidomics and novel strategies for lipidomic analyses
PA	Multi-dimensional mass spectrometry-based shotgun lipidomics and novel strategies for lipidomic analyses
TG-O	Advances in Liquid Chromatography Mass Spectrometry-Based Lipidomics: A Look Ahead
OxTG	Structural analysis of hydroperoxy- and epoxy-triacylglycerols by liquid chromatography mass spectrometry
SL+O	A lipidome atlas in MS-DIAL 4
OxPE	Mass spectrometry analysis of oxidized phospholipids
OxFA	Oxidized fatty acid analysis by charge-switch derivatization, selected reaction monitoring, and accurate mass quantitation
MMPE	Targeting Modified Lipids during Routine Lipidomics Analysis using

	HILIC and C30 Reverse Phase Liquid Chromatography coupled to Mass Spectrometry
DMPE	Targeting Modified Lipids during Routine Lipidomics Analysis using HILIC and C30 Reverse Phase Liquid Chromatography coupled to Mass Spectrometry
NAE	A lipidome atlas in MS-DIAL 4
MGMG	Comprehensive Lipidomic Analysis of Three Edible Brown Seaweeds Based on Reversed-Phase Liquid Chromatography Coupled with Quadrupole Time-of-Flight Mass Spectrometry
MGDG	Comprehensive Lipidomic Analysis of Three Edible Brown Seaweeds Based on Reversed-Phase Liquid Chromatography Coupled with Quadrupole Time-of-Flight Mass Spectrometry
MG	Comprehensive Lipidomic Analysis of Three Edible Brown Seaweeds Based on Reversed-Phase Liquid Chromatography Coupled with Quadrupole Time-of-Flight Mass Spectrometry
LPS	Multi-dimensional mass spectrometry-based shotgun lipidomics and novel strategies for lipidomic analyses
LPI	Multi-dimensional mass spectrometry-based shotgun lipidomics and novel strategies for lipidomic analyses
LPG	Multi-dimensional mass spectrometry-based shotgun lipidomics and novel strategies for lipidomic analyses
LPE	Multi-dimensional mass spectrometry-based shotgun lipidomics and novel strategies for lipidomic analyses
LPA	Multi-dimensional mass spectrometry-based shotgun lipidomics and novel strategies for lipidomic analyses
LPC	Multi-dimensional mass spectrometry-based shotgun lipidomics and novel strategies for lipidomic analyses
LDGCC	A lipidome atlas in MS-DIAL 4
LDGTS	A lipidome atlas in MS-DIAL 4
LDGTS	A lipidome atlas in MS-DIAL 4
MLCL	Identification of unique cardiolipin and monolysocardiolipin species in Acinetobacter baumannii
HexCer	Photoinduced Online Enrichment–Deglycosylation of Glycolipids for Enhancing Lipid Coverage and Identification in Single-Cell Mass Spectrometry
HexCer	Photoinduced Online Enrichment–Deglycosylation of Glycolipids for Enhancing Lipid Coverage and Identification in Single-Cell Mass Spectrometry
HexCer	Photoinduced Online Enrichment–Deglycosylation of Glycolipids for Enhancing Lipid Coverage and Identification in Single-Cell Mass Spectrometry
HexCer	Photoinduced Online Enrichment–Deglycosylation of Glycolipids for Enhancing Lipid Coverage and Identification in Single-Cell Mass Spectrometry

HBMP	A lipidome atlas in MS-DIAL 4
GPNAE	A lipidome atlas in MS-DIAL 4
FA	Mass Spectrometry Imaging for the Characterization of C=C Localization in Unsaturated Lipid Isomers at the Single-Cell Level
FAHFA	In-Silico-Generated Library for Sensitive Detection of 2-Dimethylaminoethylamine Derivatized FAHFA Lipids Using HighResolution Tandem Mass Spectrometry
PS-O	Multi-dimensional mass spectrometry-based shotgun lipidomics and novel strategies for lipidomic analyses
PI-O	Multi-dimensional mass spectrometry-based shotgun lipidomics and novel strategies for lipidomic analyses
PG-O	Multi-dimensional mass spectrometry-based shotgun lipidomics and novel strategies for lipidomic analyses
PE-O	Multi-dimensional mass spectrometry-based shotgun lipidomics and novel strategies for lipidomic analyses
PE-P	Multi-dimensional mass spectrometry-based shotgun lipidomics and novel strategies for lipidomic analyses
PC-O	Multi-dimensional mass spectrometry-based shotgun lipidomics and novel strategies for lipidomic analyses
MGDG-O	Comprehensive Lipidomic Analysis of Three Edible Brown Seaweeds Based on Reversed-Phase Liquid Chromatography Coupled with Quadrupole Time-of-Flight Mass Spectrometry
LPG-O	Multi-dimensional mass spectrometry-based shotgun lipidomics and novel strategies for lipidomic analyses
LPE-O	Multi-dimensional mass spectrometry-based shotgun lipidomics and novel strategies for lipidomic analyses
LPC-O	Multi-dimensional mass spectrometry-based shotgun lipidomics and novel strategies for lipidomic analyses
DGDG-O	A novel ether-linked phytol-containing digalactosylglycerolipid in the marine green alga, Ulva pertusa
DG-O	Multidimensional mass spectrometry-based shotgun lipidomics analysis of vinyl ether diglycerides
DLCL	Identification of unusual phospholipids from bovine heart mitochondria by HPLC-MS/MS
Hex2Cer	Photoinduced Online Enrichment–Deglycosylation of Glycolipids for Enhancing Lipid Coverage and Identification in Single-Cell Mass Spectrometry
DGDG	A novel ether-linked phytol-containing digalactosylglycerolipid in the marine green alga, Ulva pertusa
DGCC	Lipidomes of phylogenetically different symbiotic dinoflagellates of corals
DGTS	Identification and discrimination of lilii bulbos origins based on lipidomics using UHPLC–QE-OrbitrapMSMS combined with chemometrics analysis

DGGA	A lipidome atlas in MS-DIAL 4
DG	Advances in Liquid Chromatography Mass Spectrometry-Based Lipidomics: A Look Ahead
CoQ	In vitro construction of the COQ metabolon unveils the molecular determinants of coenzyme Q biosynthesis
CE	A Facile LC-MS Method for Profiling Cholesterol and Cholesteryl Esters in Mammalian Cells and Tissues
Cer	Photoinduced Online Enrichment–Deglycosylation of Glycolipids for Enhancing Lipid Coverage and Identification in Single-Cell Mass Spectrometry
Cer	Photoinduced Online Enrichment–Deglycosylation of Glycolipids for Enhancing Lipid Coverage and Identification in Single-Cell Mass Spectrometry
Cer	Photoinduced Online Enrichment–Deglycosylation of Glycolipids for Enhancing Lipid Coverage and Identification in Single-Cell Mass Spectrometry
Cer	Photoinduced Online Enrichment–Deglycosylation of Glycolipids for Enhancing Lipid Coverage and Identification in Single-Cell Mass Spectrometry
Cer	Photoinduced Online Enrichment–Deglycosylation of Glycolipids for Enhancing Lipid Coverage and Identification in Single-Cell Mass Spectrometry
CerP	Photoinduced Online Enrichment–Deglycosylation of Glycolipids for Enhancing Lipid Coverage and Identification in Single-Cell Mass Spectrometry
CL	Multi-dimensional mass spectrometry-based shotgun lipidomics and novel strategies for lipidomic analyses
BMP	PLD3 and PLD4 synthesize S,S-BMP, a key phospholipid enabling lipid degradation in lysosomes
OxPC	Mass spectrometry analysis of oxidized phospholipids
OxPS	Mass spectrometry analysis of oxidized phospholipids
OxPI	Mass spectrometry analysis of oxidized phospholipids
OxPG	Mass spectrometry analysis of oxidized phospholipids
SQDG	LC-ESI-MS/MS Analysis of Sulfolipids and Galactolipids in Green and Red Lettuce (Lactuca sativa L.) as Influenced by Sulfur Nutrition
ADGGA	A lipidome atlas in MS-DIAL 4
ADGGA-O	A lipidome atlas in MS-DIAL 4
AHexBRS-ST	A lipidome atlas in MS-DIAL 4
AHexCAS-ST	A lipidome atlas in MS-DIAL 4
AHexCer-O	A lipidome atlas in MS-DIAL 4
AHexCS-ST	A lipidome atlas in MS-DIAL 4
AHexSIS-ST	A lipidome atlas in MS-DIAL 4
AHexSTS-ST	A lipidome atlas in MS-DIAL 4
ASM	A lipidome atlas in MS-DIAL 4

BASulfate-ST	A lipidome atlas in MS-DIAL 4
BRSE	A lipidome atlas in MS-DIAL 4
CAR	Enhanced acylcarnitine annotation in high-resolution mass spectrometry data: fragmentation analysis for the classification and annotation of acylcarnitines
CASE	A lipidome atlas in MS-DIAL 4
Cholesterol-ST	A lipidome atlas in MS-DIAL 4
DCAE-SE	A lipidome atlas in MS-DIAL 4
DEGSE	A lipidome atlas in MS-DIAL 4
GD1a	Manipulation of Ion Types via Gas-Phase Ion/Ion Chemistry for the Structural Characterization of the Glycan Moiety on Gangliosides
GD1b	Manipulation of Ion Types via Gas-Phase Ion/Ion Chemistry for the Structural Characterization of the Glycan Moiety on Gangliosides
GM1	A large-scale genome–lipid association map guides lipid identification
GM3	A large-scale genome–lipid association map guides lipid identification
GQ1b	A lipidome atlas in MS-DIAL 4
GT1b	AP-MALDI Mass Spectrometry Imaging of Gangliosides Using 2,6-Dihydroxyacetophenone Using 2,6-Dihydroxyacetophenone
LNAPE	Hyphenation of Liquid Chromatography and Trapped Ion Mobility–Mass Spectrometry for Characterization of Isomeric Phosphatidylethanolamines with Focus on N-Acylated Species
LNAPS	A lipidome atlas in MS-DIAL 4
NAGly	A lipidome atlas in MS-DIAL 4
NAGlySer	A lipidome atlas in MS-DIAL 4
NAOrn	A lipidome atlas in MS-DIAL 4
PE-Cer	Ceramide lipids in alive and thermally stressed mussels: an investigation by hydrophilic interaction liquid chromatography-electrospray ionization Fourier transform mass spectrometry
PE-Cer+O	A lipidome atlas in MS-DIAL 4
PI-Cer	A lipidome atlas in MS-DIAL 4
PI-Cer+O	A lipidome atlas in MS-DIAL 4
SHexCer	A lipidome atlas in MS-DIAL 4
SHexCer+O	A lipidome atlas in MS-DIAL 4
SHex-ST	A lipidome atlas in MS-DIAL 4
SMGDG-O	Comprehensive Lipidomic Analysis of Three Edible Brown Seaweeds Based on Reversed-Phase Liquid Chromatography Coupled with Quadrupole Time-of-Flight Mass Spectrometry
Solated_cysteinolide-A	Structure Revision of a Widespread Marine Sulfonolipid Class Based on Isolation and Total Synthesis
Solated_cysteinolide-B	Structure Revision of a Widespread Marine Sulfonolipid Class Based on Isolation and Total Synthesis
SSulfate-ST	UPLC–MS/MS Identification of Sterol Sulfates in Marine Diatoms

Synthetic-SAL	Structure Revision of a Widespread Marine Sulfonolipid Class Based on Isolation and Total Synthesis
TG-EST	Characterization of Triacylglycerol Estolide Isomers Using High-Resolution Tandem Mass Spectrometry with Nanoelectrospray Ionization
Vitamin-D	Identification of Vitamin D3 Oxidation Products Using High-Resolution and Tandem Mass Spectrometry
Vitamin-E	Vitamin E analysis by ultra-performance convergence chromatography and structural elucidation of novel α -tocodienol by high-resolution mass spectrometry

Comment 11. MS-DIAL, LipidSearch, and LipidMatch, are stated as matching experimental spectra to library spectra which is very vague, and the limitations mentioned depend on the algorithm employed. For example while MS-DIAL includes dot-product type matching, retention index, etc. LipidMatch includes a rule-based approach based on fragment occurrence (intensity independent). Therefore, software such as LipidMatch, ALEX, etc. are generally instrument/matrix independent but do rely on low abundance peaks (as mentioned as a limitation). The problem is no matter what the model (machine learning or rule based) without these low abundant peaks which are sometimes the only ones necessary for structural annotation, structure cannot be assigned correctly.

Response: Thank you for your professional advice! Here we add a description of the algorithms for the individual tools, as shown in Table R11, and revised the manuscript in Line 58 and Table. S1.

Table R11. Comparison of LipidIN with other lipidomics software tools

Software name	Ion mobility data support	MS/MS similarity calculation	Decision tree annotation	Hierarchical library
LipidIN	Yes	Yes	Yes	Yes
MS-DIAL 5.1	Yes	Yes	Yes	No
LipidMatch 2.0.2	No	No	Yes	No
Entropy Search	No	Yes	No	No
LipidSearch 4.2	Yes	Yes	Yes	No

In addition, as you mentioned, low-abundance peaks significantly impact the accuracy of annotations. However, our focus here is to highlight the substantial effect that different algorithms have on the coverage of deep lipid identification. As shown in Figures R9 d-f, LipidIN outperforms all other annotation software, yielding more than twice the number of annotations for fatty acids (FA) in the SRM1950 dataset compared to other tools.

Next, I will provide some examples of how LipidIN can enhance algorithm coverage, which I hope will be helpful. For instance, consider a theoretical medium spectrum in SM anion mode with 5 characteristic peaks, including two peaks at 78.959 and 168.043. These peaks represent the head group structure of SM, while the neutral loss peaks of the precursor ion and the precursor ion peak itself convey important precursor ion information. In a similarity matching algorithm, such as the one using the Jaccard similarity coefficient (The Jaccard similarity coefficient is a metric used to gauge the similarity and diversity of sample sets. It is defined as the size of the intersection divided by the size of the union of two sets.), the presence of these two peaks, along with the precursor ion and neutral loss peak, can yield a higher similarity score (greater than or equal to 0.8). Typically, a spectral similarity score above 0.8 is regarded as a reliable match. However, this can lead to inaccuracies, as such annotations may often provide incorrect chain composition assignments. In contrast, LipidIN recognizes that having only the head group and precursor ion information does not ensure accurate fatty acid (FA) composition. Consequently, the similarity score in LipidIN will not exceed

0.5, reflecting a more cautious and precise assessment of the data.

In the positive ion spectrum of triglycerides (TG), we can identify four characteristic fragment peaks: one precursor ion peak and three characteristic peaks corresponding to fatty acids (FA). In operational research, this can be approached as a simple linear programming problem. Knowing the m/z of the precursor ion allows us to determine the C:DB of TG. With the information for two FA compositions, we can then infer the third FA composition. This illustrates the advantage of combining the hierarchical library with the EQ module, which enhances the coverage of our annotations, allowing us to annotate more lipids. However, this inference may also lead to an increase in false positive annotation results.

Comment 12. Furthermore, a comparison of these software suggest false positive rates are probably <5% at least at the lipid class level, so close to this reported confidence reported here, showing that the diverse algorithms perform similarly in many cases regardless of matrix/training set. <https://pmc.ncbi.nlm.nih.gov/articles/PMC7142889/>

Response: Thank you for your professional advice. We fully agree that the false discovery rate (FDR) of lipid subclass annotations is generally lower, regardless of the algorithm used. However, based on the SRM1950 benchmark experiments, we suggest that LipidIN can identify nearly twice as many lipids in fatty acid (FA) composition annotations compared to other tools. Previous validation experiments have also demonstrated that LipidIN effectively annotates more lipids across multiple organisms while maintaining a lower FDR.

We believe it is reasonable to expect that LipidIN will achieve a lower FDR when the number of annotations is comparable to that of other software. Notably, the FDR for other tools matches that of LipidIN only at smaller annotation counts and coarser levels of classification, making it inevitable that the FDR will increase as the number of annotations rises.

Comment 13. Line 126, please reference which community libraries were used, as I

believe reference 30 does not have all libraries used.

Response: Thank you for your suggestion, we have removed many reference articles on lipid structure during the previous submission process considering the article requirements, which have now been replenished. We present the other references in Table. R9 and Table. R10. We also revised manuscript in Line 132 and Table. S2 and cite all references in Supporting Information.

Comment 14. Line 129 Define “fragment fingerprints”, also afterward you start talking about each level and referring to the various levels which becomes complicated to follow. Rather, I would describe each “level” in a paragraph or few sentences, and then summarize all of them.

Response: Thank you for your professional advice. We have revised the definition and description of “fingerprint fragment” in the manuscript for a better understanding. We revised manuscript in Line 129.

Comment 15. 166.3 million theoretical lipid fragments across 76 lipid sub-classes is a lot, but the coverage is actually smaller than some of the most commonly used lipidomics software. For example, MS-DIAL has the greatest coverage with 117 lipid sub-classes, and LipidMatch has over 100 different lipid sub-classes as well.

Response: Thank you for your valuable suggestion. We have expanded our database to include 121 lipid subclasses, four of which belong to the Sulfonolipid Class (as noted in the article titled Structure Revision of a Widespread Marine Sulfonolipid Class Based on Isolation and Total Synthesis), all information shown in Table. R9 and Table. R10. We also revised manuscript in Line 132 and Table. S2 and cite all references in Supporting Information.

Comment 16. Line 175 described “median conformity” it would be good to get some easily understood metrics such as the +/- tolerance in minutes for the 95 and 99
92

percentile of lipids, and whether there are error rates calculated for individual retention time values.

Response: Thank you for your insightful question. In our article, the term “conformity” refers to a metric that assesses how well each individual trend aligns with the overall trend of a set of annotated results. This assessment is done on the same scale of total unsaturation for a given subcategory or on the scale of distributional saturation combinations.

As you suggested, we use the 95% confidence interval of the overall trend for this judgment. To illustrate this, let’s consider an example from the PC subclass. Suppose we have a total unsaturation level of 3 with 10 annotation results. We first fit a quadratic polynomial to the retention time and mass-to-charge ratio of these 10 points, creating a fitting function. We then calculate the upper and lower bounds of the 95% confidence interval for this function. (Simple Explanation of Fitting and Confidence Intervals: **a.** Fitting: This is a statistical method used to create a model that describes the relationship between variables. For example, if we have data points, we might use a curve or line to represent the general trend of those points. **b.** Confidence Interval: This is a range of values, derived from the data, that is likely to contain the true value of the parameter we are estimating. A 95% confidence interval means that if we were to take many samples and calculate the interval each time, about 95% of those intervals would contain the true value.) By using these concepts, we can make more informed judgments about the data we analyze. Next, we assess whether each of the 10 annotation results falls within this confidence interval. If they do, we consider them to be a good fit and were conforming to the ECN or ESCN rules. However, it’s important to note that if the difference between the upper and lower bounds of the confidence interval exceeds 2 minutes, we will classify that annotation result as not conforming to the ECN or ESCN rules.

As you mentioned, we were initially lacking a clear display of error values or error rates to better demonstrate the prevalence of ECN and ESCN. In response to your professional suggestion, we have now included statistics on retention time error and

retention time error rate, as shown in Fig. R13. We calculate the absolute error by taking the absolute value of the difference between the predicted trend value and the true retention time of the data, as illustrated in Fig. R13.a. The dashed line indicates the 95% quantile, suggesting that at least 95% of the annotated results have a retention time deviation of approximately 0.5 minutes, which is considered acceptable. Additionally, the absolute retention time deviation rate is an important metric. We derive this by dividing the absolute retention time deviation by the retention time of the annotation results. Fig. R13.b and Fig. R13.c present the absolute retention time deviation rates for ECN and ESCN according to lipid subclass. From the figures, we observe that the average absolute retention time deviation rates for the ECN and ESCN rules are 1.44% and 1.28%, respectively. This further supports the universality of these rules. However, it is worth noting that the lipid subclasses BA (Bile acids) and ST have larger deviation values due to insufficient detailed classification in the published dataset. We also revised manuscript in Line187, Supplementary Fig.4 and Source Data Supplementary Fig.4.

Fig. R13. Display of error values and error rates for ECN and ESCN. **a.** Absolute retention time error statistics: red for ECNs, blue for ESCNs, with a dotted line representing the 95% quantile. **b-c.** Box plots of absolute retention time deviation rates

for ECN and ESCN by lipid subclass, with the black horizontal line indicating the population mean.

Comment 17. “approach using EQ and LCI identified more 471 unique potential lipids” incorrect English

Response: Thank you for your expert advice. We have revised the phrase “approach using EQ and LCI identified more 471 unique potential lipids” to “Our approach using EQ and LCI modules annotated 471 unique lipids from the mixture dataset compared with other tools”. We also revised manuscript in Line 260.

Comment 18. “high-confidence potential lipids.” Is strange language, can you be high confident and just a potential lipid at the same time?

Response: Thank you for your valuable feedback. The omission of this sentence indeed could lead to confusion. We intended to convey that the lipids not annotated by other methods are considered potential lipids, which can be annotated and assigned a high score in LipidIN. We have revised the phrase “but also annotate more high-confidence potential lipids” to “but also annotate more high-confidence lipids, which were unannotated by all other methods we tested”. We also revised manuscript in Line 266.

Comment 19. The WMYn model was validated against 15 lipid standards, whereas there are 100s and 100s of lipid standards available (see Avanti, Nu-Check prep, etc.), what is the rationale for using such a limited number of standards for validation, especially as these cannot cover a great majority of the lipid classes you have included in your library?

Response: Thank you for your professional advice. Although our lipidomics data was against 15 lipid reference standards for validation of the WMYn model, moreover, we utilized the known lipids from NIST SRM 1950 as our lipid reference as you suggested.

Fortunately, our cohort 3 also employed Agilent's mass spectrometry system, which is consistent with the NIST SRM 1950 data. As a result, we were able to select 87 lipids from both positive and negative ion modes, encompassing most common lipid classes. Taking into account the 15 lipid reference standards from previous benchmark experiments, we tested the effect of WMYn on over 100 “known lipids” or lipid reference standards. These lipids were annotated in previous experiments annotated in at least two tools and achieving a top rank of 1 (Fig. R9), and detailed statistics are presented in Table. R12.

WMYn was trained on all samples from cohort 3, with 3000 epochs set and an MS2 tolerance of 0.01 Da for calculating cosine similarity. The final cosine similarity between the predicted and experimental spectra, shown in Fig. R14, reveals a mean spectral similarity above 0.96 in both positive and negative ionization modes, demonstrating the universality of the WMYn model. We have also added additional explanatory text and images at the relevant benchmark experiment locations in the main text. We also revised manuscript in Line373, Supplementary Fig.12 and Source Data Supplementary Fig.12.

Figure.S14 related to Fig.S7

Fig. R14. Dotted line plot of WMYn predicted NIST SRM 1950 spectrogram similarity using serum cohort 3 samples. **a.** Cosine similarity of 44 lipid profiles to known lipid profiles predicted using WMYn in positive ionization mode. **b.** Cosine similarity of 43 lipid profiles to known lipid profiles predicted using WMYn in negative ionization mode.

Table R12. Detailed statistics results of WMYn in 87 “known lipids”.

Lipids (FA Composition)	Cosine Similarity	ESI mode
TG 18:1_18:1_18:1	0.9779	POS
TG 18:0_18:1_18:2	0.7729	POS
TG 18:0_18:1_18:1	0.9905	POS
TG 18:0_18:0_18:1	0.9955	POS
TG 17:1_18:1_18:2	0.9810	POS
TG 16:1_18:2_18:3	0.9857	POS
TG 16:1_18:2_18:2	0.9660	POS
TG 16:1_18:1_18:2	0.8141	POS
TG 16:1_18:1_18:1	0.9876	POS
TG 16:1_16:1_18:2	0.9967	POS
TG 16:0_18:1_20:1	0.9873	POS
TG 16:0_18:1_19:1	0.9899	POS
TG 16:0_18:1_18:2	0.9994	POS
TG 16:0_18:1_18:1	0.9853	POS
TG 16:0_17:1_18:1	0.9840	POS
TG 16:0_16:1_18:3	0.9923	POS
TG 16:0_16:1_18:2	0.9901	POS
TG 16:0_16:1_18:1	0.9959	POS
TG 16:0_16:0_18:2	0.9941	POS
TG 16:0_16:0_18:1	0.9857	POS
TG 14:0_16:1_18:2	0.9952	POS
TG 14:0_16:1_18:1	0.9775	POS
TG 14:0_16:0_18:2	0.9998	POS
TG 14:0_16:0_16:1	0.9925	POS
PC 16:0_18:1	0.9911	POS
LPC 18:2	0.9734	POS
LPC 18:0	0.9814	POS
LPC 16:0	0.9989	POS
Cer 18:1;2O/24:0	0.6870	POS
Cer 18:1;2O/23:0	0.9928	POS
Cer 18:1;2O/22:0	0.9876	POS
CE 18:2	0.8391	POS
DG 16:0_18:2	0.9879	POS
DG 18:1_18:1	0.9956	POS
DG 18:1_18:2	0.9934	POS
DG 18:2_18:2	0.9933	POS
Hex2Cer 18:1;2O/16:0	0.9986	POS
HexCer 18:1;2O/22:0	0.9820	POS
TG 15:1_16:0_18:1	0.9892	POS
TG 16:1_18:0_18:1	0.9963	POS
TG 17:0_18:1_18:1	0.9912	POS

TG 18:1_18:2_18:3	0.9914	POS
TG 18:1_18:1_20:1	0.9877	POS
TG 18:0_18:1_20:3	0.9906	POS
PE-O 18:1_20:4	0.9965	NEG
PE-O 16:1_20:4	0.9801	NEG
PC 18:1_20:4	0.9863	NEG
PC 18:1_18:2	1.0000	NEG
PC 18:0_22:6	1.0000	NEG
PC 18:0_22:4	0.9897	NEG
PC 18:0_20:4	0.9946	NEG
PC 18:0_20:3	0.9956	NEG
PC 18:0_18:2	1.0000	NEG
PC 16:0_22:6	1.0000	NEG
PC 16:0_22:5	0.9986	NEG
PC 16:0_22:4	0.9910	NEG
PC 16:0_20:4	0.9863	NEG
PC 16:0_20:3	0.9925	NEG
PC 16:0_20:1	0.9947	NEG
PC 16:0_18:3	1.0000	NEG
PC 16:0_18:2	1.0000	NEG
PC 16:0_18:1	1.0000	NEG
PC 16:0_16:1	0.9893	NEG
PC 16:0_16:0	1.0000	NEG
PC 14:0_18:1	0.9988	NEG
FA 22:5	0.9915	NEG
FA 19:1	0.9928	NEG
FA 16:1	0.9985	NEG
Cer 19:1;2O/24:0	0.9963	NEG
Cer 18:2;2O/22:0	0.9972	NEG
Cer 18:1;2O/26:0	1.0000	NEG
Cer 18:1;2O/24:0	0.9977	NEG
Cer 18:1;2O/23:0	0.9919	NEG
Cer 18:1;2O/22:0	0.9969	NEG
Cer 18:0;2O/24:0	0.9990	NEG
Cer 18:0;2O/22:0	0.9971	NEG
Cer 16:1;2O/24:0	0.9972	NEG
Cer 18:1;2O/16:0	1.0000	NEG
FA 18:2	1.0000	NEG
LPE 18:0	0.9798	NEG
PC 14:0_18:2	0.9820	NEG
PC 17:0_18:2	1.0000	NEG
PC 18:2_18:2	0.9938	NEG
PC 18:1_20:3	0.9868	NEG
PC-O 18:1_20:4	0.9863	NEG

PE 16:0_18:2	1.0000	NEG
PS-O 20:0_20:4	0.9179	NEG

Comment 20. Throughout the manuscript what does “higher recall” mean in context of your lipid annotations? Please clarify all the computer program and model specific language for the average lipidomics user. Please define these terms before use. Other terms to define for the average lipidomics user “expeditious querying”, “hierarchical library”, “fragment fingerprints”, etc.

Response: Thank you for your professional advice. “Recall” (Intersection Count of Tool and Article Annotations / Total Article Annotations) defined as the number of elements annotated both in the tool and published result, divided by the total number of compositions shown in the article. It is important to note that this requires the annotated results to have same composition annotations, which is a more in-depth annotation than C:DB, with a retention time error of less than 0.15 minutes. A “higher recall” represent a larger intersection count between the tools’ annotations and article’s. We also revised manuscript in Line 225, 266.

“Fingerprint spectra” also known as “fragment fingerprints”, refer to unique mass spectrometry profiles that capture specific features of lipid substances, in addition to the common characteristic spectra shared across various mass spectrometry systems. We also revised manuscript in Line 130.

“Querying” between spectra refers to the process of searching for and retrieving specific information from spectral data, which is often used in fields like mass spectrometry, analytical chemistry, and metabolomics. The “expeditious querying” module is a fast theoretical spectra search tool capable of performing millions of searches per second. It optimizes data structures using methods such as dichotomy and hash tables. By employing these methods, the EQ module enhances its search efficiency, enabling rapid and accurate spectral matching. We also revised manuscript in Line 110.

“ Hierarchical library” is a mass spectrometry database categorized by the researcher based on the analytical significance of each MS2 characteristic peak. The specific meaning of each level is detailed in the article.

For the four terms above, we have explained them in the corresponding places in the manuscript. We have addressed the terms in the article that may be challenging for readers to understand by incorporating them into a terminology explanation table, as shown in Table R4. We hope this will help readers gain a more comprehensive understanding of the concepts and algorithms presented in the article.

Reviewer #3 (Remarks on code availability):

Comment 21. I reviewed the code briefly but did not set up an environment and test due to time constraints on my side. There was a ReadMe and the code seemed well commented.

Response: Thank you for your attention and interest in LipidIN. We apologize for any delays in usability. In response to your suggestions, we have added sample videos (<https://github.com/LinShuhaiLAB/LipidIN>) and a web interface (<https://www.lifemetabolomics.cn/software>). We invite you to test these new features.

References

1. Stein, S.E. & Scott, D.R. Optimization and testing of mass spectral library search algorithms for compound identification. *Journal of the American Society for Mass Spectrometry* **5**, 859-866 (1994).
2. Elapavalore, A. et al. Adding open spectral data to MassBank and PubChem using open source tools to support non-targeted exposomics of mixtures. *Environmental Science: Processes & Impacts* **25**, 1788-1801 (2023).
3. Murphy, R.C., Fiedler, J. & Hevko, J. Analysis of Nonvolatile Lipids by Mass Spectrometry. *Chemical Reviews* **101**, 479-526 (2001).
4. Giuffrida, F., Destailats, F., Skibsted, L.H. & Dionisi, F. Structural analysis of hydroperoxy- and epoxy-triacylglycerols by liquid chromatography mass spectrometry. *Chem Phys Lipids* **131**, 41-49 (2004).
5. Merrill, A.H., Sullards, M.C., Allegood, J.C., Kelly, S. & Wang, E. Sphingolipidomics: High-throughput, structure-specific, and quantitative analysis of sphingolipids by liquid chromatography tandem mass spectrometry. *Methods* **36**, 207-224 (2005).
6. Domingues, M.R., Reis, A. & Domingues, P. Mass spectrometry analysis of oxidized phospholipids. *Chem Phys Lipids* **156**, 1-12 (2008).
7. Han, X., Yang, K. & Gross, R.W. Multi-dimensional mass spectrometry-based shotgun lipidomics and novel strategies for lipidomic analyses. *Mass Spectrom Rev* **31**, 134-178 (2012).
8. Liu, X. et al. Oxidized fatty acid analysis by charge-switch derivatization, selected reaction monitoring, and accurate mass quantitation. *Anal Biochem* **442**, 40-50 (2013).
9. Pham, T.H. et al. Targeting Modified Lipids during Routine Lipidomics Analysis using HILIC and C30 Reverse Phase Liquid Chromatography coupled to Mass Spectrometry. *Sci Rep* **9**, 5048 (2019).
10. Long, N.P. et al. Advances in Liquid Chromatography–Mass Spectrometry–Based Lipidomics: A Look Ahead. *Journal of Analysis and Testing* **4**, 183-197 (2020).
11. Tsugawa, H. et al. A lipidome atlas in MS-DIAL 4. *Nat Biotechnol* **38**, 1159-1163 (2020).
12. Zhou, Y. et al. Photoinduced Online Enrichment–Deglycosylation of Glycolipids for Enhancing Lipid Coverage and Identification in Single-Cell Mass Spectrometry. *Analytical Chemistry* **96**, 17576-17585 (2024).
13. Ishibashi, Y. et al. A novel ether-linked phytol-containing digalactosylglycerolipid in the marine green alga, *Ulva pertusa*. *Biochem Biophys Res Commun* **452**, 873-880 (2014).
14. van der Hooft, J.J., Ridder, L., Barrett, M.P. & Burgess, K.E. Enhanced acylcarnitine annotation in high-resolution mass spectrometry data: fragmentation analysis for the classification and annotation of acylcarnitines. *Front Bioeng Biotechnol* **3**, 26 (2015).
15. Yang, K., Jenkins, C.M., Dilthey, B. & Gross, R.W. Multidimensional mass spectrometry-based shotgun lipidomics analysis of vinyl ether diglycerides. *Anal Bioanal Chem* **407**, 5199-5210 (2015).
16. Facchini, L., Losito, I., Cataldi, T.R. & Palmisano, F. Ceramide lipids in alive and thermally stressed mussels: an investigation by hydrophilic interaction liquid chromatography-electrospray ionization Fourier transform mass spectrometry. *J Mass Spectrom* **51**, 768-781 (2016).
17. Gee, P.T., Liew, C.Y., Thong, M.C. & Gay, M.C. Vitamin E analysis by ultra-performance

- convergence chromatography and structural elucidation of novel α -tocodienol by high-resolution mass spectrometry. *Food Chem* **196**, 367-373 (2016).
18. Lopalco, P., Stahl, J., Annese, C., Averhoff, B. & Corcelli, A. Identification of unique cardiolipin and monolysocardiolipin species in *Acinetobacter baumannii*. *Sci Rep* **7**, 2972 (2017).
 19. Jackson, S.N. et al. AP-MALDI Mass Spectrometry Imaging of Gangliosides Using 2,6-Dihydroxyacetophenone. *J Am Soc Mass Spectrom* **29**, 1463-1472 (2018).
 20. Mahmoodani, F. et al. Identification of Vitamin D3 Oxidation Products Using High-Resolution and Tandem Mass Spectrometry. *J Am Soc Mass Spectrom* **29**, 1442-1455 (2018).
 21. Nuzzo, G. et al. UPLC-MS/MS Identification of Sterol Sulfates in Marine Diatoms. *Mar Drugs* **17** (2018).
 22. Ding, J. et al. In-Silico-Generated Library for Sensitive Detection of 2-Dimethylaminoethylamine Derivatized FAHFA Lipids Using High-Resolution Tandem Mass Spectrometry. *Anal Chem* **92**, 5960-5968 (2020).
 23. Kim, J. & Hoppel, C.L. Identification of unusual phospholipids from bovine heart mitochondria by HPLC-MS/MS. *J Lipid Res* **61**, 1707-1719 (2020).
 24. Linke, V. et al. A large-scale genome-lipid association map guides lipid identification. *Nat Metab* **2**, 1149-1162 (2020).
 25. Chao, H.C. & McLuckey, S.A. Manipulation of Ion Types via Gas-Phase Ion/Ion Chemistry for the Structural Characterization of the Glycan Moiety on Gangliosides. *Anal Chem* **93**, 15752-15760 (2021).
 26. Sikorskaya, T.V., Efimova, K.V. & Imbs, A.B. Lipidomes of phylogenetically different symbiotic dinoflagellates of corals. *Phytochemistry* **181**, 112579 (2021).
 27. Wang, H. et al. Comprehensive Lipidomic Analysis of Three Edible Brown Seaweeds Based on Reversed-Phase Liquid Chromatography Coupled with Quadrupole Time-of-Flight Mass Spectrometry. *J Agric Food Chem* **70**, 4138-4151 (2022).
 28. Cudlman, L. et al. Characterization of Triacylglycerol Estolide Isomers Using High-Resolution Tandem Mass Spectrometry with Nano-electrospray Ionization. *Biomolecules* **13** (2023).
 29. Körber, T.T., Sitz, T., Abdalla, M.A., Mühling, K.H. & Rohn, S. LC-ESI-MS/MS Analysis of Sulfolipids and Galactolipids in Green and Red Lettuce (*Lactuca sativa* L.) as Influenced by Sulfur Nutrition. *Int J Mol Sci* **24** (2023).
 30. Zhou, L. et al. Identification and discrimination of lili bulb origins based on lipidomics using UHPLC-QE-Orbitrap/MS/MS combined with chemometrics analysis. *Journal of Food Composition and Analysis* **123**, 105512 (2023).
 31. Chandramouli, A. & Kamat, S.S. A Facile LC-MS Method for Profiling Cholesterol and Cholesteryl Esters in Mammalian Cells and Tissues. *Biochemistry* **63**, 2300-2309 (2024).
 32. Nicoll, C.R. et al. In vitro construction of the COQ metabolon unveils the molecular determinants of coenzyme Q biosynthesis. *Nature Catalysis* **7**, 148-160 (2024).
 33. Qi, C. et al. Mass Spectrometry Imaging for the Characterization of C=C Localization in Unsaturated Lipid Isomers at the Single-Cell Level. *Anal Chem* (2024).
 34. Roman, D. et al. Structure Revision of a Widespread Marine Sulfonolipid Class Based on Isolation and Total Synthesis. *Angew Chem Int Ed Engl* **63**, e202401195 (2024).
 35. Rudt, E., Schneider, S. & Hayen, H. Hyphenation of Liquid Chromatography and Trapped Ion Mobility - Mass Spectrometry for Characterization of Isomeric Phosphatidylethanolamines with Focus on N-Acylated Species. *J Am Soc Mass Spectrom* **35**, 1584-1593 (2024).

36. Singh, S. et al. PLD3 and PLD4 synthesize S,S-BMP, a key phospholipid enabling lipid degradation in lysosomes. *bioRxiv* (2024).
37. Koelmel, J.P. et al. LipidMatch: an automated workflow for rule-based lipid identification using untargeted high-resolution tandem mass spectrometry data. *BMC Bioinformatics* **18**, 331 (2017).

Comments on NCOMMS-24-49156A

The authors responded to the Reviewer’s comments. However, the formulation and definition of the machine learning architecture still need to be clarified. The current manuscript is not suitable for publication in Nature Communications.. The questions and comments are listed below.

1 On the response to Comment 1

First, the Kolmogorov–Arnold representation theorem (KART) is shown for the discussion below. The authors defined KART as

$$f(X) = f(x_1, x_2, \dots, x_n) = \sum_{q=1}^{2n+2} \Phi_q \left(\sum_{p=1}^n \phi_{q,p}(x_p) \right), \quad (1)$$

where $X \in \mathbb{R}^n$, $x_p \in \mathbb{R}$, $\phi_{q,p} : \mathbb{R} \rightarrow \mathbb{R}$, and $\Phi_q : \mathbb{R} \rightarrow \mathbb{R}$. In general, the functions are different for different indices, i.e., $\phi_{q,p} \neq \phi_{q',p'}$ for $q \neq q'$ or $p \neq p'$ and $\Phi_q \neq \Phi_{q'}$ for $q \neq q'$.

In the response, the authors described that they tried to obtain the “true spectrum” \mathbf{S} as a function of which arguments are mass spectrometry conditions (Y_1), chromatography conditions (Y_2), ionization voltage (Y_3), the nature of the compound Y_4 , and other properties: $\mathbf{S} = \Phi(Y_1, Y_2, \dots, Y_n)$. In this case, Y_1, \dots, Y_n are the inputs of the function \mathbf{S} .

1.1. What do the authors represent by the term “true spectrum?” Do they want to express the spectrum obtained by an experiment?

On the other hand, the authors described “ Y_1, \dots, Y_n denote the output matrix, corresponding to the theoretical spectra in the chromatographic and mass spectrometric conditions” in the revised manuscript. They clearly expressed that Y_1, \dots, Y_n are the output.

1.2. Whether are Y_1, \dots, Y_n inputs or outputs? The authors need to clarify Y_1, \dots, Y_n are input for what and output from what.

1.3. Are \mathbf{S} , Y_n real numbers? Is it possible to represent a spectrum by a real number? Is the “chromatography conditions” represented by a real number? What did the authors represent using the bold italic style of \mathbf{S} ?

In the response, the authors represented \mathbf{S} as

$$\mathbf{S} = \sum_{J=1}^m \Phi_J, \quad (2)$$

and they called it “a finite unitary function.”

1.4. What is the “finite unitary function?”

The authors also mentioned that “ m is a finite constant.” I believe that m is an integer because this appears as the maximum number of the index J in the summation in Eq. (2).

1.5. Were there any assumptions on m ? How is it determined? What is the relationship between the m in Eq. (2) and the n in KART (1)?

They introduced the following function:

$$\mathbf{F} = \sum_{i=1}^n f_i(WX + B). \quad (3)$$

The dimension of X has not been mentioned when it was introduced. But it could be found after showing the relationship between Φ and $f_i(WX + B)$, i.e., X , W , and B are matrices. Then, the argument of f_i is a matrix. Usually, the function of a matrix argument is defined by the element-wise function, like

$$f_i(Y) := \begin{pmatrix} f_i(Y_{11}) & f_i(Y_{12}) & & f_i(Y_{1\ell'}) \\ f_i(Y_{21}) & f_i(Y_{22}) & & f_i(Y_{2\ell'}) \\ \vdots & \vdots & \ddots & \vdots \\ f_i(Y_{\ell'1}) & f_i(Y_{\ell'2}) & & f_i(Y_{\ell'\ell'}) \end{pmatrix}, \quad (4)$$

where Y is an $\ell \times \ell'$ matrix. Therefore, \mathbf{F} is expected to be a matrix.

1.6. What are the dimensions of \mathbf{F} , X , W and B ? What did the authors represent by the bold italic style of \mathbf{F} ?

1.7. What is the relationship between n in Eq. (3) and the dimension of X ?

The authors defined the following relation:

$$\Phi(Y_1, \dots, Y_n) = \sum_{j=1}^{2n+1} \Phi_j \left(\sum_{i=1}^n f_i(WX + B) \right), \quad (5)$$

where X represents the input matrix, consisting of x_1, \dots, x_n , where each x_i is a spectrum peak.

1.8. What is the “spectrum peak?” Do they have units?

1.9. What are the dimensions of X and x_i ?

1.10. What is the relationship between the matrix element of X and x_i ? For example,

$$X_{12} = x_2. \quad (6)$$

1.11. What is the relationship between Y_p and x_p ? $\Phi(Y_1, Y_2, \dots, Y_n)$ means a function of the real number arguments Y_p ($p = 1, 2, \dots, n$). However, the authors identified that X is the input, and X consists of x_1, \dots, x_n . What is the input of Φ ? Is Φ the function of x_1, \dots, x_n ? What are the mathematical definitions of Y_p ?

The authors revealed that f_i is the sigmoid linear unit (SiLU) function. If f_i is SiLU, the suffix i of f_i can be omitted, i.e.,

$$f_i(x) = f_S(x), \quad (7)$$

where f_S is the SiLU function and it does not has the “ i ” index. We can represent \mathbf{F} as

$$\mathbf{F} = n f_S(WX + B). \quad (8)$$

Using this representation, Eq. (5) is rewritten as

$$\Phi(Y_1, \dots, Y_n) = \sum_{j=1}^{2n+1} \Phi_j(n f_S(WX + B)). \quad (9)$$

The right-hand side of Eq. (8) does not have any indexes.

I should consider the possibility of a typo in the definition of W , X , and B . If they are matrices, $\sum_{i=1}^n f_i(WX + B)$ or $\sum_{i=1}^n f_S(WX + B)$ cannot be a number. If it is not a number, we cannot discuss KART. Therefore, I assume that W is a $n \times n$ matrix and X and B are n -dimensional vector, In this case, \mathbf{F} is a n -dimensional vector defined as

$$\mathbf{F} = \sum_{i=1}^n f_S(z_i), \quad (10)$$

where z_i is a matrix element of $WX + B$, i.e.,

$$Z = WX + B, \quad (11)$$

$$Z = \begin{pmatrix} z_1 \\ z_2 \\ \vdots \\ z_n \end{pmatrix}. \quad (12)$$

Finally, we obtain the following equation:

$$\Phi(Y_1, \dots, Y_n) = \sum_{j=1}^{2n+1} \Phi_j \left(\sum_{i=1}^n f_S(z_i) \right). \quad (13)$$

I additionally assume that the input is not Y_i but x_i to clarify the input of the function, although the relationship between Y_i and x_i is still unclear.

$$\Phi(x_1, \dots, x_n) = \sum_{j=1}^{2n+1} \Phi_j \left(\sum_{i=1}^n f_S(z_i) \right). \quad (14)$$

Here, I introduce a new representation of $\sigma(z_i)$ to clarify the dependency of the input x_i as

$$\phi_i(x_1, x_2, \dots, x_n) = f_S(z_i) = f_S\left(\sum_{k=1}^n w_{ik}x_k + b_k\right). \quad (15)$$

Finally, the following representation is obtained,

$$\Phi(x_1, \dots, x_n) = \sum_{j=1}^{2n+1} \Phi_j\left(\sum_{i=1}^n \phi_i(x_1, x_2, \dots, x_n)\right). \quad (16)$$

Comparing Eqs. (1) and (16), we can recognize the critical differences between them:

- ϕ_i in Eq. (16) is not a univariate function.
- ϕ_i in Eq. (16) does not have the j index.

These differences are critical for KART, which requires univariate functions and a j index for each function to distinguish between different nonlinear transformations in the model. Therefore, it appears that the proposed function $\Phi(x_1, \dots, x_n)$ does not directly correspond to KART. **From the discussion above, it is difficult to claim that Φ is a kind of KART-inspired function.**

1.12. I recommend deleting the description of KART from the manuscript.

Despite the discussion of whether Φ is a kind of KART, the implementation of the function Φ_j should be identified. The authors claimed that “ Φ_j is a learnable nonlinear activation.”

1.13. What is the implementation of Φ_j ? Is this an artificial neural network?

I continue the discussion with the assumption that W is the $n \times n$ matrix and X and B are the n -dimensional vectors. I additionally assume that Φ_j is realized by an artificial neural network (ANN). I employ the ANN with a single hidden layer to clarify the discussion. This assumption does not lose generality, and the extension to the multi-layer perceptron is obvious. In this case,

$$\begin{pmatrix} \Phi_1(x) \\ \Phi_2(x) \\ \vdots \\ \Phi_{2n+1}(x) \end{pmatrix} = f\left(W^{(1)}x + B^{(1)}\right), \quad (17)$$

where $W^{(1)}$ a $(2n+1 \times 1)$ -dimensional weight matrix (vector), $B^{(1)}$ is a $(2n+1)$ -dimensional bias vector, and f is an activation function. On the other hand, we can introduce another expression of the argument of Φ_j in Eq. (14),

$$\sum_{i=1}^n f_S(z_i) = I_{(n)}f_S(Z) = I_{(n)}f_S(WX + B) \quad (18)$$

$$I_{(n)} = (1 \quad 1 \quad \dots \quad 1), \quad (19)$$

where the dimension of $I_{(n)}$ is n . Using Eq. (18), Eq. (14) can be represented as

$$\Phi(x_1, x_2, \dots, x_n) = I_{(2n+1)} f\left(W^{(1)} (I_{(n)} f_S(WX + B)) + B^{(1)}\right), \quad (20)$$

$$I_{(2n+1)} = \begin{pmatrix} 1 & 1 & \dots & 1 \end{pmatrix}, \quad (21)$$

where the dimension of $I_{(2n+1)}$ is $2n + 1$.

$$\Phi(x_1, x_2, \dots, x_n) = f_L\left(I_{(2n+1)} f\left(W^{(1)} f_L(I_{(n)} f_S(WX + B) + 0) + B^{(1)}\right) + 0\right), \quad (22)$$

where f_L is the linear activation function, i.e., $x = f_L(x)$. It is obvious that Eq. (22) is a special case of the ANN with four hidden layers, which is represented as

$$\Phi(X) = f^{(4)}\left(\mathcal{W}^{(4)} f^{(3)}\left(\mathcal{W}^{(3)} f^{(2)}\left(\mathcal{W}^{(2)} f^{(1)}\left(\mathcal{W}^{(1)} X + \mathcal{B}^{(1)}\right) + \mathcal{B}^{(2)}\right) + \mathcal{B}^{(3)}\right) + \mathcal{B}^{(4)}\right). \quad (23)$$

Actually, Eq. (22) is recovered from Eq. (23) with the following equations:

$$\mathcal{W}^{(1)} = W, \quad (24)$$

$$\mathcal{W}^{(2)} = I_{(n)}, \quad (25)$$

$$\mathcal{W}^{(3)} = W^{(1)}, \quad (26)$$

$$\mathcal{W}^{(4)} = I_{(2n+1)}, \quad (27)$$

$$\mathcal{B}^{(1)} = B, \quad (28)$$

$$\mathcal{B}^{(2)} = 0, \quad (29)$$

$$\mathcal{B}^{(3)} = B^{(1)}, \quad (30)$$

$$\mathcal{B}^{(4)} = 0, \quad (31)$$

$$f^{(1)} = f_S, \quad (32)$$

$$f^{(2)} = f_L, \quad (33)$$

$$f^{(3)} = f, \quad (34)$$

$$f^{(4)} = f_L. \quad (35)$$

From the discussion above, WMYn does not have a KART-like architecture but has a restricted ANN. This means that the WNYn architecture has less representability rather than the usual ANN due to the restrictions of Eqs. (25), (27), (29), (31), (33), and (35).

2 On the response to Comment 2

The authors explained the details of the WNYn architecture. However, it is still confusing.

They showed that “the first layer” of WNYn is defined as

$$H_1 = \alpha_1 ((W_1 X + B_1) \sigma(W_1 X + B_1)), \quad (36)$$

where H_1 is the latent matrix, W_1 and B_1 are the weight and bias matrices, α_1 and β_1 are the learnable scaling parameter and the learnable bias term, and σ is the SiLU activation function.

2.1. What are the dimensions of H_1 , X , W_1 , B_1 , α_1 , and β_1 ?

2.2. What is the relationship between H_1 and Eq. (5)? What do the authors mean by the term “the first layer?” The right hand side of Eq. (36) cannot be represented as $\sum_{i=1}^n \sigma(WX + B)$.

The authors introduced “the second layer”:

$$H_2 = \alpha_2 ((W_2 H_1 + B_2) \sigma(W_2 H_1 + B_2)), \quad (37)$$

where H_2 is the latent matrix, W_2 and B_2 are the weight and bias matrices, α_2 and β_2 are the learnable scaling parameter and the learnable bias term, and σ is the SiLU activation function.

2.3. What are the dimensions of H_2 , W_2 , B_2 , α_2 , and β_2 ?

2.4. What is the relationship between H_2 and Eq. (5)? What do the authors mean by the term “the second layer?”

The authors also introduced H_3 .

2.5. What is the relationship between H_3 and Eq. (5)?

3 On the response to Comment 3

The authors showed the results of the comparison between WNYn(KART) and WNYn(MLP).

3.1. The performance of an ANN depends on its architecture. A detailed architecture of WNYn(MLP) must be shown. The reason why the author chose the architecture for comparison with WNYn(KART) is also required.

3.2. WNYn(KART) is expected to have less representability of a nonlinear function than WNYn (MLP). Why is WNYn(KART) superior to WNYn(MLP)?

I appreciate all the in-depth responses to my comments. I especially was impressed and appreciated the table generated containing all technical terms and the corresponding definitions.

To complete the review and accept this manuscript, I would like to be able to test the software. Currently, I have not been able to successfully run the software on my own data (which I would need for just a simple validation that the resulting annotations are realistic for a new test data), nor could I get the software to work on the test data using the exact same parameters and zip file provided.

<https://www.lifemetabolomics.cn/LipidIN>

The user interface was relatively straight forward.

I am not sure how mzML conversion parameters (e.g. compressed versus not compressed files) effect how the software runs so a tutorial for mzML conversion on the page may be helpful.

Note you mention including your file as yzip format, only zip format exists, must be a misspelling?

Note “Working” in all portions of the webpage, which are the buttons to click when running the data, should be changed to “Run” or “Process Data”

Also note that the 2GB limit makes sense given that you are potentially running the algorithms on your own server? But this limit will significantly limit the usecase as some people will have 100 of GBs worth of files. At this point maybe LipidIN does not have to support 100 of GBs worth of files because it only needs representative MS/MS files (2 GB is still to small, at least 10 GB), but for the quant workflow you mentioned it would definitely need to have higher GB available. The quant workflow could benefit from being integrated into the WebApp...

I ran one single mzML file that was not too large, and I got the following error:

“系统接口请求超时”

“System interface request timeout”

I had it formatted correctly I believe: data.zip/data/ddMS28387_Pos.mzML and I converted with MSConvert using the following parameters:

peakPicking: vendor msLevel=1-2

threshold: absolute 1000 most-intense

I also tried a parameterless conversion to mzML and your own demo data (just downloaded, reuploaded, and got the same error)

And using the demo file that you provided and got the same error. I tested my internet connection and speed and it was fine.

step0 step1 step2 step3 step4

Complete process

data. zip should be less than 2G

MS2: a value of 0-1, You can define according to your needs.

ESI:

'p' for positive ionization mode,

'n1' for negative ionization mode [M+COOH]-,

'n2' for negative ionization mode [M+CH3COO]-.

ppm1: MS1 m/z tolerance at parts per million (ppm).You can define according to your needs.

ppm2: MS2 m/z tolerance at parts per million (ppm).You can define according to your needs

The file must be data.zip.

param key

ESI

param value

'n'

param key

MS2

param value

.01

param key

PPM1

param value

10

param key

PPM2

param value

40

File upload

data.zip

upload

100%

Only one file in yzip format can be uploaded,with a fixed name data.zip.

working

download code

12/22/2024 10:39am EST was the time ran

REVIEWER COMMENTS

Reviewer #1 (Remarks to the Author):

We sincerely appreciate your thorough review and valuable feedback. Based on the provided evidence, we revised and improved the text in the Reverse Lipidomics section to ensure clarity. Thank you again for your constructive input!

First, we define Y_1, \dots, Y_n , which represent the factors influencing the spectra, such as the instrumental systems, the stationary phase materials in high-performance liquid chromatography (HPLC) analysis, and the applied voltages, and other factors. Here, we use $\Phi(Y_1, \dots, Y_n)$ to denote the ground truth intensity value at a specific m/z in the spectrum. For example, in Fig. R1.a, the intensity value of 80.08 corresponds to m/z 756.5907, emphasizing that this value is influenced by multiple variables. Additionally, it is important to note that Y_1, \dots, Y_n and Φ are formal representations and are not intended as inputs or outputs of the model. In another word, Φ is a pseudo-function used solely to emphasize the multivariate nature of the factors influencing the spectrum.

Figure R1. An example of a spectrum. a. The spectrum of lipid PC O-18:1-18:1 ([M+FA-H]⁻) retrieved from the PubChem (<https://pubchem.ncbi.nlm.nih.gov>). **b.** Numerical values of the m/z (mass-to-charge ratio) and intensity from the spectrum. It should be noted that only Top 5 peaks are shown in this spectrum.

Secondly, we define the experimental spectrum x_i and the input matrix X . Each experimental spectrum x_i can be expressed as:

$$x_i = \begin{pmatrix} X_{1i} \\ X_{2i} \\ \vdots \\ X_{mi} \end{pmatrix}. \quad (1)$$

For the example in Fig.R1, $x_i = \begin{pmatrix} 0 \\ \vdots \\ 100 \\ \vdots \\ 80.08 \\ \vdots \end{pmatrix}$. It is important to note that each x_i must

be aligned with other spectra based on the m/z , making x_i a sparse vector.

Here, we assume that there are n spectra as input, and the merged m/z values amount to m . It is important to ensure that the merged m/z do not have other m/z values within a certain tolerance, typically set to 10 ppm. Thus, $X = (x_1, \dots, x_n)$ can be represented as:

$$X = \begin{pmatrix} X_{11} & \cdots & X_{1n} \\ \vdots & \ddots & \vdots \\ X_{m1} & \cdots & X_{mn} \end{pmatrix}_{m \times n}. \quad (2)$$

Next, we reintroduce the network structure of the first stage of WMYn and provide additional explanations and proofs. In this framework, we introduce two learnable parameters, α and β . Here, α is a learnable scaling parameter initialized to 1, and β is a learnable shifting parameter initialized to 0. In the implementation, both α and β are represented as tensors and wrapped using `nn.Parameter`, making them trainable and updatable by the optimizer during training. We assume that these two parameters are not constants and are neither fixed at 0 nor 1. Instead, they are dynamic, learnable values that can adapt during training. When the input spectrum matrix X is fed into the model, a bias adjustment is applied using these parameters. Additionally, the hidden layer in the first stage of the network is set to 512 units:

$$WX + B = \begin{pmatrix} W_{1,1} & \cdots & W_{1,m} \\ \vdots & \ddots & \vdots \\ W_{512,1} & \cdots & W_{512,m} \end{pmatrix}_{512 \times m} \begin{pmatrix} X_{1,1} & \cdots & X_{1,n} \\ \vdots & \ddots & \vdots \\ X_{m,1} & \cdots & X_{m,n} \end{pmatrix}_{m \times n} + \begin{pmatrix} B_1 \\ \vdots \\ B_{512} \end{pmatrix}, \quad (3)$$

the element in the i row and j column of the result after applying the bias adjustment can be expressed as:

$$(WX + B)_{i,j} = \sum_{p=1}^m W_{i,p} X_{p,j} + B_i, \quad \text{where } i = 1, \dots, 512, j = 1, \dots, n, \quad (4)$$

where, W is the weight matrix, and B is the bias vector. Both W and B are learnable parameters within the network.

At any node in the first layer, the bias-adjusted result will be passed through the SiLU activation function:

$$H_1 = \alpha_1(\text{SiLU}(W_1X + B_1)) + \beta_1 = \begin{pmatrix} h_{1,1}^{(1)} & \cdots & h_{1,n}^{(1)} \\ \vdots & \ddots & \vdots \\ h_{512,1}^{(1)} & \cdots & h_{512,n}^{(1)} \end{pmatrix}, \quad (5)$$

expanding the above equation for the element at the i row and j column after applying the SiLU activation function:

$$h_{i,j}^{(1)} = \alpha_1 \text{SiLU}\left(\sum_{p=1}^m W_{i,p}^{(1)} X_{p,j} + B_i^{(1)}\right) + \beta_1, \quad (6)$$

where, $h_{i,j}^{(1)}$ represents the result of the element in the i row and j column of the matrix after the first bias adjustment, followed by the application of the SiLU activation function, and the scaling by the learnable parameters α_1 and β_1 . $W_{i,p}^{(1)}$ represents the element in the i row and p column of the weight matrix W_1 from the first layer of the network. $B_i^{(1)}$ represents the i row in B_1 bias term.

Similar to the processing in the first layer, the result after the second layer can be expressed as:

$$H_2 = \alpha_2(\text{SiLU}(W_2H_1 + B_2)) + \beta_2 = \begin{pmatrix} h_{1,1}^{(2)} & \cdots & h_{1,n}^{(2)} \\ \vdots & \ddots & \vdots \\ h_{512,1}^{(2)} & \cdots & h_{512,n}^{(2)} \end{pmatrix}, \quad (7)$$

In matrix form, $W_2H_1 + B_2$ can be expressed as:

$$W_2H_1 + B_2 = \begin{pmatrix} W_{1,1}^{(2)} & \cdots & W_{1,512}^{(2)} \\ \vdots & \ddots & \vdots \\ W_{512,1}^{(2)} & \cdots & W_{512,512}^{(2)} \end{pmatrix} \begin{pmatrix} h_{1,1}^{(1)} & \cdots & h_{1,n}^{(1)} \\ \vdots & \ddots & \vdots \\ h_{512,1}^{(1)} & \cdots & h_{512,n}^{(1)} \end{pmatrix} + \begin{pmatrix} B_1^{(2)} \\ \vdots \\ B_{512}^{(2)} \end{pmatrix}. \quad (8)$$

The expression for the i row and j column element of H_2 , denoted as $h_{i,j}^{(2)}$, can be written as:

$$h_{i,j}^{(2)} = \alpha_2 \text{SiLU}\left(\sum_{q=1}^{512} W_{i,q}^{(2)} (\alpha_1 \text{SiLU}\left(\sum_{p=1}^m W_{i,p}^{(1)} X_{p,j} + B_i^{(1)}\right) + \beta_1) + B_i^{(2)}\right) + \beta_2, \quad (9)$$

where, $W_{i,q}^{(2)}$ represents the weight connecting the q column in the second hidden

layer to the i row in the second hidden layer, The bias term $B_i^{(2)}$ allows the model to independently shift the activation output of the i row, enhancing the network's flexibility. Additionally, the learnable scaling parameter α_2 , dynamically adjusts the magnitude of the activations in the second layer during training, while the learnable shifting parameter β_2 , provides the ability to shift the output as needed, further improving the model's adaptability.

We denote $\Phi_{q,i,j}(\sum_{p=1}^m f_{p,i,j}(X_{p,j}))$, $f_{p,i,j}(X_{p,j})$ and $\psi_{p,q}(X_{p,j})$:

$$f_{p,i,j}(X_{p,j}) = W_{i,p}^{(1)} X_{p,j}, \quad (10)$$

$$\psi_{p,q}(X_{p,j}) = I(q \in \{1, \dots, 512\}) f_{p,i,j}(X_{p,j}), \quad (11)$$

$$\Phi_{q,i,j}(\sum_{p=1}^m \psi_{p,q}(X_{p,j})) = W_{i,q}^{(2)} (\alpha_1 \text{SiLU}(\sum_{p=1}^m I(q) f_{p,i,j}(X_{p,j}) + b_i^{(1)}) + \beta_1), \quad (12)$$

here, $I(q \in \{1, \dots, 512\})$ is an indicator function, which equals 1 when $q \in \{1, \dots, 512\}$, and 0 otherwise.

Based on the definitions above, $h_{i,j}^{(2)}$ can be expressed as:

$$h_{i,j}^{(2)} = \alpha_2 \text{SiLU}(\sum_{q=1}^{512} \Phi_{q,i,j}(\sum_{p=1}^m \psi_{p,q}(X_{p,j})) + b_i^{(2)}) + \beta_2, \quad (13)$$

where $\psi_{p,q}(X_{p,j})$ is a univariate function with an index q . Referring to the KART framework:

$$f(X) = f(x_1, \dots, x_n) = \sum_{q=1}^{2n+1} \Phi_q(\sum_{j=1}^n \varphi_{q,p}(x_p)), \quad (14)$$

we can conclude that the intensity $f(X_{1,j}, \dots, X_{n,j})$ of the spectrum at the j m/z , which is influenced by the multivariate factors Y_1, \dots, Y_n , can be represented by a finite number of univariate functions:

$$f(X_{1,j}, \dots, X_{n,j}) = \alpha_2 \text{SiLU}(\sum_{q=1}^{512} \Phi_{q,i,j}(\sum_{p=1}^m \psi_{p,q}(X_{p,j})) + b_i^{(2)}) + \beta_2, \quad (15)$$

our proposed network architecture is inspired by KART, utilizing a two-layer structure to perform feature learning for spectrum data.

Comment 1. What do the authors represent by the term ‘‘true spectrum?’’ Do they want to express the spectrum obtained by an experiment?

Response: Thank you for your interest. In this context, the term “true spectrum” refers to spectra obtained using authentic standard references under specific liquid chromatography-mass spectrometry (LC-MS) conditions.

In our study, we use spectra obtained from lipid standards under identical LC-MS conditions as the ground truth in our model. Additionally, the “true spectrum” in this context does not equate to a spectrum obtained from a single arbitrary experiment, as spectra derived from mixed samples contain various background noises. Only spectra obtained from individual standard compounds are considered relatively reliable.

From the perspective of model input, an example of a “true spectrum” is the spectrum of PC O-18:1_18:1, as shown in Fig.R1.a. The spectral information can be represented as a data frame, where the m/z values serve as row names and the corresponding intensities are represented as numerical vectors, as illustrated in FigR1.b. By taking the union of the row names within a specified m/z tolerance, we can ultimately obtain the vector x_i . The “true spectrum” of Fig.R1.a can thus be represented as:

$$x_i = \begin{pmatrix} 0 \\ \vdots \\ 100 \\ \vdots \\ 80.08 \\ \vdots \\ 30.03 \\ \vdots \\ 20.02 \\ 20.02 \\ \vdots \end{pmatrix}. \quad (16)$$

Comment 2. Whether are Y_1, \dots, Y_n inputs or outputs? The authors need to clarify Y_1, \dots, Y_n input for what and output from what.

Response: Thank you for your reminder. We apologize for the incorrect representation of Y_1, \dots, Y_n in Methods section of the main text. Y_1, \dots, Y_n is neither an input nor an output of the model, and $\Phi(Y_1, \dots, Y_n)$ is a pseudo-function. This notation is intended to emphasize that the predicted spectrum is influenced by multiple variables. Therefore,

we have revised lines 675–721 of the main manuscript.

Comment 3. Are \mathbf{S} , Y_n real numbers? Is it possible to represent a spectrum by a real number? Is the “chromatography conditions” represented by a real number? What did the authors represent using the bold italic style of \mathbf{S} ?

Response: Thank you for your question. Here, both \mathbf{S} and Y_n are not simple numerical values. Specifically, \mathbf{S} represents a true spectrum. For example, as shown in Fig.R1.a, the spectrum of \mathbf{S} is depicted, and within the model, it is structured as illustrated in Fig.R1.b. In this structure, the first column contains the row names (m/z values), and the second column contains the corresponding intensity values. In the manuscript, \mathbf{S} refers to an entire spectrum, whereas $S_{i,j}$ denotes an individual value within the spectrum, such as 100 or 80.08.

“Chromatography conditions” cannot be represented numerically. Instead, they may refer to aspects such as the stationary phase materials used in reverse-phase high-performance liquid chromatography (HPLC) analysis or the chromatography gradient employed. Consequently, any intensity value in the true spectrum $S_{i,j}$ can be expressed as a multivariate function that accounts for these variables:

$$S_{i,j} = \Phi(Y_1, Y_2, \dots, Y_n). \quad (21)$$

Furthermore, we use a bold italic \mathbf{S} here to emphasize to the reader that the true spectrum is a vector of significant importance.

Comment 4. What is the “finite unitary function?”

Response: Thank you for your professional and detailed suggestions. In our response, the expression was a typo error and should be “finite univariate function.” What we intended to convey is that, according to KART, any multivariate continuous function $S_{i,j} = \Phi(Y_1, Y_2, \dots, Y_n)$ can be represented as a finite number of single-variable continuous functions within a two-layer nested summation.

Comment 5. Were there any assumptions on m ? How is it determined? What is the

relationship between the m in Eq. (2) and the n in KART (1)?

Response: We sincerely apologize for the misleading statement in our previous response regarding m . The description of m was incorrect. Please omit the assumptions on m . Furthermore, the m mentioned earlier should correspond to $2n+1$ in equation KART (1), which is a finite integer. In the actual model, m represents the dimension of the output of the second layer in Stage 1 and is related to the dimensional settings of the hidden layer.

Comment 6. What are the dimensions of F , X , W and B ? What did the authors represent by the bold italic style of F ?

Response: F , X , W , and B do not have fixed dimensions, however, their dimensions are interrelated. Specifically, the dimensions of F , W , and B depend on the dimensions of the input matrix X and the dimensions of the hidden layer. For the purposes of this study, we assume that X is an $m \times n$ matrix:

$$X = \begin{pmatrix} X_{11} & \cdots & X_{1n} \\ \vdots & \ddots & \vdots \\ X_{m1} & \cdots & X_{mn} \end{pmatrix}_{m \times n}. \quad (22)$$

In the WMYn model, the hidden layer is set to 512 units. Consequently, the weight matrix W is defined as follows:

$$W = \begin{pmatrix} W_{11} & \cdots & W_{1m} \\ \vdots & \ddots & \vdots \\ W_{512,1} & \cdots & W_{512m} \end{pmatrix}_{512 \times m}. \quad (23)$$

Similarly, the bias vector B is defined as:

$$B = \begin{pmatrix} b_1 \\ \vdots \\ b_{512} \end{pmatrix}_{512 \times 1}. \quad (24)$$

Here, F represents a matrix that results from transforming the input matrix X .

$$F_{i,j} = \alpha_2 \text{SiLU} \left(\sum_{q=1}^{512} \Phi_{q,i,j} \left(\sum_{p=1}^m \psi_{p,q}(X_{p,j}) \right) + b_i^{(2)} \right) + \beta_2. \quad (25)$$

Furthermore, we use a bold italic F here to emphasize to the reader that the true spectrum is a matrix of significant importance.

Comment 7. What is the relationship between n in Eq. (3) and the dimension of X ?

Response: Here, X represents the entire set of input spectra. When there are n input

spectra and the merged m/z values amount to m , the dimension of X is $m \times n$, where n denotes the number of input spectra. The value 512 indicates the number of units in the hidden layer that we have configured.

Comment 8. What is the “spectrum peak?” Do they have units?

Response: A “spectrum peak” refers to the position of an intensity signal peak in the mass spectrum, which arises from the characteristics or phenomena of specific substances, as illustrated in Fig.R1.a. In the spectrum, the x-axis represents the mass-to-charge ratio (m/z), typically measured in unified atomic mass units (u), while the y-axis denotes intensity or relative intensity. We can also replace “spectrum peak” with “ion peak” for a better understanding. They have no units.

Comment 9. What are the dimensions of X and x_i ?

Response: Thank you for your attention to the details. Assuming that the input matrix X has dimensions $m \times n$, each individual x_i has dimensions $m \times 1$.

Comment 10. What is the relationship between the matrix element of X and x_i ? For example, $X_{12} = x_2$ (6)

Response: Thank you for your attention to the details. The relationship between X and x_i is defined as $X = (x_1, x_2, \dots, x_n)$. In our model, $X_{12} \neq x_2$. We illustrate the relationship between X and x_i in Fig.R2, utilizing color blocks for differentiation. Specifically, X_{12} is highlighted within a red block, while x_2 is enclosed in a blue block.

a

$$X = \begin{pmatrix} x_1 & x_2 & \dots & x_n \\ X_{11} & X_{12} & \dots & X_{1n} \\ X_{21} & X_{22} & \dots & X_{2n} \\ \vdots & \vdots & \ddots & \vdots \\ X_{m1} & X_{m2} & \dots & X_{mn} \end{pmatrix}$$

Figure R2: Schematic diagram illustrating the relationship between X and x_i .

Comment 11. What is the relationship between Y_p and x_p ? $\Phi(Y_1, \dots, Y_n)$ means a function of the real number arguments Y_p ($p = 1, 2, \dots, n$). However, the authors identified that X is the input, and X consists of x_1, \dots, x_n . What is the input of Φ ? Is Φ the function of x_1, \dots, x_n ? What are the mathematical definitions of Y_p ?

Response: Thank you for your questions! On one hand, Y_p represents a series of variables that influence the spectrum and serve as inputs to the pseudo-function Φ . These variables are neither inputs nor outputs of the model itself but are used to emphasize that the spectrum is affected by multiple factors.

As aforementioned above, on the other hand, X serves as an input and provided a detailed explanation of how each element within X is processed within the model.

Comment 12. I recommend deleting the description of KART from the manuscript.

Response: We sincerely appreciate your thorough reading and insightful feedback. In our manuscript, Equation (14) and KART's Equation (13) demonstrate that the intensity $f(X_{1,j}, \dots, X_{n,j})$ of the j m/z in the spectrum, influenced by multiple variables Y_1, \dots, Y_n , can be expressed using a finite number of univariate functions $f(X_{1,j}, \dots, X_{n,j}) = \alpha_2 \text{SiLU}\left(\sum_{q=1}^{512} \Phi_{q,i,j}(\sum_{p=1}^m \psi_{p,q}(X_{p,j})) + b_i^{(2)}\right) + \beta_2$. Our network architecture is inspired by KART, employing a two-layer structure for spectral feature learning. Therefore, we intend to retain this portion of the description. Furthermore, our two-layer structure is merely inspired by KART, and we have not strictly implemented KART's methodology.

Comment 13. What is the implementation of Φ_j ? Is this an artificial neural network?

Response: We sincerely appreciate your comprehensive analysis and proof. The specific expression and derivation of Φ_j can be found in Equations (1) through (14). While we acknowledge your conclusion that this framework constitutes an artificial neural network, we believe that the inclusion of the learnable parameters α , and β

ensures that its nonlinear expressive capabilities are not limited.

Comment 14. What are the dimensions of H_1 , X , W_1 , B_1 , α_1 , and β_1 ?

Response: Thank you for your question. Firstly, α_1 , and β_1 are scalar parameters. The input in this context is X , which does not have fixed dimensions. We assume its dimensions to be $m \times n$, where n represents the number of input spectra and m denotes the combined number of m/z values. We have set the dimension of the hidden layer to 512, resulting in a weight matrix W_1 with dimensions $512 \times m$ and a bias vector B_1 with dimensions 512×1 . In practice, within the neural network, B_1 is broadcast to a $512 \times n$ dimension, and the final output H_1 has dimensions $512 \times n$.

Comment 15. What is the relationship between H_1 and Eq. (5)? What do the authors mean by the term “the first layer?” The right hand side of Eq. (36) cannot be represented as $\sum_{i=1}^n \sigma(WX + B)$.

Response: Thank you for your insightful question. We have corrected the previous formula and redefined it as Equation (14) to more clearly illustrate the relationship between H_1 and our final output. H_1 represents the first layer of processing in the first stage of our network, as shown in Fig.1.g of manuscript. This redefinition allows us to clearly observe the relationship between $WX + B$ and H_1 . Specifically, the data is first processed through $WX + B$ and then further processed using H_1 . The related mathematical expressions are provided in Equations (5) and (6). Expanding H_1 for the element at the i row and j column after applying the SiLU activation function, and we get $h_{i,j}^{(1)} = \alpha_1 \text{SiLU}\left(\sum_{p=1}^m W_{i,p}^{(1)} X_{p,j} + B_i^{(1)}\right) + \beta_1$.

Comment 16. What are the dimensions of H_2 , W_2 , B_2 , α_2 , and β_2 ?

Response: Thank you for your question. Firstly, α_2 , and β_2 are scalar parameters. The input in this context is H_1 , which has dimensions $512 \times n$. We have set the dimension of the hidden layer to 512, resulting in a weight matrix W_2 with dimensions

512×512 and a bias vector B_2 with dimensions 512×1 . In practice, within the neural network, B_2 is broadcast to a $512 \times n$ dimension, and the final output H_2 has dimensions $512 \times n$.

Comment 17. What is the relationship between H_2 and Eq. (5)? What do the authors mean by the term “the second layer?”

Response: Thank you for your insightful question. Similar to Question 15, there is a clear relationship between H_2 and Equation (14). Specifically, after the data has been processed by H_1 , it undergoes an additional bias operation $W_2H_1 + B_2$. The result is then further processed using the function $\alpha_2(\text{SiLU}(W_2H_1 + B_2)) + \beta_2$, yielding H_2 represents the transformation performed by the second layer of the first stage of our model.

Comment 18. What is the relationship between H_3 and Eq. (5)?

Response: Due to the presence of negative intensity values in the outputs when using our previously employed two-layer network structure to predict spectra, we have introduced a third layer, H_3 . This layer utilizes a ReLU (Rectified Linear Unit) function to set any negative values to 0, ensuring that all predicted intensities are non-negative.

Comment 19. The performance of an ANN depends on its architecture. A detailed architecture of WMYn (MLP) must be shown. The reason why the author chose the architecture for comparison with WMYn (KART) is also required.

Response: Thank you for your feedback. In Fig.1 of manuscript, we present the network architecture of WMYn (KART). In Fig.R3 below, we display the network architecture of WMYn (MLP) and compare it with WMYn (KART). Specifically, we replaced all structures in WMYn inspired by KART with equally layered MLP structures, while retaining the other network components. We evaluated the results of

both WMYn (MLP) and WMYn (KART) using cosine similarity with standard spectra.

Figure R3. Comparison of Network Architectures between WMYn (KART) and WMYn (MLP).

The hyperparameters, total number of parameters, and number of iterations for the models are set as follows:

Model	WMYn (KART)	WMYn (MLP)
Hidden Layer Dimension	512	512
Number of Heads	8	8
Number of Encoder Layers	6	6
Dropout Probability	0.1	0.1
Total Parameters (Input is 512×512)	~8.7M	~8.9M

The core structural parameters, including hidden layer dimensions, number of stacked layers, dropout probabilities, and number of heads, are identical between the two models, ensuring similarity in complexity. Additionally, the total number of parameters differs by only approximately 2.2%, maintaining close parameter counts. Combined with consistent inputs, identical numbers of iterations, and uniform evaluation metrics, we ensure a fair comparison between the two models.

Comment 20. WMYn (KART) is expected to have less representability of a nonlinear function than WMYn (MLP). Why is WMYn (KART) superior to WMYn (MLP)?

Response: Firstly, WMYn (KART) introduces additional learnable parameters α and β , while theoretically the universal approximation capacity of MLP may not be surpassed. Due to the non-linear nature of the SiLU transformation, α and β being

learnable parameters allow the function space of Equations (5) and (7) to encompass a wide range of non-linear mappings. This dynamic adjustment of the model's non-linear mapping capabilities enables it to adapt to different data distributions, thereby enhancing its generalization ability.

Secondly, we conducted comparative experiments by replacing all KART- inspired structures in WMYn with equally layered MLP structures. The results indicate that WMYn (KART) outperforms WMYn (MLP) in spectral learning and generation tasks within relatively small sample spaces.

A more layer MLP may have more powerful than WMYn (KART) in nonlinear function. In our study, we used two layers from either MLP architecture or WMYn (KART) network, and figured out WMYn (KART) is superior to WMYn (MLP) in cosine similarity and small-sample learning by using lipid references.

Reviewer #3 (Remarks to the Author):

I appreciate all the in-depth responses to my comments. I especially was impressed and appreciated the table generated containing all technical terms and the corresponding definitions.

To complete the review and accept this manuscript, I would like to be able to test the software. Currently, I have not been able to successfully run the software on my own data (which I would need for just a simple validation that the resulting annotations are realistic for a new test data), nor could I get the software to work on the test data using the exact same parameters and zip file provided.

<https://www.lifemetabolomics.cn/LipidIN>

The user interface was relatively straight forward. I am not sure how mzML conversion parameters (e.g. compressed versus not compressed files) effect how the software runs so a tutorial for mzML conversion on the page may be helpful.

Comment 1. Note you mention including your file as yzip format, only zip format exists, must be a misspelling?
--

Response: Thank you very much for your positive comments, and we appreciate your interest in using our website for lipid annotation. The 'yzip' was indeed a typo, and we have corrected it. The corrected web page was shown following.

a

Figure R4. Modified Web Interface.

Comment 2. Note “Working” in all portions of the webpage, which are the buttons to click when running the data, should be changed to “Run” or “Process Data”

Response: Thank you for your professional advice. We have replaced all the functional buttons on the webpage with a 'Run' button, and the corrected web page was shown above.

Comment 3. Also note that the 2GB limit makes sense given that you are potentially running the algorithms on your own server? But this limit will significantly limit the usecase as some people will have 100 of GBs worth of files. At this point maybe LipidIN does not have to support 100 of GBs worth of files because it only needs representative MS/MS files (2 GB is still to small, at least 10 GB), but for the quant workflow you mentioned it would definitely need to have higher GB available. The quant workflow could benefit from being integrated into the WebApp...

Response: Thank you for testing our website. The file upload limit has been carefully determined based on our current server configuration. To address the challenges identified, we have developed a user-friendly interface using R Shiny. The interface is illustrated in the figure below, and a detailed tutorial on how to use it has been uploaded

to GitHub (<https://github.com/LinShuhaiLAB/LipidIN/tree/main/LipidIN%20GUI>). Additionally, we have stored our projects on Code Ocean (<https://codeocean.com/articles>) to facilitate reproducibility for reviewers.

Figure R5. Two new methods for using and testing LipidIN. a-b. Downloading and installation of the R-based UI interface from GitHub. **c-d.** Using Code Ocean for reproducible testing.

Comment 4. I ran one single mzML file that was not too large, and I got the following error:

“系统接口请求超时”

“System interface request timeout”

I had it formatted correctly I believe: data.zip/data/ddMS28387_Pos.mzML and I converted with MSConvert using the following parameters: peakPicking:

vendor msLevel=1-2

threshold: absolute 1000 most-intense

I also tried a parameterless conversion to mzML and your own demo data (just downloaded, reuploaded, and got the same error)

And using the demo file that you provided and got the same error. I tested my internet

connection and speed and it was fine.

12/22/2024 10:39am EST was the time ran

Response: Thank you for testing our website. Upon analyzing the testing results, we noticed an error in the data upload process. After carefully reviewing the background information and considering the reviewers' comments, we confirmed that there were no records of data uploads during the specified time period, indicating a failure in the upload process.

Further investigation revealed that the website's stability still requires optimization. To address this, we recommend using either the UI interface or Code Ocean to replicate and upload the data more reliably. During this round of testing, we engaged 20 researchers to perform the software successfully by using UI interface or R codes with Windows system or MacOS. To ensure transparency, we have detailed their computer configurations and R programming environments in the attached image for your reference. For your convenience, we have also prepared a step-by-step user manual for the UI interface, which is included below. Furthermore, due to the current instability of the website and the inability to resolve these issues in the near term, we have decided to remove all website-related information from the main text. Line 821 was revised as following:

The code for LipidIN can be found at GitHub
<https://github.com/LinShuhaiLAB/LipidIN>, code ocean
(<https://codeocean.com/capsule/3229548/tree> and
<https://codeocean.com/capsule/3481249/tree>), and Zenedo
<https://doi.org/10.5281/zenodo.14600201>.

LipidIN Installation and Launch Guide

After downloading the **LipidIN** project from GitHub, navigate to the LipidIN GUI folder. Open the file ‘code for launch UI.R’ using Rstudio.

How to install R and RStudio: <https://rstudio-education.github.io/hopr/starting.html>;
https://blog.csdn.net/W_chuanqi/article/details/123626811.

How to install Rtools: <https://cran.rstudio.com/bin/windows/Rtools/rtools40.html>.

1. **Check Required Packages:** Lines 1–10 of the script are used to check whether the required R packages are installed. Select these lines, then click **Run** in the top-right corner of your R console. Wait for the necessary packages to be installed.
2. **Modify File Path:** Once all the dependencies are installed, update the file path on **line 15** of the script. Replace the placeholder file path with the path to the downloaded ‘LipidIN_2.0.0.1.tar.gz’ file.
3. **Install LipidIN:** After updating the file path, run **lines 12–19** to install LipidIN. During the installation process, a pop-up window will prompt you to confirm the installation of private packages. Click **Yes** to proceed.
4. **Launch the LipidIN UI:** Finally, run **lines 22–23** of the script. This will launch the LipidIN user interface.

Parameter input and precautions

1. **Step 1: Input mzML File Path:** In the first input field of the UI, enter the file path of the mzML file you wish to annotate. If necessary, you can use the **MSConvert** software to convert your files into the mzML format. The default parameters of MSConvert are sufficient for this conversion, and additional operations such as centralization are not required. Please ensure that there are no spaces or special characters in the file path.
2. **Step 2: Input Spectral Library Path:** In the second input field, provide the file path to the spectral library to be used for annotation. Note that the spectral library must be in **RDA format**. For instructions on converting an MSP-format spectral library to RDA format, refer to the guide: How to Convert Your MSP Format Spectral Library to RDA Format

(<https://github.com/LinShuhaiLAB/LipidIN/tree/main/How%20to%20Convert%20Your%20MSP%20Format%20Spectral%20Library%20to%20RDA%20Format>). Alternatively, you can use the pre-compiled RDA spectral library available on our GitHub repository: LipidIN/LipidIN 4-level hierarchical library/pos_ALL.rda. As with the mzML file path, ensure that the spectral library path does not contain spaces or special characters.

- 3. Step 3: Select Filtering Threshold:** Choose a filtering threshold to exclude peaks with intensities lower than a certain percentage of the maximum intensity in the spectrum. Recommended values for this threshold are **0.01**, **0.05**, or **0.1**.
- 4. Step 4: Select Ionization Mode:** Select the ionization mode for your analysis. The available options are: Positive ionization mode, Negative ionization mode ($[M+HCOO]^-$) and Negative ionization mode ($[M+CH_3COO]^-$).
- 5. Step 5: Set Mass Tolerance for Precursor Ions:** Specify the mass-to-charge ratio (m/z) tolerance for precursor ions during spectral library matching. Input the desired tolerance value in this step.
- 6. Step 6: Set Mass Tolerance for Fragment Ions:** Specify the mass-to-charge ratio (m/z) tolerance for fragment ions during spectral library matching. Input the desired tolerance value in this step.
- 7. Step 7: Start the Annotation Process:** After verifying the inputs for all the steps above, click the **Start** button to initiate the lipid annotation process.

Package Loading and Installation

After clicking "Run," the first message (Message 1) will indicate that the system is checking for missing packages and loading them. If you are running the script for the first time, this process might take longer, as it will automatically install RTools if necessary. Please be patient during this step.

The installation process is as follows: 1. When encountering the error message "**Error in SourceCpp**", it indicates that **Rtools** needs to be installed. 2. Click the red dot in the upper-right corner of the image to terminate the LipidIN operation. 3. Afterward, an installation prompt for **Rtools** will appear. 4. Click **Yes** to begin the download and installation process.

1. File Format Conversion: Once all required packages are loaded, the second

message (Message 2) will notify you that LipidIN is converting mzML files to rda files. If you encounter errors at this step, please ensure the following: Your mzML files contain MS2 spectra, and the input file path does not contain spaces or special characters.

2. **Library Loading:** After file conversion, the system will load the spectral library, which will be indicated by the third message (Message 3). This step can take a significant amount of time, especially if your spectral library contains millions of entries.
3. **Spectral Matching:** Once the library is loaded, spectral matching will begin, and the fourth message (Message 4) will appear. This process is typically very fast. If it becomes stuck or an error occurs, check the format and content of your spectral library for accuracy.
4. **Lipid Retention Time Analysis:** After spectral matching, the system will perform lipid retention time prediction, indicated by the fifth message (Message 5). This step takes approximately 1–2 minutes.
5. **Completion:** Finally, when all annotation processes are complete, the system will display the sixth message (Message 6).

Comments on NCOMMS-24-49156A

First, I appreciate the author’s sincere responses to all the comments. However, I still have many comments on the revised manuscript. Therefore, I consider the current manuscript unsuitable for publication in Nature Communications. The detailed comments are shown below.

The authors showed the detailed definition of the WMYn architecture shown in Eqs. (2)–(13). The definition is very clear. After the definition, they discussed the correspondence between KART (Eq. (14)) and WMYn (Eq. (15)). However, it is just a “formal correspondence.” To be precise, it is not a formal correspondence.

First of all, $\psi_{p,q}$ must have the i index as $\psi_{p,q,i}$ by the defined Eq. (11).

In Eq. (14) (KART), $f(X) = f(x_1, \dots, x_n)$ is represented by the univalent functions Φ_q and $\varphi_{q,p}$. From the perspective of machine learning, the function $f(X)$ can fit arbitrary function $f_t(X)$ by a training data-set $\{(X_1, f_t(X_1)), \dots, (X_N, f_t(X_N))\}$. If the training succeeds, the function $f(X)$ approximates $f_t(X)$. In the training, the univalent functions Φ_q and $\varphi_{q,p}$ are trained, i.e., these functions must be univalent trainable functions. If $f_t(X)$ is a nonlinear function, Φ_q or $\varphi_{q,p}$ are nonlinear. In some cases, Φ_q or $\varphi_{q,p}$ are linear even if $f_t(X)$ is nonlinear. In most cases, Φ_q and $\varphi_{q,p}$ are trainable nonlinear functions. However, $f_{p,i,j}(X_{p,j})$ defined in Eq. (10) is linear for $X_{p,j}$. This is a very special case. The authors need to prove that the linear function is enough to construct a nonlinear function $f(X)$ in KART.

Furthermore, the authors claimed that $\psi_{p,q}(X_{p,j})$ has q dependence. However, it is not correct. From Eqs. (11) and (13), we can eliminate q dependence from $\psi_{p,q}$. We can make the same function with the following definitions:

$$h_{i,j}^{(2)} = \alpha_2 \text{SiLU} \left(\sum_{q=1}^{512} \phi_{q,i,j} \left(\sum_{p=1}^m f_{p,i,j}(X_{p,j}) \right) + b^{(2)} \right) + \beta_2. \quad (\text{a})$$

This equation exactly equals to Eq. (13). Obviously, this definition is entirely different from the definition of KART (Eq. (14)). Actually, the authors have shown in Eq. (9) that $W_{i,q}^{(2)}$ is the only term that has the q index.

In addition, this definition has the three-layered structure: $f_{p,i,j}(x)$, $\phi_{q,i,j}(x)$, and $\alpha_2 \text{SiLU}(x_{i,j} + b_i^{(2)}) + \beta_2$ correspond to the first, second, and third layers. This structure is obviously different from the two-layer structure of KAN shown in Eq. (14), i.e., $\varphi_{q,p}(x_p)$ and $\phi_q(x_q)$ corresponds to the first and second layers, respectively.

From the discussion above, WMYn does not have the KART structure. On the other hand, it can be shown that WMYn is a restricted multi-layer perceptron again. Using Eqs. (5) and (7), Eq. (9) can be represented as

$$\begin{aligned} H_2 &= \alpha_2 (\text{SiLU}(W_2 H_1 + B_2) + \beta_2) \\ &= \alpha_2 (\text{SiLU}(W_2 (\alpha_1 (\text{SiLU}(W_1 X + B_1) + \beta_1) + B_2) + \beta_2) \\ &= f_4(A_2 f_3(W_2 f_2(A_1 f_1(W_1 X + B_1) + \beta_1) + \gamma_1) + B_2) + \beta_2) + \gamma_2, \end{aligned} \quad (\text{b})$$

where

$$f_1(X) = \text{SiLU}(X), \quad (\text{c})$$

$$f_2(X) = \text{Linear}(X) = X, \quad (\text{d})$$

$$f_3(X) = \text{SiLU}(X), \quad (\text{e})$$

$$f_4(X) = \text{Linear}(X) = X, \quad (\text{f})$$

$$A_1 = \alpha_1 I_{512} = \alpha_1 \begin{pmatrix} 1 & 0 & \dots & 0 \\ 0 & 1 & \dots & 0 \\ \vdots & \vdots & \ddots & \vdots \\ 0 & 0 & \dots & 1 \end{pmatrix}, \quad (\text{g})$$

$$A_2 = \alpha_2 I_{512} = \alpha_2 \begin{pmatrix} 1 & 0 & \dots & 0 \\ 0 & 1 & \dots & 0 \\ \vdots & \vdots & \ddots & \vdots \\ 0 & 0 & \dots & 1 \end{pmatrix}, \quad (\text{h})$$

$$\gamma_1 = 0, \quad (\text{i})$$

$$\gamma_2 = 0, \quad (\text{j})$$

I_{512} is the 512×512 identity matrix. The Eq. (b) implies that WMYn is a restricted multi-layer perceptron with batch input

$$X = (x_1 \quad x_2 \quad \dots \quad x_n). \quad (\text{K})$$

The restrictions are Eqs. (d), (f), (g), (h), (i), and (j).

From the discussion above, WMYn is not related to KART but is a multi-layer perceptron.

A. I recommend removing the sentences about KART in the text and SI.

1 On the response to Comment 1

Thank you for the response. I understood the “true spectrum” defined by the authors.

B. Is “true spectrum” is a popular term in this research field? I recommend that the authors use this term in the text if it improves the readability of this manuscript.

The representation of x_i in Eq. (6) may represent that the spectrum shown Fig.R.1.a. I expected the indices of x_i represents discretized m/z value.

C. Please show the definition of the indices of x_i because the way to discretize the m/z value is not obvious.

2 On the response to Comment 2

I confirmed that the authors modified the related part in the Method section. However, the authors need to modify the sentences related to KART. The sentence “Kolmogorov–Arnold representation theorem (KART) states that any multivariate continuous function $\Phi(Y_1, \dots, Y_n)$ can be represented as a finite sum of univariate continuous functions.” is incorrect. This sentence represents the following equation,

$$\Phi(Y_1, \dots, Y_n) = \sum_{i=1}^N \sum_{j=1}^n \varphi_{i,j}(Y_j). \quad (1)$$

D. I recommend that the authors directly show KART as an equation as follows if they need to show in the text:

$$\Phi(Y_1, \dots, Y_n) = \sum_{q=1}^{2n+2} \psi_q \left(\sum_{p=1}^n \phi_{q,p}(Y_p) \right). \quad (2)$$

D. I think that the authors directly show KART as an equation as follows if they need to show in the text. However, I recommend removing the sentences about $\Phi(Y_1, \dots, Y_n)$, because it does not help understanding the WMYn architecture. See the following responses.

3 On the response to Comment 3

The author explained that \mathbf{S} represents a vector. However, they also explained that \mathbf{S} have elements expressed by $S_{i,j}$. From the representation of the element, \mathbf{S} is expected as a matrix. In addition, in Eq. (16), the authors showed an example of a vector x_i representing a spectrum.

E. Is the form of $S_{i,j}$ different from that of x_i (vector)? If so, why does S have two indices, i and j ? What are the indices i and j ? They cannot be found on the right-hand side of Eq. (21).

Again, I recommend to remove the discussion about \mathbf{S} .

4 On the response to Comments 4 and 5

I would appreciate the author’s response, which helped me to understand the manuscript.

5 On the response to Comment 6

F. The bold style is not usually used to express “significant importance.” Please use the font styles to distinguish different mathematical objects, e.g., scalars in italic style and vectors in bold style.

6 On the response to Comment 7

I would appreciate the author’s response, which helped me to understand the manuscript.

7 On the response to Comment 8

G. Please use the term “peak position” or a similar term to represent the value on the horizontal axis in spectrum plots.

8 On the response to Comments 9–18

I would appreciate the author’s response, which helped me to understand the manuscript.

9 On the response to Comments 19 and 20

H. Please explicitly show the architecture of $WNY_n(\text{MLP})$, like Eq. (B), with the explicit dimensions of the weights and biases and the explicit activation functions. Please discuss the efficiency of $WNY_n(\text{KART})$ based on the explicit architectures of $WNY_n(\text{MLP})$ and $WNY_n(\text{KART})$. In addition, I recommend that the authors change the name of $WNY_n(\text{KART})$.

I was able to now run the code in RStudio given the instructions provided (which I assume are also on the website).

Pros:

- 1) the software ran extremely fast!
- 2) The software was relatively straightforward to use.
- 3) The software was relatively accurate: Spot checking the MS/MS spectra and RT trends, as well as benchmarking against LipidMatch, showed that the software was quite accurate in annotations.
- 4) Coverage was high, for example using the same file (adipose tissue) via LipidMatch 235 unique m/zs (two decimal points) were annotated whereas in LipidIN 243 unique m/zs were annotated showing similar coverage.

Cons:

- 1) Results are highly sensitive to the Abundance filter, and setting the abundance filter high creates many false negatives and low creates several false positives without any numbers allowed between. Coverage was relatively low when putting the abundance filter to the recommends 0.05, for example using the same file (adipose tissue) via LipidMatch 235 unique m/zs (two decimal points) were detected whereas only 160 were detected by LipidIN in positive mode. LipidIN missed some predominant lipids including PC(16:0_18:1) (one of the major common lipids) as well as all other PCs, due to their low acyl fragments. Hence the abundance filter was set to 0 (there was only one option 0.01 or 0, could not put 0.0001 as desired... At 0 several false positives occurred including MGDGs (plant lipids) in adipose (MGDG 22:2 | MGDG 8:2_14:0).
- 2) Retention times reported are for MS/MS scans not for peaks, with peak RTs and abundances shown
- 3) Most aspects of the workflow were not covered by the implementation recommended by the author to me. Peak picking and alignment across samples, gap filling, deisotoping, blank filtering, combining negative and positive mode, are some essential steps in the lipidomics workflow not covered by LipidIN. This will limit usability.

More detailed feedback:

The GUI does not like long directories (they don't paste fully and will cause errors).

You could distribute R packages and an R version with LipidIN to reduce future dependency issues.

The following installation in the R Code:

```
install.packages(  
  pkgs = '~/LipidIN-main/LipidIN GUI/LipidIN_2.0.0.1.tar.gz',  
  lib = .libPaths()[length(.libPaths())],  
  repos = NULL,  
  dependencies = T  
)
```

Errors when you put the exact directory (as mentioned in the instructions).

I needed to go to the same documents folder it was looking for and put the tar.gz there for successful installation, so this could be modified to be able to take a full path.

The retention times reported are from the MS/MS scans (time of MS/MS scan) which is not that retention time of the peak itself. For example, for the two TGs below RTs of 14.90 and 15.39 were reported, whereas the RTs at the peak maximum were not reported (14.96 and 15.31).

RT: 14.3668 - 16.2238

This may be why when I plotted RTs against m/z for the same lipid class and degree of unsaturation the trend was not perfect (albeit it's not always expected to be perfect). For reference I have included RT vs m/z trends from another software for a different dataset to show how clear they often are. Are the RTs which are used for retention time analysis the retention times reported in pos_ALL_final_output.csv, because if they are these are the incorrect RTs (the MS/MS scan RT is not the RT of the peak itself). Furthermore, if the reported RTs are not the ones used, the RTs of the peaks themselves should be reported.

Above is from LipidIN for a TG (0 unsaturation) profile in adipose tissue

Above is from LipidMatch for a TG (0 unsaturation) profile in cell lines

Above is from LipidIN for a TG (1 unsaturation) profile in adipose tissue

Above is from LipidMatch for a TG (1 unsaturation) profile in cell lines

I did a quick comparison to LipidMatch. I was able to get 160 unique annotations base on a unique m/z in lipidIN in positive mode to two decimal points. I was able to get 235 unique m/zs annotated in LipidMatch with high confidence in positive mode to two decimal points. I spot checked a few TGs and both LipidMatch and lipidIN gave the same top annotation.

Comments on NCOMMS-24-49156A

Reviewer #1 (Remarks to the Author):

Response: Thank you very much for your professional and detailed comments! Your insightful feedback improved the quality of our manuscript. We have carefully addressed each of your suggestions and have revised both the manuscript and supporting information. We provide our point-by-point responses to your comments along with the specific changes implemented below.

Comment A. I recommend removing the sentences about KART in the text and SI.

Response: We sincerely thank you for your careful derivation and proof demonstrating the formal relationship between KART (Eq. (14)) and WMY_n (Eq. (15)). We have removed the statements about KART from manuscript and supporting information. Additionally, we termed “WMY_n (SiLU)” and “WMY_n (ReLU)” in the comparison of SiLU and ReLU activation functions. As a result, we found that SiLU is more flexibility and higher accuracy than ReLU activation function by small-sample learning (Fig. R1).

Fig. R1. Cosine similarity performance comparison between WMYn (SiLU) and WMYn (ReLU). a. The cosine similarity between predicted spectra trained with different sample sizes and experimental detection spectra. The red bars represent WMYn (ReLU), the blue bars represent WMYn (SiLU), and the dotted line indicates the mean cosine similarity. b. The cosine similarity between the predicted spectra and experimental spectra for 15 lipid reference standards under epoch = 3000, with an input sample size of 10. Error bars represent standard deviation, and a T-test was used for significance testing.

Comment B. Is “true spectrum” is a popular term in this research field? I recommend that the authors use this term in the text if it improves the readability of this manuscript.

Response: We sincerely appreciate your thoughtful suggestion. While “true spectrum” is not a commonly used term in mass spectrometry, where “standard reference spectra” is typically preferred. We used the term “true spectrum” to align with the concept of “Ground Truth” in machine learning. This choice was made to enhance readability and facilitate understanding for readers from interdisciplinary backgrounds, particularly those familiar with machine learning terminology.

Comment C. Please show the definition of the indices of x_i because the way to discretize the m/z value is not obvious.

Response: We sincerely thank you for your careful review and valuable comments on our manuscript. Regarding the definition of the indices x_i we will add the following clarification to the revised manuscript: “Mass spectrometry data are discretized in the m/z dimension at intervals of 0.0001 Da, based on the mass spectral resolution. Specifically, the raw continuous m/z values are rounded to the nearest multiple of 0.0001 Da.”

Comment D. I think that the authors directly show KART as an equation as follows if they need to show in the text However, I recommend removing the sentences about $\Phi(Y_1, \dots, Y_n)$, because it does not help understanding the WMYn architecture. See the following responses.

Response: We sincerely thank you for your detailed suggestion. As you have previously demonstrated, KART (Eq. (14)) and WMYn (Eq. (15)) are formally related. In light of your suggestion, we have removed the sentences about $\Phi(Y_1, \dots, Y_n)$ and KART from the manuscript.

Comment E. Is the form of $S_{i,j}$ different from that of x_i (vector)? If so, why does S have two indices, i and j ? What are the indices i and j ? They cannot be found on the right-hand side of Eq. (21). Again, I recommend to remove the discussion about S .

Response: We sincerely thank you for your careful reading and insightful observation. To clarify, the final output of S is indeed a vector. However, in the first stage of the network, S is initially represented in matrix form and undergoes further processing in subsequent stages, particularly in the third stage, to transform it into a vector. We apologize for any confusion caused by the lack of clarity in the original manuscript.

As your suggestion, we have removed the discussion about S from both the manuscript and SI to avoid ambiguity and ensure focus on the core aspects of the methodology.

Comment E. The bold style is not usually used to express “significant importance.” Please use the font styles to distinguish different mathematical objects, e.g., scalars in italic style and vectors in bold style.

Response: We sincerely thank you for pointing out this important detail. We have carefully followed your advice and thoroughly reviewed the manuscript and SI to ensure that scalars are consistently presented in italic style and vectors in bold style.

Comment G. Please use the term “peak position” or a similar term to represent the value on the horizontal axis in spectrum plots.

Response: Thanks to your suggestion, we have standardized the term “spectrum peak” instead of “peak position” throughout the manuscript and SI.

Comment H. Please explicitly show the architecture of WMYn (MLP), like Eq. (b), with the explicit dimensions of the weights and biases and the explicit activation functions. Please discuss the efficiency of WMYn (KART) based on the explicit architectures of WMYn (MLP) and WMYn (KART). In addition, I recommend that the authors change the name of WMYn (KART).

Response: Thanks to your suggestion, following is the architecture of WMYn (MLP):

$$\begin{aligned}
 H_2 &= \text{ReLU}(W_2 H_1 + B_2) = \text{ReLU}(W_2 (\text{ReLU}(W_1 X + B_1)) + B_2) \\
 &= f_2(W_2 (f_1(W_1 X + B_1)) + B_2)
 \end{aligned}$$

Previous derivations proved that WMYn (KART) is in fact a multi-layer perceptron, and thus WMYn (KART) differs from WMYn (MLP) only in the activation function used and the scaling factor.

We therefore used WMYn (SiLU) to replace WMYn (KART) and WMYn (ReLU) to replace WMYn (MLP), and revised the sentences in the manuscript and SI.

The hyperparameters, total number of parameters, and number of iterations for the models are set as follows:

Model	WMYn (SiLU)	WMYn (ReLU)
Hidden Layer Dimension	512	512
Number of Heads	8	8
Number of Encoder Layers	6	6
Dropout Probability	0.1	0.1
Total Parameters (Input is 512×512)	~8.7M	~8.9M

Reviewer #3 (Remarks to the Author):

I was able to now run the code in RStudio given the instructions provided (which I assume are also on the website).

Pros:

- 1) the software ran extremely fast!
- 2) The software was relatively straightforward to use.
- 3) The software was relatively accurate: Spot checking the MS/MS spectra and RT trends, as well as benchmarking against LipidMatch, showed that the software was quite accurate in annotations.
- 4) Coverage was high, for example using the same file (adipose tissue) via LipidMatch 235 unique m/zs (two decimal points) were annotated whereas in LipidIN 243 unique m/zs were annotated showing similar coverage.

Response: We sincerely thank you for positive comments and constructive feedback! We provided point-by-point response and made clarifications to your comments.

Comment 1) Results are highly sensitive to the Abundance filter, and setting the abundance filter high creates many false negatives and low creates several false positives without any numbers allowed between. Coverage was relatively low when putting the abundance filter to the recommends 0.05, for example using the same file (adipose tissue) via LipidMatch 235 unique m/zs (two decimal points) were detected whereas only 160 were detected by LipidIN in positive mode. LipidIN missed some predominant lipids including PC(16:0_18:1) (one of the major common lipids) as well as all other PCs, due to their low acyl fragments. Hence the abundance filter was set to 0 (there was only one option 0.01 or 0, could not put 0.0001 as desired... At 0 several false positives occurred including MGDGs (plant lipids) in adipose (MGDG 22:2 | MGDG 8:2_14:0).

Response: We sincerely thank you for your detailed testing and valuable feedback. We have addressed the limitation in the user interface (UI) by allowing users to set the abundance filter to values with up to 10 decimal places (Fig. R2). This update provides

greater flexibility and precision in filtering low abundance peaks as requested.

- Folder Path:** e.g., D:/mzML_Files
- Enter the Full Path of the File:** e.g., C:/Users/YourName/Documents/library_positive.rda
- Abundance Filter (0-1):** 0.0100000001
- Ionization Mode:** Positive Ion Mode (dropdown menu)
- MS1 Tolerance (ppm):** 5
- MS2 Tolerance (ppm):** 20
- Use MS1 Rt or MS2 scan time for LCI:** MS2 scan time (dropdown menu)

Fig. R2. Abundance filter to values with up to 10 decimal places.

The abundance filter is indeed a critical parameter in lipid annotation, and we appreciate your insights into its impact on false positives and negatives.

Firstly, as you rightly pointed out, low abundance peaks play a crucial role in structural annotation, as highlighted in your earlier comment: “Therefore, software such as LipidMatch, ALEX, etc. are generally instrument/matrix independent but do rely on low abundance peaks (as mentioned as a limitation). The problem is no matter what the model (machine learning or rule based) without these low abundant peaks which are sometimes the only ones necessary for structural annotation, structure cannot be assigned correctly.” To address this, we recommend users combine the annotation results from both positive and negative ion modes to minimize false positives and false negatives, ensuring a more robust annotation process.

Secondly, we understand that setting the abundance filter to 0 introduced false positives, such as MGDGs in adipose tissue. To mitigate this, we recommend users customize the hierarchical library by removing species-specific lipids (e.g., MGDGs) that are not relevant to their samples.

Comment 2) Retention times reported are for MS/MS scans not for peaks, with peak RTs and abundances shown.

Response: We sincerely thank you for pointing out this issue. In the new version of LipidIN we have given the user the option to use MS1 retention time or MS/MS scan time, where the step of obtaining the MS1 retention time for peak lifting is done with the assistance of R package XCMS 4.0.1[DOI: 10.18129/B9.bioc.xcms] (Fig. R3).

Fig. R3. More options for MS1 retention time or MS/MS scan time.

Modern mass spectrometers are capable of rapid switching, with the delay from MS1 scans to MS2 scans typically ranging from milliseconds to a few seconds. To streamline

the extraction of primary peaks (a computationally intensive step) and to avoid the loss of useful information due to variations in peak extraction parameters, we opted not to perform peak extraction during the lipid annotation step in per version LipidIN. Of course it is more accurate to use the MS1 retention time, but this need extra time and the quality of the lifted peaks is affected by the parameters and there will be many missing peaks.

However, we apologize for not clearly clarifying the distinction between MS/MS scan times and peak retention times in the earlier version of the manuscript. To address this, we have added relevant statements in the revised text to ensure clarity and avoid any potential misunderstandings.

Comment 3) Most aspects of the workflow were not covered by the implementation recommended by the author to me. Peak picking and alignment across samples, gap filling, deisotoping, blank filtering, combining negative and positive mode, are some essential steps in the lipidomics workflow not covered by LipidIN. This will limit usability.

Response: We sincerely appreciate your valuable feedback regarding the workflow coverage of LipidIN.

To enhance workflow support, the updated version of LipidIN incorporates several improvements beyond its novel lipid annotation strategy. These enhancements include utilizing XCMS for peak picking, aligning quantitative results across samples based on MS1 retention times, mass-to-charge ratios, and LipidIN annotation results. We have also enhanced the extraction of TIC plots with the assistance of XCMS, implemented gap filling for results that couldn't be quantitatively quantified, and removed isotopes using the CAMERA 1.58.0 package [DOI: 10.18129/B9.bioc.CAMERA].

For blank filtering, combining negative and positive modes, batch effect removal, and data quality control, users can use tools MZmine and Startarget to reprocessing. We are also eager to propose more advanced solutions to these challenges in the future and to incorporate them into upcoming versions of LipidIN.

Of note, in this manuscript, we focus on lipid annotation rather than quantitative analysis. I still would like to highlight the innovation of this manuscript. First, LipidIN features a largest lipid fragmentation hierarchical library that encompasses all potential chain compositions and carbon-carbon double bond locations based on the Paternò-Büchi reaction and electron-activated dissociation. Second, EQ module could query around 100 billion spectra within 1 second for ALL mass spectral libraries by using a general personal computer, which was up to 60,000 times faster than flash entropy search in a million of spectral library querying task. Third, the accuracy of LCI module generally outperforms the existing tools like MS-DIAL. Four, reverse lipidomics exhibited four advantages: (1) regeneration of lipid fingerprints as the 5th-level library for high-coverage and high-confidence annotation, (2) platform-independent lipidomic analysis for enhanced platform transferability, (3) enhancement of high resolution MS data for higher accuracy annotation, and (4) an interactive interface that empowers the broad exploration of reverse lipidomics with other spectral querying environments like Entropy Search.

Finally, we would like to emphasize that the primary focus of this study is on lipid annotation and fingerprint spectrum regeneration, rather than quantitative analysis or correction of quantitative results, we recognize the importance of a complete workflow and are actively exploring ways to expand its functionality in future updates. We appreciate your insightful comments, which will guide us in improving the usability and versatility of LipidIN.

Comment 4) The GUI does not like long directories (they don't paste fully and will cause errors).

Response: We sincerely thank you for bringing this issue to our attention. The limitation you encountered with long directory paths in the RShiny-based UI is indeed a known constraint. While we have not observed errors related to long file addresses in our current tests, issues may arise if the directory path contains special characters or is incorrectly formatted.

As you pointed out, the UI interface is not optimized for handling excessively long addresses. To address this, we have added a special note in the user guide, advising users to avoid long directory paths and providing best practices for organizing their files. We appreciate your feedback and are committed to improving the usability of the software.

Comment 5) You could distribute R packages and an R version with LipidIN to reduce future dependency issues.

Response: We sincerely thank you for your valuable suggestion. We have added a clear restriction requirement regarding the R language version in the usage guide to ensure compatibility.

In addition, we have packaged and uploaded all necessary dependency packages to Zenodo. Users can simply download and copy these packages into their R library directory, ensuring consistency in the required R packages.

We appreciate your thoughtful recommendation and are committed to improving the accessibility and reliability of LipidIN.

Comment 6) The following installation in the R Code:

```
install.packages(  
  pkgs = '~/LipidIN-main/LipidIN GUI/LipidIN_2.0.0.1.tar.gz',  
  lib = .libPaths()[length(.libPaths())],  
  repos = NULL,  
  dependencies = T )
```

Errors when you put the exact directory (as mentioned in the instructions).

I needed to go to the same documents folder it was looking for and put the tar.gz there for successful installation, so this could be modified to be able to take a full path.

Response: We sincerely thank you for bringing this issue to our attention and for your valuable suggestion. To address the installation problem and improve user experience, we have revised the manual and sample code to emphasize the importance of using the exact directory path for installation.

```
install.packages(  
  pkgs = 'D:/bio_inf/LipidIN-main/LipidIN GUI/LipidIN_2.0.0.1.tar.gz',  
  # Note that here you need to write the full address  
  # of LipidIN_2.0.0.1.tar.gz, the above is only an example,  
  # please modify the address according to the actual situation.  
  lib = .libPaths()[length(.libPaths())],  
  repos = NULL,  
  dependencies = T  
)
```

Changes and comments in the codes

Fig. R4. Revised user manual in Github and zenodo.

Comment 7) The retention times reported are from the MS/MS scans (time of MS/MS scan) which is not that retention time of the peak itself. For example, for the two TGs below RTs of 14.90 and 15.39 were reported, whereas the RTs at the peak maximum were not reported (14.96 and 15.31).

Response: We sincerely thank you for your detailed observation regarding the retention times (RTs) reported in our analysis. In the new version of LipidIN we have given the user the option to use MS1 retention time or MS/MS scan time, where the step of obtaining the MS1 retention time is done with the help of XCMS package.

We appreciate your understanding and valuable feedback, which have helped us improve the clarity of our work.

Comment 8) This may be why when I plotted RTs against m/z for the same lipid class and degree of unsaturation the trend was not perfect (albeit it's not always expected to be perfect). For reference I have included RT vs m/z trends from another software for a different dataset to show how clear they often are. Are the RTs which are used for retention time analysis the retention times reported in pos_ALL_final_output.csv, because if they are these are the incorrect RTs (the MS/MS scan RT is not the RT of the peak itself). Furthermore, if the reported RTs are not the ones used, the RTs of the peaks themselves should be reported.

Above is from LipidIN for a TG (0 unsaturation) profile in adipose tissue

Above is from LipidMatch for a TG (0 unsaturation) profile in cell lines

Above is from LipidIN for a TG (1 unsaturation) profile in adipose tissue

Above is from LipidMatch for a TG (1 unsaturation) profile in cell lines

Response: Thank you for your insightful observation. Firstly, in the updated version of LipidIN, we have provided users with the option to choose between MS1 retention time or MS/MS scan time. The process of obtaining MS1 retention time is facilitated by XCMS 4.0.1.

Additionally, we hypothesize that retention time biases may arise from isomerism. Isomers were commonly present, especially in TGs with complex and diverse chain compositions. To illustrate this, we use the example of TG annotated in the dataset ST003398 (available at Metabolomics Workbench), where the RT and m/z trends also

show some deviation, and we can use ESCN and IUP to solve this situation. Furthermore, we compared the number of annotations in both datasets and demonstrated that LipidIN can annotate a broader range of lipids.

Fig. R5. RTs against m/z trends of TGs with degree of unsaturation at 0 and 4.

Comment 9) I did a quick comparison to LipidMatch. I was able to get 160 unique annotations base on a unique m/z in lipidIN in positive mode to two decimal points. I was able to get 235 unique m/z s annotated in LipidMatch with high confidence in positive mode to two decimal points. I spot checked a few TGs and both LipidMatch and lipidIN gave the same top annotation.

Response: Thank you for your testing and feedback! As previously stated, low abundance peaks have an important role to play and when the abundance filter is set at 0.05 many low abundance PCs etc. will not be accurately annotated due to the absence of important FAs. In addition, we believe that the consistency of the TG results provides good evidence of the accuracy of our annotations.

REVIEWER COMMENTS

Reviewer #3 (Remarks to the Author):

Thank you for your in-depth response and patience in this back and forth, I am excited to see all the improvements in the software via this discussion.

Comment 1) I appreciate the emphasis of "reverse lipidomics" and focus on annotation. Indeed, this could allow for targeted peak picking afterwards. This could be emphasized in the text in the intro/conclusion, that this is a reverse lipidomics approach allowing annotation without peak picking first (similar to LipidSearch). Then leave the community to find a good automated targeted peak picking solution, which is highly needed. I have heard good things about El Maven, and MZMine 2 (not 3 or 4) worked very well for targeted peak picking for non drifted data, but was very slow. I am not aware of high performing targeted peak picking workflows which are mostly automated (but this is not for you to solve! But just would be helpful in the future for users of your workflow to have a complete workflow).

Response: Thanks for your positive comments and professional advice. We have revised the line 109 of manuscript and emphasized "a reverse lipidomics approach allowing annotation without peak picking first" in the introduction section.

Comment 2) I do not know of good automated non-targeted peak picking software, I have not used XCMS except for non-targeted peak picking where it performs relatively poorly compared to MZMine and vendor peak picking solutions in multiple in-house benchmarking studies (including involving developers). But maybe it will suffice for just finding the right RTs of the peaks, make sure to evaluate the accuracy of XCMS for finding the peak center if you haven't already.

Response: Here we use XCMS for accurate peak retention time, and in the future we will continue to update the algorithm to achieve a good automatic non targeted peak extraction.

Comment 3) Using "Apex Peak Trigger" or similar should be recommended for your work along with a dynamic exclusion where only 1 scan is acquired per peak ideally. You should add a sentence or two on this in the paper along with the discussion of the limitation of the RT used. That way, at least the DDA data will likely have correct RTs most of the time depending on the performance of the vendors Apex Peak Trigger, I know Thermo and Agilent algorithms work nicely. Indeed, the data I used to benchmark your algorithm was Apex Peak Triggered (tries to find the peak apex, which is what you want). I understand now why you used the MS/MS RT to save time and not have to incorporate a peak picking algorithm.

Response: We appreciate your suggestion to incorporate the "Apex Peak Trigger" or a similar method, along with dynamic exclusion for acquiring only one scan per peak. We agree that this approach could improve the accuracy of retention times (RTs) in DDA data. In this regard, we have added a brief discussion in line 683-685 of LCI model in the revised manuscript.

Comment 4) Our TGs were not isomers unlike your examples, as you can see in our previous figure the RT drift is way too large, plus we pulled up examples showing it was just about using the RT from the triggered MS/MS (hence make sure the users know even with Apex trigger there will be false positives/negatives due to the MS/MS RT not being the correct RT).

Response: We understand your point regarding the comparison to isomers and appreciate the clarification. We now allow the user to choose between using MS1 retention time or the MS/MS scan time.

Overall, other than a few minor sentences added to the manuscript as mentioned above, I am happy publishing the manuscript.

Response: We already added a few sentences to the revised manuscript according to your advice. Thanks again for your kind suggestion and professional comments!

Supplemental Materials for the Review of NCOMMS-24-49156A

The authors responded to the Reviewer’s comments. However, the formulation and definition of the machine learning architecture still need to be clarified. The current manuscript is not recommended for publication in Nature Communications.. The questions and comments are listed below.

1 On the response to Comment 1

First, the Kolmogorov–Arnold representation theorem (KART) is shown for the discussion below. The authors defined KART as

$$f(X) = f(x_1, x_2, \dots, x_n) = \sum_{q=1}^{2n+2} \Phi_q \left(\sum_{p=1}^n \phi_{q,p}(x_p) \right), \quad (1)$$

where $X \in \mathbb{R}^n$, $x_p \in \mathbb{R}$, $\phi_{q,p} : \mathbb{R} \rightarrow \mathbb{R}$, and $\Phi_q : \mathbb{R} \rightarrow \mathbb{R}$. In general, the functions are different for different indices, i.e., $\phi_{q,p} \neq \phi_{q',p'}$ for $q \neq q'$ or $p \neq p'$ and $\Phi_q \neq \Phi_{q'}$ for $q \neq q'$.

In the response, the authors described that they tried to obtain the “true spectrum” \mathbf{S} as a function of which arguments are mass spectrometry conditions (Y_1), chromatography conditions (Y_2), ionization voltage (Y_3), the nature of the compound Y_4 , and other properties: $\mathbf{S} = \Phi(Y_1, Y_2, \dots, Y_n)$. In this case, Y_1, \dots, Y_n are the inputs of the function \mathbf{S} .

1.1. What do the authors represent by the term “true spectrum?” Do they want to express the spectrum obtained by an experiment?

On the other hand, the authors described “ Y_1, \dots, Y_n denote the output matrix, corresponding to the theoretical spectra in the chromatographic and mass spectrometric conditions” in the revised manuscript. They clearly expressed that Y_1, \dots, Y_n are the output.

1.2. Whether are Y_1, \dots, Y_n inputs or outputs? The authors need to clarify Y_1, \dots, Y_n are input for what and output from what.

1.3. Are \mathbf{S} , Y_n real numbers? Is it possible to represent a spectrum by a real number? Is the “chromatography conditions” represented by a real number? What did the authors represent using the bold italic style of \mathbf{S} ?

In the response, the authors represented \mathbf{S} as

$$\mathbf{S} = \sum_{J=1}^m \Phi_J, \quad (2)$$

and they called it “a finite unitary function.”

1.4. What is the “finite unitary function?”

The authors also mentioned that “ m is a finite constant.” I believe that m is an integer because this appears as the maximum number of the index J in the summation in Eq. (2).

1.5. Were there any assumptions on m ? How is it determined? What is the relationship between the m in Eq. (2) and the n in KART (1)?

They introduced the following function:

$$\mathbf{F} = \sum_{i=1}^n f_i(WX + B). \quad (3)$$

The dimension of X has not been mentioned when it was introduced. But it could be found after showing the relationship between Φ and $f_i(WX + B)$, i.e., X , W , and B are matrices. Then, the argument of f_i is a matrix. Usually, the function of a matrix argument is defined by the element-wise function, like

$$f_i(Y) := \begin{pmatrix} f_i(Y_{11}) & f_i(Y_{12}) & & f_i(Y_{1\ell'}) \\ f_i(Y_{21}) & f_i(Y_{22}) & & f_i(Y_{2\ell'}) \\ \vdots & \vdots & \ddots & \vdots \\ f_i(Y_{\ell'1}) & f_i(Y_{\ell'2}) & & f_i(Y_{\ell'\ell'}) \end{pmatrix}, \quad (4)$$

where Y is an $\ell \times \ell'$ matrix. Therefore, \mathbf{F} is expected to be a matrix.

1.6. What are the dimensions of \mathbf{F} , X , W and B ? What did the authors represent by the bold italic style of \mathbf{F} ?

1.7. What is the relationship between n in Eq. (3) and the dimension of X ?

The authors defined the following relation:

$$\Phi(Y_1, \dots, Y_n) = \sum_{j=1}^{2n+1} \Phi_j \left(\sum_{i=1}^n f_i(WX + B) \right), \quad (5)$$

where X represents the input matrix, consisting of x_1, \dots, x_n , where each x_i is a spectrum peak.

1.8. What is the “spectrum peak?” Do they have units?

1.9. What are the dimensions of X and x_i ?

1.10. What is the relationship between the matrix element of X and x_i ? For example,

$$X_{12} = x_2. \quad (6)$$

1.11. What is the relationship between Y_p and $x_{p'}$? $\Phi(Y_1, Y_2, \dots, Y_n)$ means a function of the real number arguments Y_p ($p = 1, 2, \dots, n$). However, the authors identified that X is the input, and X consists of x_1, \dots, x_n . What is the input of Φ ? Is Φ the function of x_1, \dots, x_n ? What are the mathematical definitions of Y_p ?

The authors revealed that f_i is the sigmoid linear unit (SiLU) function. If f_i is SiLU, the suffix i of f_i can be omitted, i.e.,

$$f_i(x) = \sigma_S(x), \quad (7)$$

where σ_S is the SiLU function. We can represent \mathbf{F} as

$$\mathbf{F} = n \sigma_S(WX + B). \quad (8)$$

Using this representation, Eq. (5) is rewritten as

$$\Phi(Y_1, \dots, Y_n) = \sum_{j=1}^{2n+1} \Phi_j(n \sigma_S(WX + B)). \quad (9)$$

The right-hand side of Eq. (8) does not have any indexes.

I should consider the possibility of a typo in the definition of W , X , and B . I assume that W is a $n \times n$ matrix and X and B are n -dimensional vector, In this case, \mathbf{F} is a n -dimensional vector defined as

$$\mathbf{F} = \sum_{i=1}^n \sigma_S(z_i), \quad (10)$$

where z_i is a matrix element of $WX + B$, i.e.,

$$Z = WX + B, \quad (11)$$

$$Z = \begin{pmatrix} z_1 \\ z_2 \\ \vdots \\ z_n \end{pmatrix}. \quad (12)$$

Finally, we obtain the following equation:

$$\Phi(Y_1, \dots, Y_n) = \sum_{j=1}^{2n+1} \Phi_j \left(\sum_{i=1}^n \sigma_S(z_i) \right). \quad (13)$$

I additionally assume that the input is not Y_i but x_i to clarify the input of the function, although the relationship between Y_i and x_i is still unclear.

$$\Phi(x_1, \dots, x_n) = \sum_{j=1}^{2n+1} \Phi_j \left(\sum_{i=1}^n \sigma_S(z_i) \right). \quad (14)$$

Here, I introduce a new representation of $\sigma(z_i)$ to clarify the dependency of the input x_i as

$$\phi_i(x_1, x_2, \dots, x_n) = \sigma_S(z_i) = \sigma_S\left(\sum_{k=1}^n w_{ik}x_k + b_k\right). \quad (15)$$

Finally, the following representation is obtained,

$$\Phi(x_1, \dots, x_n) = \sum_{j=1}^{2n+1} \Phi_j\left(\sum_{i=1}^n \phi_i(x_1, x_2, \dots, x_n)\right). \quad (16)$$

Comparing Eqs. (1) and (16), we can recognize the critical differences between them:

- ϕ_i in Eq. (16) is not a univariate function.
- ϕ_i in Eq. (16) does not have the j index.

These differences are critical for KART, which requires univariate functions and a j index for each function to distinguish between different nonlinear transformations in the model. Therefore, it appears that the proposed function $\Phi(x_1, \dots, x_n)$ does not directly correspond to KART. **From the discussion above, it is difficult to say that Φ is a kind of KART-inspired function.**

1.12. I recommend deleting the description of KART from the manuscript.

Despite the discussion of whether Φ is a kind of KART, the implementation of the function Φ_j should be identified. The authors claimed that “ Φ_j is a learnable nonlinear activation.”

1.13. What is the implementation of Φ_j ? Is this an artificial neural network?

I continue the discussion with the assumption that W is the $n \times n$ matrix and X and B are the n -dimensional vectors. I additionally assume that Φ_j is realized by an artificial neural network (ANN). I employ the ANN with a single hidden layer to clarify the discussion. This assumption does not lose generality, and the extension to the multi-layer perceptron is obvious. In this case,

$$\begin{pmatrix} \Phi_1(x) \\ \Phi_2(x) \\ \vdots \\ \Phi_{2n+1}(x) \end{pmatrix} = W^{(2)} \sigma\left(W^{(1)}x + B^{(1)}\right) + B^{(2)}, \quad (17)$$

where $W^{(1)}$ and $W^{(2)}$ are the $l \times 1$ and $2n+1 \times l$ weight matrices (vectors), $B^{(1)}$ is a l -dimensional bias vector and $B^{(2)}$ is a bias constant, and σ is an activation function. On the other hand, we can introduce another expression of the argument of Φ_j in Eq. (14),

$$\sum_{i=1}^n \sigma(z_i) = I_{(n)} \sigma(Z) = W' \sigma(WX + B) \quad (18)$$

$$I_{(n)} = (1 \quad 1 \quad \dots \quad 1), \quad (19)$$

where the dimension of $I_{(n)}$ is n . Using Eq. (18), Eq. (14) can be represented as

$$\Phi(x_1, x_2, \dots, x_n) = W'' W^{(2)} \sigma\left(W^{(1)} \left(W' \sigma_S(WX + B)\right) + B^{(1)}\right) + B^{(2)}, \quad (20)$$

$$I_{(2n+1)} = (1 \quad 1 \quad \dots \quad 1), \quad (21)$$

where the dimension of $I_{(2n+1)}$ is $2n + 1$.

$$\Phi(x_1, x_2, \dots, x_n) = \bar{W}^{(2)} \sigma \left(W^{(1)} \sigma_{\mathbf{I}} (I_{(n)} \sigma_{\mathbf{S}} (WX + B) + B_0) + B^{(1)} \right) + B_0, \quad (22)$$

$$\bar{W}^{(2)} = I_{(2n+1)} W^{(2)}, \quad (23)$$

$$B_0 = 0, \quad (24)$$

where $\sigma_{\mathbf{I}}$ is the identical function, i.e., $x = \sigma_{\mathbf{I}}(x)$. It is obvious that Eq. (22) is a special case of the ANN with three hidden layers, which is represented as

$$\Phi(X) = \mathcal{W}^{(\text{out})} \sigma^{(3)} \left(\mathcal{W}^{(3)} \sigma^{(2)} \left(\mathcal{W}^{(2)} \sigma^{(1)} \left(\mathcal{W}^{(1)} X + \mathcal{B}^{(1)} \right) + \mathcal{B}^{(2)} \right) + \mathcal{B}^{(3)} \right) + \mathcal{B}^{(\text{out})}. \quad (25)$$

Actually, Eq. (22) is recovered from Eq. (25) with the following equations:

$$\mathcal{W}^{(1)} = W, \quad (26)$$

$$\mathcal{W}^{(2)} = I_{(n)}, \quad (27)$$

$$\mathcal{W}^{(3)} = \bar{W}^{(2)} = I_{(2n+1)} W^{(2)}, \quad (28)$$

$$\mathcal{W}^{(\text{out})} = 1, \quad (29)$$

$$\mathcal{B}^{(1)} = B, \quad (30)$$

$$\mathcal{B}^{(2)} = B_0 = 0, \quad (31)$$

$$\mathcal{B}^{(3)} = B^{(1)}, \quad (32)$$

$$\mathcal{B}^{(\text{out})} = B_0 = 0, \quad (33)$$

$$\sigma^{(1)} = \sigma_{\mathbf{S}}, \quad (34)$$

$$\sigma^{(2)} = \sigma_{\mathbf{I}}, \quad (35)$$

$$\sigma^{(3)} = \sigma. \quad (36)$$

From the discussion above, WMYn does not have a KART-like architecture but has a restricted ANN. This means that the WNYn architecture has less representability rather than the usual ANN due to the restrictions of Eqs. (27), (28), (29), (31), (33), and (35).

2 On the response to Comment 2

2. In response to Comment 2, the authors explained the details of the WNYn architecture. However, it is still confusing.

They showed that “the first layer” of WNYn is defined as

$$H_1 = \alpha_1 ((W_1 X + B_1) \sigma(W_1 X + B_1)), \quad (37)$$

where H_1 is the latent matrix, W_1 and B_1 are the weight and bias matrices, α_1 and β_1 are the learnable scaling parameter and the learnable bias term, and σ is the SiLU activation function.

2.1. What are the dimensions of H_1 , X , W_1 , B_1 , α_1 , and β_1 ?

2.2. What is the relationship between H_1 and Eq. (5)? What do the authors mean by the term “the first layer?” The right hand side of Eq. (37) cannot be represented as $\sum_{i=1}^n \sigma(WX + B)$.

The authors introduced “the second layer”:

$$H_2 = \alpha_2 ((W_2 H_1 + B_2) \sigma(W_2 H_1 + B_2)), \quad (38)$$

where H_2 is the latent matrix, W_2 and B_2 are the weight and bias matrices, α_2 and β_2 are the learnable scaling parameter and the learnable bias term, and σ is the SiLU activation function.

2.3. What are the dimensions of H_2 , W_2 , B_2 , α_2 , and β_2 ?

2.4. What is the relationship between H_2 and Eq. (5)? What do the authors mean by the term “the second layer?”

The authors also introduced H_3 .

2.5. What is the relationship between H_3 and Eq. (5)?

3 On the response to Comment 3

The authors showed the results of the comparison between WNYn(KART) and WNYn(MLP).

3.1. The performance of an ANN depends on its architecture. A detailed architecture of WNYn(MLP) must be shown. The reason why the author chose the architecture for comparison with WNYn(KART) is also required.

3.2. WNYn(KART) is expected to have less representability of a nonlinear function than WNYn(MLP). Why is WNYn(KART) superior to WNYn(MLP)?

I appreciate all the in-depth responses to my comments. I especially was impressed and appreciated the table generated containing all technical terms and the corresponding definitions.

To complete the review and accept this manuscript, I would like to be able to test the software. Currently, I have not been able to successfully run the software on my own data (which I would need for just a simple validation that the resulting annotations are realistic for a new test data), nor could I get the software to work on the test data using the exact same parameters and zip file provided.

<https://www.lifemetabolomics.cn/LipidIN>

The user interface was relatively straight forward.

I am not sure how mzML conversion parameters (e.g. compressed versus not compressed files) effect how the software runs so a tutorial for mzML conversion on the page may be helpful.

Note you mention including your file as yzip format, only zip format exists, must be a misspelling?

Note “Working” in all portions of the webpage, which are the buttons to click when running the data, should be changed to “Run” or “Process Data”

Also note that the 2GB limit makes sense given that you are potentially running the algorithms on your own server? But this limit will significantly limit the usecase as some people will have 100 of GBs worth of files. At this point maybe LipidIN does not have to support 100 of GBs worth of files because it only needs representative MS/MS files (2 GB is still to small, at least 10 GB), but for the quant workflow you mentioned it would definitely need to have higher GB available. The quant workflow could benefit from being integrated into the WebApp...

I ran one single mzML file that was not too large, and I got the following error:

“系统接口请求超时”

“System interface request timeout”

I had it formatted correctly I believe: data.zip/data/ddMS28387_Pos.mzML and I converted with MSConvert using the following parameters:

peakPicking: vendor msLevel=1-2

threshold: absolute 1000 most-intense

I also tried a parameterless conversion to mzML and your own demo data (just downloaded, reuploaded, and got the same error)

And using the demo file that you provided and got the same error. I tested my internet connection and speed and it was fine.

step0 step1 step2 step3 step4

Complete process

data.zip should be less than 2G

MS2: a value of 0-1, You can define according to your needs.

ESI:

'p' for positive ionization mode,

'n1' for negative ionization mode [M+COOH]-,

'n2' for negative ionization mode [M+CH3COO]-.

ppm1: MS1 m/z tolerance at parts per million (ppm).You can define according to your needs.

ppm2: MS2 m/z tolerance at parts per million (ppm).You can define according to your needs

The file must be data.zip.

param key	param value
ESI	'n'
MS2	.01
PPM1	10
PPM2	40

File upload

data.zip 100%

Only one file in zip format can be uploaded,with a fixed name data.zip.

working

download code

12/22/2024 10:39am EST was the time ran

Comments on NCOMMS-24-49156A

First, I appreciate the author’s sincere responses to all the comments. However, I still have many comments on the revised manuscript. Therefore, I consider the current manuscript unsuitable for publication in Nature Communications. The detailed comments are shown below.

The authors showed the detailed definition of the WMYn architecture shown in Eqs. (2)–(13). The definition is very clear. After the definition, they discussed the correspondence between KART (Eq. (14)) and WMYn (Eq. (15)). However, it is just a “formal correspondence.” To be precise, it is not a formal correspondence.

First of all, $\psi_{p,q}$ must have the i index as $\psi_{p,q,i}$ by the defined Eq. (11).

In Eq. (14) (KART), $f(X) = f(x_1, \dots, x_n)$ is represented by the univalent functions Φ_q and $\varphi_{q,p}$. From the perspective of machine learning, the function $f(X)$ can fit arbitrary function $f_t(X)$ by a training data-set $\{(X_1, f_t(X_1)), \dots, (X_N, f_t(X_N))\}$. If the training succeeds, the function $f(X)$ approximates $f_t(X)$. In the training, the univalent functions Φ_q and $\varphi_{q,p}$ are trained, i.e., these functions must be univalent trainable functions. If $f_t(X)$ is a nonlinear function, Φ_q or $\varphi_{q,p}$ are nonlinear. In some cases, Φ_q or $\varphi_{q,p}$ are linear even if $f_t(X)$ is nonlinear. In most cases, Φ_q and $\varphi_{q,p}$ are trainable nonlinear functions. However, $f_{p,i,j}(X_{p,j})$ defined in Eq. (10) is linear for $X_{p,j}$. This is a very special case. The authors need to prove that the linear function is enough to construct a nonlinear function $f(X)$ in KART.

Furthermore, the authors claimed that $\psi_{p,q}(X_{p,j})$ has q dependence. However, it is not correct. From Eqs. (11) and (13), we can eliminate q dependence from $\psi_{p,q}$. We can make the same function with the following definitions:

$$h_{i,j}^{(2)} = \alpha_2 \text{SiLU} \left(\sum_{q=1}^{512} \phi_{q,i,j} \left(\sum_{p=1}^m f_{p,i,j}(X_{p,j}) \right) + b^{(2)} \right) + \beta_2. \quad (\text{a})$$

This equation exactly equals to Eq. (13). Obviously, this definition is entirely different from the definition of KART (Eq. (14)). Actually, the authors have shown in Eq. (9) that $W_{i,q}^{(2)}$ is the only term that has the q index.

In addition, this definition has the three-layered structure: $f_{p,i,j}(x)$, $\phi_{q,i,j}(x)$, and $\alpha_2 \text{SiLU}(x_{i,j} + b_i^{(2)}) + \beta_2$ correspond to the first, second, and third layers. This structure is obviously different from the two-layer structure of KAN shown in Eq. (14), i.e., $\varphi_{q,p}(x_p)$ and $\phi_q(x_q)$ corresponds to the first and second layers, respectively.

From the discussion above, WMYn does not have the KART structure. On the other hand, it can be shown that WMYn is a restricted multi-layer perceptron again. Using Eqs. (5) and (7), Eq. (9) can be represented as

$$\begin{aligned} H_2 &= \alpha_2 (\text{SiLU}(W_2 H_1 + B_2) + \beta_2) \\ &= \alpha_2 (\text{SiLU}(W_2 (\alpha_1 (\text{SiLU}(W_1 X + B_1) + \beta_1) + B_2) + \beta_2) \\ &= f_4(A_2 f_3(W_2 f_2(A_1 f_1(W_1 X + B_1) + \beta_1) + \gamma_1) + B_2) + \beta_2) + \gamma_2, \end{aligned} \quad (\text{b})$$

where

$$f_1(X) = \text{SiLU}(X), \quad (\text{c})$$

$$f_2(X) = \text{Linear}(X) = X, \quad (\text{d})$$

$$f_3(X) = \text{SiLU}(X), \quad (\text{e})$$

$$f_4(X) = \text{Linear}(X) = X, \quad (\text{f})$$

$$A_1 = \alpha_1 I_{512} = \alpha_1 \begin{pmatrix} 1 & 0 & \dots & 0 \\ 0 & 1 & \dots & 0 \\ \vdots & \vdots & \ddots & \vdots \\ 0 & 0 & \dots & 1 \end{pmatrix}, \quad (\text{g})$$

$$A_2 = \alpha_2 I_{512} = \alpha_2 \begin{pmatrix} 1 & 0 & \dots & 0 \\ 0 & 1 & \dots & 0 \\ \vdots & \vdots & \ddots & \vdots \\ 0 & 0 & \dots & 1 \end{pmatrix}, \quad (\text{h})$$

$$\gamma_1 = 0, \quad (\text{i})$$

$$\gamma_2 = 0, \quad (\text{j})$$

I_{512} is the 512×512 identity matrix. The Eq. (b) implies that WMYn is a restricted multi-layer perceptron with batch input

$$X = (x_1 \quad x_2 \quad \dots \quad x_n). \quad (\text{K})$$

The restrictions are Eqs. (d), (f), (g), (h), (i), and (j).

From the discussion above, WMYn is not related to KART but is a multi-layer perceptron.

A. I recommend removing the sentences about KART in the text and SI.

1 On the response to Comment 1

Thank you for the response. I understood the “true spectrum” defined by the authors.

B. Is “true spectrum” is a popular term in this research field? I recommend that the authors use this term in the text if it improves the readability of this manuscript.

The representation of x_i in Eq. (6) may represent that the spectrum shown Fig.R.1.a. I expected the indices of x_i represents discretized m/z value.

C. Please show the definition of the indices of x_i because the way to discretize the m/z value is not obvious.

2 On the response to Comment 2

I confirmed that the authors modified the related part in the Method section. However, the authors need to modify the sentences related to KART. The sentence “Kolmogorov–Arnold representation theorem (KART) states that any multivariate continuous function $\Phi(Y_1, \dots, Y_n)$ can be represented as a finite sum of univariate continuous functions.” is incorrect. This sentence represents the following equation,

$$\Phi(Y_1, \dots, Y_n) = \sum_{i=1}^N \sum_{j=1}^n \varphi_{i,j}(Y_j). \quad (1)$$

D. I recommend that the authors directly show KART as an equation as follows if they need to show in the text:

$$\Phi(Y_1, \dots, Y_n) = \sum_{q=1}^{2n+2} \psi_q \left(\sum_{p=1}^n \phi_{q,p}(Y_p) \right). \quad (2)$$

D. I think that the authors directly show KART as an equation as follows if they need to show in the text. However, I recommend removing the sentences about $\Phi(Y_1, \dots, Y_n)$, because it does not help understanding the WMYn architecture. See the following responses.

3 On the response to Comment 3

The author explained that \mathbf{S} represents a vector. However, they also explained that \mathbf{S} have elements expressed by $S_{i,j}$. From the representation of the element, \mathbf{S} is expected as a matrix. In addition, in Eq. (16), the authors showed an example of a vector x_i representing a spectrum.

E. Is the form of $S_{i,j}$ different from that of x_i (vector)? If so, why does S have two indices, i and j ? What are the indices i and j ? They cannot be found on the right-hand side of Eq. (21).

Again, I recommend to remove the discussion about \mathbf{S} .

4 On the response to Comments 4 and 5

I would appreciate the author’s response, which helped me to understand the manuscript.

5 On the response to Comment 6

F. The bold style is not usually used to express “significant importance.” Please use the font styles to distinguish different mathematical objects, e.g., scalars in italic style and vectors in bold style.

6 On the response to Comment 7

I would appreciate the author's response, which helped me to understand the manuscript.

7 On the response to Comment 8

G. Please use the term “peak position” or a similar term to represent the value on the horizontal axis in spectrum plots.

8 On the response to Comments 9–18

I would appreciate the author's response, which helped me to understand the manuscript.

9 On the response to Comments 19 and 20

H. Please explicitly show the architecture of $WNY_n(\text{MLP})$, like Eq. (B), with the explicit dimensions of the weights and biases and the explicit activation functions. Please discuss the efficiency of $WNY_n(\text{KART})$ based on the explicit architectures of $WNY_n(\text{MLP})$ and $WNY_n(\text{KART})$. In addition, I recommend that the authors change the name of $WNY_n(\text{KART})$.

I was able to now run the code in RStudio given the instructions provided (which I assume are also on the website).

Pros:

- 1) the software ran extremely fast!
- 2) The software was relatively straightforward to use.
- 3) The software was relatively accurate: Spot checking the MS/MS spectra and RT trends, as well as benchmarking against LipidMatch, showed that the software was quite accurate in annotations.
- 4) Coverage was high, for example using the same file (adipose tissue) via LipidMatch 235 unique m/zs (two decimal points) were annotated whereas in LipidIN 243 unique m/zs were annotated showing similar coverage.

Cons:

- 1) Results are highly sensitive to the Abundance filter, and setting the abundance filter high creates many false negatives and low creates several false positives without any numbers allowed between. Coverage was relatively low when putting the abundance filter to the recommends 0.05, for example using the same file (adipose tissue) via LipidMatch 235 unique m/zs (two decimal points) were detected whereas only 160 were detected by LipidIN in positive mode. LipidIN missed some predominant lipids including PC(16:0_18:1) (one of the major common lipids) as well as all other PCs, due to their low acyl fragments. Hence the abundance filter was set to 0 (there was only one option 0.01 or 0, could not put 0.0001 as desired... At 0 several false positives occurred including MGDGs (plant lipids) in adipose (MGDG 22:2 | MGDG 8:2_14:0).
- 2) Retention times reported are for MS/MS scans not for peaks, with peak RTs and abundances shown
- 3) Most aspects of the workflow were not covered by the implementation recommended by the author to me. Peak picking and alignment across samples, gap filling, deisotoping, blank filtering, combining negative and positive mode, are some essential steps in the lipidomics workflow not covered by LipidIN. This will limit usability.

More detailed feedback:

The GUI does not like long directories (they don't paste fully and will cause errors).

You could distribute R packages and an R version with LipidIN to reduce future dependency issues.

The following installation in the R Code:

```
install.packages(  
  pkgs = '~/LipidIN-main/LipidIN GUI/LipidIN_2.0.0.1.tar.gz',  
  lib = .libPaths()[length(.libPaths())],  
  repos = NULL,  
  dependencies = T  
)
```

Errors when you put the exact directory (as mentioned in the instructions).

I needed to go to the same documents folder it was looking for and put the tar.gz there for successful installation, so this could be modified to be able to take a full path.

The retention times reported are from the MS/MS scans (time of MS/MS scan) which is not that retention time of the peak itself. For example, for the two TGs below RTs of 14.90 and 15.39 were reported, whereas the RTs at the peak maximum were not reported (14.96 and 15.31).

RT: 14.3668 - 16.2238

This may be why when I plotted RTs against m/z for the same lipid class and degree of unsaturation the trend was not perfect (albeit it's not always expected to be perfect). For reference I have included RT vs m/z trends from another software for a different dataset to show how clear they often are. Are the RTs which are used for retention time analysis the retention times reported in pos_ALL_final_output.csv, because if they are these are the incorrect RTs (the MS/MS scan RT is not the RT of the peak itself). Furthermore, if the reported RTs are not the ones used, the RTs of the peaks themselves should be reported.

Above is from LipidIN for a TG (0 unsaturation) profile in adipose tissue

Above is from LipidMatch for a TG (0 unsaturation) profile in cell lines

Above is from LipidIN for a TG (1 unsaturation) profile in adipose tissue

Above is from LipidMatch for a TG (1 unsaturation) profile in cell lines

I did a quick comparison to LipidMatch. I was able to get 160 unique annotations base on a unique m/z in lipidIN in positive mode to two decimal points. I was able to get 235 unique m/zs annotated in LipidMatch with high confidence in positive mode to two decimal points. I spot checked a few TGs and both LipidMatch and lipidIN gave the same top annotation.